# A multimodal cell census and atlas of the mammalian primary motor cortex

BRAIN Initiative Cell Census Network (BICCN)[1]*

Here we report the generation of a multimodal cell census and atlas of the mammalian primary motor cortex as the initial product of the BRAIN Initiative Cell Census Network (BICCN). This was achieved by coordinated large-scale analyses of single-cell transcriptomes, chromatin accessibility, DNA methylomes, spatially resolved single-cell transcriptomes, morphological and electrophysiological properties and cellular resolution input–output mapping, integrated through cross-modal computational analysis. Our results advance the collective knowledge and understanding of brain cell-type organization[1–5]. First, our study reveals a unified molecular genetic landscape of cortical cell types that integrates their transcriptome, open chromatin and DNA methylation maps. Second, cross-species analysis achieves a consensus taxonomy of transcriptomic types and their hierarchical organization that is conserved from mouse to marmoset and human. Third, in situ single-cell transcriptomics provides a spatially resolved cell-type atlas of the motor cortex. Fourth, cross-modal analysis provides compelling evidence for the transcriptomic, epigenomic and gene regulatory basis of neuronal phenotypes such as their physiological and anatomical properties, demonstrating the biological validity and genomic underpinning of neuron types. We further present an extensive genetic toolset for targeting glutamatergic neuron types towards linking their molecular and developmental identity to their circuit function. Together, our results establish a unifying and mechanistic framework of neuronal cell-type organization that integrates multi-layered molecular genetic and spatial information with multi-faceted phenotypic properties.

Unique among body organs, the human brain is a vast network of information processing units, comprising billions of neurons interconnected through trillions of synapses. Diverse neuronal and non-neuronal cells display a wide range of molecular, anatomical, and physiological properties that together shape the network dynamics and computations underlying mental activities and behaviour. Brain networks self-assemble during development, leveraging genomic information shaped by evolution to build a set of stereotyped network scaffolds that are largely identical among individuals; life experiences then customize neural circuits in each individual. An essential step towards understanding the architecture, development, function and diseases of the brain is to discover and map its constituent elements of neurons and other cell types.

The notion of a 'neuron type', with similar properties among its members, as the basic unit of brain circuits has been an important concept for over a century; however, rigorous and quantitative definitions have remained surprisingly elusive[1–5]. Neurons are remarkably complex and heterogeneous, both locally and in their long-range axonal projections, which can span the entire brain and connect to many target regions. Many conventional techniques analyse one neuron at a time, and often study only one or two cellular phenotypes in an incomplete way (for example, missing axonal arbours in distant targets). As a result, despite major advances in past decades, phenotypic analyses of neuron types have remained severely limited in resolution, robustness, comprehensiveness and throughput. Complexities in the relationship between different cellular phenotypes (multi-modal correspondence) have fuelled long-standing debates on neuronal classification[6].

Single-cell genomics technologies provide unprecedented resolution and throughput to measure the transcriptomic and epigenomic profiles of individual cells and have rapidly influenced many areas of biology including neuroscience, promising to catalyse a transformation from phenotypic description and classification to a mechanistic and explanatory molecular genetic framework for the cellular basis of brain organization. The application of single-cell RNA sequencing (scRNA-seq) to the neocortex and other brain regions has revealed a complex but tractable hierarchical organization of transcriptomic cell types that are consistent overall with knowledge from decades of anatomical, physiological and developmental studies but with an unmatched level of granularity[7–11]. Similarly, single-cell DNA methylation and chromatin accessibility studies have begun to reveal cell-type-specific genome-wide epigenetic landscapes and gene regulatory networks in the brain[12–15]. Notably, the scalability and high information content of these methods enable comprehensive quantitative analysis and classification of all cell types, which are readily applicable

*Lists of authors and their affiliations appear at the end of the paper.

to brain tissues across species and provide a quantitative means of comparative analysis[16,17].

Other recent technological advances provide the resolution and throughput to analyse whole-brain neuronal morphology and comprehensive projection mapping[18,19]. Imaging-based single-cell transcriptomics and its combination with functional imaging, and integration of electrophysiology and single-cell sequencing, enable mapping of the spatial organization and key phenotypic properties of molecularly defined cell types[20–24]. Finally, molecular classification of cell types enables genetic access to specific cell types using transgenic mice[25–27] and, more recently, enhancer-based viral vectors[28–32]. All of these methods have been applied to brain tissues in independent studies, but not yet in a coordinated fashion to establish how different modalities correspond with one another, and whether a molecular genetic framework is explanatory for other functionally important cellular phenotypes.

The overarching goal of the BRAIN Initiative Cell Census Network (BICCN) is to leverage these technologies to generate an open-access reference brain cell atlas that integrates molecular, spatial, morphological, connectional and functional data for describing cell types in mouse, human and non-human primate[33]. A key concept is the Brain Cell Census, similar conceptually to a population census, that defines the constituent neuronal and non-neuronal cell types and their proportions, spatial distributions and defining phenotypic characteristics. This cell-type classification, organized as a taxonomy, should aim for consensus across modalities and across mammalian species for conserved types. Beyond the cell census, a Brain Cell Atlas would be embedded in a 3D common coordinate framework (CCF) of the brain[34], in which the precise location and distribution of all cell types and their multi-modal features are registered and displayed. This spatial framework facilitates integration, interpretation and navigation of various types of information for understanding brain network organization and function.

Here we present the cell census and atlas of cell types in the primary motor cortex (MOp, referred to as M1 in primates) of mouse, marmoset and human (Extended Data Fig. 1, Extended Data Table 1). MOp is important in the control of complex movement and is well conserved across species, with a rich history of anatomical, physiological and functional studies to aid interpretation of this cell-type information[35,36]. We describe a synthesis of eleven companion studies through a coordinated multi-laboratory effort. In these studies, we derive a cross-species consensus molecular taxonomy of cell types using scRNA-seq or single-nucleus RNA sequencing (snRNA-seq), DNA methylation and chromatin accessibility data[37–40]. In mouse, we map the spatial cellular organization by multiplexed error-robust fluorescence in situ hybridization (MERFISH)[41], characterize morphological and electrophysiological properties by multimodal profiling using patch clamp recording, biocytin staining and scRNA-seq (Patch-seq)[42,43], describe the cellular input–output wiring diagrams by anterograde and retrograde tracing[44], identify glutamatergic neuron axon projection patterns by Epi-retro-seq[45], Retro-MERFISH[41] and single-neuron complete morphology reconstruction[46], and describe transgenic driver lines targeting glutamatergic cell types on the basis of marker genes and lineages[47]. Finally, we integrate this information into a cohesive description of cell types in MOp. These datasets are organized by the BRAIN Cell Data Center (BCDC) and made public through the BICCN web portal (https://www.biccn.org). Key concepts and terms are described in Extended Data Table 2, including anatomical terms for input and output brain regions for MOp, and hierarchical cell class, subclass and type definitions.

Major findings:
- Combined single-cell transcriptomic and epigenomic analysis reveals a unified molecular genetic landscape of cortical cell types that integrates gene expression, chromatin state and DNA methylation.
- A combination of single-cell 'omics, MERFISH-based spatially resolved single-cell transcriptomics and Patch-seq generates a census of cell types, including their proportions and spatial distribution across cortical layers and sublayers.
- Comparative analysis of mouse, marmoset and human transcriptomic types describes a conserved cross-species taxonomy of cortical cell types with hierarchical organization that reflects developmental origins; the transcriptional similarity of cell type granularity across species varies as a function of evolutionary distance.
- We observed highly conserved transcriptomic and epigenomic signatures of cell identity across species, as well as a large set of species-enriched cell-type gene expression profiles that suggests a high degree of evolutionary specialization.
- Correspondence among molecular, anatomical and physiological datasets reinforces the transcriptomic classification of neuronal subclasses and distinctive types, demonstrating their biological validity and genomic underpinnings, and also reveals continuously varying properties along these axes for some neuronal subclasses and types.
- Anatomical studies yield a cellular-resolution wiring diagram of mouse MOp anchored on major transcriptome-defined projection types, including input–output connectivity at the subpopulation level and output pathways at a genetically defined single-cell level.
- Long-range axon projection patterns of individual glutamatergic excitatory neurons exhibit a complex and diverse range of relationships with transcriptomic and epigenetic types (between one-to-one and many-to-many), suggesting another level of regulation in defining single-cell connectional specificity.
- Cell-type transcriptional and epigenetic signatures guide the generation of genetic tools for targeting glutamatergic pyramidal neuron types and fate mapping their progenitor types.
- Multi-site coordination within BICCN and data archives enabled a high degree of standardization, computational integration and creation of open data resources for community dissemination of data, tools and knowledge.

## Results

### Molecular definition of cell types in MOp

A mouse MOp molecular taxonomy was derived from seven scRNA-seq and snRNA-seq (sc/snRNA-seq) datasets and single-nucleus methylcytosine sequencing (snmC-seq2) and single-nucleus assay for transposase-accessible chromatin using sequencing (snATAC-seq) datasets[37]. The combined sc/snRNA-seq datasets contained a large number of cells profiled using both droplet-based and deep full-length sequencing methods (Extended Data Table 1), resulting in a consensus transcriptomic taxonomy with the greatest resolution compared with other data types, including 90 neuronal and 116 total clusters or transcriptomic types (t-types)[37]. We used this mouse MOp transcriptomic taxonomy as the anchor for comparison and cross-correlation of cell-type classification results across all data types. We further applied two computational approaches, SingleCellFusion (SCF) and LIGER, to combine the transcriptomic and epigenomic datasets and derive an integrated molecular taxonomy consisting of 56 neuronal cell types (corresponding to the 90 transcriptomic neuronal types)[37] (Fig. 1a). This integrated taxonomy linked RNA transcripts with epigenomic marks identifying potential cell-type-specific *cis*-regulatory elements (CREs) and transcriptional regulatory networks. Similarly, we established M1 cell-type taxonomies for human (127 t-types) and marmoset (94 t-types) by unsupervised clustering of snRNA-seq data, followed by integration with epigenomic datasets[38].

To establish a consensus classification of MOp and M1 cell types among mouse, human and marmoset, we integrated snRNA-seq datasets across species and identified 45 conserved t-types, including 24 GABAergic (γ-aminobutyric acid-producing), 13 glutamatergic and 8 non-neuronal types (Extended Data Fig. 2a). The similarity between types was represented as a consensus taxonomy, with branch robustness quantified by using different subsets of genes with variable expression (Fig. 1b). These

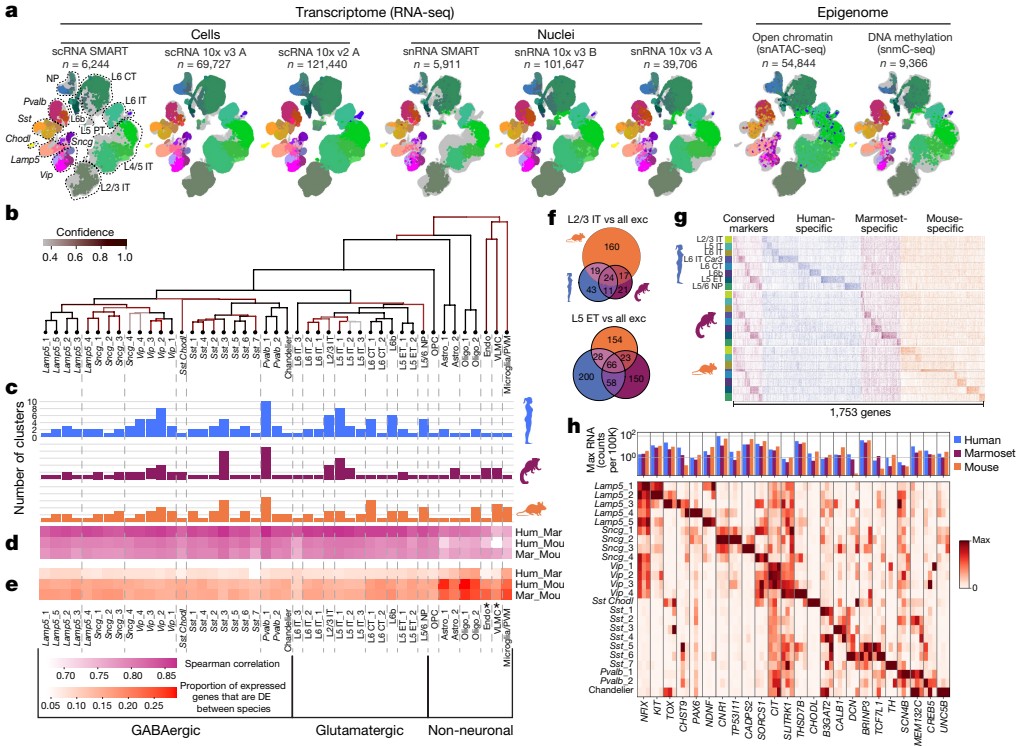

**Fig. 1 | MOp consensus cell-type taxonomy. a**, Integrated transcriptomic and epigenomic datasets using SCF show consistent molecular cell-type signatures as revealed by a low-dimensional embedding in mouse MOp. Each UMAP plot represents one dataset. Colours indicate cell subclasses. **b**, Dendrogram of integrated human (hum), marmoset (mar) and mouse (mou) cell types based on snRNA-seq datasets (10x v3). Branch colour denotes confidence after 10,000 bootstrap iterations. **c**, Number of within-species clusters included in each cross-species cluster. **d**, **e**, For each cross-species cluster, correlations (**d**) and differentially expressed genes (Wald two-sided test, adjusted *P*-value < 0.05, fold-change > 4) (**e**) between pairs of species. Asterisks denote non-neuronal

populations that were under-sampled in human. DE, differentially expressed. **f**, Venn diagrams of shared differentially expressed genes between species for L2/3 IT and L5 ET subclasses. **g**, Conserved and species-enriched differentially expressed genes for all glutamatergic subclasses shown in an expression heat map. **h**, Conserved markers of GABAergic neuron types across three species. Data may be viewed at NeMO Analytics. Marmoset silhouette is adapted from www.phylopic.org (public domain). Exc, excitatory; max, maximum; astro, astrocyte; endo, endothelial cell; oligo, oligodendrocyte; OPC, oligodendrocyte progenitor cell; PVM, perivascular macrophage.

types were grouped into broader subclasses on the basis of shared developmental origins for GABAergic inhibitory neurons (that is, three caudal ganglionic eminence (CGE)-derived subclasses (*Lamp5*, *Sncg* and *Vip*) and three medial ganglionic eminence (MGE)-derived subclasses (*Sst Chodl*, *Sst* and *Pvalb*)), layer and projection pattern in mouse for glutamatergic excitatory neurons (that is, intratelencephalic (IT), extratelencephalic (ET), corticothalamic (CT), near-projecting (NP) and layer 6b (L6b)), and non-neuronal functional subclasses (for example, oligodendrocytes and astrocytes) (Extended Data Table 2). Note that the layer 5 extratelencephalic (L5 ET) neurons have been called pyramidal tract (PT) or subcerebral projection neurons (SCPN)[48,49]; here we use the name L5 ET to be more accurate across cortical areas and species (Methods).

The resolution of this cross-species consensus taxonomy was lower than that derived from each species alone, owing to variation in gene expression across species. The degree of species alignments varied across consensus types (Fig. 1c); some types could be aligned one-to-one (for example, *Lamp5*_1 and L6 IT_3), whereas others aligned several-to-several (for example, *Pvalb*_1, L2/3 IT and L5 IT_1). This may reflect over- or under-clustering, limitations in aligning highly similar cell types, or species-specific expansion of cell-type diversity.

We expected that cell types from more recent common ancestors would share more similar gene expression profiles. Indeed, transcriptomic profiles of consensus cell types were more correlated between human and marmoset, and had 25–50% fewer differentially expressed genes than between primate and mouse (Fig. 1d, e). The one exception was the vascular leptomeningeal cell (VLMC) type, which had greater Spearman correlations of overall gene expression (Fig. 1d)

between marmoset and mouse. However, this probably reflects that rare non-neuronal cells in human (*n* = 40 nuclei) were under-sampled compared with marmoset (*n* = 463) and mouse (*n* = 2,329), and average expression was not adequately estimated[38].

Glutamatergic subclasses expressed 50–450 marker genes and, unexpectedly, the majority of markers were species-enriched (Fig. 1f, g). This evolutionary divergence of marker gene expression may reflect species adaptations or relaxed constraints on genes that can be substituted with others for related cellular functions. Glutamatergic subclasses also had a core set of 5–65 markers that were conserved across all three species (Fig. 1g); these genes are candidates for conserved cell identity and function, and are useful for consistent labelling across species. GABAergic subclasses expressed 50–325 markers in each species, and 18–55 markers were conserved. At a finer level, GABAergic consensus types also expressed conserved markers with similar expression levels across species and relatively type-specific expression (Fig. 1h). Some marker genes also showed evidence for cell-type-specific enhancers located in regions of open chromatin and DNA hypomethylation in both human and mouse (Extended Data Fig. 2b, c).

### Spatially resolved cell atlas of mouse MOp
We used MERFISH, a single-cell transcriptome imaging method[50,51], to identify cell types in situ and map their spatial organization. We selected a panel of 258 genes (254 of which passed quality control) on the basis of prior knowledge of marker genes for major cortical cell types and genes identified using sc/snRNA-seq data, and we imaged approximately 300,000 individual cells across MOp and adjacent areas[41].

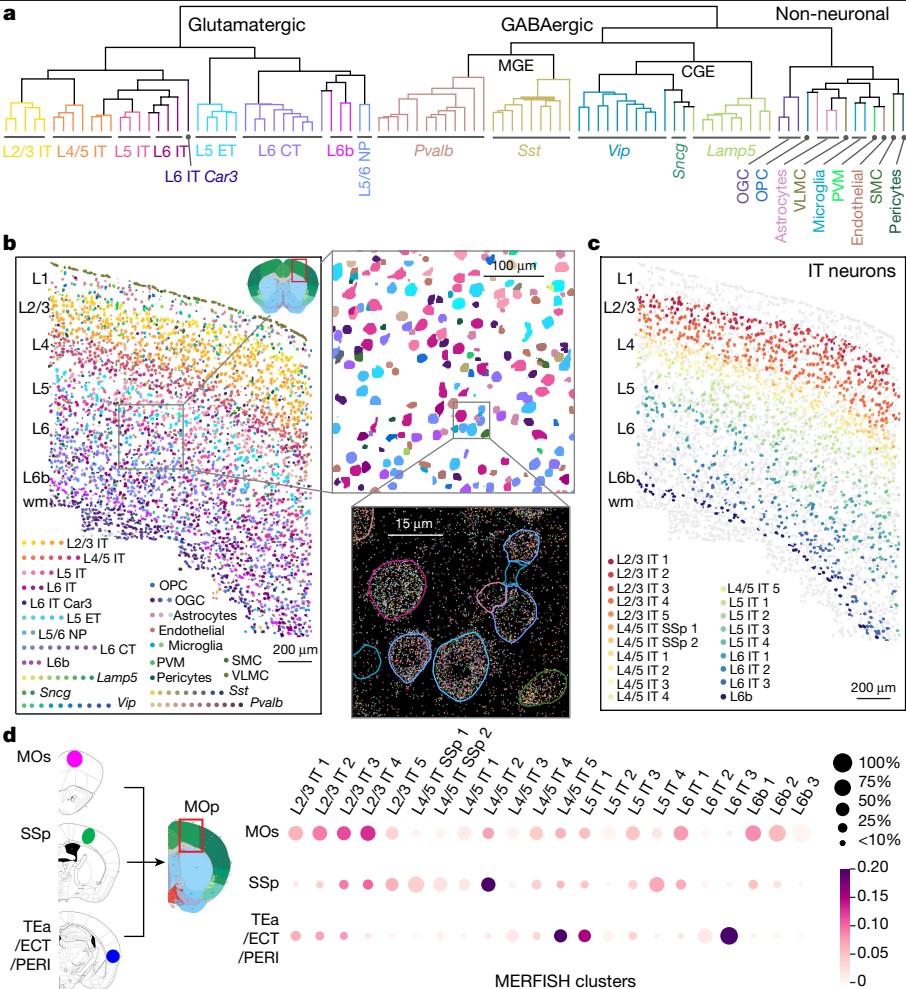

**Fig. 2 | In situ cell-type identification, spatial mapping and projection mapping of individual cells in MOp by MERFISH. a**, Dendrogram showing the hierarchical relationship among the subclasses and clusters in the mouse MOp identified by MERFISH, coloured by the subclass each cluster belongs to. **b**, Left, spatial map of cell clusters identified in a coronal slice (bregma ~+0.9), with cells coloured by their cluster identity as shown in the colour index. Top right, zoomed-in map of the boxed region of the left panel. Bottom right, spatial localization of individual RNA molecules in the boxed region of the top right panel, coloured by their gene identity. The segmented cell boundaries are coloured according to the cell clusters they belong to. **c**, IT neurons in the same coronal slice as shown in **b**. The IT neurons are coloured by their cluster identity, as shown in the colour index, together with L6b cells in dark blue to

mark the bottom border of the cortex. All other cells are shown in grey. **d**, Projection patterns of the MOp neurons into three other regions of the brain, MOs, SSp and TEa–ECT–PERI. Left, CTb was used as retrograde tracer and injected into these three regions. The CTb signals and the MERFISH gene panel were imaged in MOp to determine both the cluster identities and projection targets of individual cells. Projection of MOp neurons to the target regions are displayed as a dot plot, where the size of the dot represents the fraction of cells projecting to each indicated target among all CTb-positive, single-projecting cells in a cluster, and the colour represents the fraction of cells a target received from each indicated cluster. Data may be viewed at NeMO Analytics. OGC, oligodendrocyte; SMC, smooth muscle cell.

Clustering analysis of the MERFISH-derived single-cell expression profiles resulted in a total of 95 cell clusters in MOp (42 GABAergic, 39 glutamatergic and 14 non-neuronal) (Fig. 2a), which showed excellent, essentially one-to-one correspondence to the consensus sc/snRNA-seq taxonomy at subclass level (for example, glutamatergic IT, ET, NP, CT and L6b subclasses, and GABAergic *Lamp5*, *Sncg*, *Vip*, *Sst* and *Pvalb* subclasses) and good correspondence at cluster level[41].

Spatial distribution of the MERFISH clusters showed a complex, laminar pattern in MOp (Fig. 2b). Many glutamatergic clusters showed narrow distributions along cortical depth that subdivided individual cortical layers, although frequently without discrete boundaries[41]. Notably, IT cells, the largest branch of neurons in the MOp, formed a largely continuous gradient of cells with correlated gradual changes between their expression profiles and their cortical depths[41] (Fig. 2c). Many GABAergic clusters also showed laminar distribution, preferentially residing within one or two layers[41]. Among the non-neuronal

cell clusters, VLMCs formed the outermost layer of cells of the cortex, whereas mature oligodendrocytes and some astrocytes were enriched in white matter. Other subclasses of non-neuronal cells were largely dispersed across all layers. MERFISH analysis also revealed interesting spatial distribution of cell types along the medial–lateral and anterior–posterior axes[41]. Overall, the neuronal and non-neuronal cell clusters in MOp form a complex spatial organization refining traditionally defined cortical layers.

Integration of retrograde tracing with MERFISH (Retro-MERFISH) identified projection targets of different neuron types in the MOp[41] (Fig. 2d). Retrograde tracers were injected into secondary motor cortex (MOs), primary somatosensory cortex (SSp), and temporal association (TEa) and neighbouring ectorhinal (ECT) and perirhinal (PERI) areas, and retrograde labels were imaged together with the MERFISH gene panel in the MOp (approximately 190,000 cells were imaged). Each of the three target regions received inputs from multiple cell clusters in

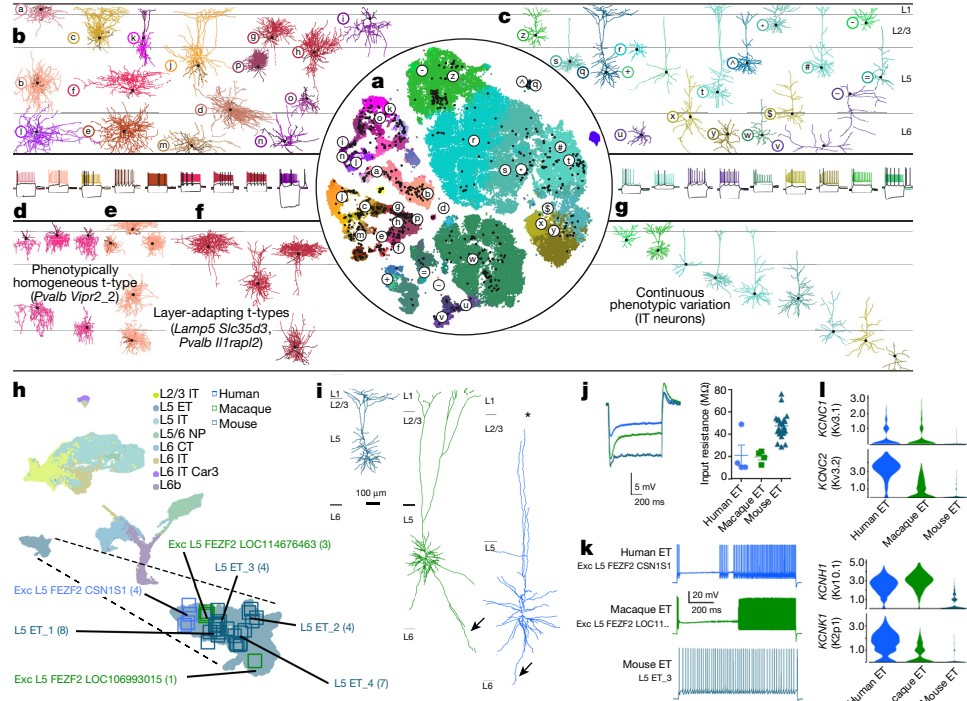

**Fig. 3 | Correspondence between transcriptomic and morpho-electrical properties of mouse MOp neurons by Patch-seq, and cross-species comparison of L5 ET neurons. a**, *t*-Distributed stochastic neighbor embedding (*t*-SNE) of the scRNA-seq 10x v2 dataset with the superimposed Patch-seq neurons[71] (black dots). **b**, **c**, Examples of GABAergic interneuron (**b**) and glutamatergic excitatory neuron (**c**) morphologies and electrophysiological recordings. Letters and symbols refer to cells marked in **a**. Three voltage traces are shown in each cell: the hyperpolarization trace obtained with the smallest current stimulation, the first depolarization trace that elicited at least one action potential, and the depolarization trace showing maximal firing rate. Stimulation length, 600 ms. **d**, Example of a phenotypically homogeneous t-type (*Pvalb Vipr2*_2, chandelier neurons). **e**, **f**, Two examples of t-types showing layer-adapting morphologies: *Lamp5 Slc35d3*, neurogliaform cells (**e**) and *Pvalb Il1rapl2*, fast-spiking basket cells (**f**). **g**, Example of a transcriptomic subclass (excitatory IT neurons) that shows continuous within-subclass co-variation between distances in transcriptomic space and morphological space, as seen in similar colour ordering in **a**

(right) and **g**. **h**, UMAP visualization of cross-species integration of snRNA-seq data for glutamatergic neurons isolated from mouse, macaque and human, with colours corresponding to cell subclass. Patch-seq samples mapping to various ET neuron types are denoted by squares, colour-coded by species. **i**, Dendritic reconstructions of L5 ET neurons. The human and macaque neurons display classical Betz cell features including taproot dendrites (arrows). Note that the human neuron is truncated (asterisk) before reaching the pial surface. **j**, Voltage response of mouse, macaque and human ET neurons to a 1 s, −300 pA current injection (left) and input resistance (mean ± s.e.m.; macaque *n* = 4, human *n* = 4, mouse *n* = 22) (right). False-discovery rate (FDR)-corrected two-sided Wilcoxon ranked-sum test (human versus mouse *W* = 12, *P* = 0.31, *d* = 2.09; human versus macaque *W* = 5, *P* = 0.49, *d* = 0.08; macaque versus mouse *W* = 0, *P* = 0.0004, *D* = 2.5). **k**, Example spike trains in response to a 10-s suprathreshold current injection. **l**, Violin plots of enriched potassium channel gene expression in human, macaque and mouse L5 ET neurons. Data may be viewed at NeMO Analytics.

the MOp, primarily from IT cells; each IT cluster projected to multiple regions, with each region receiving input from a different composition of IT clusters[41] (Fig. 2d). Overall, projections of MOp neurons do not follow a simple 'one cell type to one target region' pattern, but rather form a complex multiple-to-multiple network.

### Multimodal analysis of cell types with Patch-seq

We used Patch-seq to characterize the electrophysiological and morphological phenotypes and laminar location of t-types. We patched more than 1,300 neurons in MOp of adult mice, recorded their electrophysiological responses to a set of current steps, filled them with biocytin to recover their morphologies (around 50% of cells) and obtained their transcriptomes using Smart-seq2 sequencing[42]. We mapped these cells to the mouse MOp transcriptomic taxonomy[37] (Fig. 1a). Cells were assigned to 77 t-types (Fig. 3a), thereby characterizing the morpho-electric phenotypes of most glutamatergic and GABAergic t-types (examples in Fig. 3b, c).

We found that morpho-electric phenotypes were largely determined by transcriptomic subclasses, with different subclasses having distinct phenotypes. For example, *Sst* interneurons were often characterized

by large membrane time constants, pronounced hyperpolarization sag, and rebound firing after stimulation offset. However, within each subclass, there was substantial variation in morpho-electric properties between t-types. This variation was not random but organized such that transcriptomically similar t-types had more similar morpho-electric properties than distant t-types. For example, excitatory t-types from the IT subclasses with more similar transcriptomes were also located at adjacent cortical depths, suggesting that distance in t-space co-varied with anatomical distance[42], even within a layer (Fig. 3g), in line with the above MERFISH results (Fig. 2c). Similarly, electrophysiological properties of *Sst* interneurons varied continuously across the transcriptomic landscape[42]. Thus, within major transcriptomic subclasses, morpho-electric phenotypes and/or soma depth frequently varied smoothly across neighbouring t-types, indicating that transcriptomic neighbourhood relationships in many cases corresponded to similarities in other modalities.

At the level of single t-types, some t-types showed layer-adapting morphologies in different layers (Fig. 3e, f) or even considerable within-type morpho-electric variability within a layer. For example, *Vip Mybpc1*_2 neurons had variable rebound firing strength after

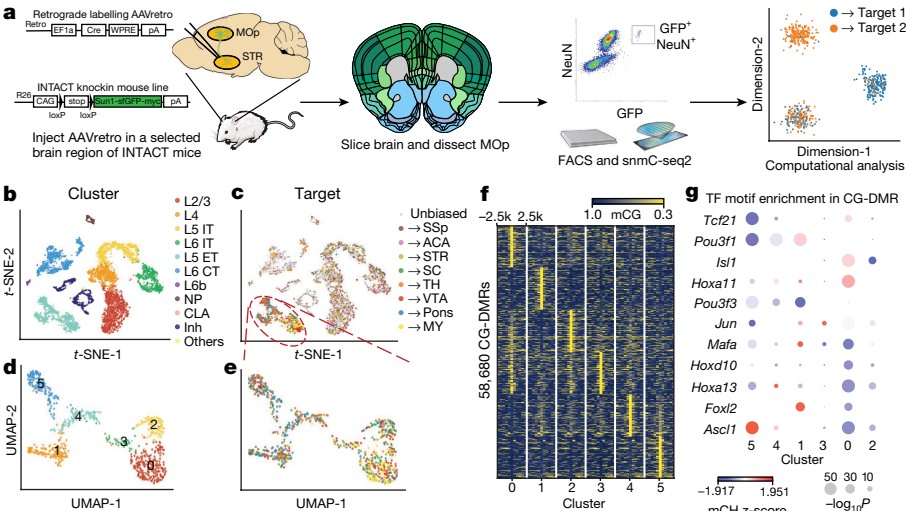

**Fig. 4 | Epi-retro-seq links molecular cell types with distal projection targets. a**, Workflow of Epi-retro-seq. Mouse brain sagittal panel is adapted from https://commons.wikimedia.org/wiki/File:Mouse_brain_sagittal.svg (public domain). Coronal reference plane is adapted from Allen Reference Atlas with permission. FACS, fluorescence activated cell sorting. **b**, **c**, *t*-SNE of MOp cells profiled by Epi-retro-seq (*n* = 2,115) and unbiased snmC-seq2 (*n* = 4,871) computed with 100-kb-bin-level mCH, coloured by subclasses (**b**) or projection targets (**c**). **d**, **e**, UMAP embedding of L5 ET cells in MOp profiled by

Epi-retro-seq (*n* = 848) computed with 100-kb-bin-level mCH, coloured by clusters (**d**) or projection targets (**e**). **f**, mCG levels at CG-DMRs identified between the six clusters and their flanking 2.5-kb regions. Top 100 CG-DMRs in each cluster are shown. **g**, Transcription factor motif enrichment in CG-DMRs in each cluster. Colour represents *z*-scored gene-body mCH level of the transcription factors, and size represents −log₁₀ *P* value (computed with Homer, using one-sided binomial tests) of motif enrichment in the CG-DMRs. CLA, claustrum; inh, inhibitory.

stimulation offset. Surprisingly few t-types were entirely homogeneous with regard to the measured morpho-electric properties (Fig. 3d).

Patch-seq also enables direct comparison of the morpho-electric properties of homologous cell types across species. Here we analysed the gigantocellular Betz cells found in M1 of primates and large carnivores, which are predicted to be in the L5 ET subclass[38], as are the mouse corticospinal-projecting L5 ET neurons. We first created a joint embedding of excitatory neurons in mouse, macaque and human, which showed strong homology across all three species for the L5 ET subclass (Fig. 3h). Patch-seq recordings were made from L5 neurons in acute and cultured slice preparations of mouse MOp and macaque M1. We also capitalized on a unique opportunity to record from neurosurgical tissue excised from human premotor cortex—which also contains Betz cells—during surgery to treat epilepsy. To enable visualization of cells in heavily myelinated macaque M1 and human premotor cortex, we used adeno-associated viruses (AAVs) to drive fluorophore expression in glutamatergic neurons in slice culture.

Patch-seq cells in each species that mapped to the L5 ET subclass (Fig. 3h) were all large L5 neurons that sent apical dendrites to the pial surface (Fig. 3i). Macaque and human L5 ET neurons were much larger, with hallmark Betz cell long 'taproot' basal dendrites[52]. Subthreshold membrane properties were relatively well conserved across species. For example, L5 ET neurons in all three species had low input resistances, although they were exceptionally low in macaque and human (Fig. 3j). Conversely, suprathreshold properties of macaque and human Betz ET neurons were highly specialized; they responded to prolonged suprathreshold current injections with biphasic firing in which a pause in firing early in the sweep was followed by a marked increase in firing later (Fig. 3k). Intriguingly, several genes encoding ion channels were enriched in macaque and human L5 ET neurons compared with mouse (Fig. 3l), and may contribute to the distinctive primate suprathreshold properties. These results indicate that primate Betz cells are homologous to mouse thick-tufted L5 ET neurons, but display species specializations in their morphology, physiology and gene expression.

## Multimodal correspondence by Epi-retro-seq

To understand molecular diversity among projection neurons, we developed Epi-retro-seq[45]—which combines retrograde tracing and epigenomic profiling—and applied it to mouse MOp neurons projecting to each of the eight selected brain regions receiving inputs from MOp (Fig. 4a). Th-target regions included two cortical areas, SSp and anterior cingulate area (ACA), and six subcortical areas, striatum (STR), thalamus (TH), superior colliculus (SC), ventral tegmental area and substantia nigra (VTA+SN), pons and medulla (MY).

We obtained methylomes for 2,115 MOp projection neurons. Co-clustering them with MOp neurons collected without enrichment of specific projections, we observed precise agreement among all major cell subclasses (Fig. 4b, c). We observed enrichment of cortico-cortical and cortico-striatal projecting neurons in IT subclasses (L2/3, L4, L5 IT, L6 IT and L6 IT *Car3*), and cortico-subcortical projecting neurons in L5 ET. Many cortico-thalamic projecting neurons were also observed in L6 CT (Extended Data Fig. 3a). Consistent with the specificity of retrograde labelling, quantitative comparisons with unbiased collection of neurons in MOp suggest at least 30-fold (IT) or 200-fold (ET) enrichment of neurons in the expected subclasses (Methods).

Enrichment of L5 ET neurons with Epi-retro-seq (40.2% versus 5.62% in unbiased profiling of MOp using snmC-seq2) enabled investigation of subtypes of L5 ET neurons known to project to multiple subcortical targets in TH, VTA+SN, pons and MY[48]. The 848 L5 ET neurons were segregated into 6 clusters (Fig. 4d, e). MY-projecting neurons showed clear enrichment for L5 ET cluster 0 (Fig. 4e, Extended Data Fig. 3b), in agreement with scRNA-seq data for anterolateral motor cortex (ALM), part of MOs[9,53]. We used gene body non-CG methylation (mCH) levels to integrate the L5 ET Epi-retro-seq cells with the ALM Retro-seq cells and observed enrichment of MY-projecting cells in the same cluster[45].

The presence of mCH in gene bodies is strongly anti-correlated with gene expression in neurons, whereas promoter-distal differentially CG-methylated regions (CG-DMRs) are reliable markers of regulatory elements such as enhancers[12]. We identified 511 differentially CH-methylated genes (CH-DMGs) and 58,680 CG-DMRs across the L5 ET clusters (Fig. 4f). We also inferred transcription factors that

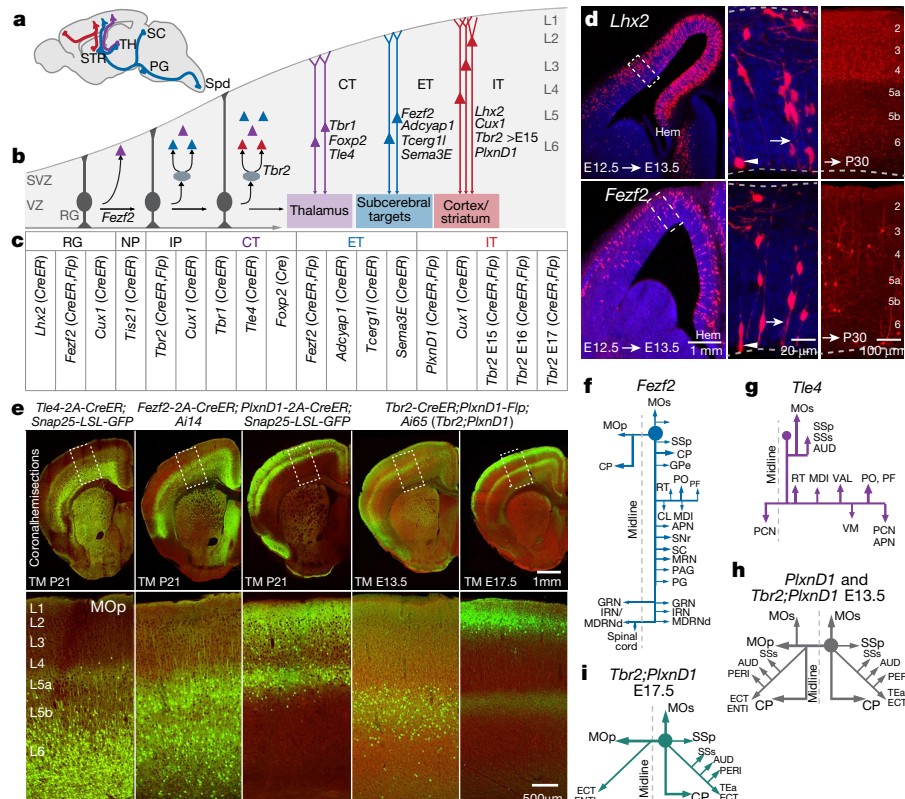

**Fig. 5 | Genetic tools for targeting cortical glutamatergic projection neuron types. a**, Major PyN projection classes mediating IT (red) and cortical output channels (ET, blue; CT, purple). PG, pontine grey; Spd, spinal cord. **b**, Developmental trajectory from progenitors to mature PyNs. Genes specify lineage and projection types. VZ, ventricular zone; SVZ, subventricular zone. **c**, New gene knockin driver mouse lines targeting RG, neurogenic precursor (NP), IP and broad projection types. **d**, Pulse-chase of E12.5 *Lhx2-2A-CreER;Ai14* (*Lhx2*) (top row) and *Fezf2-2A-CreER;Ai14* (*Fezf2*) (bottom row) embryos for 24 hours densely labelled RGs throughout dorsal neuroepithelium (left). Middle, boxed areas shown left, magnified, showing RG*Lhx2+* and RG*Fezf2+*. Long pulse-chase (right) of E12.5 RGs generates PyNs across layers at postnatal day (P)30. Arrows show endfeet and arrowheads show dividing soma. Hem, cortical hem. **e**, Driver–reporter recombination patterns (reporter, pseudo-coloured green; background, red) from five PyN subpopulations defined by *Tle4*, *Fezf2*, *PlxnD1* and *Tbr2;PlxnD1* with tamoxifen (TM) induction times. Combinatorial definition of PyN*PlxnD1* subtypes by lineage, birth time and anatomical location achieved by *Tbr2;PlxnD1* intersection: tamoxifen at E13.5 and at E17.5 labelled

different *Tbr2*-expressing IP-derived PyN*PlxnD1* cohorts. Boxed areas in MOp (top row) are shown in the bottom row. **f–i**, Main PyN subpopulation projection targets from MOp. Drivers were crossed with mouse reporter lines, *Rosa26-CAG-LSL-Flp* (Cre-dependent) or *Rosa26-CAG*-dual-*(LSL-FSF)*-tTA (Cre-AND-Flp-dependent), and tamoxifen induction was performed to convert transient CreER to constitutive reporter expression for anterograde tracing with Flp- or tTA-dependent AAV vector (AAV8-CAG-fDIO-TVA-EGFP or AAV-TRE-3g-TVA-EGFP). Filled circle shows MOp injection site. For full names of projection target acronyms, see refs. [34,47]. APN, anterior pretectal nucleus; AUD, auditory cortex; CL, central lateral nucleus; ENTI, entorhinal area, lateral part; GPe, globus pallidus, external segment; GRN, gigantocellular reticular nucleus; IRN, intermediate reticular nucleus; MDl, mediodorsal nucleus, lateral part; MDRNd, medullary reticular nucleus, dorsal part; MRN, midbrain reticular nucleus; PAG, periaqueductal grey; PCN, paracentral nucleus; PF, parafascicular nucleus; RT, reticular nucleus; SNr, substantia nigra, reticular part; SSs, supplemental somatosensory cortex.

may contribute to defining the cell clusters by identifying enriched transcription factor-binding DNA sequence motifs within CG-DMRs (Fig. 4g). For example, *Ascl1* is a transcription factor whose motif was significantly enriched in the MY-projecting cluster. In addition, 230 hypo-CH-DMGs were identified between the MY-projecting cluster and other projection neurons. One of the most differentially methylated genes is *Ptprg* (Extended Data Fig. 3c), which encodes the receptor tyrosine phosphatase-γ, which interacts with contactin proteins to mediate neural projection development[54]. Thus, these epigenomic mapping data for projection neurons facilitate the understanding of gene regulation in establishing neuronal identity and connectivity.

## Cell-type-targeting tools

Genetic access to specific neural subpopulations and progenitors is necessary for multi-modal analyses to validate t-types, fate-map their developmental trajectories, and study their function in circuit operation[25]. Here we present a genetic toolkit for dissecting and fate-mapping

glutamatergic pyramidal neuron (PyN) subpopulations largely on the basis of their developmental genetic programs.

Along the lineage progression of neural progenitors during corticogenesis in the embryonic dorsal telencephalon, radial glial progenitors (RGs) generate PyNs either directly or indirectly through intermediate progenitors (IPs)[55] (Fig. 5a, b). Temporal expression of transcription factors gates sequential developmental decisions to shape hierarchically organized PyN subpopulations[47,56]. The LIM-homeodomain protein LHX2 and zinc-finger transcription factor FEZF2 act at multiple stages of neurogenesis[55,57], and IPs specifically express the T-box transcription factor *Tbr2* during indirect neurogenesis[58]. We generated temporally inducible *Lhx2-CreER*, *Fezf2-CreER*, *Tbr2-CreER*, *Fezf2-Flp* and *Tbr2-FlpER* driver lines (Fig. 5c) that faithfully recapitulate the spatiotemporal expression of these transcription factors and enable fate-mapping of associated RG and IP pools[47]. For example, *Lhx2-CreER* and *FezF2-CreER* drivers captured embryonic day (E)12.5 RGs in the dorsal neuroepithelium, distributed along a medial-high and lateral-low

gradient, consistent with their mRNA expression at this stage[59,60]. These RGs generated PyNs across all cortical layers, suggesting multipotency (Fig. 5d).

We also generated 15 Cre and Flp driver lines targeting PyN subpopulations, including the CT, PT and IT subclasses, and subpopulations within these subclasses (Fig. 5b, c). These driver lines precisely recapitulated endogenous expression patterns, highlighted here with three representative lines (Fig. 5e): L2/3 and L5a for IT-*Plxnd1* (IT*Plxnd1*), L5b and L6 for ET-*Fezf2* (ET*Fezf2*), L6 for CT-*Tle4* (CT*Tle4*). Anterograde projection tracing in MOp of adult animals demonstrated that IT*Plxnd1* projected to multiple ipsilateral and contralateral cortical areas and to STR/caudate putamen (CP); ET*Fezf2* projected robustly to several ipsilateral cortical sites, CP and numerous subcortical targets including TH, MY and corticospinal tract; CT*Tle4* projected specifically to a set of thalamic nuclei[47] (Fig. 5f–h).

We further developed a combinatorial method to target PyN subtypes on the basis of their lineage, birth order and anatomical features. For example, the PyN*PlxnD1* population localizes to L5a, L3 and L2 and projects to many ipsilateral and contralateral cortical and striatal targets[47] (Fig. 5e, h). Based on the knowledge that most IT PyNs are generated from IPs[61], we generated *PlxnD1-Flp;Tbr2-CreER;Ai65* compound mice in which the inducible *Tbr2-CreER* allele was used to birth date IT*PlxnD1*. Tamoxifen induction at E13.5 and 17.5 selectively labelled L5a and L2 IT*PlxnD1*, respectively, across cortical areas (Fig. 5e). To reveal their projection patterns, we bred the *PlxnD1-Flp;Tbr2-CreER;*dual-tTA mice for tTA-dependent viral tracing in MOp. We found that E13.5-born IT*PlxnD1(E13.5)* neurons resided in L5a and projected ipsilaterally to multiple cortical areas, contralaterally to homotypic and heterotypic areas, and bilaterally to CP (Fig. 5h). By contrast, E17.5-born IT*PlxnD1(E17.5)* neurons resided in L2; although they also projected to ipsilateral cortical and striatal targets, and to homotypic contralateral cortex, they extended minimal projections to heterotypic contralateral cortex and CP (Fig. 5i). Together, this set of PyN driver lines provides much-improved specificity, robustness, reliability and coverage, and demonstrates feasibility to target highly specific PyN subtypes.

## MOp input–output wiring diagram

A comprehensive cellular resolution input–output MOp wiring diagram was generated by combining classic tracers, genetic viral labelling in Cre driver lines and single-neuron reconstructions with high-resolution, brain-wide imaging, precise 3D registration to CCF and computational analyses[44].

We first systematically characterized the global inputs and outputs of MOp upper limb (MOp-ul) region using classic anterograde (*Phaseolus vulgaris* leucoagglutinin (PHAL)) and retrograde (cholera toxin b (CTb)) tract tracing[44] (Fig. 6a). MOp-ul projects to more than 110 grey matter regions and spinal cord, and around 60 structures in the cerebral cortex and TH project back to MOp-ul.

We generated a fine-grained areal and laminar distribution map of multiple MOp-ul projection neuron populations using retrograde tracing[44] (Extended Data Fig. 4a). In parallel with these tracer-labelled, projection- and layer-defined cell populations, we characterized the distribution patterns in MOp-ul of neuronal populations labelled in 28 Cre driver lines, including those from different IT (for example, *Cux2*, *Plxnd1* and *Tlx3* driver lines), ET (*Rbp4*, *Sim1* and *Fezf2*) and CT (*Ntsr1* and *Tle4*) subclasses with distinct laminar distributions[47,62].

Viral tracers were used to systematically examine MOp-ul cell subclass-specific inputs and outputs[44] (Extended Data Fig. 4b). Neurons projecting to Cre-defined starter cells were labelled using trans-synaptic rabies viral tracers. Projections from MOp were labelled following AAV-GFP injection into wild-type mice, revealing patterns consistent with PHAL tracing (Fig. 6a). Projections from L2/3 IT, L4 IT, L5 IT, L5 ET and L6 CT cells were mapped following injections of Cre-dependent viral tracers into Cre lines selective for these laminar and projection cell subclasses[63]. Most Cre line anterograde tracing

experiments revealed a component of the overall output pathway. This result is consistent with labelling from retrograde injections in various thalamic nuclei (posterior complex (PO), ventral anterior-lateral complex (VAL) and ventral medial nucleus (VM)) and cortical areas such as MOs and SSp.

We systematically characterized axonal projections of more than 300 single MOp excitatory neurons, by combining sparse labelling, high-resolution whole-brain imaging, complete axonal reconstruction and quantitative analysis[44,46], augmented with publicly available single-cell reconstructions from the Janelia Mouselight project[18]. Additional analysis was also conducted using BARseq[44,64]. This analysis revealed a rich diversity of projection patterns within the IT, ET and CT subclasses (Fig. 6b). Individual L6 neurons display several distinct axonal arborization targets that likely contribute to the composite subpopulation output described for the *Ntsr1* and *Tle4* diver lines. Individual IT cells across L2–L6 also generate richly diverse axonal trajectories. Confirming and extending previous reports[53], we characterized detailed axon projections of the MY-projecting and non-MY-projecting L5 ET neurons, revealing complex axon collaterals in TH and midbrain regions[44,46].

## Multimodal characterization of L4 IT neurons in MOp

Traditionally MOp has been considered an agranular cortical area, defined by the lack of a cytoarchitectonic layer 4, which usually contains spiny stellate or star pyramid excitatory neurons. However, previous studies have suggested that L4 neurons similar to those typically found in sensory cortical areas are also present in mouse MOp and macaque M1[65,66]. Here we present multimodal evidence to confirm the presence of L4-like neurons in mouse MOp and primate M1 (Fig. 7).

We performed a joint clustering (Methods) and uniform manifold approximation and projection (UMAP) embedding of all IT neurons (excluding the highly distinct L6 IT Car3 cells) from 11 mouse molecular datasets, including 6 sc/snRNA-seq datasets, and the snmC-seq2, snATAC-seq, Epi-retro-seq, MERFISH and Patch-seq data (Fig. 7a). This resulted in five joint clusters, mostly along a continuous variation axis from L2/3 to L4/5 to L5 to L6 in line with the above MERFISH and Patch-seq results. The joint clustering enabled linkage of the cells independently profiled by each individual modality and cross-correlation of these disparate properties. Consequently, we identified epigenomic peaks linked to cluster-specific marker genes—*Cux2* for L2/3 IT and L4/5 IT (1), *Rspo1* for L4/5 IT (1), *Htr2c* for L4/5 IT (2-3), and *Rorb* for L4/5 IT and L5 IT (Fig. 7b, cluster names from SCF). MERFISH data showed that L4/5 IT and L5 IT cells occupied distinct layers in MOp, and the L4/5 IT type expressed *Rspo1* (Fig. 7c), a L4 cell-type marker in sensory cortical areas identified in previous studies[9]. There are fewer *Rspo1*+ L4 cells in MOp than in the neighbouring SSp. Transcriptomic IT types from mouse corresponded well with those from human and marmoset at subclass level, whereas substantial ambiguities existed at cluster level (Fig. 7d), probably owing to the gene expression variation between rodents and primates (Fig. 1).

We further compared the L4 cells in mouse MOp with those from mouse primary visual cortex (VISp)[9] after co-clustering all the SMART-seq glutamatergic transcriptomes from both regions (Fig. 7e). In UMAP, L4/5 IT cells in MOp occupied a subspace of the L4 IT co-cluster defined by the intersection of marker genes *Cux2* and *Rorb*, suggesting that L4 cells in MOp are similar to a subset of L4 cells in VISp, while the L4 cells in VISp have additional diversity and specificity.

L4 IT cells in MOp also exhibited morphological features characteristic of traditionally defined L4 excitatory neurons. In Patch-seq[42], cells from the L4/5 IT_1 type had no or minimal apical dendrites without tufts in L1, in contrast to cells from the L2/3 IT, L4/5 IT_2 and L5 IT types, which had tufted apical dendrites (Fig. 7f). We obtained complete morphological reconstructions of excitatory neurons with their somas located in L2, L3 or L4 in MOp or MOs from fMOST imaging of *Cux2-CreERT2;Ai166* mice[46]. The reconstructed MOp or MOs neurons with somas in putative

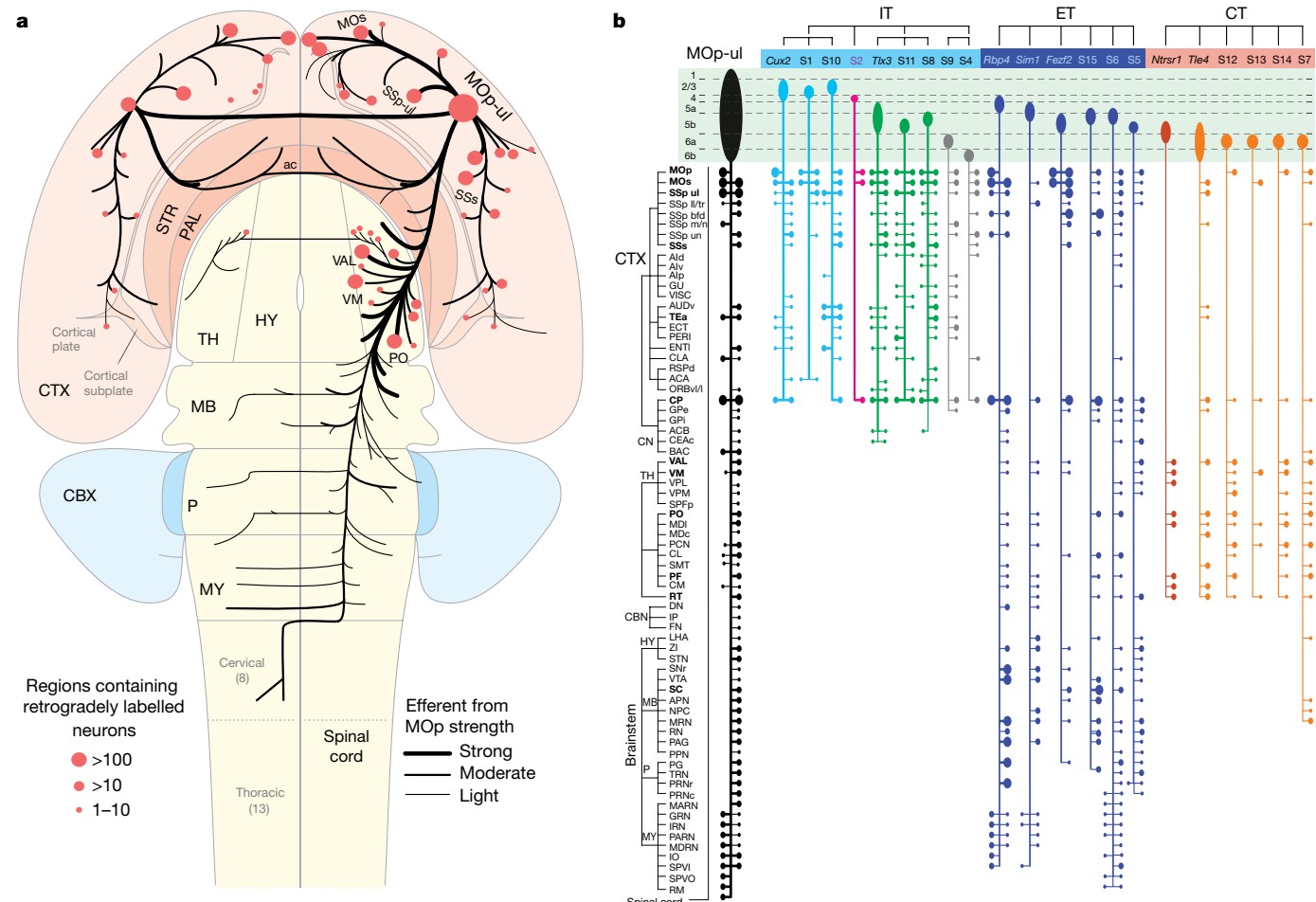

**Fig. 6 | Global wiring diagram and anatomical characterization of MOp-ul neuron types. a**, Flat map representation of the MOp-ul input–output wiring diagram. Black lines and red dots indicate axonal projections (outputs) and retrograde labelling sources (inputs), respectively, with line thickness and dot sizes representing relative connection strengths. Most MOp-ul projection targets in the cortex and TH also contain input sources, suggesting bi-directional connections. Numbers in parentheses indicate numbers of cervical or thoracic segments in spinal cord. **b**, Projection patterns arising from excitatory cell subclasses, IT, ET and CT, with corresponding Cre line assignment and somatic laminar location, compared with the overall projection pattern from the MOp-ul region (left, black). Along each vertical output pathway, horizontal bars on the right and left sides represent ipsilateral and contralateral collaterals, respectively, with dot sizes indicating the strength of axonal termination in different targets. For full names of projection target acronyms, see refs. [34,44]. ac, anterior commissure; ACB, nucleus accumbens; AId, v, p, agranular insular cortex, dorsal, ventral, posterior part; AUDv, ventral auditory cortex; BAC, bed nucleus of the anterior commissure; CBN, cerebellar nuclei; CBX, cerebellar cortex; CEAc, central

amygdalar nucleus, capsular part; CM, central medial nucleus; CN, cerebral nuclei; CTX, cerebral cortex; DN, dentate nucleus; FN, fastigial nucleus; GPi, globus pallidus, internal segment; GU, gustatory cortex; HY, hypothalamus; IO, inferior olivary complex; IP, interposed nucleus; LHA, lateral hypothalamic area; MARN, magnocellular reticular nucleus; MB, midbrain; MDc, mediodorsal nucleus, central part; MDRN, medullary reticular nucleus; NPC, nucleus of the posterior commissure; ORBvl, l, orbital cortex, ventrolateral, lateral part; P, pons; PAL, pallidum; PARN, parvicellular reticular nucleus; PPN, pedunculopontine nucleus; PRNr, pontine reticular nucleus; PRNc, pontine reticular nucleus, caudal part; RM, nucleus raphe magnus; RN, red nucleus; RSPd, retrosplenial cortex, dorsal part; SMT, submedial nucleus; SPFp, subparafascicular nucleus, parvicellular part; SPVI, spinal nucleus of the trigeminal, interpolar part; SPVO, spinal nucleus of the trigeminal, oral part; SSp-ul, -ll, -tr, -bfd, -m, -n, -un, primary somatosensory cortex upper limb, lower limb, trunk, barrel field, mouth, nose, unassigned; STN, subthalamic nucleus; TRN, tegmental reticular nucleus; VISC, visceral cortex; VPL, ventral posterolateral nucleus; VPM, ventral posteromedial nucleus; ZI, zona incerta.

L4 (between L2/3 and L5) exhibited two local morphological features typical of L4 neurons from sensory cortices (Fig. 7g). First, the dendrites of the L4 neurons were simple and untufted, whereas those of the L2/3 neurons all had extensive tufts. Second, the local axons of L4 neurons mostly projected upward into L2/3 in addition to collateral projections, whereas the local axons of L2/3 neurons had axon branches projecting downward into L5. These local projection patterns are consistent with the canonical feedforward pathways within a cortical column observed in somatosensory and visual cortices, with the first feedforward step from L4 to L2/3 and the second feedforward step from L2/3 to L5[67]. We also found that the MOp or MOs L4 neurons had intracortical long-range projections similar to the L2/3 neurons[46] (Fig. 6b).

## Multimodal characterization of L5 ET neurons in MOp

Previous studies showed that in mouse ALM, L5 ET neurons have two transcriptomically distinct projection types that may be involved in different motor control functions: the TH-projecting type in movement planning and the MY-projecting type in movement initiation[53]. Here we demonstrate that L5 ET neurons in mouse MOp also have MY-projecting and non-MY-projecting types, with distinct gene markers, epigenomic elements, laminar distribution, genetic targeting tools and corresponding types in human and marmoset.

Compared with the previous VISp–ALM transcriptomic taxonomy[9], mouse MOp L5 ET_1 type corresponded to the ALM MY-projecting type, whereas MOp L5 ET_2-4 types corresponded to the ALM TH-projecting

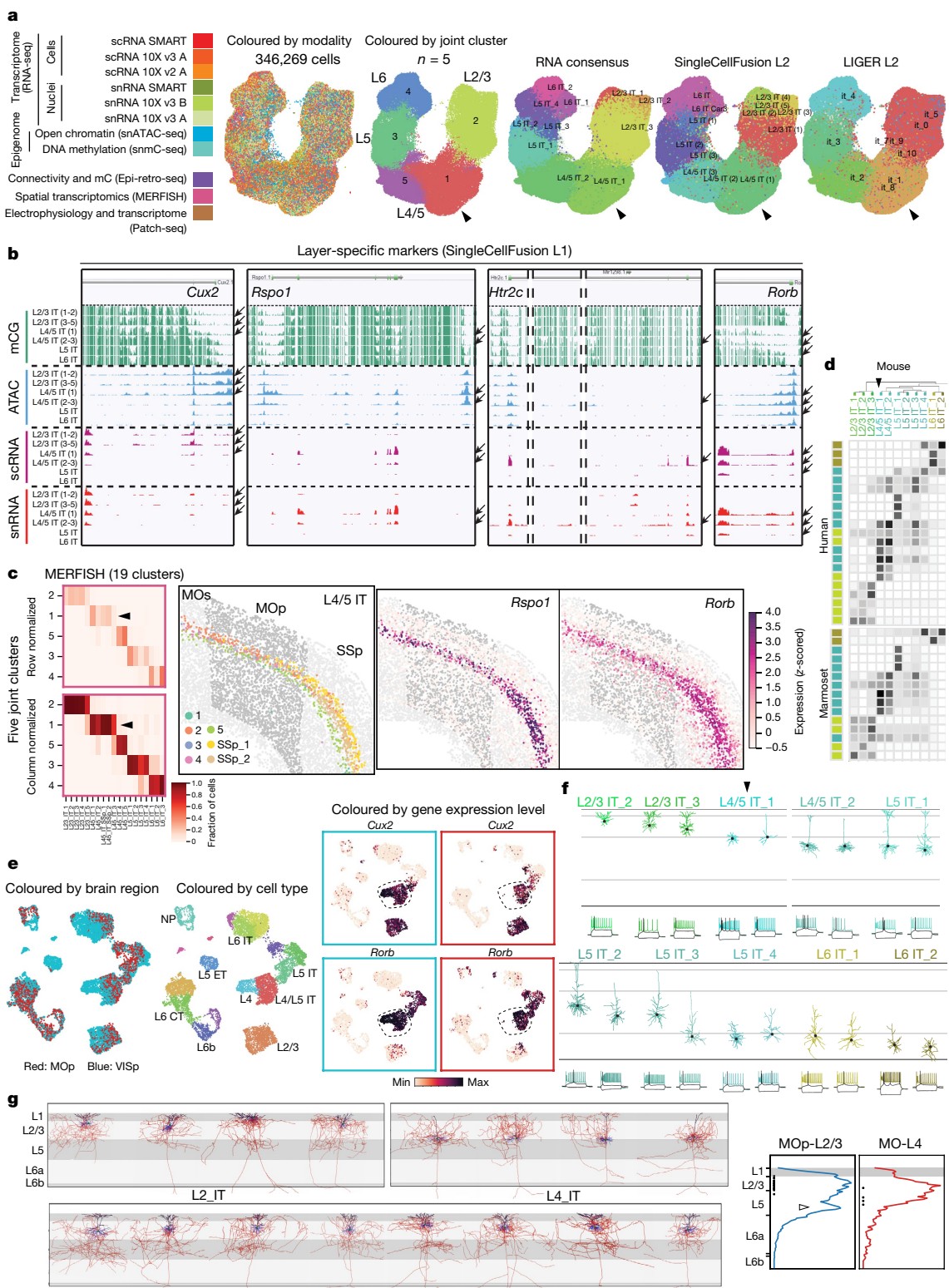

**Fig. 7 | Existence of L4 excitatory neurons in MOp. a**, UMAP embedding of IT cells from 11 datasets. Cells are coloured by modalities, by cluster identities from the 11-dataset joint clustering, and by cluster identities generated from other consensus clustering[37]. **b**, Genome browser view of layer-specific gene markers—from L2/3 to L5—across IT cell types (SCF L1)[37]. Arrows indicate cell types with correlated transcription and epigenomic signatures of the specific marker gene. **c**, MERFISH IT clusters correspond well with the joint clusters from **a** (confusion matrices, left), and reveal a group of L4 specific clusters (L45_IT) between L2/3 and L5 and marked by genes *Rspo1* and *Rorb* (right). **d**, Correspondence between mouse and human or marmoset transcriptomic IT types. **e**, UMAP embedding of excitatory cells from MOp and VISp. Gene expression levels are log₁₀(transcripts per million + 1). **f**, Dendritic morphologies and spiking patterns of mouse Patch-seq cells from L2/3-6 IT types. Arrowheads in **a**, **c**, **d**, **f** indicate the L4/5 IT_1 type. **g**, Left, local dendritic and axonal morphologies of fully reconstructed IT neurons with somas located in L2, L3 and L4. Black, apical dendrites. Blue, basal dendrites. Red, axons. Right, quantitative vertical profiles showing average distribution of local axons along cortical depth for L2/3 or L4 neurons. Dots indicate soma locations and the open arrowhead points to L2/3 neuron axon projections down to L5. Layer marking is approximate owing to the variable thickness of layers in different parts of MOp.

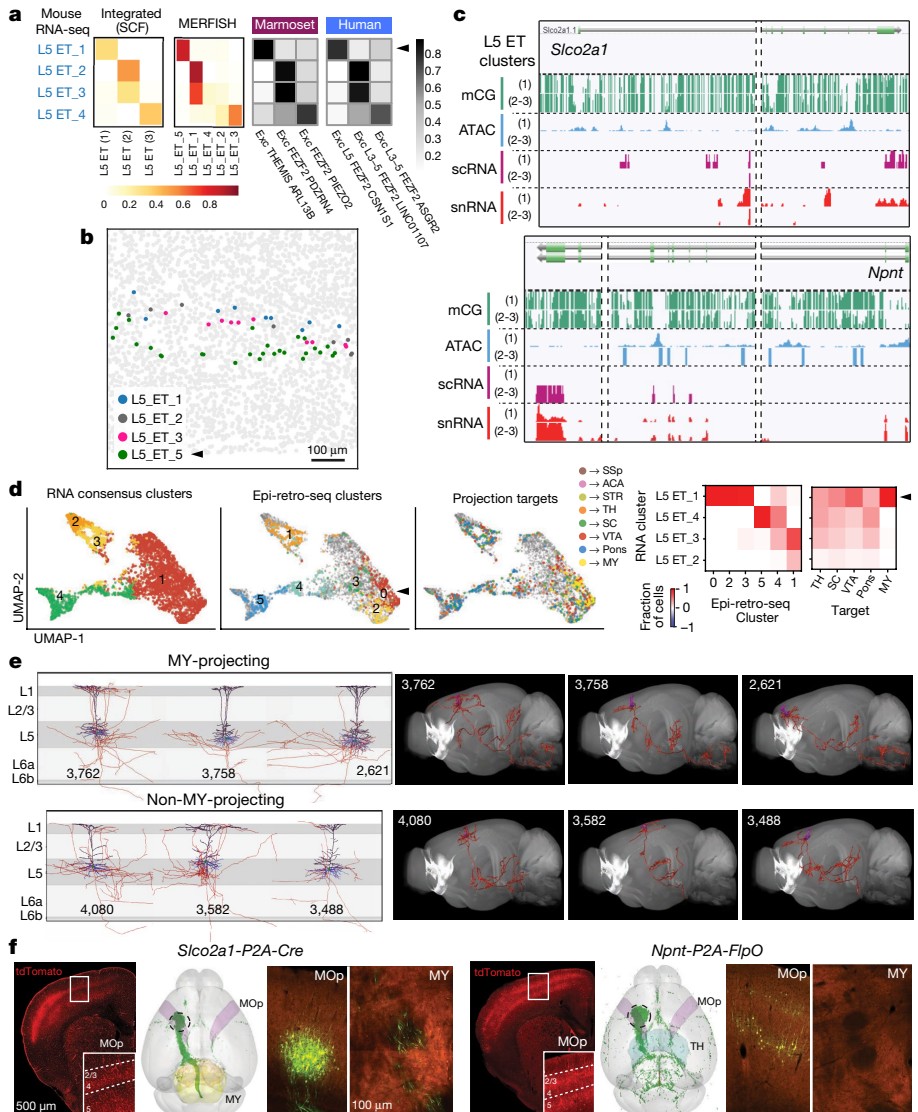

**Fig. 8 | Two distinct L5 ET projection neuron types in MOp. a**, Confusion matrices between mouse L5 ET RNA-seq clusters and SCF, MERFISH, human or marmoset clusters, with the fraction of cells in each of the other datasets mapped to mouse transcriptomic clusters. **b**, Distribution of MERFISH L5 ET cells in upper or lower L5. L5_ET_4 cells are not found in this section. **c**, Genome browser view of gene markers for the MY-projecting (*Slco2a1*) and non-MY-projecting (*Npnt*) L5 ET neurons. **d**, Left, integration UMAP panels between L5 ET Epi-retro-seq clusters and consensus transcriptomic clusters using the snRNA 10x v3 B dataset with the largest number of L5 ET cells (more than 4,000). Right, confusion matrices (normalized by columns). **e**, Local dendritic and axonal morphologies (left) and brain-wide axon projections (right) of fully reconstructed MY-projecting and non-MY-projecting L5 ET neurons. Black, apical dendrites; blue, basal dendrites; red, axons. Numbers are cell IDs. **f**, Characterization of two L5 ET driver lines. For each line, the first panel shows tdTomato reporter expression from a *Slco2a1-P2A-Cre;Ai14* or *Npnt-P2A-FlpO;Ai65F* mouse. The second panel shows a whole-brain view of the projection pattern from a *Slco2a1-P2A-Cre* mouse injected with AAV-pCAG-FLEX-EGFP-WPRE or a *Npnt-P2A-FlpO* mouse injected with AAV-pCAG-fDIO-mNeonGreen-WPRE (MOp (purple), TH (blue) and MY (yellow); injection sites indicated by the dashed circles). Last two panels show GFP- or mNeonGreen-labelled neurons at the injection site (MOp) and axon fibres in contralateral MY (seen only in *Slco2a1* but not *Npnt*).

types[37]. Here we show that this distinction is consistent across all molecular datasets (Fig. 8a). L5 ET_1 or L5 ET_2-4 types corresponded well with SCF type L5 ET (1) or L5 ET (2-3) and MERFISH cluster L5_ET_5 or L5_ET_1-4, respectively, as well as with different L5 ET types from human and marmoset. The laminar distribution of these two groups was revealed by MERFISH, with L5_ET_1-4 cells intermingled in the upper part of L5 and L5_ET_5 cells located distinctly in lower L5 (Fig. 8b). The two groups were further distinguished by epigenomic peaks associated with specific marker genes, *Slco2a1* for SCF L5 ET (1) type and *Npnt* for SCF L5 ET (2-3) types (Fig. 8c).

Epi-retro-seq revealed more complex long-range projection patterns among the 6 epigenetic L5-ET clusters identified, with MY projection cells predominantly in cluster 0 but also in clusters 2 and 3 (Extended Data Fig. 3b). We co-clustered L5 ET cells from the Epi-retro-seq data and the snRNA-seq 10x v3 B data[37], and found that the consensus transcriptomic cluster L5 ET_1 corresponded to Epi-retro-seq clusters 0, 2 and 3, whereas transcriptomic clusters L5 ET_2-4 corresponded to Epi-retro-seq clusters 1, 4 and 5, which contain almost no MY-projecting neurons (Fig. 8d).

We identified multiple full-morphology reconstructions of MOp L5 ET neurons from fMOST imaging of *Fezf2-CreER;Ai166* and *Pvalb-T2A-CreERT2;Ai166* transgenic mice, which were clustered into MY-projecting and non-MY projecting morphological types but also exhibited extensive morphological and projectional variability among individual cells[46] (Fig. 8e), although this was not directly linked

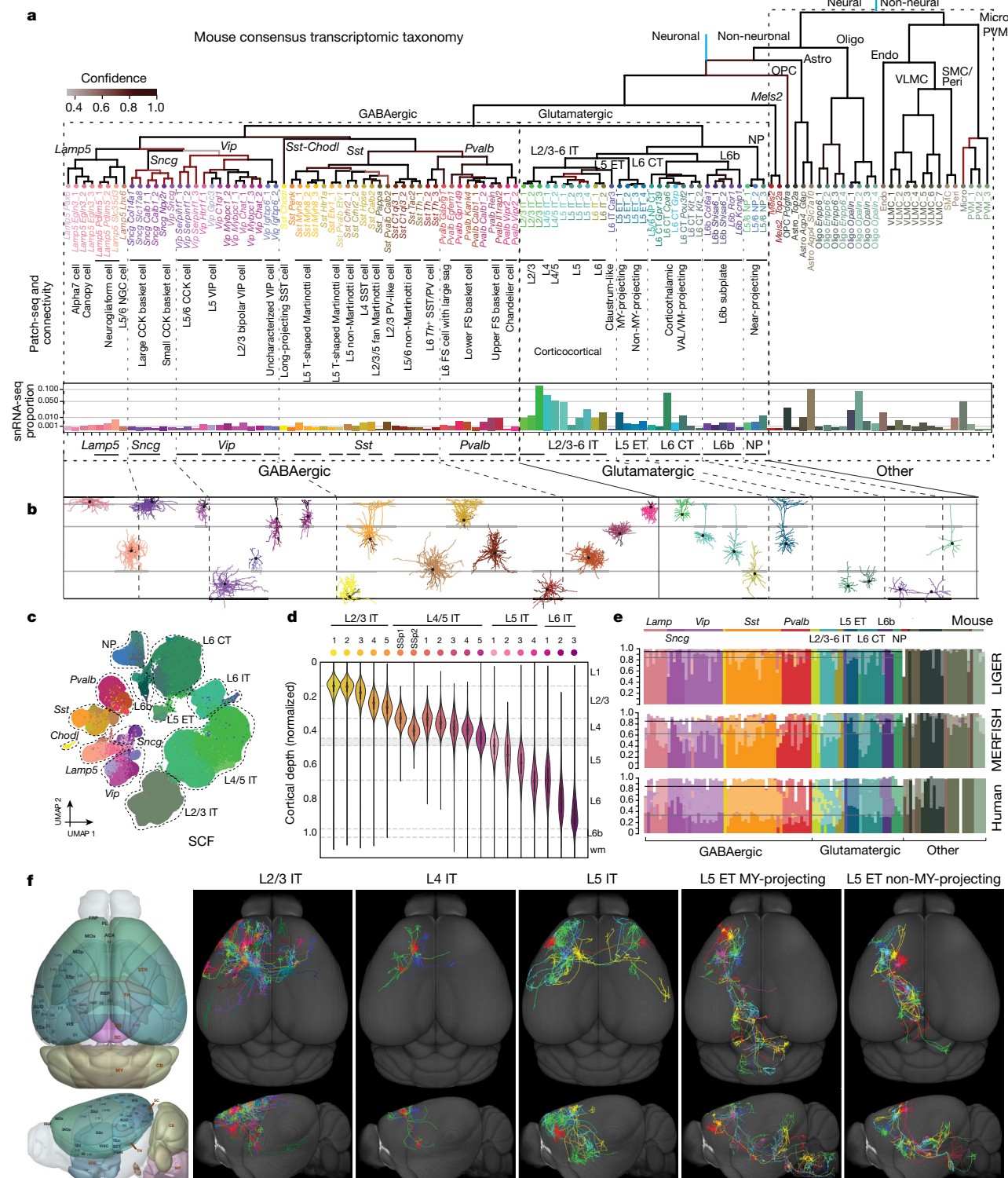

**Fig. 9 | An integrated multimodal census and atlas of MOp cell types.**
**a**, Mouse MOp consensus transcriptomic taxonomy at the top is used to anchor cell-type features in all the other modalities. Major cellular divisions, class and subclass labels are shown above major branches and cluster labels are shown below each leaf node. Using Patch-seq and connectivity studies, many transcriptomic neuron types or subclasses are annotated and correlated with known cortical neuron types. No Patch-seq data were collected for the 'uncharacterized' *Vip* types. Relative proportions of all cell types are calculated from the snRNA-seq 10x v3 B data (bar graph). **b**, Representative local dendritic and axonal morphologies of GABAergic and glutamatergic neuron types from Patch-seq data. **c**, UMAP representation of the mouse transcriptomic–epigenomic integrated molecular taxonomy (SCF version). **d**, Gradual transition of MERFISH IT clusters across cortical layers and depth.

**e**, Percentage of LIGER, MERFISH and human cells assigned to mouse consensus transcriptomic cell types (Methods). Darker subclass colours indicate an exact match to the cluster/type, while lighter-coloured stacked bars indicate a match to taxonomic neighbours within the same subclass or, occasionally, a neighbouring subclass. Grey line, mean exact type match over neuronal types; black line, mean subclass match. **f**, Single-neuron full morphology reconstructions show distinct long-range axon projection patterns between glutamatergic subclasses and cell-to-cell variations within each subclass: L2/3 IT (17 cells), L4 IT (3 cells), L5 IT (5 cells), L5 ET MY-projecting (6 cells) and L5 ET non-MY-projecting (6 cells). Left, Allen mouse brain CCFv3 as an anatomical reference. CCK, cholecystokinin; FS, fast spiking; NGC, neurogliaform cell; PV, parvalbumin; SST, somatostatin; VIP, vasoactive intestinal peptide.

to t-types. Both groups of cells had thick-tufted dendrites that were similar to each other (Fig. 8e), consistent with the Patch-seq study[42].

We used CRISPR–Cas9 gene editing to generate transgenic mice in which Cre or Flp recombinase was targeted to *Slco2a1* or *Npnt*, marker genes for the MY-projecting or non-MY-projecting L5 ET type, respectively (Fig. 8c). Cre- and Flp-dependent tdTomato reporter in *Slco2a1-P2A-Cre;Ai14* and *Npnt-P2A-FlpO;Ai65F* mice labelled cortical L5 neurons, as well as vascular cells in *Slco2a1* mice and L2/3 cells in *Npnt* mice (Fig. 8f). *Slco2a1*-labelled cells occupy a deeper sub-lamina of L5 than those targeted by *Npnt*, consistent with the MERFISH result (Fig. 8b). To test the projection specificity of labelled neurons, we injected AAV vectors encoding a Cre- or Flp-dependent EGFP reporter into L5 in the MOp of these mice. GFP-labelled axon terminals were found in MY of *Slco2a1* but not *Npnt* mice, demonstrating cell-type specificity of these new driver lines (Fig. 8f).

## An integrated synthesis of MOp cell types

As the conclusion of this series of studies from the BICCN, we present an overview and integrated synthesis of the multimodal census and atlas of cell types in the primary motor cortex of mouse, non-human primate and human (Fig. 9).

This integrated synthesis uses the mouse MOp consensus transcriptomic taxonomy[37] as the anchor (Fig. 9a) because it was derived from the largest datasets and was the reference taxonomy for nearly all the cross-modality and cross-species comparisons. This taxonomy has a hierarchical organization, with major divisions first between neural and non-neural cell types, then between neuronal and non-neuronal types within the neural branch, and finally between GABAergic and glutamatergic types within the neuronal branch.

Correspondence matrices show that the mouse MERFISH-based spatial transcriptomic taxonomy[41], the transcriptomic–epigenomic integrated mouse molecular taxonomies using either SCF or LIGER[37] and the human and marmoset transcriptomic taxonomies[38] all aligned largely consistently with the mouse consensus transcriptomic taxonomy (Fig. 9e, Extended Data Fig. 5, Supplementary Table 1). The alignments are highly consistent at subclass level, but disagreements exist at individual-cluster level and increase with cross-species comparison (Fig. 9e), suggesting that differential variations exist in different data types and consistency, in particular that across species, may be more appropriately described at an intermediate level of granularity. We developed a standardized nomenclature system to track cell types described in different modalities (Supplementary Table 2).

Through integrative approaches such as Patch-seq[42], Epi-retro-seq[45] and axon projection mapping[44,46], we related many t-types or subclasses to cortical neuron types traditionally defined by electrophysiological, morphological and connectional properties (Fig. 9a, b, f), thus bridging the cell-type taxonomy with historical knowledge. We derived the relative proportion of each cell type in mouse MOp using either snRNA-seq or MERFISH data. The MERFISH data[41] also revealed the spatial distribution pattern of each cell type, showing that many glutamatergic or GABAergic neuron types adopt narrow distributions along the cortical-depth direction, often occupying predominantly a single layer or a sublayer, and related types (for example, the L2/3-6 IT excitatory types) display a largely gradual transition across cortical depth or layers (Fig. 9d).

Finally, we demonstrate the potential to elucidate gene regulatory mechanisms by discovering candidate CREs (cCREs) and master transcription factors specific to neuronal subclasses in the combined transcriptomic and epigenomic datasets (Fig. 9c). We found 7,245 distal (more than 1 kbp from the transcription start site) cCRE–gene pairs in MOp neurons that showed a positive correlation between accessibility at 6,280 cCREs and expression level of 2,490 putative target genes (Methods)[37,40]. We grouped these putative enhancers into modules based on accessibility across cell clusters (Extended Data Fig. 6) and identified a large number of enhancer–gene pairs for each subclass of

neurons (Extended Data Fig. 5). Similarly, we identified transcription factors showing cell-type specificity supported by both RNA expression and DNA-binding motif enrichment in cell subclasses[37,39] (Methods) (Extended Data Fig. 7).

## Discussion

### A cell census and atlas of primary motor cortex

Understanding the principles of brain circuit organization requires a detailed understanding of its basic components. The current effort combines a wide array of single-cell-based techniques to derive a robust and comprehensive molecular cell-type classification and census of the primary motor cortex of mouse, marmoset and human, coupled with a spatial atlas of cell types and an anatomical input–output wiring diagram in mouse. We demonstrate the robustness and validity of this classification through strong correlations across cellular phenotypes, and strong conservation across species. Together these data comprise a cell atlas of the primary motor cortex that encompasses a comprehensive reference catalogue of cell types, their proportions, spatial distributions, anatomical and physiological characteristics, and molecular genetic profiles, registered into a CCF. This cell atlas establishes a foundation for an integrative study of the architecture and function of cortical circuits akin to reference genomes for studying gene function and genome regulatory architecture. Furthermore, it provides a map of the genes that contribute to cellular phenotypes and their epigenetic regulation. These data resources and associated tools enabling genetic access for manipulative experimentation are publicly available. This body of work provides a roadmap for exploring cellular diversity and organization across brain regions, organ systems and species.

### Principles of cortical cell-type organization

Substantiating previous studies[9,10], our multimodal cross-species study of the primary motor cortex suggests that a general principle of cortical cell-type organization is its hierarchical relationship, whereby high-level classes linked by major branches comprise progressively finer subpopulations connected by minor branches. In this scheme, the higher-level classes and subclasses are categorically and concordantly distinct from each other across modalities, are conserved across species, and probably arise from different developmental programs, such as GABAergic neuron derivatives of different zones of the ganglionic eminences or the layer-selective glutamatergic neurons derived sequentially from progenitors of the cortical plate. At the lower branch levels (types or clusters), however, while certain cell types are highly distinct (for example, *Pvalb* chandelier cells), distinctions and boundaries among many other clusters can be ambiguous and vary among different modalities.

In this context, another important finding, consistent with and building on multiple other studies[9,11,68,69], is the coexistence of discrete and continuous variations of cell features across modalities at the lower branch level. A compelling example is the continuous and concordant variation of transcriptomic, anatomical and physiological properties along cortical depth within multiple cell populations, including the glutamatergic L2/3–L6 IT and GABAergic *Sst* and *Pvalb* subclasses. Although some of the variations may result from technical factors, such as differences in the resolution of measurements across data modalities (with transcriptomics providing the highest granularity at present), a major source of these continuous variations may reflect true biology, supported by the coordinated variation across transcriptomic, spatial, morphological and physiological properties as shown by MERFISH and Patch-seq. Therefore, another emerging principle of cell type organization is the coexistence of discrete and continuous variations that underlie cell-type diversity.

Together, the principles of hierarchical organization comprising discrete classes and types as well as continuum within and across

subpopulations represent a more nuanced and biologically realistic description of cell-type landscape, with implications in cell classification and census. For example, the multimodal variations at finer granularity may preclude a fully discretized representation of cell types with consistency across cell phenotypes, and may explain some of the discrepancies in estimated numbers of cell types using different approaches. An intriguing question is whether continuous variations of cell features will increase further or become more discretized in the context of neural circuit operation, converging to a set of distinct functional elements from a more continuous cellular landscape. An example of this is regionalization. We identify a MOp-specific input–output wiring diagram—however, transcriptomic cell types are generally shared between MOp and its neighbouring cortical areas[11]. Region-specific connectivity patterns of similar molecular types may be a major factor defining the functional specificity of the primary motor cortex.

## Perspectives on cell-type classification

Our findings have major implications for understanding the biological basis of cellular identity towards a more rigorous, quantitative and satisfying definition and classification of cell types. First and foremost, our discovery of the compelling correspondence across molecular genetic, anatomical and physiological features of hierarchically organized cell populations, reflecting developmental origins and mainly conserved across mammalian species, demonstrates the biological validity and genomic underpinning of major cell types. These findings establish a unifying and mechanistic framework of cell-type classification that integrates multi-layered molecular genetic information with multi-faceted phenotypic properties. Thus, single-cell transcriptomics and epigenomics can serve as powerful approaches for establishing a foundational framework of cell types, owing to not only their unparalleled scalability but also to their representation of the underlying molecular genetic programs rooted in development and evolution. Physiological, morphological and connectional characterizations assign functional attributes to cells; their concordance with molecular identities provides strong validation to the molecularly defined cell types, whereas their differential variations reveal additional, probably network- and activity-driven factors that contribute to further refinement of cell types.

While the higher levels of the hierarchy comprise ~around 25 subclasses (16 neuronal and 9 non-neuronal) that are identified with remarkable consistency across multiple species and experimental modalities, many finer levels of cell properties do not neatly segregate into discrete and consistent sets of cell types with perfect correspondence among data modalities. These include aspects of continuous distributions, species specializations and mismatches between molecular and anatomical phenotypes that may result from developmental events no longer represented in the adult. Different methods provide somewhat different granularity of clustering, and thus different numbers of putative cell types. For example, single-cell transcriptomics identifies around 100 clusters representing the terminal leaves of this hierarchically branched organization[37]. Looking ahead, it is important to note that at more refined levels, the number of cell types that can be distinguished will probably change with additional cellular features characterized at greater breadth and depth using new methods and approaches.

Overall, the landscape of cell types appears to be generated from a combination of specification through evolutionarily driven and developmentally regulated genetic mechanisms, and refinement of cellular identities through intercellular interactions within the network in which the cells are embedded. In this scenario, genetic mechanisms drive intrinsic or cell-autonomous determination of cell fate, as well as progressive temporal generation of cell types from common progenitor pools that explain global similarities and continuous features of cellular phenotypes reflecting developmental gradients. Network influences can drive further phenotypic refinement that may not be reflected in the adult genetic signature—for example for axonal projection and synaptic connectivity that may reflect transient or stochastic developmental events, region or circuit-specific and/or activity or plasticity-dependent modification to form and reshape functionally specific circuits. Future studies focusing on these mechanisms and testing of the ensuing hypotheses will enable a deeper understanding of the nature of variability among related cell types in the mammalian brain.

## Cell-type conservation and divergence

Evolutionary conservation is strong evidence of functional significance. The demonstrated conservation of cell types from mouse, marmoset, macaque and human suggests that these conserved types have important roles in cortical circuitry and function in mammals and even more distantly related species. We also find that similarity of cell types varies as a function of evolutionary distance, with substantial species differences that represent either adaptive specialization or genetic drift. For the most part, species specializations tend to appear at the finer branches of the hierarchical taxonomy. This result is consistent with a recent hypothesis in which cell types are defined by common evolutionary descent and evolve independently, such that new cell types are generally derived from existing genetic programs and appear as specializations at the finer levels of the taxonomic tree[70].

A surprising finding across all homologous cell types was the relatively high degree of divergence for genes with cell-type-specific expression in a given species. This observation provides a clear path to identify core conserved genes underlying the canonical identity and features of those cell types. Furthermore, it highlights the need to understand species adaptations superimposed on the conserved program, as many specific cellular phenotypes may vary across species including gene expression, epigenetic regulation, morphology, connectivity and physiological properties. As we illustrated in the Betz cells, there is clear homology across species in the L5 ET subclass, but variation in many measurable properties across species.

## Linking model organisms to human biology and disease

Our findings have major implications for the consideration of model organisms to understand human brain function and disease. Despite major investments, animal models of neuropsychiatric disorders have often been characterized by 'loss of translation', fuelling heated debates about the utility of model organisms in the development of treatments for human diseases. Cell census information aligned across species will be highly valuable for making rational choices about the best models for each disease and therapeutic target. For example, the characterization of cell types and their properties shown in Fig. 9 can be used to infer the main characteristics of homologous cell types in humans and other mammalian species, which would be difficult to obtain otherwise. They can also reveal potential limitations of model organisms and the necessity to study human and other primates to understand the specific cell-type features that contribute to human brain function and diseases. This reductionist dissection of the cellular components provides a foundation for understanding the general principles of neural circuit organization and computation that underlie mental activities and brain disorders.

## Future directions

The approach we took to generate a cell census and atlas through a systematic dissection of cell types opens up numerous avenues for future work. The MOp census and atlas provides a foundational platform for the broad neuroscience community to accumulate and integrate cell-type information across species. Classification of cell types based on their molecular, spatial and connectional properties in the adult sets the stage for developmental studies to understand the molecular genetic programs underlying cell-type specification, maturation and circuit assembly. The molecular genetic information promises to deliver tools for genetic access to many brain cell types via transgenic

and enhancer virus strategies. A combination of single-cell transcriptomics and functional measurements may further elucidate the roles of distinct cell types in circuit computation during behaviour, bridging the gap between molecular and functional definition of cell types. The systematic, multi-modal strategy described here can be extended to the whole brain, and major efforts are underway in the BICCN to generate a brain-wide cell census and atlas in the mouse with increasing coverage of human and non-human primates.

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

**BRAIN Initiative Cell Census Network (BICCN)**

**Corresponding authors**
Edward M. Callaway[1], Hong-Wei Dong[2,93]✉, Joseph R. Ecker[3,4]✉, Michael J. Hawrylycz[5]✉, Z. Josh Huang[6,7]✉, Ed S. Lein[5]✉, John Ngai[8,9,94]✉, Pavel Osten[6]✉, Bing Ren[10,11]✉, Andreas Savas Tolias[12,13]✉, Owen White[14]✉, Hongkui Zeng[5]✉ & Xiaowei Zhuang[15]✉

**BICCN contributing principal investigators**
Giorgio A. Ascoli[16,17], M. Margarita Behrens[18], Edward M. Callaway[1], Jerold Chun[19], Hong-Wei Dong[2,93], Joseph R. Ecker[3,4], Guoping Feng[20,21,22], James C. Gee[23], Satrajit S. Ghosh[24], Yaroslav O. Halchenko[25], Michael J. Hawrylycz[5], Ronna Hertzano[26,14], Z. Josh Huang[6,7], Ed S. Lein[5], Byung Kook Lim[27], Maryann E. Martone[28,29], Lydia Ng[5], John Ngai[8,9,94], Pavel Osten[6], Lior Pachter[30], Alexander J. Ropelewski[31], Timothy L. Tickle[32], Andreas Savas Tolias[12,13], Owen White[14], X. William Yang[33,34], Hongkui Zeng[5], Kun Zhang[35] & Xiaowei Zhuang[15]

**Principal manuscript editors**
Z. Josh Huang[6,7], Ed S. Lein[5] & Hongkui Zeng[5]

**Manuscript writing and figure generation**
Giorgio A. Ascoli[16,17], Trygve E. Bakken[5], Philipp Berens[36,37,38,39], Edward M. Callaway[1], Tanya L. Daigle[5], Hong-Wei Dong[2,93], Joseph R. Ecker[3,4], Julie A. Harris[5,95], Michael J. Hawrylycz[5], Z. Josh Huang[6,7], Nikolas L. Jorstad[5], Brian E. Kalmbach[5,40], Dmitry Kobak[36], Ed S. Lein[5], Yang Eric Li[11], Hanqing Liu[3,41], Katherine S. Matho[6], Eran A. Mukamel[42], Maitham Naeemi[5], John Ngai[8,9,94], Pavel Osten[6], Bing Ren[10,11], Federico Scala[12,13], Pengcheng Tan[3,43], Jonathan T. Ting[5,40], Andreas Savas Tolias[12,13], Fangming Xie[44], Hongkui Zeng[5], Meng Zhang[15], Zhuzhu Zhang[3], Jingtian Zhou[3,45], Xiaowei Zhuang[15] & Brian Zingg[2,93]

**Analysis coordination**
Trygve E. Bakken[5], Edward M. Callaway[1], Hong-Wei Dong[2,93], Joseph R. Ecker[3,4], Julie A. Harris[5,95], Michael J. Hawrylycz[5], Z. Josh Huang[6,7], Ed S. Lein[5], Eran A. Mukamel[42], John Ngai[8,9,94], Pavel Osten[6], Bing Ren[10,11], Andreas Savas Tolias[12,13], Hongkui Zeng[5] & Xiaowei Zhuang[15]

**Integrated data analysis**
Ethan Armand[42], Trygve E. Bakken[5], Philipp Berens[36,37,38,39], Hong-Wei Dong[2,93], Joseph R. Ecker[3,4], Julie A. Harris[5,95], Z. Josh Huang[6,7], Nikolas L. Jorstad[5], Brian E. Kalmbach[5,40], Dmitry Kobak[36], Ed S. Lein[5], Yang Eric Li[11], Hanqing Liu[3,41], Eran A. Mukamel[42], Bing Ren[10,11], Federico Scala[12,13], Pengcheng Tan[3,43], Andreas Savas Tolias[12,13], Fangming Xie[44], Zizhen Yao[5], Hongkui Zeng[5], Meng Zhang[15], Zhuzhu Zhang[3], Jingtian Zhou[3,45] & Xiaowei Zhuang[15]

**scRNA-seq and snRNA-seq data generation and processing**
Darren Bertagnolli[5], Tamara Casper[5], Jerold Chun[19], Kirsten Crichton[5], Nick Dee[5], Dinh Diep[35], Song-Lin Ding[5], Weixiu Dong[35], Elizabeth L. Dougherty[46], Guoping Feng[20,21,22], Olivia Fong[5], Melissa Goldman[47], Jeff Goldy[5], Rebecca D. Hodge[5], Lijuan Hu[48], C. Dirk Keene[49], Fenna M. Krienen[47], Matthew Kroll[5], Blue B. Lake[35], Kanan Lathia[35], Ed S. Lein[5], Sten Linnarsson[48], Christine S. Liu[19,50], Evan Z. Macosko[46], Steven A. McCarroll[46,47], Delissa McMillen[5], Naeem M. Nadaf[46], Thuc Nghi Nguyen[5,95], Carter R. Palmer[19,50], Thanh Pham[5], Nongluk Plongthongkum[35], Nora M. Reed[47], Aviv Regev[46,51,96], Christine Rimorin[5], William J. Romanow[19], Steven Savoia[6], Kimberly Siletti[48], Kimberly Smith[5], Josef Sulc[5], Bosiljka Tasic[5], Michael Tieu[5], Amy Torkelson[5], Herman Tung[5], Cindy T. J. van Velthoven[5], Charles R. Vanderburg[46], Anna Marie Yanny[5], Hongkui Zeng[5] & Kun Zhang[35]

**ATAC-seq data generation and processing**
M. Margarita Behrens[18], Jerold Chun[19], Dinh Diep[35], Weixiu Dong[35], Rongxin Fang[45], Xiaomeng Hou[10], Blue B. Lake[35], Yang Eric Li[11], Christine S. Liu[19,50], Jacinta D. Lucero[18], Julia K. Osteen[18], Carter R. Palmer[19,50], Antonio Pinto-Duarte[18], Nongluk Plongthongkum[35], Olivier Poirion[10], Sebastian Preissl[10], Bing Ren[10,11], William J. Romanow[19], Xinxin Wang[10,97] & Kun Zhang[35]

**Methylcytosine data production and analysis**
Andrew I. Aldridge[3], Anna Bartlett[3], M. Margarita Behrens[18], Lara Boggeman[52], Carolyn O'Connor[52], Rosa G. Castanon[3], Huaming Chen[3], Joseph R. Ecker[3,4], Conor Fitzpatrick[52], Hanqing Liu[3,41], Jacinta D. Lucero[18], Chongyuan Luo[3,4,98], Joseph R. Nery[3], Michael Nunn[3], Julia K. Osteen[18], Antonio Pinto-Duarte[18], Angeline C. Rivkin[3], Wei Tian[3] & Jingtian Zhou[3,45]

**Epi-retro-seq data generation and processing**
Anna Bartlett[3], M. Margarita Behrens[18], Lara Boggeman[52], Edward M. Callaway[1], Carolyn O'Connor[52], Rosa G. Castanon[3], Bertha Dominguez[53], Joseph R. Ecker[3,4], Conor Fitzpatrick[52], Tony Ito-Cole[1], Matthew Jacobs[1], Xin Jin[54,99,100], Cheng-Ta Lee[53], Kuo-Fen Lee[53], Paula Assakura Miyazaki[1], Eran A. Mukamel[42], Joseph R. Nery[3], Michael Nunn[3], Yan Pang[1], Antonio Pinto-Duarte[18], Mohammad Rashid[1], Angeline C. Rivkin[3], Jared B. Smith[54], Pengcheng Tan[3,43], Minh Vu[1], Elora Williams[54], Zhuzhu Zhang[3] & Jingtian Zhou[3,45]

**'Omics data analysis**
Ethan Armand[42], Trygve E. Bakken[5], Tommaso Biancalani[46], A. Sina Booeshaghi[30], Megan Crow[55], Dinh Diep[35], Sandrine Dudoit[56], Joseph R. Ecker[3,4], Rongxin Fang[45], Stephan Fischer[55], Olivia Fong[5], Jesse Gillis[55], Jeff Goldy[5], Qiwen Hu[57], Nikolas L. Jorstad[5], Peter V. Kharchenko[57], Fenna M. Krienen[47], Blue B. Lake[35], Ed S. Lein[5], Yang Eric Li[11], Sten Linnarsson[48], Hanqing Liu[3,41], Evan Z. Macosko[46], Eran A. Mukamel[42], John Ngai[8,9,94], Sheng-Yong Niu[3], Vasilis Ntranos[58], Lior Pachter[30], Olivier Poirion[10], Elizabeth Purdom[59], Aviv Regev[46,51,96], Davide Risso[60], Hector Roux de Bézieux[61], Kimberly Siletti[48], Kimberly Smith[5], Saroja Somasundaram[5], Kelly Street[62], Valentine Svensson[30], Bosiljka Tasic[5], Wei Tian[3], Eeshit Dhaval Vaishnav[46], Koen Van den Berge[59,63], Cindy T. J. van Velthoven[5], Joshua D. Welch[64], Fangming Xie[44], Zizhen Yao[5], Hongkui Zeng[5] & Jingtian Zhou[3,45]

**Tracing and connectivity data generation**
Xu An[6,7], Helen S. Bateup[8,9,65], Ian Bowman[2,93], Rebecca K. Chance[8], Hong-Wei Dong[2,93], Nicholas N. Foster[2,93], William Galbavy[6,66], Hui Gong[67,68], Lin Gou[2,93], Julie A. Harris[5,95], Joshua T. Hatfield[6,7], Houri Hintiryan[2,93], Karla E. Hirokawa[5,95], Z. Josh Huang[6,7], Gukhan Kim[6], Daniel J. Kramer[8], Anan Li[67,68], Xiangning Li[67], Byung Kook Lim[27], Qingming Luo[69,68], Katherine S. Matho[6], Rodrigo Muñoz-Castañeda[6], Lydia Ng[5], John Ngai[8,9,94], David A. Stafford[8], Hongkui Zeng[5] & Brian Zingg[2,93]

**Morphology data generation and reconstruction**
Tanya L. Daigle[5], Hong-Wei Dong[2,93], Zhao Feng[67], Hui Gong[67,68], Julie A. Harris[5,95], Karla E. Hirokawa[5,95], Z. Josh Huang[6,7], Xueyan Jia[68], Shengdian Jiang[70], Tao Jiang[68], Xiuli Kuang[71], Rachael Larsen[5], Phil Lesnar[5], Xiangning Li[67], Yaoyao Li[71], Yuanyuan Li[72], Lijuan Liu[70], Qingming Luo[69,68], Hanchuan Peng[70], Lei Qu[73], Miao Ren[69], Zongcai Ruan[70], Elise Shen[5], Yuanyuan Song[70], Wayne Wakeman[5], Peng Wang[74], Yimin Wang[75], Yun Wang[5], Lulu Yin[70], Jing Yuan[67,68], Hongkui Zeng[5], Sujun Zhao[70] & Xuan Zhao[70]

**OLST/STPT and other data generation**
Xu An[6,7], William Galbavy[6,66], Joshua T. Hatfield[6,7], Z. Josh Huang[6,7], Gukhan Kim[6], Katherine S. Matho[6], Rodrigo Muñoz-Castañeda[6], Arun Narasimhan[6], Pavel Osten[6] & Ramesh Palaniswamy[6]

**Morphology, connectivity and imaging analysis**
Xu An[6,7], Giorgio A. Ascoli[16,17], Samik Banerjee[6], Liya Ding[70], Hong-Wei Dong[2,93], Zhao Feng[67], Nicholas N. Foster[2,93], William Galbavy[6,66], Hui Gong[67,68], Julie A. Harris[5,95], Joshua T. Hatfield[6,7], Z. Josh Huang[6,7], Dhananjay Huilgol[6,7], Bingxing Huo[6], Xueyan Jia[68], Gukhan Kim[6], Hsien-Chi Kuo[5], Sophie Laturnus[36], Anan Li[67,68], Xu Li[6], Katherine S. Matho[6], Partha P. Mitra[6], Judith Mizrachi[6], Rodrigo Muñoz-Castañeda[6], Maitham Naeemi[5], Arun Narasimhan[6], Lydia Ng[5], Pavel Osten[6], Ramesh Palaniswamy[6], Hanchuan Peng[70], Quanxin Wang[5], Yimin Wang[75], Yun Wang[5], Peng Xie[70], Feng Xiong[70], Yang Yu[5] & Hongkui Zeng[5]

**Spatially resolved single-cell transcriptomics (MERFISH)**
Hong-Wei Dong[2,93], Stephen W. Eichhorn[15], Zizhen Yao[5], Hongkui Zeng[5], Meng Zhang[15], Xiaowei Zhuang[15] & Brian Zingg[2,93]

**Multimodal profiling (Patch-seq)**
Philipp Berens[36,37,38,39], Jim Berg[5], Matteo Bernabucci[12,13], Yves Bernaerts[36], Cathryn René Cadwell[77], Jesus Ramon Castro[12,13], Rachel Dalley[5], Leonard Hartmanis[78], Gregory D. Horwitz[40,79], Xiaolong Jiang[12,13,80], Brian E. Kalmbach[5,40], C. Dirk Keene[49], Andrew L. Ko[81,82], Dmitry Kobak[36], Sophie Laturnus[36], Ed S. Lein[5], Elanine Miranda[12,13], Shalaka Mulherkar[12,13], Philip R. Nicovich[5,95], Scott F. Owen[5,101], Rickard Sandberg[78], Federico Scala[12,13], Kimberly Smith[5], Staci A. Sorensen[5], Zheng Huan Tan[12,13], Jonathan T. Ting[5,40], Andreas Savas Tolias[12,13] & Hongkui Zeng[5]

**Transgenic tools**
Shona Allen[8], Xu An[6,7], Helen S. Bateup[8,9,65], Rebecca K. Chance[8], Tanya L. Daigle[5], William Galbavy[6,66], Joshua T. Hatfield[6,7], Dirk Hockemeyer[8,65,83], Z. Josh Huang[6,7], Dhananjay Huilgol[6,7], Gukhan Kim[6], Daniel J. Kramer[8], Angus Y. Lee[84], Katherine S. Matho[6], John Ngai[8,9,94], David A. Stafford[8], Bosiljka Tasic[5], Matthew B. Veldman[85,86], X. William Yang[33,34], Zizhen Yao[5] & Hongkui Zeng[5]

**NeMO archive and analytics**
Ricky S. Adkins[14], Seth A. Ament[14,87], Héctor Corrada Bravo[88], Robert Carter[14], Apaala Chatterjee[14], Carlo Colantuoni[14,89], Jonathan Crabtree[14], Heather Creasy[14], Victor Felix[14], Michelle Giglio[14], Brian R. Herb[14], Ronna Hertzano[26,14], Jayaram Kancherla[88], Anup Mahurkar[14], Carrie McCracken[14], Lance Nickel[14], Dustin Olley[14], Joshua Orvis[14], Michael Schor[14] & Owen White[14]

**Brain Image Library (BIL) archive**
Greg Hood[31] & Alexander J. Ropelewski[31]

**DANDI archive**

Benjamin Dichter[90], Satrajit S. Ghosh[24], Michael Grauer[91], Yaroslav O. Halchenko[25] & Brian Helba[91]

**Brain Cell Data Center (BCDC)**

Anita Bandrowski[28], Nikolaos Barkas[92], Benjamin Carlin[92], Florence D. D'Orazi[5], Kylee Degatano[92], James C. Gee[23], Thomas H. Gillespie[28], Michael J. Hawrylycz[5], Farzaneh Khajouei[32], Kishori Konwar[32], Maryann E. Martone[28,29], Lydia Ng[5], Carol Thompson[5] & Timothy L. Tickle[32]

**Project management**

Florence D. D'Orazi[5], Hui Gong[67,68], Houri Hintiryan[2,93], Kathleen Kelly[6], Blue B. Lake[35], Katherine S. Matho[6], Stephanie Mok[5], Michael Nunn[3], Susan Sunkin[5] & Carol Thompson[5]

[1]Systems Neurobiology Laboratories, The Salk Institute for Biological Studies, La Jolla, CA, USA. [2]UCLA Brain Research and Artificial Intelligence Nexus, Department of Neurobiology, David Geffen School of Medicine, University of California Los Angeles, Los Angeles, CA, USA. [3]Genomic Analysis Laboratory, The Salk Institute for Biological Studies, La Jolla, CA, USA. [4]Howard Hughes Medical Institute, The Salk Institute for Biological Studies, La Jolla, CA, USA. [5]Allen Institute for Brain Science, Seattle, WA, USA. [6]Cold Spring Harbor Laboratory, Cold Spring Harbor, NY, USA. [7]Department of Neurobiology, Duke University School of Medicine, Durham, NC, USA. [8]Department of Molecular and Cell Biology, University of California Berkeley, Berkeley, CA, USA. [9]Helen Wills Neuroscience Institute, University of California Berkeley, Berkeley, CA, USA. [10]Center for Epigenomics, Department of Cellular and Molecular Medicine, University of California San Diego School of Medicine, La Jolla, CA, USA. [11]Ludwig Institute for Cancer Research, La Jolla, CA, USA. [12]Department of Neuroscience, Baylor College of Medicine, Houston, TX, USA. [13]Center for Neuroscience and Artificial Intelligence, Baylor College of Medicine, Houston, TX, USA. [14]Institute for Genome Sciences, University of Maryland School of Medicine, Baltimore, MD, USA. [15]Howard Hughes Medical Institute, Department of Chemistry and Chemical Biology, Department of Physics, Harvard University, Cambridge, MA, USA. [16]Center for Neural Informatics, Krasnow Institute for Advanced Study, George Mason University, Fairfax, VA, USA. [17]Bioengineering Department, George Mason University, Fairfax, VA, USA. [18]Computational Neurobiology Laboratory, The Salk Institute for Biological Studies, La Jolla, CA, USA. [19]Sanford Burnham Prebys Medical Discovery Institute, La Jolla, CA, USA. [20]McGovern Institute for Brain Research, Massachusetts Institute of Technology, Cambridge, MA, USA. [21]Department of Brain and Cognitive Sciences, Massachusetts Institute of Technology, Cambridge, MA, USA. [22]Stanley Center for Psychiatric Research, Broad Institute of MIT and Harvard, Cambridge, MA, USA. [23]Department of Radiology, University of Pennsylvania, Philadelphia, PA, USA. [24]Massachusetts Institute of Technology, Cambridge, MA, USA. [25]Dartmouth College, Hannover, NH, USA. [26]Department of Otorhinolaryngology, Anatomy and Neurobiology, University of Maryland School of Medicine, Baltimore, MD, USA. [27]Division of Biological Science, Neurobiology Section, University of California San Diego, La Jolla, CA, USA. [28]Department of Neurosciences, University of California San Diego, La Jolla, CA, USA. [29]SciCrunch, Inc., San Diego, CA, USA. [30]California Institute of Technology, Pasadena, CA, USA. [31]Pittsburgh Supercomputing Center, Carnegie Mellon University, Pittsburgh, PA, USA. [32]Klarman Cell Observatory and Data Sciences Platform, Broad Institute of MIT and Harvard, Cambridge, MA, USA. [33]Center for Neurobehavioral Genetics, Jane and Terry Semel Institute for Neuroscience and Human Behaviors, University of California Los Angeles, Los Angeles, CA, USA. [34]Department of Psychiatry and Biobehavioral Science, David Geffen School of Medicine, University of California Los Angeles, Los Angeles, CA, USA. [35]Department of Bioengineering, University of California San Diego, La Jolla, CA, USA. [36]Institute for Ophthalmic Research, University of Tübingen, Tübingen, Germany. [37]Center for Integrative Neuroscience, University of Tübingen, Tübingen, Germany. [38]Institute for Bioinformatics and Medical Informatics, University of Tübingen, Tübingen, Germany. [39]Bernstein Center for Computational Neuroscience, University of Tübingen, Tübingen, Germany. [40]Department of Physiology and Biophysics, University of Washington, Seattle, WA, USA. [41]Division of Biological Sciences, University of California San Diego, La Jolla, CA, USA. [42]Department of Cognitive Science, University of California San Diego, La Jolla, CA, USA. [43]School of Pharmaceutical Sciences, Tsinghua University, Beijing, China. [44]Department of Physics, University of California San Diego, La Jolla, CA, USA. [45]Bioinformatics and Systems Biology Graduate Program, University of California San Diego, La Jolla, CA, USA. [46]Broad Institute of MIT and Harvard, Cambridge, MA, USA.

[47]Department of Genetics, Harvard Medical School, Boston, MA, USA. [48]Division of Molecular Neurobiology, Department of Medical Biochemistry and Biophysics, Karolinska Institute, Stockholm, Sweden. [49]Department of Laboratory Medicine and Pathology, University of Washington, Seattle, WA, USA. [50]Biomedical Sciences Program, School of Medicine, University of California San Diego, La Jolla, CA, USA. [51]Howard Hughes Medical Institute, Department of Biology, MIT, Cambridge, MA, USA. [52]Flow Cytometry Core Facility, The Salk Institute for Biological Studies, La Jolla, CA, USA. [53]Peptide Biology Laboratories, The Salk Institute for Biological Studies, La Jolla, CA, USA. [54]Molecular Neurobiology Laboratory, The Salk Institute for Biological Studies, La Jolla, CA, USA. [55]Stanley Institute for Cognitive Genomics, Cold Spring Harbor Laboratory, Cold Spring Harbor, NY, USA. [56]Department of Statistics and Division of Biostatistics, University of California Berkeley, Berkeley, CA, USA. [57]Department of Biomedical Informatics, Harvard Medical School, Boston, MA, USA. [58]University of California San Francisco, San Francisco, CA, USA. [59]Department of Statistics, University of California Berkeley, Berkeley, CA, USA. [60]Department of Statistical Sciences, University of Padova, Padua, Italy. [61]Division of Biostatistics, School of Public Health, University of California Berkeley, Berkeley, CA, USA. [62]Department of Data Sciences, Dana-Farber Cancer Institute, Boston, MA, USA. [63]Department of Applied Mathematics, Computer Science and Statistics, Ghent University, Ghent, Belgium. [64]Department of Computational Medicine and Bioinformatics, University of Michigan, Ann Arbor, MI, USA. [65]Chan Zuckerberg Biohub, San Francisco, CA, USA. [66]Program in Neuroscience, Department of Neurobiology and Behavior, Stony Brook University, Stony Brook, NY, USA. [67]Britton Chance Center for Biomedical Photonics, Wuhan National Laboratory for Optoelectronics, MoE Key Laboratory for Biomedical Photonics, Huazhong University of Science and Technology, Wuhan, Hubei, China. [68]HUST-Suzhou Institute for Brainsmatics, Suzhou, Jiangsu, China. [69]School of Biomedical Engineering, Hainan University, Haikou, Hainan, China. [70]SEU-ALLEN Joint Center, Institute for Brain and Intelligence, Southeast University, Nanjing, Jiangsu, China. [71]School of Optometry and Ophthalmology, Wenzhou Medical University, Wenzhou, Zhejiang, China. [72]Key Laboratory of Intelligent Computation & Signal Processing, Ministry of Education, Anhui University, Hefei, Anhui, China. [73]Anhui University, Hefei, Anhui, China. [74]Shanghai University, Shanghai, China. [75]School of Computer Engineering and Science, Shanghai University, Shanghai, China. [76]Stanford University, School of Medicine, Stanford, CA, USA. [77]Department of Pathology, University of California San Francisco, San Francisco, CA, USA. [78]Department of Cell and Molecular Biology, Karolinska Institutet, Stockholm, Sweden. [79]Washington National Primate Research Center, University of Washington, Seattle, WA, USA. [80]Jan and Dan Duncan Neurological research Institute, Houston, TX, USA. [81]Department of Neurological Surgery, University of Washington School of Medicine, Seattle, WA, USA. [82]Regional Epilepsy Center at Harborview Medical Center, Seattle, WA, USA. [83]Innovative Genomics Institute, University of California Berkeley, Berkeley, CA, USA. [84]Cancer Research Laboratory, University of California Berkeley, Berkeley, CA, USA. [85]Semel Institute & Department of Psychiatry and Biobehavioral Science, University of California Los Angeles, Los Angeles, CA, USA. [86]David Geffen School of Medicine, University of California Los Angeles, Los Angeles, CA, USA. [87]Department of Psychiatry, University of Maryland School of Medicine, Baltimore, MD, USA. [88]Center for Bioinformatics and Computational Biology, University of Maryland College Park, College Park, MD, USA. [89]Department Neurology & Department Neuroscience, Johns Hopkins School of Medicine, Baltimore, MD, USA. [90]CatalystNeuro, Benicia, CA, USA. [91]Kitware Inc., Clifton Park, NY, USA. [92]Data Sciences Platform, Broad Institute of MIT and Harvard, Cambridge, MA, USA. [93]Center for Integrative Connectomics, USC Mark and Mary Stevens Neuroimaging and Informatics Institute, Department of Neurology, Zilkha Neurogenetic Institute, Keck School of Medicine at USC, University of Southern California, Los Angeles, CA, USA. . [94]Present address: National Institute of Neurological Disorders and Stroke, National Institutes of Health, Bethesda, MD, USA. [95]Present address: Cajal Neuroscience, Seattle, WA, USA. [96]Present address: Genentech, South San Francisco, CA, USA. [97]Present address: McDonnell Genome Institute, Washington University School of Medicine, St Louis, MO, USA. [98]Present address: Department of Human Genetics, University of California Los Angeles, Los Angeles, CA, USA. [99]Present address: Center for Motor Control and Disease, Key Laboratory of Brain Functional Genomics, East China Normal University, Shanghai, China . [100]Present address: NYU-ECNU Institute of Brain and Cognitive Science, New York University Shanghai, Shanghai, China. [101]Present address: Department of Neurosurgery, Stanford University School of Medicine, Stanford, CA, USA. [✉]e-mail: CALLAWAY@SALK.EDU; HongWeiD@mednet.ucla.edu; ecker@salk.edu; MikeH@alleninstitute.org; josh.huang@duke.edu; EdL@alleninstitute.org; john.ngai@nih.gov; osten@cshl.org; biren@ucsd.edu; astolias@bcm.edu; Owhite@som.umaryland.edu; hongkuiz@alleninstitute.org; zhuang@chemistry.harvard.edu

# Methods

## Nomenclature of the L5 ET subclass of glutamatergic neurons

In this manuscript we have adopted a nomenclature for major sub-classes of cortical glutamatergic excitatory neurons, which have long-range projections both within and outside of the cortex, following a long tradition of naming conventions that often classify neurons based on their projection targets. This nomenclature is based on our de novo transcriptomic taxonomy (Fig. 9) that organizes cell types hierarchically and validates the naming of the primary branches of glutamatergic neurons by their major long-range projection targets. At these levels, glutamatergic neurons are clearly divided into several subclasses, the cortico-cortical and cortico-striatal only projecting IT neurons that are distributed across nearly all layers (L2/3 IT, L4/5 IT, L5 IT, L6 IT and L6 IT *Car3*), the layer 5 neurons projecting to extratelencephalic targets (L5 ET), the CT-projecting neurons in layer 6 (L6 CT), the NP neurons found in layers 5 and 6, and the L6b neurons whose projection patterns remain largely unknown.

While the IT, CT, NP and L6b neurons have been consistently labelled as such in the field, the L5 ET neurons have not been named consistently in the literature, largely owing to their large variety of projection targets and other phenotypic features that vary depending on cortical areas and species. Here we use the term L5 ET (layer 5 extratelencephalic) to refer to this prominent and distinct subclass of neurons as a standard name that can be accurately used across cortical regions and across species, and we provide our rationale below.

It has long been appreciated that cortical layer 5 contains two distinct populations of neurons that can be distinguished, not only based on the presence or absence of projections to ET targets (ET and IT cells), but also based on their predominant soma locations, dendritic morphologies and intrinsic physiology[48]. Accordingly, various names incorporating these features have been adopted to refer to L5 ET versus L5 IT cells, such as L5b versus L5a, thick-tufted versus thin-tufted and burst-firing versus regular-firing. The most common term used to refer to L5 ET cells residing in motor cortical areas has been PT, which refers to neurons projecting to the pyramidal tract. As accurately stated in Wikipedia, "The pyramidal tracts include both the corticobulbar tract and the corticospinal tract. These are aggregations of efferent nerve fibers from the upper motor neurons that travel from the cerebral cortex and terminate either in the brainstem (corticobulbar) or spinal cord (corticospinal) and are involved in the control of motor functions of the body."

Owing to the past wide use of the term PT, we do not take the decision to use L5 ET rather than PT lightly. However, in the face of multiple lines of evidence that have accumulated over the last several years[72,73] and prominently highlighted in this manuscript, it is now clear that PT represents only a subset of L5 ET cells and is thus unable to accurately encompass the entire L5 ET subclass. This realization is informed by comparisons across species and cortical areas, and by single-cell transcriptomics and descriptions of the projections of single neurons, as well as studies linking transcriptional clusters to projection targets.

As noted above, the overall transcriptomic relationships between cortical neurons are well-described by a hierarchical tree that closely matches developmental lineage relationships as neurons become progressively restricted in their adult fates[37,38] (Fig. 9). The cortical excitatory neurons are a major branch, distinct from inhibitory, glial and epithelial cells. Subsequent splitting of the excitatory neurons reveals several major excitatory neuron subclasses—IT, L5 ET, L6 CT, NP and L6b. These major subclasses are conserved across mammalian species[9,10], as well as across all cortical areas as shown in mouse[11]. It is therefore clear that names are needed that both accurately incorporate and accurately distinguish between neurons in these subclasses, and which are applicable across all cortical areas.

Also as noted above, a widely used alternative to L5 ET is PT. Further, this term is traditionally used along with CT to distinguish between cells with these different projections. The two main observations that make these alternative nomenclatures untenable are: (1) PT refers to motor neurons that project into MY or spinal cord, but in many cortical areas (for example, visual and auditory areas) none of the L5 ET cells are motor neurons; and (2) even in the motor cortex many cells in the L5 ET subclass do not project to the pyramidal tract and instead project solely to the TH (or to TH and other non-PT targets). This is revealed by single-neuron reconstructions[18,46,53] (Figs. 6, 8), BARseq[64], projections from neuron populations with known gene expression and anatomical position in mouse lines[63], and studies directly linking projections to transcriptomics[9,41] and epigenetics[45] (Figs. 4, 8). The term PT therefore is not inclusive of the entire L5 ET subclass. Furthermore, the L5 CT cells within the L5 ET subclass are largely continuous with PT cells (or 'PT-like' cells), not only genetically but also anatomically[41,42] (Figs. 2, 3), as a majority of L5 ET cells project to multiple targets, typically including both the TH and the PT structures (for example, MY and spinal cord), as well as the midbrain[46] (Figs. 6, 8). Thus, the L5 ET subclass should neither be split into PT and CT, nor should the CT-only cells be omitted by use of the term PT. These facts also inform us that it is important to maintain a distinction between L5 CT (a type of L5 ET) and L6 CT (a major subclass of cortical excitatory neurons that is highly distinct from L5 ET, despite the presence of some L6 CT cells at the bottom of layer 5)[41]. CT can be accurately used as a generic term, but CT neurons do not belong to a single subclass of cortical excitatory neurons.

We recognize that another name that has been used to describe L5 ET cells is subcerebral projection neuron (SCPN)[49]. Given that the telencephalon is equivalent to the cerebrum, ET and subcerebral have the same meaning and the term L5-SCPN would be an accurate and equivalent alternative. But the 'L5' qualifier is crucial in either case to distinguish these cells from the L6 CT subclass. We favour the use of ET because SCPN has not been widely adopted and due to symmetry with the widely used 'IT' nomenclature. Alternatively, given their evidence that "unlike pyramidal tract neurons in the motor cortex, these neurons in the auditory cortex do not project to the spinal cord", Chen et al[64] used the term 'pyramidal tract-like' (PT-l). We also favour L5 ET over L5 PT-l which clings to an inaccurate and now outdated nomenclature.

## Integrating 10x v3 snRNA-seq datasets across species

To identify homologous cell types across species, human, marmoset and mouse 10x v3 snRNA-seq datasets were integrated using Seurat's SCTransform workflow. Each major cell class (glutamatergic, GABAergic and non-neuronal cells) was integrated separately across species. Expression matrices were reduced to 14,870 one-to-one orthologues across the three species (NCBI Homologene; 22 November 2019). Nuclei were downsampled to have approximately equivalent numbers at the subclass level across species. Marker genes were identified for each species cluster using Seurat's FindAllMarkers function with test.use set to 'roc', > 0.7 classification power. Markers were used as input to guide alignment and anchor-finding during integration steps. For full methods see ref. [38]. Code for generating Figs. 1b–h, 3, Extended Data Fig. 2 is available at http://data.nemoarchive.org/publication_release/Lein_2020_M1_study_analysis/Transcriptomics/flagship/. Analysis was performed in RStudio using R version 3.5.3, R packages: Seurat 3.1.1, ggplot2 3.2.1 and scrattch.hicat 0.0.22.

## Estimation of cell-type homology

To establish a robust cross-species cell type taxonomy, we applied a tree-based clustering method on integrated class-level datasets (https://github.com/AllenInstitute/BICCN_M1_Evo). The integrated space (from the previously mentioned Seurat integration) was over-clustering into small sets of highly similar nuclei for each class (about 500 clusters per class). Clusters were aggregated into metacells, then hierarchical clustering was performed based on the metacell gene expression matrix using Ward's method. Hierarchical trees were then assessed for cluster size, species mixing and branch stability by subsampling the dataset

100 times with 95% of nuclei. Finally, we recursively searched every node of the tree, and if certain heuristic criteria were not sufficient for a node below the upper node, all nodes below the upper node were pruned and nuclei belonging to this subtree were merged into one homologous group. We identified 24 GABAergic, 13 glutamatergic and 8 non-neuronal cross-species consensus clusters that were highly mixed across species and robust. For full methods see ref. [38]. A final dendrogram of consensus cell types was constructed by transforming the raw unique molecular identifier (UMI) counts to $\log_2$(counts per million (CPM)) normalized counts. Up to 50 marker genes per cross-species cluster were identified by using the scrattch.hicat (v0.0.22) (https://github.com/AllenInstitute/scrattch.hicat) display_cl and select_markers functions with the following parameters; q1.th = 0.4, q.diff.th = 0.5, de.score.th = 80. Median cross-species cluster $\log_2$ CPM expression of these genes were then used as input for scrattch.hicat's build_dend function. This analysis was bootstrapped 10,000 times with branch colour denoting confidence. Branch robustness was assessed by rebuilding the dendrogram 10,000 times with a random 80% subset of variable genes across clusters and calculating the proportion of iterations that clusters were present on the same branch. Consensus taxonomy agreement in Fig. 9e is determined by selecting maximum frequency leaf match with stacked bars indicating assigned consensus cell types in the centred neighbourhood.

## Cross-species differential gene expression and correlations

Expression matrices were subsetted to include one-to-one orthologous genes across all three species. Spearman correlations shown in Fig. 1d were performed by comparing cross-species cluster median $\log_2$ CPM expression of all orthologous genes for each species pair. To calculate the number of differentially expressed genes between each species pair for each cross-species cluster, we used a pseudobulk comparison method[74] from DESeq2 (v1.30.0). For a given cross-species cluster, each sample was split by species and donor, then a Wald test was performed between each species pair. Genes with adjusted $P$-values < 0.05 and $\log_2$ fold-changes greater than 2 in either direction were counted and reported in Fig. 1e.

## Generation of Epi-retro-seq data

We injected retrograde tracer rAAV2-retro-Cre[75] into a target region in INTACT mice[76], which turned on Cre-dependent GFP expression in the nuclei of MOp neurons projecting to the injected target region. Individual GFP-labelled nuclei of MOp projection neurons were then isolated using fluorescence-activated nucleus sorting (FANS) (box outlines selected cells in Fig. 4a). snmC-seq2[77] was performed to profile the DNA methylation (mC) of each single nucleus.

## Evaluation of contamination in Epi-retro-seq

The methods used to evaluate contamination level and potential reasons are described in detail in ref. [45]. Specifically, we quantified the ratio between the number of cells in expected on-target subclasses (for example, L5 ET cluster for ET-projecting neurons) versus in expected off-target subclasses (for example, IT clusters for ET-projecting neurons), denoted as $r_p$, and compared the ratio with the one expected from the unbiased data without enrichment for specific projections, denoted as $r_u$. This provides an estimation of signal-to-noise ratio of each FANS experiment. For IT projections, we used IT subclasses as on-target and L6 CT + inhibitory as off-target, and for ET projections, we used L5 ET as on-target and IT + inhibitory as off-target. For the MOp neurons without enrichment of projections, the expected ratio between cells in IT subclasses and in L6 CT + inhibitory are $r_u$ = 2,652:1,775, whereas the expected ratio between cells in L5 ET subclass and in IT + inhibitory are $r_u$ = 202:3,434. The fold enrichment in the text was computed by $r_p/r_u$ for each FANS run separately and averaged across IT or ET targets respectively.

We want to point out that, in addition to this computational method, other methods are available to evaluate and minimize potential contamination in Epi-retro-seq. In cases in which differences in expected results from on- versus off-target populations are unknown, other available methods would need to be used to eliminate cases in which injections might have directly labelled cells outside the intended target region, such as examination of labelling along the injection electrode track.

## Integration of L5 ET cells from Epi-retro-seq and 10x snRNA-seq

For snRNA-seq, the 4,515 cells from 10x v3 B dataset labelled as L5 ET by SCF were selected[37]. The read counts were normalized by the total read counts per cell and log transformed. Top 5,000 highly variable genes were identified with Scanpy[78] (v1.8.1) and $z$-score was scaled across all the cells. For Epi-retro-seq, the posterior methylation levels of 12,261 genes in the 848 L5 ET cells were computed[45]. Top 5,000 highly variable genes were identified with AllCools[79] and $z$-score was scaled across all the cells. The 1,512 genes as the intersection between the two highly variable gene lists were used in Scanorama[80] (v1.7.1) to integrate the $z$-scored expression matrix and minus $z$-scored methylation matrix with sigma equal to 100.

## Integrating mouse transcriptomic, spatially resolved transcriptomic, and epigenomic datasets

To integrate IT cell types from different mouse datasets, we first take all cells that are labelled as IT, except for L6_IT_Car3, from the 11 datasets as listed in Fig. 7a. These cell labels are either from dataset-specific analyses[41,45], or from the integrated clustering of multiple datasets[37]. The integrated clustering and embedding of the 11 datasets are then generated by projecting all datasets into the 10x v2 scRNA-seq dataset using SingleCellFusion[37,79]. Genome browser views of IT and ET cell types (Figs. 7b, 8c) are taken from the corresponding cell types of the brainome portal[37] (https://brainome.ucsd.edu/BICCN_MOp). MERFISH data were analysed using custom Python code, which is available at https://github.com/ZhuangLab/MERlin.

## Identification of cCREs

For peak calling in the snATAC-seq data, we extracted all the fragments for each cluster, and then performed peak calling on each aggregate profile using MACS2[81] v2.2.7.1. using Python 3.6 with parameter: "--nomodel --shift −100 --ext 200 --qval 1e-2 −B --SPMR". First, we extended peak summits by 250 bp on either side to a final width of 501 bp. Then, to account for differences in performance of MACS2 based on read depth and/or number of nuclei in individual clusters, we converted MACS2 peak scores ($-\log_{10}(q$-value)) to 'score per million'[82]. Next, a union peak set was obtained by applying an iterative overlap peak-merging procedure, which avoids daisy-chaining and still allows for use of fixed-width peaks. Finally, we filtered peaks by choosing a score per million cut-off of 5 as cCREs for downstream analysis.

## Predicting enhancer–promoter interactions

First, co-accessible cCREs are identified for all open regions in all neuron types (cell clusters with less than 100 nuclei from snATAC-seq are excluded) using Cicero[83] with the following parameters: aggregation k = 50, window size = 500 kb, distance constraint = 250 kb. In order to find an optimal co-accessibility threshold, we generated a random shuffled cCRE-by-cell matrix as background and calculated co-accessible scores from this shuffled matrix. We fitted the distribution of co-accessibility scores from random shuffled background into a normal distribution model by using the R package fitdistrplus[84]. Next, we tested every co-accessible cCRE pair and set the cut-off at co-accessibility score with an empirically defined significance threshold of FDR < 0.01. The cCREs outside of ±1 kb of transcriptional start sites in GENCODE mm10 (v16) were considered distal. Next, we assigned co-accessibility pairs to three groups: proximal-to-proximal, distal-to-distal and distal-to-proximal. In this study, we focus only on distal-to-proximal pairs. We calculated the Pearson's correlation

coefficient (PCC) between gene expression (scRNA SMART-seq) and cCRE accessibility across the joint clusters to examine the relationships between the distal cCREs and target genes as predicted by the co-accessibility pairs. To do so, we first aggregated all nuclei or cells from scRNA-seq and snATAC-seq for every joint cluster to calculate accessibility scores ($\log_2$ CPM) and relative expression levels ($\log_2$ transcripts per million). Then, PCC was calculated for every gene-cCRE pair within a 1-Mbp window centred on the transcriptional start sites for every gene. We also generated a set of background pairs by randomly selecting regions from different chromosomes and shuffling the cluster labels. Finally, we fit a normal distribution model on background and defined a cut-off at PCC score with an empirically defined significance threshold of FDR < 0.01, in order to select significant positively correlated cCRE-gene pairs.

## Identification of *cis*-regulatory modules

We used nonnegative matrix factorization (NMF) to group cCREs into *cis*-regulatory modules based on their relative accessibility across cell clusters. We adapted NMF (Python package: sklearn v.0.24.2) to decompose the cluster-by-cCRE matrix $V$ ($N \times M$, $N$ rows: cCRE, $M$ columns: cell clusters) into a coefficient matrix $H$ ($R \times M$, $R$ rows: number of modules) and a basis matrix $W$ ($N \times R$), with a given rank $R$: $V \approx WH$.

The basis matrix defines module related accessible cCREs, and the coefficient matrix defines the cell cluster components and their weights in each module. The key issue to decompose the occupancy profile matrix was to find a reasonable value for the rank $R$ (that is, the number of modules). Several criteria have been proposed to decide whether a given rank $R$ decomposes the occupancy profile matrix into meaningful clusters. Here we applied a measurement called sparseness[85] to evaluate the clustering result. Median values were calculated from 100 times for NMF runs at each given rank with a random seed, which will ensure the measurements are stable. Next, we used the coefficient matrix to associate modules with distinct cell clusters. In the coefficient matrix, each row represents a module and each column represents a cell cluster. The values in the matrix indicate the weights of clusters in their corresponding module. The coefficient matrix was then scaled by column (cluster) from 0 to 1. Subsequently, we used a coefficient > 0.1 (~95th percentile of the whole matrix) as a threshold to associate a cluster with a module. Similarly, we associated each module with accessible elements using the basis matrix. For each element and each module, we derived a basis coefficient score, which represents the accessible signal contributed by all clusters in the defined module.

## Identification of subclass-selective transcription factors by both RNA expression and motif enrichment

All analyses for this section were at the subclass level. For RNA expression, we used the scSMART-seq dataset and compared each subclass with the rest of the population through a one-tailed Wilcoxon test and FDR correction to select significantly differentially expressed transcription factors (adjusted *P*-value < 0.05, cluster average fold change > 2). To perform the motif enrichment analysis, we used known motifs from the JASPAR 2020 database[86] and the subclass specific hypo-CG-DMR identified in Yao et al.[37]. The AME software from the MEME suite (v5.1.1)[87] was used to identify significant motif enrichment (adjusted *P*-value < $10^{-3}$, odds ratio > 1.3) using default parameters and the same background region set as described[37]. All genes in Extended Data Fig. 7 were both significantly expressed and had their motif enriched in at least one of the subclasses.

## Generation and use of new knockin mouse lines

All experimental procedures were approved by the Institutional Animal Care and Use Committees (IACUC) of Cold Spring Harbor Laboratory, University of California Berkeley and Allen Institute, in accordance with NIH guidelines. Mouse knockin driver lines are being deposited to the Jackson Laboratory for wide distribution.

## Generation and use of *Tle4-2A-CreER*, *Fezf2-2A-CreER*, *PlexinD1-2A-CreER*, *PlexinD1-2A-Flp*, *Tbr2-2A-CreER* and dual-tTA mouse lines

Driver and reporter mouse lines were generated using a PCR-based cloning. Knockin mouse lines *Tle4-2A-CreER*, *Fezf2-2A-CreER*, *PlexinD1-2A-CreER*, *PlexinD1-2A-Flp* and *Tbr2-2A-CreER* were generated by inserting a 2A-CreER or 2A-Flp cassette in-frame before the STOP codon of the targeted gene. Targeting vectors were generated using a PCR-based cloning approach[27,47]. In brief, for each gene of interest, two partially overlapping BAC clones from the RPCI-23&24 library (made from C57BL/b mice) were chosen from the Mouse Genome Browser. 5′ and 3′ homology arms were PCR amplified (2–5 kb upstream and downstream, respectively) using the BAC DNA as template and cloned into a building vector to flank the 2A-CreERT2 or 2A-Flp expressing cassette as described[27]. These targeting vectors were purified, tested for integrity by enzyme restriction and PCR sequencing. Linearized targeting vectors were electroporated into a 129SVj/B6 hybrid ES cell line (v.6.5). ES cell clones were first screened by PCR and then confirmed by Southern blotting using appropriate probes. DIG-labelled Southern probes were generated by PCR, subcloned and tested on wild-type genomic DNA to verify that they give clear and expected results. Positive v6.5 ES cell clones were used for tetraploid complementation to obtain male heterozygous mice following standard procedures. The $F_0$ males and subsequent generations were bred with reporter lines (*Ai14*, Snap25-LSL-EGFP, *Ai65*) and induced with tamoxifen at the appropriate ages to characterize the resulting genetically targeted recombination patterns. Drivers *Tle4-2A-CreER*, *Fezf2-2A-CreER* and *PlexinD1-2A-CreER* were additionally crossed with reporter Rosa26-CAG-LSL-Flp and *Tbr2-2A-CreER;PlexinD1-2A-Flp* with reporter dual-tTA, and induced with tamoxifen at the appropriate age to perform anterograde viral tracing, with Flp- or tTA-dependent AAV vector expressing EGFP (AAV8-CAG-fDIO-TVA-EGFP or AAV-TRE-3g-TVA-EGFP), to characterize the resulting axon projection patterns.

## Generation of *Npnt-P2A-FlpO* and *Slco2a1-P2A-Cre* mouse lines

To generate lines bearing in-frame genomic insertions of P2A-FlpO or P2A-Cre, we engineered double-strand breaks at the stop codons of *Npnt* and *Slco2a1*, respectively, using ribonucleoprotein (RNP) complexes composed of SpCas9-NLS protein and in vitro transcribed sgRNA (*Npnt*: GATGATGTGAGCTTGAAAAG and *Slco2a1*: CAGTCTGCAGGA-GAATGCCT). These RNP complexes were nucleofected into $10^6$ v6.5 mouse embryonic stem cells (C57/BL6;129/sv; a gift from R. Jaenisch) along with repair constructs in which P2A-FlpO or P2A-Cre was flanked with the following sequences homologous to the target site, thereby enabling homology-directed repair.

*Npnt-P2A-FlpO*: TGGCCCTTGAGCTCTAGTGTTCCCACTTGCCATAG AAATCTGATCTTCGGTTTGGGGGAAGGGTTGCCTTACCATGCTCCATG AGTGAGCACTGGGAAAAGGGGCAGAGGAGGCCTGACCAGTGTATACG TTCTCTCCCTAGGTCATCTTCAAAGGTGAAAAAAGGCGTGGTCACACGG GGGAGATTGGATTGGATGATGTGAGCTTGAAGCGCGGAAGATGTGGAA GCGGAGCTACTAACTTCAGCCTGCTGAAGCAGGCTGGAGACGTGGAG GAGAACCCTGGACCTATGGCTCCTAAGAAGAAGAGGAAGGTGATGAGC CAGTTCGACATCCTGTGCAAGACCCCGCCGAAGGTGCTGGTGCGGCAG TTCGTGGAGAGATTCGAGAGGCCCAGCGGCGAAAAGATCGCCAGCTGT GCCGCCGAGCTGACCTACCTGTGCTGGATGATCACCCACAACGGCACC GCGATCAAGAGGGCCACCTTCATGAGTTATAACACCATCATCAGCAACA GCCTGAGTTTTGACATCGTGAACAAGAGCCTGCAGTTCAAGTACAAGAC CCAGAAGGCCACCATCCTGGAGGCCAGCCTGAAGAAGCTGATCCCCG CATGGGAGTTCACGATTATCCCTTACAACGGCCAGAAGCACCAGAGCG ACATCACCGACATCGTGTCCAGCCTGCAGCTGCAGTTCGAAAGCAGCG AGGAGGCCGACAAGGGGAATAGCCACAGCAAGAAGATGCTGAAGGCC CTGCTGTCCGAAGGCGAGAGCATCTGGGAGATTACCGAGAAGATCCTG AACAGCTTCGAGTACACCAGCAGATTTACCAAAACGAAGACCCTGTACC AGTTCCTGTTCCTGGCCACATTCATCAACTGCGGCAGGTTCAGCGCACA

TCAAGAACGTGGACCCGAAGAGCTTCAAGCTCGTCCAGAACAAGTATC
TGGGCGTGATCATTCAGTGCCTGGTCACGGAGACCAAGACAAGCGTGT
CCAGGCACATCTACTTTTTTCAGCGCCAGAGGCAGGATCGACCCCCTGG
TGTACCTGGACGAGTTCCTGAGGAACAGCGAGCCCGTGCTGAAGAGA
GTGAACAGGACCGGCAACAGCAGCAGCAACAAGCAGGAGTACCAGCTG
CTGAAGGACAACCTGGTGCGCAGCTACAACAAGGCCCTGAAGAAGAA
CGCCCCCTACCCCATCTTCGCTATTAAAAACGGCCCTAAGAGCCACATC
GGCAGGCACCTGATGACCAGCTTTCTGAGCATGAAGGGCCTGACCGAG
CTGACAAACGTGGTGGGCAACTGGAGCGACAAGAGGGCCTCCGCCGT
GGCCAGGACCACCTACACCCACCAGATCACCGCCATCCCCGACCACTAC
TTCGCCCTGGTGTCCAGGTACTACGCCTACGACCCCATCAGTAAGGAGA
TGATCGCCCTGAAGGACGAGACCAACCCCATCGAGGAGTGGCAGCACA
TCGAGCAGCTGAAGGGCAGCGCCGAGGGCAGCATCAGATACCCCGCCT
GGAACGGCATTATAAGCCAGGAGGTGCTGGACTACCTGAGCAGCTACAT
CAACAGGCGGATCTGAAAGAGGTCGCTGCTGAGAAGACCCCTGGCAG
CTCCCGAGCTAGCAGTGAATTTGTCGCTCTCCCTCATTTCCCAATGCTT
GCCCTCTTGTCTCCCTCTTATCAGGCCTAGGGCAGGAGTGGGTCAGGA
GGAAGGTTGCTTGGTGACTCGGGTCTCGGTGGCCTGTTTTGGTGCAAT
CCCAGTGAACAGTGACACTCTCGAAGTACAGGAGCATCTGGAGACACCT
CCGGGCCCTTCTG

*Slco2a1-P2A-Cre*: TGCCCCTGGGCCTCACCATACCTGTCTCTTCCTGCC
TCATAGGTACCTGGGCCTACAGGTAATCTACAAGGTCTTGGGCACACT
GCTGCTCTTCTTCATCAGCTGGAGGGTGAAGAAGAACAGGGAATACAG
TCTGCAGGAGAATGCTTCCGGATTGATTGGAAGCGGAGCTACTAACTTC
TCCCTGTTGAAACAAGCAGGGGATGTCGAAGAGAATCCTGGACCTATG
GCTCCTAAGAAGAAGAGGAAGGTGATGAGCCAGTTCGACATCCTGTGCA
AGACTCCTCCAAAGGTGCTGGTGCGGCAGTTCGTGGAGAGATTCGAGA
GGCCCAGCGGCGAGAAGATCGCCAGCTGTGCCGCCGAGCTGACCTACC
TGTGCTGGATGATCACCCACAACGGCACCGCCATCAAGAGGGCCACCT
TCATGAGCTACAACACCATCATCAGCAACAGCCTGAGCTTCGACATCG
TGAACAAGAGCCTGCAGTTCAAGTACAAGACCCAGAAGGCCACCATCC
TGGAGGCCAGCCTGAAGAAGCTGATCCCCGCCTGGGAGTTCACCATCA
TCCCTTACAACGGCCAGAAGCACCAGAGCGACATCACCGACATCGTG
TCCAGCCTGCAGCTGCAGTTCGAGAGCAGCGAGGAGGCCGACAAGG
GCAACAGCCACAGCAAGAAGATGCTGAAGGCCCTGCTGTCCGAGGGC
GAGAGCATCTGGGAGATCACCGAGAAGATCCTGAACAGCTTCGAGTACA
CCAGCAGGTTCACCAAGACCAAGACCCTGTACCAGTTCCTGTTCCTG
GCCACATTCATCAACTGCGGCAGGTTCAGCGACATCAAGAACGTGGA
CCCCAAGAGCTTCAAGCTGGTGCAGAACAAGTACCTGGGCGTGATCATT
CAGTGCCTGGTGACCGAGACCAAGACAAGCGTGTCCAGGCACATCTAC
TTTTTTCAGCGCCAGAGGCAGGATCGACCCCCTGGTGTACCTGGACGAG
TTCCTGAGGAACAGCGAGCCCGTGCTGAAGAGAGTGAACAGGACCGGC
AACAGCAGCAGCAACAAGCAGGAGTACCAGCTGCTGAAGGACAACCT
GGTGCGCAGCTACAACAAGGCCCTGAAGAAGAACGCCCCCTACCCCA
TCTTCGCTATCAAGAACGGCCCTAAGAGCCACATCGGCAGGCACCTGA
TGACCAGCTTTCTGAGCATGAAGGGCCTGACCGAGCTGACAAACGTGG
TGGGCAACTGGAGCGACAAGAGGGCCTCCGCCGTGGCCAGGACCACCT
ACACCCACCAGATCACCGCCATCCCCGACCACTACTTCGCCCTGGTG
TCCAGGTACTACGCCTACGACCCCATCAGCAAGGAGATGATCGCCCTG
AAGGACGAGACCAACCCCATCGAGGAGTGGCAGCACATCGAGCAGCT
GAAGGGCAGCGCCGAGGGCAGCATCAGATACCCCGCCTGGAACGGCA
TCATCAGCCAGGAGGTGCTGGACTACCTGAGCAGCTACATCAACAGG
CGGATCTGACCTTCAGCTGGGACTACTGCCCTGCCCCAGAGACTGGAT
ATCCTACCCCTCCACACCTACCTATATTAACTAATGTTAGCATGCCTTCC
TCCTCCTTCC

Transfected cells were cultured and resulting colonies directly screened by PCR for correct integration using the following genotyping primers: flanking primer ATGCATTGCTTCATGCCATA and internal recombinase primer CCTTCAGCAGCTGGTACTCC for *Npnt-P2A-FlpO* left homology arm; GATTGAGGTCAGGCCAGAAG and TCGACATCGTGA ACAAGAGC for *Npnt-P2A-FlpO* right homology arm; CTGGTGAAAGGG GAACTCTTGCT and GATCCCTGAACATGTCCATCAGG for *Slco2a1-P2A-Cre* left homology arm; TACAGCATCCCTGACAAACACCA and TAGCACCGCAGGTGTAGAGAAGG for *Slco2a1-P2A-Cre* right homology arm.

The inserted transgenes were fully sequenced and candidate lines were analysed for normal karyotype. Lines passing quality control were aggregated with albino morulae and implanted into pseudopregnant females, producing germline-competent chimeric founders which in turn were crossed with the appropriate reporter lines on the C57/BL6 background.

## Ethics oversight

All experimental procedures using live animals were performed according to protocols approved by Institutional Animal Care and Use Committees (IACUC) of all participating institutions: Allen Institute for Brain Science, Baylor College of Medicine, Broad Institute of MIT and Harvard, Cold Spring Harbor Laboratory, Harvard University, Salk Institute for Biological Studies, University of California Berkeley, University of California San Diego and University of Southern California. Macaque experiments were performed on animals designated for euthanasia via the Washington National Primate Research Center's Tissue Distribution Program.

Postmortem adult human brain tissue collection was performed in accordance with the provisions of the United States Uniform Anatomical Gift Act of 2006 described in the California Health and Safety Code section 7150 (effective 1 January 2008) and other applicable state and federal laws and regulations. The Western Institutional Review Board reviewed tissue collection processes and determined that they did not constitute human subjects research requiring institutional review board (IRB) review. Before commencing the human Patch-seq, the donor provided informed consent and experimental procedures were approved by the hospital institute review board.

## Reporting summary

Further information on research design is available in the Nature Research Reporting Summary linked to this paper.

## Data availability

Primary data are accessible through the Brain Cell Data Center and data archives. Brain Cell Data Center (BCDC), Overall BICCN organization and data, www.biccn.org. Neuroscience Multi-omic Data Archive (NeMO), RRID:SCR_016152. Brain Image Library (BIL), RRID:SCR_017272. Distributed Archives for Neurophysiology Data Integration (DANDI), RRID:SCR_017571. Publicly used databases in study: NCBI Homologene, 11/22/2019, https://www.ncbi.nlm.nih.gov/homologene, GENCODE mm10 (v16), https://www.gencodegenes.org, JASPAR 2020 database, http://jaspar.genereg.net. All data resources associated with this publication are available as listed at: https://github.com/BICCN/CellCensusMotorCortex and https://doi.org/10.5281/zenodo.4726182.

## Code availability

All code and libraries used in the manuscript are available at https://github.com/BICCN/CellCensusMotorCortex and https://doi.org/10.5281/zenodo.4726182.

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

**Acknowledgements** We thank additional members of our laboratories and institutions who contributed to the experimental and analytical components of this project. This work was supported by grants from the National Institute of Mental Health (NIMH) of the National Institutes of Health (NIH) under: U24MH114827, U19MH114821, U19MH114830, U19MH114831, U01MH117072, U01MH114829, U01MH121282, U01MH117023, U01MH114825, U01MH114819, U01MH114812, U01MH121260, U01MH114824, U01MH117079, U01MH116990, U01MH114828, R24MH117295, R24MH114793, R24MH114788, R24MH114815. We thank NIH BICCN program officers, in particular Yong Yao, for their guidance and support throughout this study. Additional support: NIH grants R01NS39600 and R01NS86082 to G.A.A. H.S.B. is a Chan Zuckerberg Biohub Investigator. Deutsche Forschungsgemeinschaft through a Heisenberg Professorship (BE5601/4-1), the Cluster of Excellence Machine Learning—New Perspectives for Science (EXC 2064, project number 390727645) and the Collaborative Research Center 1233 Robust Vision (project number 276693517), the German Federal Ministry of Education and Research (FKZ 01GQ1601 and 01IS18039A) to P.B. This work was supported in part by the Flow Cytometry Core Facility of the Salk Institute with funding from NIH-NCI CCSG: P30 014195 and Shared Instrumentation Grant S10-OD023689. NIH grant R01MH094360 to H.-W.D. We thank M. Becerra, T. Boesen, C. Cao, M. Fayzullina, K. Cotter, L. Gao, L. Gacia, L. Korobkova, D. Lo, C. Mun, S. Yamashita and M Zhu for their technical and informatics support. Hearing Health Foundation Hearing Restoration Project grant to R.H. NIH grant OD010425 to G.D.H. NIH grant RF1MH114126 to E.S.L. and J.T.T. National Natural Science Foundation of China (NNSFC) grant 61890953 to H.G. NNSFC grant 81827901 to Q.L. This project was supported in part by NIH grants P51OD010425 from the Office of Research Infrastructure Programs (ORIP) and UL1TR000423 from the National Center for Advancing Translational Sciences (NCATS). Its contents are solely the responsibility of the authors and do not necessarily represent the official view of NIH, ORIP, NCATS or the Institute of Translational Health Sciences at the Washington National Primate Research Center. NNSFC grant 61871411 and the University Synergy Innovation Program of Anhui Province GXXT-2019-008 to L.Q. Howard Hughes Medical Institute and the Klarman Cell Observatory for A.R. Howard Hughes Medical Institute for J.R.E. and X. Zhuang. NNSFC Grant 32071367 and NSF Shanghai Grant 20ZR1420100 to Yimin Wang. NIH grants R01EY023173 and U01MH105982 to H.Z. Researchers from Allen Institute for Brain Science wish to thank the Allen Institute founder, P. G. Allen, for his vision, encouragement and support.

**Author contributions** BICCN contributing principal investigators: G.A.A., M.M.B., E.M.C., J. Chun, J.R.E., G.F., J.C.G., S.S.G., Y.O.H., M.J.H., R.H., H.-W.D., Z.J.H., E.S.L., B.K.L., M.E.M., L. Ng, P.O., L.P., A.J.R., T.L.T., A.S.T., O.W., X.W.Y., H.Z., K.Z., X. Zhuang and J.N. Principal manuscript editors: Z.J.H., E.S.L. and H.Z. Manuscript writing and figure generation: G.A.A., T.E.B., P.B., E.M.C., T.L.D., J.A.H., J.R.E., M.J.H., H.-W.D., Z.J.H., N.L.J., B.E.K., D.K., E.S.L., Y.E.L., H.L., K.S.M., E.A.M., M. Naeemi, B.Z., P.O., B.R., F.S., P.T., J.T.T., A.S.T., F. Xie, H.Z., M.Z., Z.Z., J.Z., X. Zhuang and J.N. Analysis coordination: T.E.B., E.M.C., J.A.H., J.R.E., M.J.H., H.-W.D., Z.J.H., E.S.L., E.A.M., P.O., B.R., A.S.T., H.Z., X. Zhuang and J.N. Integrated data analysis: E.A., T.E.B., P.B., J.A.H., J.R.E., H.-W.D., Z.J.H., N.L.J., B.E.K., D.K., E.S.L., Y.E.L., H.L., E.A.M., P.O., B.R., F.S., P.T., A.S.T., F. Xie, Z.Y., H.Z., M.Z., Z.Z., J.Z. and X. Zhuang. scRNA-seq and snRNA-seq data generation and processing: D.B., T.N.N., T.C., J. Chun, K.C., N.D., D.D., S.D., W.D., E.L.D., G.F., O.F., M. Goldman, J. Goldy, R.D.H., L. Hu, C.D.K., F.M.K., M.K., B.B.L., K.L., E.S.L., S. Linnarsson, C.S.L., S.A.M., D.M., N.M.N., C.R.P., T.P., N.P., N.M.R., A.R., C.R., W.J.R., S. Savoia, K. Siletti, K. Smith, J.S., B.T., M.T., A.T., H.T., C.T.J.v.V., C.R.V., A.M.Y., H.Z. and K.Z. ATAC-seq data generation and processing: M.M.B., J. Chun, D.D., W.D., R.F., X.H., B.B.L., Y.E.L., C.S.L., J.D.L., J.K.O., C.R.P., A.P-D., N.P., O.P., S.P., B.R., W.J.R., X.W. and K.Z. Methylcytosine data production and analysis: A.I.A., A. Bartlett, M.M.B., L.B., C.O., R.G.C., H.C., J.R.E., C.F., C.L., H.L., J.D.L., J.R.N., M. Nunn, J.K.O., A.P-D., A.C.R., W.T. and J.Z. Epi-retro-seq data generation and processing: A. Bartlett, M.M.B., L.B., E.M.C., C.O., R.G.C., B. Dominguez, J.R.E., C.F., T.I.-C., M.J., X. Jin, C.L., K.L., P.A.M., E.A.M., J.R.N., M. Nunn, Y.P., A.P-D., M. Rashid, A.C.R., J.B.S., P.T., M.V., E.W., Z.Z. and J.Z. 'Omics data analysis: E.A., T.E.B., T.B., A.S.B., M.C., D.D., S.D., J.R.E., R.F., S.F., O.F., J. Gillis, J. Goldy, Q.H., N.L.J., P.V.K., F.M.K., B.B.L., E.S.L., Y.E.L., S. Linnarsson, H.L., E.Z.M., E.A.M., S.-Y.N., V.N., L.P., O.P., E.P., A.R., D.R., H.R.d.B., K. Siletti, K. Smith, S. Somasundaram, K. Street, V.S., B.T., W.T., E.D.V., K.V.d.B., C.T.J.v.V., J.D.W., F. Xie, Z.Y., H.Z., J.Z. and J.N. Tracing and connectivity data generation: X.A., H.S.B., R.K.C., J.A.H., K.E.H., W.G., H.G., J.T.H., I.B., H.-W.D., Z.J.H., G.K., D.J.K., A.L., Xiangning Li, B.K.L., Q.L., K.S.M., L. Ng, L.G., H.H., B.Z., R.M.-C., D.A.S., H.Z. and J.N. Morphology data generation and reconstruction: T.L.D., J.A.H., Z.F., K.E.H., H.G., H.-W.D., Z.J.H., X. Jia, S.J., T.J., X.K., R.L., P.L., Xiangning Li, Yaoyao Li, Yuanyuan Li, L.L., Q.L., H.P., L.Q., M. Ren, Z.R., E.S., Y.S., W.W., P.W., Yimin Wang, Yun Wang, L.Y., J.Y., H.Z., S.Z. and X. Zhao. OLST/STPT and other data generation: X.A., W.G., J.T.H., Z.J.H., G.K., K.S.M., A.N., P.O., R.P. and R.M.-C. Morphology, connectivity and imaging analysis: X.A., G.A.A., S.B., L.D., J.A.H., Z.F., W.G., H.G., J.T.H., H.-W.D., Z.J.H., D. Huilgol, B. Huo, X. Jia, G.K., H.-C.K., S. Laturnus, A.L., Xu Li, N.N.F., K.S.M., P.P.M., J.M., M. Naeemi, A.N., L. Ng, P.O., R.P., H.P., R.M.-C., Q.W., Yimin Wang, Yun Wang, P.X., F. Xiong, Y.Y. and H.Z. Spatially resolved single-cell transcriptomics (MERFISH): M.Z., S.W.E., B.Z., Z.Y., H.Z., H.-W.D. and X. Zhuang. Multimodal profiling (Patch-seq): P.B., J.B., M.B., Y.B., C.R.C., J.R.C., R.D., P.R.N., L. Hartmanis, G.D.H., X. Jiang, B.E.K., C.D.K., A.L.K., D.K., S. Laturnus, E.S.L., E.M., S. Mulherkar, S.F.O., R.S., F.S., K. Smith, S.A.S., Z.H.T., J.T.T., A.S.T. and H.Z. Transgenic tools: S.A., X.A., H.S.B., R.K.C., T.L.D., W.G., J.T.H., D. Hockemeyer, Z.J.H., D. Huilgol, G.K., D.J.K., A.Y.L., K.S.M., D.A.S., B.T., M.B.V., X.W.Y., Z.Y., H.Z. and J.N. NeMO archive and analytics: R.S.A., S.A.A., H.C.B., R.C., A.C., C.C., J. Crabtree, H.C., V.F., M. Giglio, B.R.H., R.H., J.K., A.M., C.M., L. Nickel, D.O., J.O., M.S. and O.W. Brain Image Library (BIL) archive: G.H. and A.J.R. DANDI archive: B. Dichter, S.S.G., M. Grauer, Y.O.H. and B. Helba. Brain Cell Data Center (BCDC): A. Bandrowski, N.B., B.C., F.D.D., K.D., J.C.G., T.H.G., M.J.H., F.K., K. Konwar, M.E.M., L. Ng, C.T. and T.L.T. Project management: F.D.D., H.G., K. Kelly, B.B.L., K.S.M., S. Mok, H.H., M. Nunn, S. Sunkin and C.T. Manuscript correspondence: H.Z.

**Competing interests** A. Bandrowski is a cofounder of SciCrunch, a company devoted to improving scientific communication. J.R.E. is a member of Zymo Research SAB. J.A.H., K.E.H., T.N.N. and P.R.N. are currently employed by Cajal Neuroscience. P.V.K. serves on the Scientific Advisory Board of Celsius Therapeutics Inc. M.E.M. is a founder and CSO of SciCrunch Inc., a UCSD tech start up that produces tools in support of reproducibility including RRIDs. A.R. is a founder and equity holder of Celsius Therapeutics, an equity holder in Immunitas Therapeutics and until 31 August 2020 was a member of the scientific advisory board of Syros Pharmaceuticals, Neogene Therapeutics, Asimov and ThermoFisher Scientific. From 1 August 2020, A.R. has been an employee of Genentech. B.R. is a co-founder of Arima Genomics, Inc. and Epigenome Technologies, Inc. K.Z. is a co-founder, equity holder and serves on the Scientific Advisor Board of Singlera Genomics. X. Zhuang is a co-founder and consultant of Vizgen.

**Additional information**
**Correspondence and requests for materials** should be addressed to Hongkui Zeng (lead contact), Edward M. Callaway, Hong-Wei Dong, Joseph R. Ecker, Michael J. Hawrylycz, Z. Josh Huang, Ed S. Lein, John Ngai, Pavel Osten, Bing Ren, Andreas Savas Tolias, Owen White, or Xiaowei Zhuang.

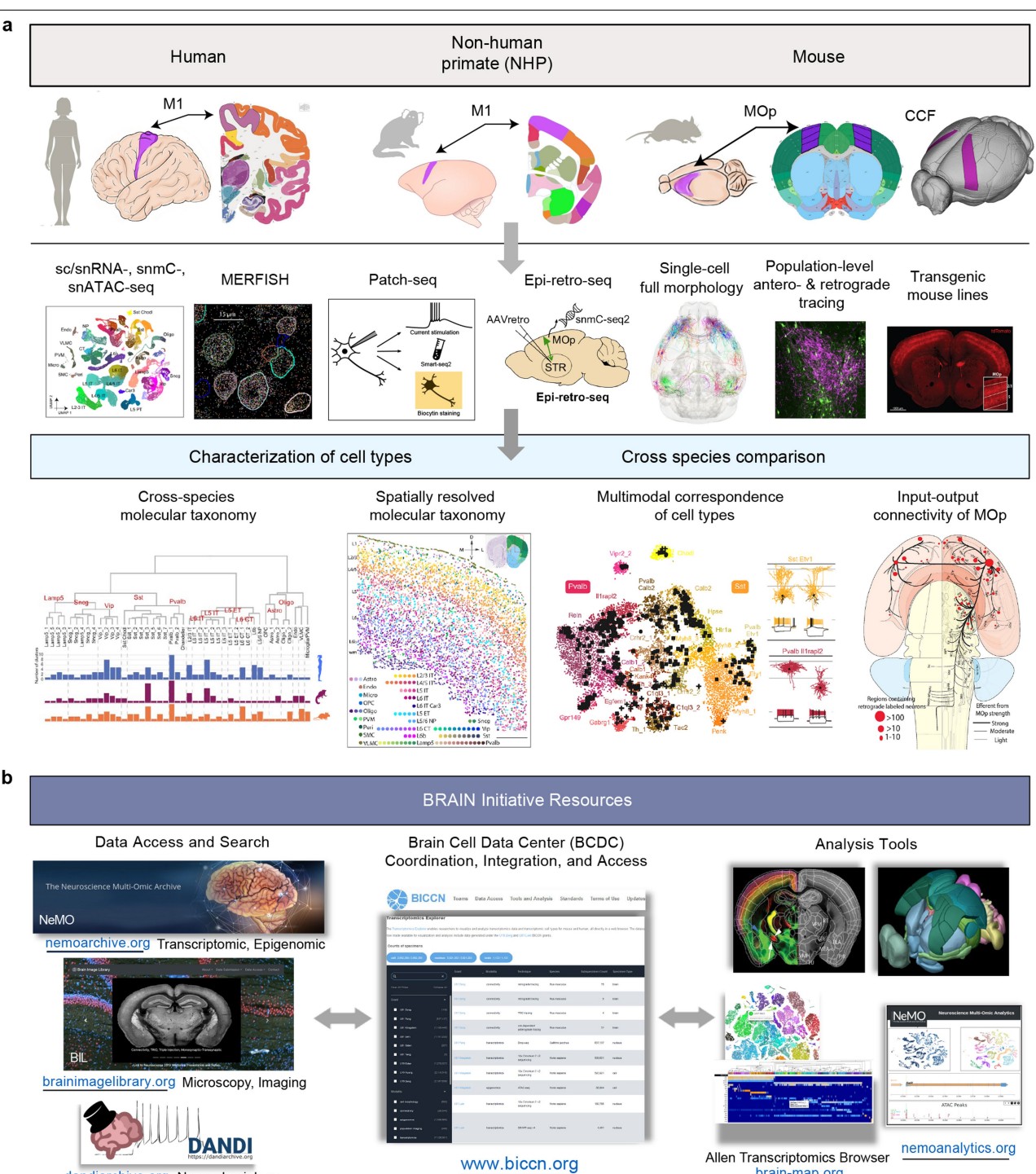

**Extended Data Fig. 1 | Summary of experimental and computational approaches taken and community resources generated by BICCN.**
**a**, Comprehensive characterization of cell types in the primary motor cortex (MOp or M1) of three mammalian species using multiple approaches spanning molecular, genetic, physiological and anatomical domains. Integration of these datasets leads to a cohesive multimodal description of cell types in the mouse MOp and a cross-species molecular taxonomy of MOp cell types. **b**, The multimodal datasets are organized by the Brain Cell Data Center (BCDC), archived in the Neuroscience Multi-omic (NeMO) Archive (for molecular datasets), Brain Image Library (BIL, for imaging datasets) and Distributed

Archive for Neurophysiology Data Integration (DANDI, for electrophysiology data), and made publicly available through the BICCN web portal www.biccn. org and resource page DOI:10.5281/zenodo.4726182. Human and mouse icons and brains are credited to Anna Hupalowska at Broad Institute. Marmoset icon and brain are modified from unrestricted use purchase from Shutterstock. Allen mouse CCF, BCDC and transcriptomics browser images are reproduced with permission from Allen Institute. Mouse brain panel in Epi-retro-seq is adapted from https://commons.wikimedia.org/wiki/File:Mouse_brain_ sagittal.svg (public domain). DANDI artwork is licensed under CC-BY-3.0 from https://github.com/dandi/artwork.

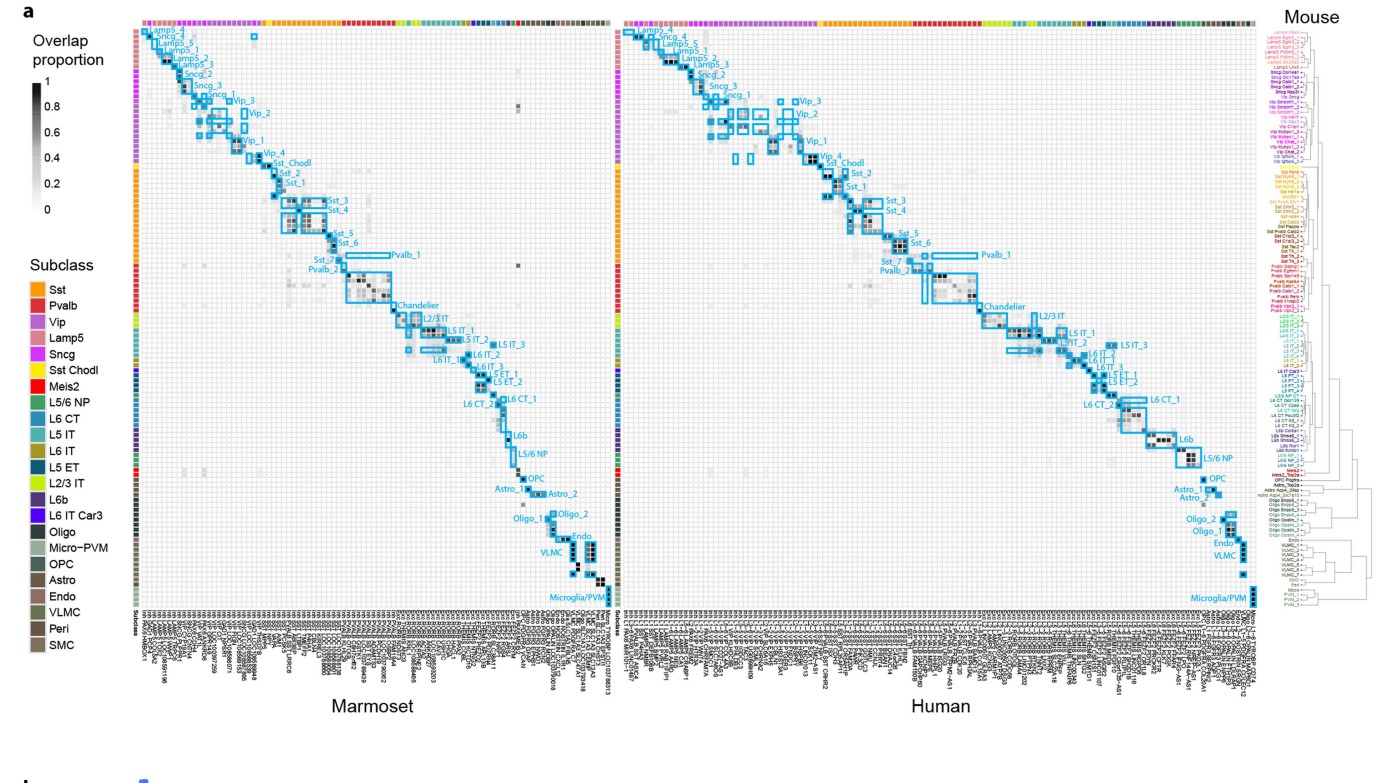

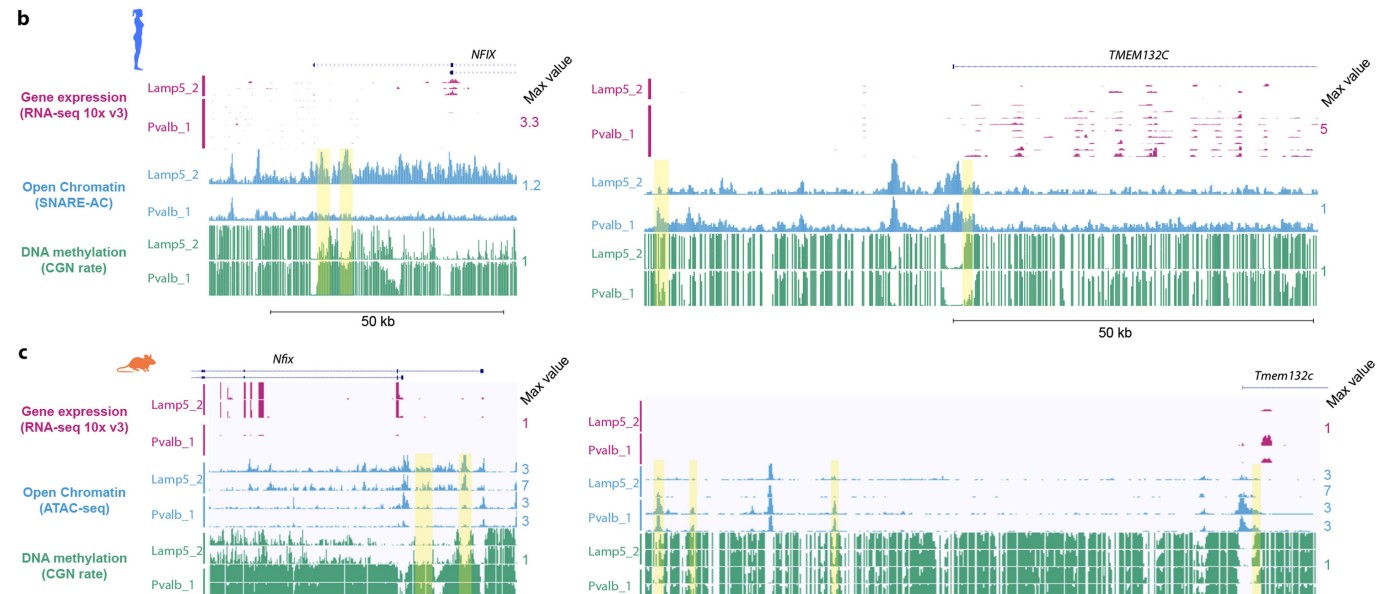

**Extended Data Fig. 2 | MOp consensus cell type taxonomy. a**, Cluster overlap heatmap showing the proportion of nuclei in each pair of species clusters that are mixed in the cross-species integrated space. Cross-species consensus clusters are indicated by labeled blue boxes. Mouse clusters (rows) are ordered by the mouse MOp transcriptomic taxonomy dendrogram[37]. Marmoset (left columns) and human (right columns) transcriptomic clusters[38] are ordered to align with mouse clusters. Color bars at top and left indicate subclasses of within-species clusters. **b-c**, Genome browser view showing transcriptomic and epigenetic signatures for gene markers of *Lamp5*_2 (*NFIX*) and *Pvalb*_1 (*TMEM132C*) GABAergic neurons in human (b) and mouse (c). Yellow bars highlight sites of open chromatin and DNA hypomethylation in the cell type with corresponding marker expression.

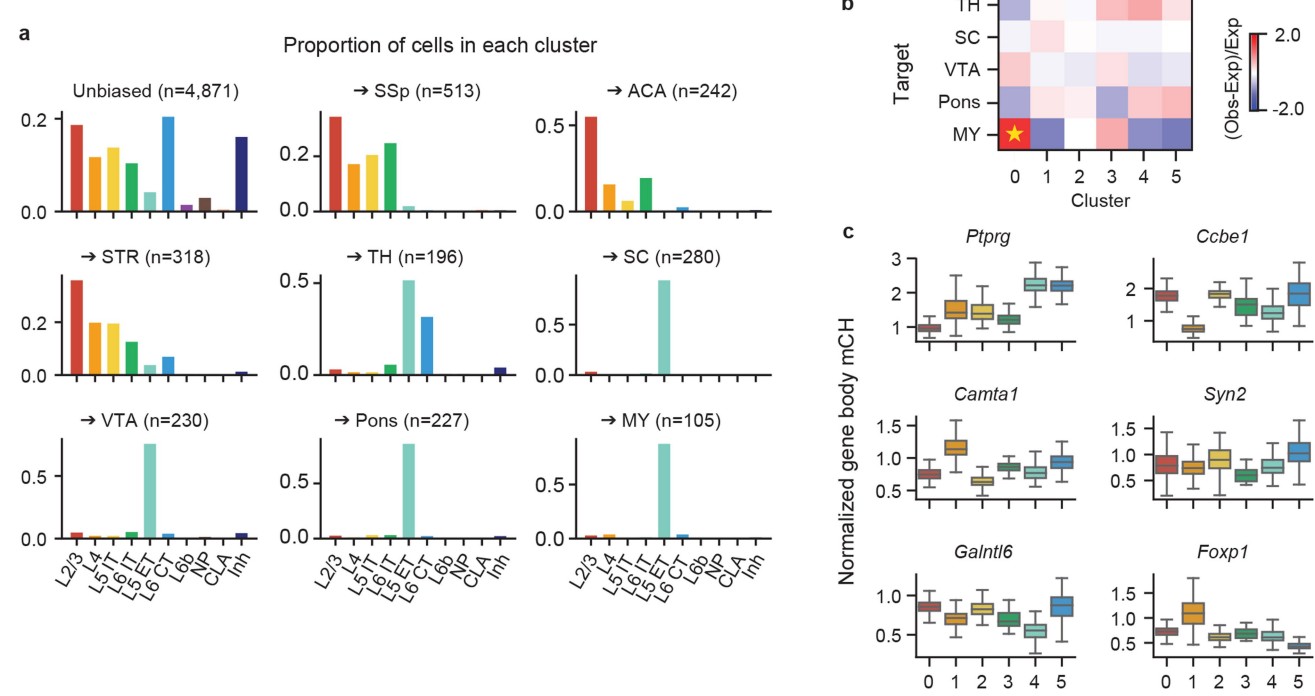

**Extended Data Fig. 3 | Epi-retro-seq links molecular cell types with distal projection targets. a**, Distribution across subclasses of neurons from unbiased snmC-seq2 and neurons projecting to each target. **b**, Enrichment of L5 ET neurons projecting to each target in each cluster. * represents FDR < 0.05, Wald test, Benjamini-Hochberg Procedure. **c**, Boxplots of normalized mCH levels at gene bodies of example CH-DMGs in the six clusters. Numbers of cells represented by the boxes are 242, 165, 118, 42, 119, and 162 for the six clusters. The elements of boxplots are defined as: center line, median; box limits, first and third quartiles; whiskers, 1.5× interquartile range.

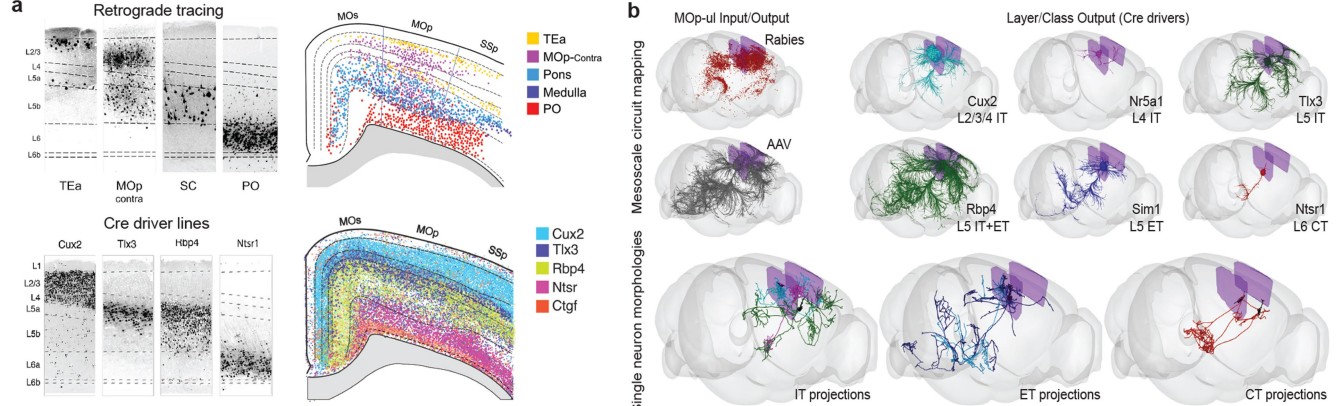

**Extended Data Fig. 4 | Anatomical characterization of MOp-ul neuron types. a**, MOp-ul neurons classified by projection targets or transgenic Cre expression. Top, retrograde tracing using CTb revealed layer-specific distributions of MOp-ul neurons with respect to their major projection targets. Representative images (left) show neurons labeled by CTb injections into cortical areas (TEa, contralateral MOp), SC in the midbrain, and PO of the thalamus. Detected cells were pseudo-colored and overlaid onto a schematic coronal section near the center of MOp-ul (right). MOp neurons that project to TEa are distributed in L2 and L5 (yellow), to the contralateral MOp in L2-L6b (purple), to targets in the pons and medulla in L5b (blue), and to thalamus in L6a (red). Bottom, distribution of neurons labeled in transgenic Cre lines was mapped in MOp and across the whole cortex. Images (left) show laminar patterns of Cre+ nuclei in MOp-ul from four driver lines (*Cux2*, *Tlx3*, *Rbp4*, and *Ntsr1*). Detected nuclei from these lines, plus the *Ctgf-Cre* line, were pseudo-colored and overlaid onto a schematic coronal section near the center

of MOp-ul (right). Cre+ nuclei are found in L2-4 in *Cux2*; L5a and superficial L5b in *Tlx3*; L5a and L5b in *Rbp4*; L6a in *Ntsr1*, and L6b in *Ctgf*. **b**, 3D views show brain-wide MOp input–output patterns at the population and single cell resolution. Top left, regional MOp inputs and outputs were mapped using retrograde (in red, example showing rabies tracing from the *Tlx3-Cre* line) and anterograde (in black, example showing AAV-EGFP) tracing methods. Top right, whole-brain axonal trajectories from 6 Cre line-defined subpopulations labeled with Cre-dependent AAV tracer injections at the same MOp-ul location. Bottom, individual projection neurons were fully reconstructed following high-resolution whole-brain imaging of sparsely labeled cells. Representative examples of IT, ET, and CT neurons are shown in each panel. The two ET examples represent distinct projection types; medulla (dark blue)- and non-medulla-projecting (light blue). 3D renderings were generated following registration of projection and reconstruction data into CCFv3 using BrainRender[88].

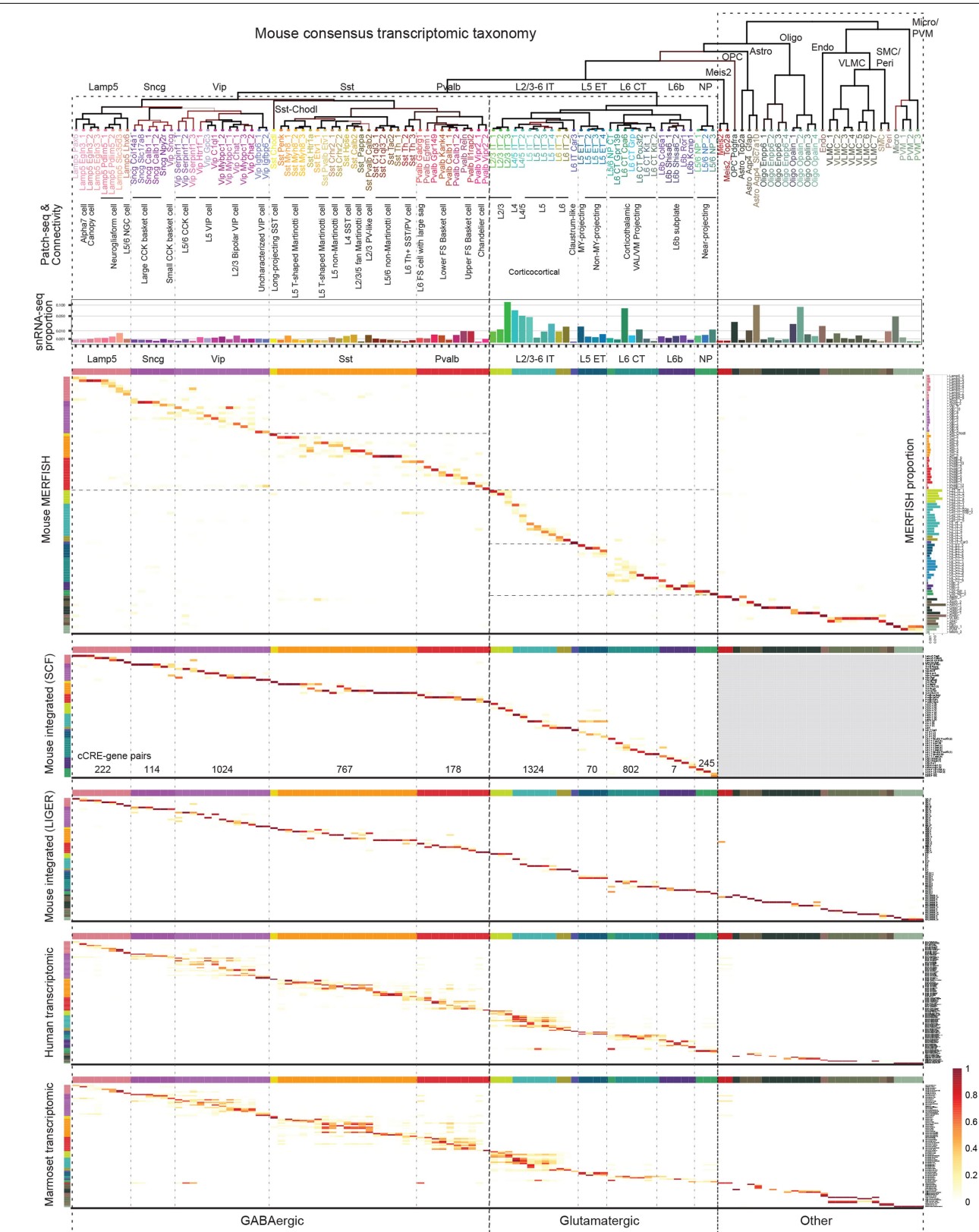

**Extended Data Fig. 5** | See next page for caption.

**Extended Data Fig. 5 | An integrated multimodal census and atlas of cell types in the primary motor cortex of mouse, marmoset and human.** The mouse MOp consensus transcriptomic taxonomy at the top is used to anchor cell type features in all the other modalities. Subclass labels are shown above major branches and cluster labels are shown below each leaf node. Confusion matrices show correspondence between the mouse MOp transcriptomic taxonomy (116 clusters) with those derived from other molecular datasets, including mouse MERFISH (95 clusters), the integrated mouse molecular taxonomies by SingleCellFusion (SCF) (56 neuronal clusters) or LIGER (71 clusters), and the human and marmoset transcriptomic taxonomies (127 and 94 clusters, respectively). Cells within each taxonomy were either mapped to the reference (MERFISH, SCF, LIGER) or shared common cells via integration (Human, Marmoset). Color code corresponds to the fraction of cells in each column mapped to or shared with each reference cluster, and each column summed up to 1. These mapping relationships between the mouse consensus transcriptomic taxonomy and other taxonomies are summarized in an overview panel in Figure 9e. Using Patch-seq and connectivity studies, many transcriptomic neuronal types or subclasses are annotated and correlated with known cortical neuron types traditionally defined by electrophysiological, morphological and connectional properties. Relative proportions of all cell types within the mouse MOp are calculated from either the snRNA-seq 10x v3 B dataset (horizontal bar graph) or the MERFISH dataset (vertical bar graph to the right of the MERFISH matrix). The numbers of cCRE-gene pairs in modules corresponding to neuronal subclasses identified by Cicero from the scRNA-seq and snATAC-seq datasets are shown at the bottom of the SCF matrix.

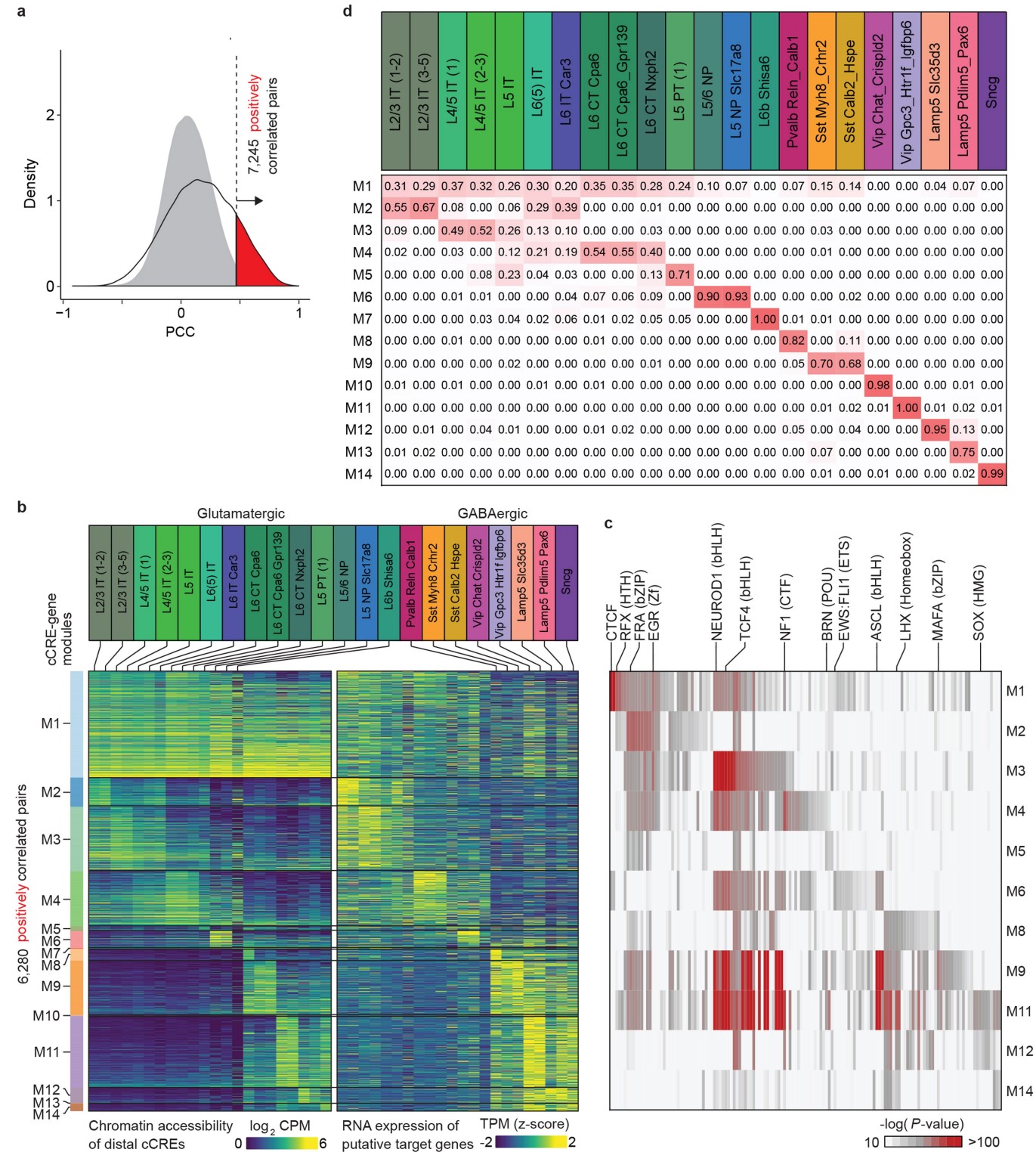

**Extended Data Fig. 6** | See next page for caption.

**Extended Data Fig. 6 | Identification of putative enhancer-gene pairs.**
**a**, Detection of putative enhancer-gene pairs. 7,245 pairs of positively correlated cCRE and genes (highlighted in red) were identified using an empirically defined significance threshold of FDR < 0.01. Grey filled curve shows the distribution of PCC for randomly shuffled cCRE-gene pairs.
**b**, Heatmap of chromatin accessibility of 6,280 putative enhancers, grouped by distinct enhancer-gene modules, across joint cell clusters (left) and expression of 2,490 target genes (right). Note genes are displayed for each putative enhancer separately. CPM: counts per million, TPM: transcripts per million. About 76% of putative enhancers showed cluster-specific chromatin accessibility and were enriched for lineage-specific TFs, while 24% were widely accessible and linked to genes expressed across neuronal clusters with the highest expression in glutamatergic neurons (module M1). Other modules (M2 to M14) of enhancer-gene pairs were active in a subclass-specific manner.
**c**, Enrichment of known TF motifs in distinct enhancer-gene modules.

Displayed are known motifs from HOMER with enrichment -log p-value > 10. In module M1, de novo motif analysis of putative enhancers showed enrichment of sequence motifs recognized by TFs CTCF and MEF2. CTCF is a widely expressed DNA binding protein with a well-established role in transcriptional insulation and chromatin organization, but recently it was also reported that CTCF can promote neurogenesis by binding to promoters and enhancers of related genes. In the L2/3 IT selective module M2, putative enhancers were enriched for the binding motif for Zinc-finger transcription factor EGR, a known master transcriptional regulator of excitatory neurons[89]. In the Pvalb selective module M8, putative enhancers were enriched for sequence motifs recognized by the MADS factor MEF2, which is associated with regulating cortical inhibitory and excitatory synapses and behaviours relevant to neurodevelopmental disorders[90]. **d**, Heatmap showing the weight of each joint cell cluster in each module, derived from the coefficient matrix. The values of each column are scaled (0–1).

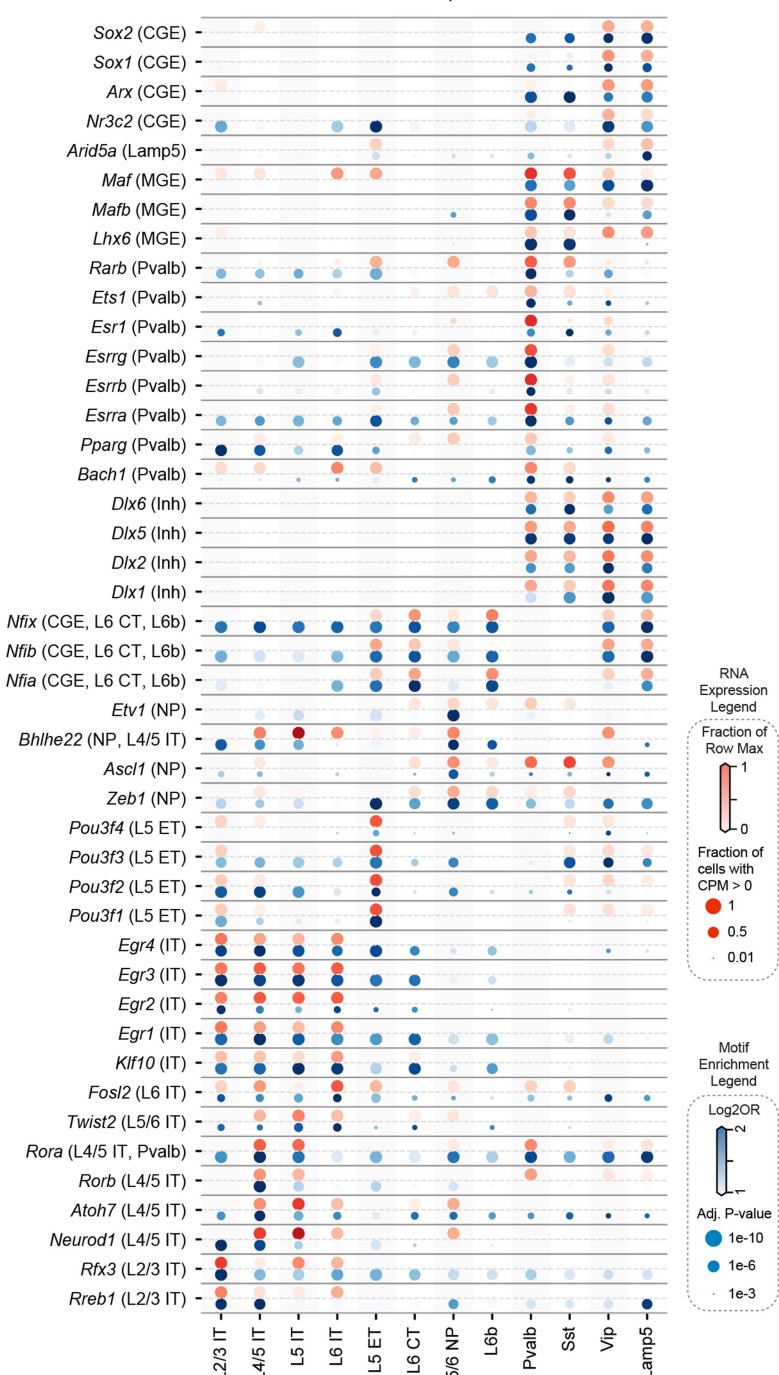

Transcription Factors RNA Expression and
Motif Enrichment in MOp Subclasses

**Extended Data Fig. 7 | Dot plot illustrating RNA expression levels (red) and hypo-CG-DMR motif enrichments (blue) of transcription factors (TFs) in mouse MOp subclasses.** The size and color of red dots indicate the proportion of expressing cells and the average expression level in each subclass, respectively. The size and color of blue dots indicate adjusted P-value (Fisher's exact test, Benjamini-Hochberg Procedure) and $log_2$(Odds Ratio) of motif enrichment analysis, respectively. Combining these two orthologous pieces of evidence identified many well-studied TFs in embryonic precursors, such as the *Dlx* family members for pan-inhibitory neurons, and *Lhx6* and *Mafb* for MGE derived inhibitory neurons. We further identified many additional TFs with more restricted patterns in specific subclasses, such as *Rfx3* and *Rreb1* (in L2/3 IT), *Atoh7* and *Rorb* (in L4/5 IT), *Pou3* family members (in L5 ET), *Etv1* (in L5/6 NP), *Esrr* family members (in *Pvalb*), and *Arid5a* (in *Lamp5*).

**Extended Data Table 1 | Experimental and computational techniques used in this study and associated datasets[91–110]**

| Feature | Experimental or analytic technique(s) | Abbreviations | References | Samples (e.g. # of cells or nuclei) in MOp/M1 | Total samples in flagship and companion papers |
|---|---|---|---|---|---|
| **Transcription** | Single-cell mRNA sequencing | scRNA-seq: SMART-Seq v4 (Smart-seq2 chemistry), 10x Chromium v2, v3 | Background: [9,11] Companion: [37] | **SMART-Seq v4:** 6,288 cells (mouse) **10x Chromium v2, v3:** 193,824 cells (mouse) | 1,163,727 cells |
| | Single-nucleus mRNA sequencing | snRNA-seq: SMART-Seq v4 (Smart-seq2 chemistry), 10x Chromium v2, v3 | Background: [10,91,92] Companion: [37,38] | **SMART-Seq v4:** 6,171 nuclei (mouse) 10,534 nuclei (human) **10x Chromium v2, v3:** 294,717 nuclei (mouse) 69,279 nuclei (marmoset) 15,842 nuclei (macaque) 76,533 nuclei (human) | 1,100,168 nuclei |
| **DNA methylation** | Single-nucleus methylcytosine sequencing 2 | snmC-seq2 | Background: [77] Companion: [37–39] | 9,941 nuclei (mouse) 5,324 nuclei (marmoset) 5,222 nuclei (human) | 110,294 nuclei |
| **Open chromatin** | Single-nucleus Assay for Transposase- Accessible Chromatin | snATAC-seq | Background: [13,93] Companion: [37,40] | 79,625 nuclei (mouse) | 813,799 nuclei |
| **Combined transcription/Open chromatin** | Single-nucleus chromatin accessibility and mRNA expression sequencing | SNARE-seq2 | Background: [94] Companion: [38] | 9,946 nuclei (marmoset) 84,178 nuclei (human) | 94,124 nuclei |
| **Spatially resolved single-cell transcriptomics** | Multiplexed error-robust fluorescence in situ hybridization | MERFISH | Background: [50,51] Companion: [41] | ~300,000 cells (mouse) | ~300,000 cells |
| **Clustering and data integration methods** | Clustering - Hierarchical iterative clustering | scrattch.hicat | Background: [9,11] Companion: [37,38] | | |
| | Clustering - Metacell hierarchical clustering with dynamic tree pruning | tree-based method | Companion: [38] | | |
| | Clustering of snATAC-seq data | SnapATAC | Background: [95] Companion: [40] | | |
| | Clustering - Leiden clustering | | Background: [96] Companion: [38] | | |
| | Multimodality and cross-species integration | LIGER, Seurat, SingleCellFusion (SCF), scrattch.hicat | Background: [11,79,92,97–99] Companion: [37,38] | | |
| **Statistical validation** | Cross-dataset replicability analysis | MetaNeighbor | Background: [100] Companion: [37,38] | | |
| **Electrophysiology, cellular morphology and transcriptomics** | Combined in vitro slice physiology, biocytin cell filling, cytoplasm extraction and RNA-sequencing | Patch-seq, Smart-seq2 | Background: [20,101,102] Companion: [38,42,43] | 1,237 cells (mouse) 6 cells (macaque) 6 cells (human) | 133 cells (mouse) 6 cells (macaque) 391 cells (human) |
| **Cellular morphology and projection** | Whole-brain single-cell full morphology reconstruction | fMOST, MouseLight | | ~300 neurons (full morphology) | 1,741 neurons (full morphology) |
| | Barcoded anatomy resolved by sequencing | BARseq | Background: [18,64,103] Companion: [44,46] | | 10,299 neurons (BARseq) |
| **Inter-areal circuit mapping** | Anterograde tracing: PHAL; Viral tracers: AAV, Cre-dependent AAV, monosynaptic anterograde AAV-Cre | AAV, PHAL | Background: [63,104–107] Companion: [44,47] | 22 experiments (mouse) | |
| | Retrograde tracing: CTB, rabies viral tracers | RV, rabies, TRIO | Background: [75,104,108,109] Companion: [44,47] | 40 experiments (mouse) | |
| **Projection-specific profiling** | Retrograde viral labeling of neurons with defined projections followed by epigenome profiling | Epi-Retro-Seq | | 2,111 cells (mouse) | 11,827 cells |
| | Combined retrograde labeling and MERFISH | Retro-MERFISH | Companion: [41,45] | ~190,000 cells (mouse) | ~190,000 cells |
| **Genetic tools** | Transgenic mouse lines | FlpO, Cre, CreER knockin lines; TIGRE-MORF/Ai166, MORF3 reporter line | Background: [110] Companion: [46,47] Stafford, Daigle, Chance et al., in preparation | 6 knock-in driver lines 1 reporter line | 26 knock-in lines |

## Extended Data Table 2 | Glossary

| Name | | Definition | InterLex Identifier |
|---|---|---|---|
| MOp (mouse), M1 (human and non-human primate) | | Primary motor cortex, the main target of cellular diversity analyses | ILX:0770115 |
| | L1 | Layers within MOp or M1 | ILX:0770170 |
| | L2 | | ILX:0778049 |
| | L3 | | ILX:0778050 |
| | L2/3 (mouse) | | ILX:0770171 |
| | L4 | | ILX:0770172 |
| | L5 | | ILX:0770179 |
| | L6 | | ILX:0770173 |
| | L6b | | ILX:0770180 |
| Brain regions receiving axonal projections from MOp targeted for retrograde labeling studies | | | ILX:0770177 |
| Cortical | SSp | Primary somatosensory cortex | ILX:0770117 |
| | MOs | Secondary motor cortex | ILX:0770116 |
| | TEa | Temporal association area | ILX:0770118 |
| | ACA | Anterior cingulate area | ILX:0770120 |
| Subcortical | STR | Striatum | ILX:0770122 |
| | TH | Thalamus | ILX:0770123 |
| | SC | Superior colliculus | ILX:0770124 |
| | VTA | Ventral tegmental area | ILX:0770137 |
| | HY | Hypothalamus | ILX:0770165 |
| | MB | Midbrain | ILX:0770126 |
| | P | Pons | ILX:0770127 |
| | MY | Medulla | ILX:0770125 |
| | CLA | Claustrum | ILX:0770128 |
| *Cell class*: Top branches of hierarchical tree | | | ILX:0770094 |
| Neural | Non-neuronal | Non-neuronal cells of neuroectoderm origin | ILX:0770099 |
| | Neuronal: GABAergic | Neurons that use GABA as a neurotransmitter and which exert an inhibitory post-synaptic effect | ILX:0770098 |
| | Neuronal: Glutamatergic | Neurons that use glutamate as a neurotransmitter and exert an excitatory post-synaptic effect | ILX:0770097 |
| Non-neural | | Cells of mesoderm, neural crest or yolk sac origin | ILX:0770187 |
| *Cell subclass*: Subset of class, major groupings with highly convergent evidence across data modalities | | | ILX:0770095 |
| GABAergic subclasses | Lamp5 | Genomic, cellular marker, phenotype and developmental origin-defined GABAergic cell sets | ILX:0770149 |
| | Sncg | | ILX:0770150 |
| | Vip | | ILX:0770151 |
| | Sst | | ILX:0770152 |
| | Sst Chodl | | ILX:0770153 |
| | Pvalb | | ILX:0770154 |
| | Meis2 | | ILX:0770155 |
| Glutamatergic subclasses | L2/3 IT | Layer 2/3 intratelencephalic-projecting | ILX:0770156 |
| | L4/5 IT | Layer 4/5 intratelencephalic-projecting | ILX:0770174 |
| | L5 IT | Layer 5 intratelencephalic-projecting | ILX:0770157 |
| | L6 IT | Layer 6 intratelencephalic-projecting | ILX:0770158 |
| | L6 IT Car3 | Layer 6 intratelencephalic-projecting, Car3-expressing | ILX:0770159 |
| | L5 ET | Layer 5 extratelencephalic-projecting | ILX:0770160 |
| | L6 CT | Layer 6 corticothalamic | ILX:0770162 |
| | L6b | Layer 6b neurons | ILX:0770163 |
| | L5/6 NP | Layer 5/6 near-projecting | ILX:0770161 |
| Neural non-neuronal subclasses | Astro | Astrocytes | ILX:0770141 |
| | Oligo | Oligodendrocytes | ILX:0770140 |
| | OPC | Oligodendrocyte progenitor cells | ILX:0770139 |
| Non-neural subclasses | Endo | Endothelial cells | ILX:0770142 |
| | VLMC | Vascular leptomeningeal cells | ILX:0770143 |
| | SMC | Smooth muscle cells | ILX:0770144 |
| | Peri | Pericytes | ILX:0770145 |
| | Micro | Microglia | ILX:0770146 |
| | PVM | Perivascular myeloid cells | ILX:0770147 |
| *Cell type*: Subset of subclass, finest resolution clustering achieved for a modality or a consensus clustering across modalities and/or species | | | ILX:0770096 |

Corresponding author(s): Edward M. Callaway, Hong-Wei Dong, Joseph R. Ecker, Mike Hawrylycz, Z. Josh Huang, Ed S. Lein, John Ngai, Pavel Osten, Bing Ren, Andreas Savas Tolias, Owen White, Hongkui Zeng, Xiaowei Zhuang

# Reporting Summary

Nature Research wishes to improve the reproducibility of the work that we publish. This form provides structure for consistency and transparency in reporting. For further information on Nature Research policies, see our Editorial Policies and the Editorial Policy Checklist.

## Statistics

For all statistical analyses, confirm that the following items are present in the figure legend, table legend, main text, or Methods section.

| n/a | Confirmed | |
|---|---|---|
| ☐ | ☒ | The exact sample size (*n*) for each experimental group/condition, given as a discrete number and unit of measurement |
| ☐ | ☒ | A statement on whether measurements were taken from distinct samples or whether the same sample was measured repeatedly |
| ☐ | ☒ | The statistical test(s) used AND whether they are one- or two-sided *Only common tests should be described solely by name; describe more complex techniques in the Methods section.* |
| ☐ | ☒ | A description of all covariates tested |
| ☐ | ☒ | A description of any assumptions or corrections, such as tests of normality and adjustment for multiple comparisons |
| ☐ | ☒ | A full description of the statistical parameters including central tendency (e.g. means) or other basic estimates (e.g. regression coefficient) AND variation (e.g. standard deviation) or associated estimates of uncertainty (e.g. confidence intervals) |
| ☐ | ☒ | For null hypothesis testing, the test statistic (e.g. *F*, *t*, *r*) with confidence intervals, effect sizes, degrees of freedom and *P* value noted *Give P values as exact values whenever suitable.* |
| ☐ | ☒ | For Bayesian analysis, information on the choice of priors and Markov chain Monte Carlo settings |
| ☐ | ☒ | For hierarchical and complex designs, identification of the appropriate level for tests and full reporting of outcomes |
| ☐ | ☒ | Estimates of effect sizes (e.g. Cohen's *d*, Pearson's *r*), indicating how they were calculated |

*Our web collection on statistics for biologists contains articles on many of the points above.*

## Software and code

Policy information about availability of computer code

| Data collection | All data for this manuscript was collected in the set of corresponding core companion papers. Please refer to these papers for data generation and quantification software.<br><br>• An integrated transcriptomic and epigenomic atlas of mouse primary motor cortex cell types, Yao et al., 2021<br><br>• Evolution of cellular diversity in primary motor cortex of human, marmoset monkey, and mouse, Bakken et al., 2021<br><br>• Molecular, spatial and projection diversity of neurons in primary motor cortex revealed by in situ single-cell transcriptomics, Zhang et al, 2021<br><br>• Phenotypic variation of transcriptomic cell types in mouse motor cortex, Scala et al, 2020<br><br>• Cellular Anatomy of the Mouse Primary Motor Cortex, Munoz-Casteneda, 2021<br><br>• Genetic dissection of the glutamatergic neuron system in cerebral cortex, Matho et al, 2021 |
|---|---|

- An atlas of gene regulatory elements in Adult Mouse Cerebrum, Li et al, 2021

- Human cortical expansion involves diversification and specialization of supragranular intratelencephalic-projecting neurons, Berg et al, 2021

- Brain-wide single neuron reconstruction reveals morphological diversity in molecularly defined striatal, thalamic, cortical and claustral neuron types, Peng et al, 2021

- DNA Methylation Atlas of the Mouse Brain at Single-Cell Resolution, Liu et al., 2021

- Epigenomic Diversity of Cortical Projection Neurons in the Mouse Brain, Zhang et al., 2021

**Data analysis**

Human, macaque, marmoset, mouse transcriptomics:  Code for generating Figure 1b-h, ED Figure 2, and Figure 3h is available at http://data.nemoarchive.org/biccn/lab/lein/2020_M1_study_analysis/Transcriptomics/flagship/. Analysis was performed in RStudio using R version 3.5.3, R packages: Seurat 3.1.1, ggplot2 3.2.1, scrattch.hicat 0.0.22.

Estimation of cell type homology:  A final dendrogram of consensus cell types was constructed by transforming the raw UMI counts to log2 CPM normalized counts. Up to 50 marker genes per cross-species cluster were identified by using the scrattch.hicat (v0.0.22) (https://github.com/AllenInstitute/scrattch.hicat) display_cl and select_markers functions with the following parameters; q1.th = 0.4, q.diff.th = 0.5, de.score.th = 80.

Cross-species differential gene expression and correlations: To calculate the number of DE genes between each species pair for each cross-species cluster, we used a pseudobulk comparison method from DESeq2 (v1.30.0)99. For a given cross-species cluster, each sample was split by species and donor, then a Wald test was performed between each species pair.

Integration of L5 ET cells from Epi-Retro-Seq and 10x snRNA-Seq: Top 5,000 snRNA-seq highly variable genes were identified with Scanpy v1.8.1 and z-score scaled across all the cells.  Top 5,000 Epi-Retro-Seq highly variable genes were identified with AllCools and z-score scaled across all the cells. The 1,512 genes as the intersection between the two highly variable gene lists were used in Scanorama v1.7.1 to integrate the z-scored expression matrix and minus z-scored methylation matrix with sigma equal to 100.

Integrating mouse transcriptomic, spatially resolved transcriptomic, and epigenomic datasets: The integrated clustering and embedding of the 11 datasets are then generated by projecting all datasets into the 10x v2 scRNA-seq dataset using SingleCellFusion.  MERFISH data was analyzed using custom Python code. This code is available at https://github.com/ZhuangLab/MERlin.

Identification of candidate cis-regulatory elements: For peak calling in the snATAC-seq data, we extracted all the fragments for each cluster, and then performed peak calling on each aggregate profile using MACS2 v2.2.7.1. using Python 3.6.

Predicting enhancer-promoter interactions: First, co-accessible cCREs are identified for all open regions in all neurons types (cell clusters with less than 100 nuclei from snATAC-seq are excluded), using Cicero v1.0.0.

Identification of cis-regulatory modules: NMF (Python package: sklearn v0.24.2) was used to decompose the cell-by-cCRE matrix V (N×M, N rows: cCRE, M columns: cell clusters) into a coefficient matrix H (R×M, R rows: number of modules) and a basis matrix W (N×R), with a given rank R.

Code availability: All code and libraries used in the manuscript are available at  https://github.com/BICCN/CellCensusMotorCortex.
 DOI: 10.5281/zenodo.4726182.

---

**Data**  All data is freely available for public use, see also "Data availability" section of the main manuscript.

Primary data is accessible through the Brain Cell Data Center and data archives.

- Brain Cell Data Center (BCDC), Overall BICCN organization and data, www.biccn.org

BRAIN Initiative Data Archives for BICCN data

- Neuroscience Multi-omic Data Archive (NeMO), RRID:SCR_016152
- Brain Image Library (BIL), RRID:SCR_017272
- Distributed Archives for Neurophysiology Data Integration (DANDI), RRID:SCR_017571

Publicly used databases in study:

- NCBI Homologene, 11/22/2019, https://www.ncbi.nlm.nih.gov/homologene
- GENCODE mm10 (v16), https://www.gencodegenes.org
- JASPAR 2020 database, http://jaspar.genereg.net

Data Availability: In addition to the raw data available through the archives all figure specific data sets are available at:

https://github.com/BICCN/CellCensusMotorCortex.      DOI: 10.5281/zenodo.4726182.

# Field-specific reporting

Please select the one below that is the best fit for your research. If you are not sure, read the appropriate sections before making your selection.

☒ Life sciences ☐ Behavioural & social sciences ☐ Ecological, evolutionary & environmental sciences

For a reference copy of the document with all sections, see nature.com/documents/nr-reporting-summary-flat.pdf

# Life sciences study design

All studies must disclose on these points even when the disclosure is negative.

| Sample size | Sample sizes for all data types are provided in Extended Data Table 1 of the manuscript. Specifically:

2,111 cells generated by Epi-Retro-Seq and 9876 cells generated by snmC-seq2 were analyzed. 79,625 nuclei generated by snATAC-seq were used.
6439 cells generated by SMART-seq and 6300 were analyzied
176584 cells generated by 10X v3 and 94170 were analyzed
145748 cells generated by 10X v2 and 122641 were analyzed
6848 nuclei were generated by SMAR-seq v4 and 6278 were analyzed.
91071 nuclei were generated by 10X v2 and 76714 were analyzed.
215823 nuclei were generated by 10X v3 (by Macoksco's lab) and 178926 were analyzed.  90266 nuclei were generated by 10X v3(by Allen Institute for Brain Science) and 40555 were analyzed.

For Epi-Retro-Seq:384 nuclei from each projection from 2 male and 2 female mice (except the MOp-SSp projection from which 768 nuclei were assayed) were analyzed. The sample size was chosen based on preliminary results from pilot studies that on average >=50 labeled projection neurons can be successfully collected from MOp of each animal.

Human, macaque, marmoset, mouse transcriptomics: For high-throughput single nucleus genomic sequencing of primate tissue, sufficient nuclei were profiled to capture the rarest neuronal cell types observed in mouse motor cortex. For lower throughput assays, sufficient nuclei were profiled to characterize all neuronal subclasses and most cell types.

MERFISH: Two replicate animals were imaged under each condition. From the two replicate animals imaged for the identification and spatial mapping of cell types, a total of ~300,000 cells were imaged, which generated a sufficient number of single-cell profiles and gave sufficient statistics for the effect sizes of interest. From the two replicate animals imaged for projection target mapping, a total of ~190,000 cells were imaged, which gave sufficient statistics for the effect sizes of interest.

Mouse Patch-seq:  Sampling strategy was determined using pre-existing knowledge of the transcriptional diversity of the mouse cortex (Tasic et al., 2018; Yao et al., 2020) and based also on the variability of morphological and electrophysiological types predicted by existing literature (Jiang et al., 2015, Gouwens et al. 2019, Scala et al., 2019).

Human Patch-seq: Sampling strategy was determined using preexisting knowledge about the size of primate L5 ET neurons.

Anatomy: The  sample sizes for different injection methods with different tracers were specified in Methods sections as described for different laboratories. In general, representative cases presented in all figures were selected from a much larger data pool. Each of the injections was repeated in at least two cases for verification purposes, and only tracing data that was validated is reported.

Transgenic line characterization: 2-3 individual animals per genotype  (note: valuation of expression patterns of genetically encoded reporters was qualitative). |

| Data exclusions | snmC-Seq2 and Epi-Retro-Seq:
Poor quality nuclei were excluded from clustering if they failed to meet the following pre-established quality control (QC) thresholds:
< 500,000 non-clonal reads
> 1% non-conversion rate

snATAC-seq:
No samples were excluded.
For analysis as pre-established only nuclei with > 1,000 reads/nucleus and transcriptional start site enrichment > 10 were selected.
Potential barcode collisions were excluded from analysis

SMART-seq v4 cells/nuclei:
Cells that met any one of the following pre-established criteria were removed: < 100,000 total reads, < 1,000 detected genes (with CPM > 0), < 75% of reads aligned to genome, or CG dinucleotide odds ratio > 0.5.

10X cells cells/nuclei:
Cells were first classified as neuron or non-neuronal cell types, and neuronal cells with more than 2000 detected genes and non-neuronal cells with more than 1000 detected genes were selected, excluding cells with doublet score greater than 0.3 as a pre-established criterion.

Human, macaque, marmoset, mouse transcriptomics:
The following criteria were pre-established. Nuclei belonging to low-quality, sex-specific, or donor-specific clusters were removed from analyses. Briefly:

Human RNA-seq (SMART-seq v4):
> 30% cDNA longer than 400 base pairs
> 500,000 reads aligned to exonic or intronic sequence
> 40% of total reads aligned
> 50% unique reads
> 0.7 TA nucleotide ratio |

Human and Macaque RNA-seq (10x v3): Pre-establihed acceptance criteria were:

> 500 (non-neuronal nuclei) or > 1000 (neuronal nuclei) genes detected
< 0.3 doublet score

Marmoset RNA-seq (10x v3):
Cell barcodes were filtered to distinguish true nuclei barcodes from empty beads and PCR artifacts by assessing proportions of ribosomal and mitochondrial reads. Pre-established criteria include ratio of intronic/exonic reads (> 50% of intronic reads), library size (> 1000 UMIs) and sequencing efficiency (true cell barcodes have higher reads/UMI).

Mouse RNA-seq (SMART-seq v4 and 10x v3): Pre-established criteria for rejection included
< 100,000 total reads, < 1,000 detected genes (CPM > 0), < 75% of reads aligned to genome, or CG dinucleotide odds ratio > 0.5. Cells were classified into broad classes of excitatory, inhibitory, and non-neuronal based on known markers, and cells with ambiguous identities were removed as doublets.

MERFISH:  No data was excluded from consideration. All images were included in the basic analysis.

Mouse Patch-seq:
Cells meeting any of the pre-established exclusion criteria described in the following were declared low quality and did not get a t-type assignment: cells with the highest correlation below 0.4 (78 cells); cells that would be assigned to non-neural t-types, presumably due to RNA contamination (14 cells); cells with the highest correlation less than 0.02 above the maximal correlation in one of the other two transcriptomic orders (5 cells). Four cells were assigned to an excitatory t-type, despite having clearly inhibitory firing, morphology, and/or soma depth location (such as L1). The most likely cause was RNA contamination from excitatory cells that are much more abundant in the mouse cortex. These four cells were excluded from all analyses and visualizations (as if they did not pass the transcriptomic quality control). In addition, one cell was likely located outside of MOp, based on the slice anatomy, and was excluded as well. For the electrophysiology, the cells were not recorded or included when seal resistance values were <1 GΩ before achieving whole-cell configuration and/or initial access resistance was >30 MΩ.  Cells were excluded from morphological analysis when the staining quality did not match pre-established criteria for inclusion.  Cells that showed low staining quality such as poor fill, excessive background, dendritic or axonal truncation were not reconstructed and not included in the dataset.

Human Patch-seq:
Patch-seq samples with a mapping confidence < 0.5  were excluded from analysis as a pre-established criterion.

Anatomy:
The best most representative injections were chosen for the analysis. The others were excluded due to off-targeting of the injection site, missing/damaged tissue, weak tracer labeling of the axons or high background, etc.   These were pre-established criteria.

Transgenic line characterization:
Data were excluded from failed experiments - i.e., in cases where no signal was detected or tissue was damaged during processing as a pre-established criterion.

Replication

Epi-Retro-Seq:
At least 2 male and 2 female mice were injected with AAV-retro-Cre for each projection target. Male and female samples were pooled separately for nuclei preparation.

snmC-seq2:
Each dissected region has at least two replicates, each replicate was pooled from 6-30 animals separately for nuclei preparation and downstream analyses.

snATAC-seq:
Experiments were performed for 2 biological replicates for each dissected region

sc/snRNA-seq:
The number of animals used for profiling in replication for each platform listed below:
10X v2 cells: 3 male
10X v2 nuclei: 2 male, 1 female
10X v3 cells: 3 male, 3 female
10X v3 nuclei: 5 male, 7 female
SMART-seq cells: 28 male, 17 female
SMART-seq nuclei: 8 male, 2 female

Human, macaque, marmoset, mouse transcriptomics:
All species clusters were examined to ensure representation from multiple donors and both sexes. Specifically, clustering reproducibility was measured by performing clustering analysis 100 times using a randomly-selected 80% of nuclei. Similarly, all cross-species clusters were examined for representation from all three species and final cluster assignment was based on 100 iterations of clustering using 95% of nuclei. Replicated findings by profiling tissue from multiple donors from each species (human, marmoset, and mouse). Donor effects are reported as Extended Data Figures in the companion manuscript.

MERFISH:
Reported results were replicated from two animals under each condition.  Reported results were replicated with replicates generating similar results.

Mouse Patch-seq:
The results of this study were not directly replicated. However, all the results were collected from multiple animals from multiple litters per wild-type and transgenic lines.  For our mouse and macaque recordings we compiled data across different animals (4 macaques; 6 mice)

Human Patch-seq:
The results were not directly replicated, but when possible data were collected from multiple subjects. The human Patch-seq data was from a rare surgical case and so replication was not possible.

Anatomy:
This study focuses on characterizing inputs/outputs of the primary motor cortex upper limb area (MOp-ul) using different tracing methods. Each of tracer injections were repeated multiple times in different animals. While the best, most representative cases were chosen for inclusion in the analysis data set, the other injections served as validation cases, demonstrating the replicability and consistency of tracer labeling.

Transgenic line characterization:
Expression patterns of genetically encoded reporters are typically representative of 2-3 animals of the same genotype.

| Randomization | There was no randomization performed as the study does not involve multiple study groups. |
|---|---|

Human, macaque, marmoset, mouse transcriptomics:
All species specimens were controls and were therefore allocated into the same experimental group. Randomization was not used. To compare datasets across species, random nuclei from each cluster were chosen to downsample each species' dataset, and ensure approximately equal representation of cell types at the subclass level. Additionally, for heatmap visualizations (Figure 2g), up to 50 random nuclei from each subclass for each species were chosen.

| Blinding | There was no blinding performed as the study does not involve multiple study groups. |
|---|---|

Human, macaque, marmoset, mouse transcriptomics:
Human specimens were de-identified and assigned a unique numerical code. Knowledge of which sample came from which species was necessary for analytical pipelines. Additionally, donor metadata was used for QC to ensure no sex- or donor-specific clusters in our cell type taxonomies.

Patch-seq:
Electrophysiological features were extracted without having information about the molecular typing of the cell. Cell type was determined by mapping patch-seq transcriptomic data onto corresponding species Cv3 or SSv4 reference dataset. Researchers were blinded to donor, but not species metadata during alignment.

# Reporting for specific materials, systems and methods

We require information from authors about some types of materials, experimental systems and methods used in many studies. Here, indicate whether each material, system or method listed is relevant to your study. If you are not sure if a list item applies to your research, read the appropriate section before selecting a response.

## Materials & experimental systems

| n/a | Involved in the study |
|---|---|
| ☐ | ☒ Antibodies |
| ☒ | ☐ Eukaryotic cell lines |
| ☒ | ☐ Palaeontology and archaeology |
| ☐ | ☒ Animals and other organisms |
| ☐ | ☒ Human research participants |
| ☒ | ☐ Clinical data |
| ☒ | ☐ Dual use research of concern |

## Methods

| n/a | Involved in the study |
|---|---|
| ☒ | ☐ ChIP-seq |
| ☒ | ☐ Flow cytometry |
| ☒ | ☐ MRI-based neuroimaging |

## Antibodies

| Antibodies used | All data for this manuscript was collected in the set of corresponding core companion papers. All antibodies used in those studies are described in the core companion papers cited in the manuscript, also listed in the above "Data collection" section. |
|---|---|
| Validation | |

## Animals and other organisms

Policy information about studies involving animals; ARRIVE guidelines recommended for reporting animal research

| Laboratory animals | Epi-Retro-Seq: 42-49 day old adult male and female INTACT mice (R26R-CAG-loxp-stop-loxp-Sun1-sfGFP-Myc maintained on C57BL/6J background) were used for Epi-Retro-Seq experiments. |
|---|---|

snmC-seq2: Adult (P56) C57BL/6J male mice

snATAC-seq: Adult (P56) C57BL/6J male mice

sc/snRNA-seq: Adult (P56) C57BL/6J male and female mice

Human, macaque, marmoset, mouse transcriptomics:
Common marmoset (Callithrix jacchus) animals were used (2 males, 1.9 years and 2.3 years; and 1 female, 3.1 years).
Pig-tailed macaque (Macaca nemestrina) animals were used (2 males, 12 and 17 years; and 1 female, 3 years).
Mouse (Mus musculus) animals were used (male and female wildtype C57Bl/6J P56 +/- 3 days).

MERFISH: Mus musculus, C57BL/6, male, P57-63

Mouse Patch-seq: Male and Female mice (median age 75 days, interquartile range 64-100, full range 35-245 days) were used in this study. Specific information about every single animal can be found in https://github.com/berenslab/mini-atlas. In particular, we used C57Bl/6 Wild type, Viaat-Cre/Ai9 mice, SOM-Cre/Ai9, VIPCre/Ai9, PV-Cre/Ai9, NPY-Cre/Ai9, Scl17a8-Cre/Ai9, Scl17a8-iCre/Ai9, Vipr2-Cre/Ai9 and Gnb4-Cre/Ai9. Detailed information about the origin of each single Cre line reported here can be find in the main text.

Anatomy: Mus musculus, male and female, 2-month old, wild type C57Bl6, Cre driver transgenics and reporters, some obtained from Jackson Laboratories.

Transgenic line generation and characterization:  Mus musculus, mixed C57BL6 background (transgenes as indicated in text), 4-8 weeks age, mixed sex.

All rodent rooms are on a 12/12 hr light/dark cycle (6am - 6pm), except that Allen Institute rodent rooms are on a 14/10 hr light/dark cycle (6am-8pm). The room temperature range is 68-72°F (20-22°C) and the humidity range is 30-70%.

| | |
|---|---|
| Wild animals | This study did not involve wild animals. |
| Field-collected samples | The study did not involve samples collected from the field. |
| Ethics oversight | All experimental procedures using live animals were performed according to protocols approved by Institutional Animal Care and Use Committees (IACUC) of all participating institutions: Allen Institute for Brain Science, Baylor College of Medicine, Broad Institute of MIT and Harvard, Cold Spring Harbor Laboratory, Harvard University, Salk Institute for Biological Studies, University of California Berkeley, University of California San Diego, University of Southern California. Macaque experiments were performed on animals designated for euthanasia via the Washington National Primate Research Center's Tissue Distribution Program. |

# Human research participants

Policy information about studies involving human research participants

| | |
|---|---|
| Population characteristics | Human transcriptomics:<br>43 y/o Iranian female with PMI 18.5 hours from mitral valve prolapse (SSv4),<br>50 y/o caucasian male with PMI 24.5 hours from cardiovascular event (SSv4),<br>54 y/o caucasian male with PMI 25 hours from cardiovascular event (SSv4),<br>60 y/o unknown female with PMI 18 hours from car accident (SSv4, Cv3, SNARE-seq2, snmC-seq2),<br>and 50 y/o unknown male with PMI 10 hours from cardiovascular event (SSv4, Cv3, SNARE-seq2, snmC-seq2).<br><br>Data type: SMART-Seqv4 (SSv4), 10x Genomics Chromium Single Cell 3' Kit v3 (Cv3), Single-Nucleus Chromatin Accessibility and mRNA Expression sequencing (SNARE-seq2), Single nucleus methyl cytosine sequencing (snmCseq2).<br><br>Human Patch-seq:<br>61 y/o caucasian female undergoing surgical resection for treatment of deep brain tumor. |
| Recruitment | Human transcriptomics:<br>Postmortem adult human brain tissue was collected after obtaining permission from decedent next-of-kin. Postmortem tissue specimens from males and females between 18 – 68 years of age with no known history of neuropsychiatric or neurological conditions ('control' cases) were considered for inclusion in this study of cell transcriptional profiles. Key conditions for exclusion were:<br>• Known brain injury, cancer or disease<br>• Known neuropsychiatric or neuropathological history<br>• Epilepsy or other seizure history<br>• Drug/alcohol dependency<br>• > 1 hour on ventilator<br>• Positive for infectious disease<br>• Prion disease<br>• Chronic renal failure<br>• Death from homicide or suicide<br>• Sleep apnea<br>• Time since death (postmortem interval, PMI) > 25 hours |

Ethics oversight

Human Patch-seq:
Tissue was collected after obtaining informed consent of the patient

Human transcriptomics:
Postmortem adult human brain tissue collection was performed in accordance with the provisions of the United States Uniform Anatomical Gift Act of 2006 described in the California Health and Safety Code section 7150 (effective 1/1/2008) and other applicable state and federal laws and regulations. The Western Institutional Review Board reviewed tissue collection processes and determined that they did not constitute human subjects research requiring institutional review board (IRB) review.

Human Patch-seq:
The patient provided informed consent and experimental procedures were approved by the hospital institute review board before commencing the study.

Note that full information on the approval of the study protocol must also be provided in the manuscript.

