## [Peer Review File · Nature]

Manuscript Title: A multimodal cell census and atlas of the mammalian primary motor cortex

Reviewer Comments & Author Rebuttals.

Reviewer Reports on the Initial Version:

Referee #1 (Remarks to the Author):

Reviewer's Report

This is the flagship paper from the BRAIN Initiative Cell Census Network. As stated at their website this initiative has at its goal: "...a comprehensive understanding of brain cell types [being] essential to understand how neural circuits generate perception and complex behaviors." It represents a "coordinated large-scale analyses [available as open-access] of single-cell transcriptomes, chromatin accessibility, DNA methylomes, spatially resolved single-cell transcriptomes, morphological and electrophysiological properties, and cellular resolution input-output mapping, integrated through cross-modal computational analysis." The brain, being at the pinnacle of the evolutionary tree, is complex in structure, function, regulation and diversity of cell types coordinating behavior. In the introduction the authors provide a roadmap of the findings by this initiative that spans this flagship article along with eleven companion papers. Central to this article is the leverage of modern technologies (e.g. single cell analyses) to realize an atlas of the cellular diversity and function in the brain. Table 1 presents a useful summary of experimental and computational techniques used in the present study and associated datasets. Because complex studies of this kind are littered with an alphabet soup of acronyms, the glossary in table 2 provides for the quick look-up for the general reader. Finally, in the introduction, a bullet-point list of the major findings by the consortium provides a useful summary and figure 1 a schematic of the of experimental and computational approaches of the study overall. In the result section, figure 2 summarizes the motor cortex (MOp) consensus cell type taxonomy. What follows is a detailed exposition of brain mapping and genomic and imaging analyses and the generation of CRISPR/Cas9 edited mice. The authors finally present in figure 10 an overview and integrated atlas of cell types in the primary motor cortex of mouse, marmoset and human. In the discussion, the authors pull all the strands together and provide future directions. The methods used are state-of-the-art, verified by numerous other studies.

I find no serious points of critique. The paper is well-written and relatively easy to follow. The discussion is particularly useful to the general reader.

Three minor points of critique (none are dealbreakers):

1. Although companion papers are available on BioRxiv, more explicit reference to, discussion of, and putting in context the findings of the other papers may help expand the atlas specifics and set the stage of the entire consortium's efforts.
2. Incorporation of brain disease genetic risk variants (for Parkinson's and Alzheimer's Diseases, for example) using the genomic single cell annotations may set the new data in context of brain abnormalities. Risk variants reside largely in DNA non-coding regions containing multiple SNPs with unclear functionality – the present atlas may inform mechanisms for these disease risks, both inter- and intra-cellularly.
3. Mechanisms involved in brain hemispheric asymmetry may be revealed and discussed using the present multidimensional atlas and could expose insight into whole-body disease asymmetry, as

well as normal functions such as handedness, for example.

Referee #2 (Remarks to the Author):

In this paper by the BRAIN initiative cell census network, the consortium gives an overall description of a massive approach to categorize cell types in the motor cortex of different mammalian species using a variety of different approaches. I will refrain from summarizing all techniques applied and datasets acquired since it would provide just a repetition of information easily accessible in the paper several times (abstract, introduction, results, discussion).

This study presents an impressive amount of data that will no doubt be extremely valuable for data mining and as a resource for many researchers interested in cell types, cortex and neuroscience. Furthermore, it points out possible entry points for future functional studies on the role of cortex and cortical cell types in behavior and computation. A main goal of the paper in my view is not to delineate exactly how many cell types there are in the mammalian motor cortex, but to provide insight into how multi-omics information can help to consolidate cell type classifications and their stability/plasticity across species. In other words, which are the most important defining features for cell types and which others are confirmatory or contradicting the dimension of one type of classification. These questions are particularly important if one wants to argue that this work is in the long term useful not only to understand the biology of motor cortex, but also to have access to human cortical cell types with the idea to intervene in disorders where information on promoters and cell types will no doubt be of key importance, but experimental access is very limited for obvious reasons. To that point, the repetitive description of the number of cell types is difficult to follow since the cell type numbers differ significantly across different sections of the paper, which makes it confusing because one begins to wonder what these numbers really mean.

Given the massive amount of different approaches, groups conducting the work and strategies to determine cell type diversity, this paper aims to be a top level paper prompting readers to other studies, each describing a facet of the big project in more details. The value of this paper beyond pointing people towards these other studies is to present and synthesize a coherent picture with the main messages conveyed by the other studies that people can then read to understand the experimental and computational basis for the claims. This study should be able to synthesize all main findings of these combined studies with Figures well capturing these findings and messages, so that readers who are less interested in the details can extract the main take home messages of this large enterprise.

Through parts of this paper, the authors do a very good job in providing these top level insights, but there is room for improvement at several places. It would be very useful if the authors could still work towards making these main insights and the arguments for how they reached them more accessible to readers only interested in getting an overview picture of the findings. This should be at the expense of some detailed information that might be less interesting for the general reader. I will comment on suggestions sequentially according to Figures shown.

(1) While the 11 companion papers are cited throughout the paper, I suggest to include a separate listing of References to go with this paper on these 11 studies. There is currently Table 1 on the applied methods citing background and companion papers but since they are not cited in sequence, one always has to go back to the reference list and extract them. Another option is to put these 11 references in sequence when this concept is first mentioned in the introduction, so they would appear in one block and have consecutive citation numbers. Similarly, if there was a way of displaying Ref numbers and which panels refer to which Figures in this paper in Figure 1, it might help the reader to follow the flow.

(2) Data presented in Figure 2 and associated text aims to determine how many cell types there

are in mouse motor cortex, and then compare these cell types to human and marmoset cell types. The accompanying text should make this point at the start and perhaps at the expense of some of the details on mouse profiling give insight into the criteria for concluding which and how many cell types are conserved and why. The authors just very shortly say that over- or underclustering may be issues but not how this is resolved. It is important that readers can follow how cell types are defined across species, much more so than the details of how many nuclei were profiled in mouse. The same goes for marker definition of main cell types and across species – what are the markers and how conserved are they? These are the main points that should come across in this section.

(3) Using MERFISH, the cell type number presented in Figure 3 and associated text is quite different from the one presented before. It is unclear how the genes that were studied here have been chosen. One would have thought that in a logical flow, the genes to study should have arisen from the data presented in Figure 2, but this does not seem to be the case. The authors at the very least should comment on the overlap of genes used in the two approaches. The fact that the cell type number differs needs an explanation. Both methods are based on transcripts, and ideally should result in matching data and cell type classification.

(4) Figure 4 and associated text presents a heroic effort to patch many cortical cells in different species, subsequently determine their morphology and assess the transcriptome. It would be useful if instead of showing single example cell ephys data in panel d, the authors provided quantitative differences between what they call a cell type based on physiology and whether they in addition take into account transcriptome data. In other words, what does morphology and/or ephys traits add to the transcriptome classification we already know? Does morphology and ephys lead to more diversity, or does it go hand-in-hand with the transcriptome data? The authors should synthesize this new data with previously presented data.

(5) Figure 5 and associated text is focused on analyzing the methylome of motor cortex neurons, while at the same time taking into account neuronal projections. From the design, it is unclear why the authors would not have acquired the methylome of all motor cortex neurons so they could directly compare this to the transcriptome data, but here introduce the additional dimension of neuronal projections. Nevertheless, for this section in particular, making links to the other sections would be very useful and is currently not well done. Again, the key question here is: what is the added value of having the methylome over the transcriptome data, do we need this information to define cell types? Is the projection information useful in addition? These points should be made clearer at the expense of some of the unnecessary details presented here.

(6) Figure 6 and associated text take a look at input and output of Mop-upper limb, as well as single cell morphology. The authors also present a number of Cre driver lines. The authors should try to link this information to the one of filling single neurons earlier in the study if possible, and they should present a coherent quantitative synthesis of target structure or input structure overlap between the different neuron types presented. While no link to the transcriptome and methylome data can be made here, at least information on layer identity and cell morphology should be commented upon.

(7) Figure 7 and associated text describe Cre and FLP driver lines for different cell types, and introduces the dimension of development into the paper. The tools generated will no doubt be extremely useful for the community, yet one misses how it links to the rest. In particular, what do we learn from the developmental dimension, i.e. how does it link to the transcriptional cell types defined in the adult in earlier Figures of this paper? Comments on how the genes were picked to generate new mouse lines and linking them to the rest of the paper would be useful (this is only shown for two mouse lines in panel A).

(8) The last figures show examples of integrative analysis across the different information channels, yet one lacks information on how starting from one source of information and adding more data contributes to the certainty of assigning cell types. The case studies presented read a

bit like a self-fulfilling prophecy when one would rather like to know how important which contribution is to the cell types profile, e.g. are we 80% certain with the transcriptome what cell type it is or do we need the single cell morphology and ephys data because we are only 30% sure after the transcriptome? If some sort of synthetic analysis like this would be presented at the overall level (e.g. for these X cell types, we are 80% confident with just the transcriptome, morphology adds another 10% etc, for these Y cell types...present in form of table should be possible), this would be extremely valuable, and would interest the general reader much more than individual case studies which might be more interesting for a specialized reader (in addition, the case studies are rather confirmatory than providing novel insight). The same comment holds for the certainty of cell type identification across species, in particular humans, which will be key for possible medical applications.

Referee #3 (Remarks to the Author):

Comments on "A multimodal cell census and atlas of the mammalian primary motor cortex"

The huge range of dataset going into the companion papers are obviously impressive and will be of great use to the community at large. These are collected by experts in the field and the experiments are done to high standards to include the data.

Reviewing this paper is a challenge since (a) the range of techniques in the companion papers is so broad as to make it tough to be an expert in all of them, (b) some of the methods in the companion papers are novel and it is uncertain whether they should be evaluated here or by reviewers for the specific papers, and (c) the concept of a flagship paper and what it should or should not include is new to me and I am not sure what to evaluate. Is it a review? Is it primary literature? I come away feeling the main point is for the reader to leave the paper understanding the basis for the Mouse Consensus Transcriptomic Taxonomy and its relationship to the primate cell types. It is my guess that, once published, those companion papers will be the standard citation for the applicable individual datasets. My internally generated expectation, then, is that this paper must:

- present a basis for the hierarchy of cell types in these three species ("Justify Figure 10");
- summarize those results in a manner that makes them accessible to a broad audience; and
- emphasize the new findings from such a large body of work

So, does it work? I think it mostly does. Though I provide a long list of minor comments, these are intended to demonstrate my respect for and interest in the work instead of as obstacles to publication. I think I understand the basis for the hierarchy. I am less convinced that the results were summarized and accessible. The paper feels at times like each companion paper got a figure to show its highlights and a section in the results, but without making the results section feel like it came from a single author (probably not unexpected with so many contributors). I think this would be improved if the explanations were briefer and provided at a higher level (this assumes all the companion papers get published). Lastly, I feel like the emphasis on new findings was limited because some of the points called out felt like they were already known. So, what is the great insight that we get? I think it's the hierarchy.

Overall, I would recommend publication or publication with minor revision.

Major point:

Lines 45+: Would you be willing to venture a definition of "cell types"? The presentation of the data seems influenced in part by biological systematics. In that field, each of the terminal taxa are generally species. Maybe there's not 100% consensus about what defines a species, but Ernst Mayr was effective in popularizing a biological definition (populations that can reproduce with one another AND are reproductively isolated from others, for example). Even if not everyone agrees, can we advance a definition so we have something to argue about? (I mean something like: 'All the cells of a given cell type must be located in a similar standardized anatomical location/cortical

area/layer in the brain of that species, share inputs and deliver outputs to the same targets, have similar intrinsic excitability, and share similar transcriptional and epigenetic state.' And then, 'similar' means ...) Or is it simply that cell types are 'cells that group together at the lowest possible hierarchical level in the integrated transcriptomic classification' with the other features (morphology, connectivity, etc) only being found in cells whose transcriptional state matches? Defining this could address whether cells in different cortical areas are the same type and whether similar cells in different layers are the same type. (Are layer 2/3 and layer 5 fast spiking PV+ interneurons the same cell type? Seems like yes in Figure 10 but no by my definition above. Are medulla-projecting layer 5 pyramids in MOp, MOs, or SSp the same cell type?) Perhaps I am rude for asking, as those paragraphs address the issue but don't come to a snappy conclusion.

(For a contrasting definition, from your reference 105: "Here, we define a cell type as 'a set of cells in an organism that change in evolution together, partially independent of other cells, and are evolutionarily more closely related to each other than to other cells'. That is, cell types are evolutionary units with the potential for independent evolutionary change." I find this definition not as helpful for neuroscience.)

Minor points (sorry for the length):

Abstract, Line 15: What does "congruently" mean here? The authors's collective vocabulary is better than mine, but I would take it to mean a perfect superposition of one thing onto the other (which would be unlikely to occur in transcriptomic data).

Introduction, Line 69: The number of citations in the intro are useful for context, but likely could be shorter for publication purposes (examples not needed from all areas where this technique was effectively used ... cortex, hippocampus, hypothalamus, etc.).

The first major finding is that "Combined single-cell transcriptomic and epigenomic analysis reveals a unified molecular genetic landscape of adult cortical cell types that integrates gene expression, chromatin state and DNA methylation maps." I am wondering what this bullet means?

Transcriptomic data give some idea of which genes are being expressed (and how highly expressed) at a given time – so I understand this will help clarify what needs to be expressed differentially to define cell types. But epigenomics is modifications of the DNA that can be expected to affect gene expression, so it is not surprising that these would describe a similar landscape. Similarly, chromatin state is expected to vary with gene expression, 'opening' up the chromatin to permit expression. DNA methylation might act, for example, to suppress gene expression. Perhaps there is some deeper meaning here, but this bullet seems redundant to me.

More subtly, it will be interesting to understand how "continuously varying properties" (Line 161) in transcriptomics relate to physiology and morphology. Or how to distinguish whether there are correlated or causal. Although I think this alludes to a gradient within a given cortical area, such as from pia to white matter within a given layer (as discussed in Berg et al., 2020 bioRxiv for some L2/3 cell types), I would also be curious if anything is known about the tangential gradient as we move across topographical areas in MOp – from limb to face to trunk. It would seem there might be some gradient needed to help maintain topography in connections to homotopic areas within M1 (well at least in primate) or other motor cortical areas or long-range targets.

What is the molecular specialization of corticomotoneuronal neurons? These are famous in primates and possibly missing in rodents (but see Grinevich, Brecht, and Osten J Neurosci 2005). Do these occur in the data?

Similarly, "Cell type transcriptional and epigenetic signatures can guide the generation of an extensive genetic toolkit for targeting glutamatergic pyramidal neuron types and fate mapping their progenitor types." (Line 171) I agree, but the idea that this is possible is not a new finding (probably more examples, but Visel et al., 2013 Cell among others). Similarly, Introduction, Line

87: Is there earlier literature (than the bioRxiv references 33-39) from other groups that have used enhancer-based approaches to target for genetic access to specific cell types in mice? I am not an expert in these approaches but I think I have seen seminars exploiting enhancers used in published mouse models since at least a couple years ago. This bullet in my opinion belongs with the publication describing the models in which this is described.

"Comparative analysis of mouse, marmoset and human transcriptomic types achieves a unified cross-species taxonomy of cortical cell types with their hierarchical organization that reflects developmental origins; transcriptional similarity of cell type granularity across species varies as a function of evolutionary distance." (Line 151+) It would be useful to know if there are exceptions to this rule for evolutionary distance. The example given is "Three non-neuronal types had greater [S]pearman correlations of overall gene expression (Fig. 2d, right columns) between marmoset and mouse likely because non-neuronal cells were undersampled in human M1 resulting in fewer rare types." (Lines 239-242) but the explanation included suggests these exceptions weren't evaluated much. This is not unexpected, but when the finding is one of the major bullets and contradictory data isn't assessed in great detail (because it's believed to be wrong), then the finding is not so surprising.

How should we interpret that many marker genes for glutamatergic subtypes are species specific? In that context, it is a bit inconsistent with the overall pattern of IT, ET, CT, and NC types that do seem to be conserved. Does this mean that no specific marker is needed to make such types, but a suite of them that is robust to substitutions?

Introduction, Line 109-111: The style of present day articles seems to include presaging the results in the end of the introduction, but here you cite the first figure and table! Only an opinion for style purposes, but to me the results begin here. "Here we present the cell census and atlas of cell types in one region of the mammalian brain, the primary motor cortex (MOp or M1) of mouse, marmoset and human, through an analysis with unprecedented scope, depth and range of approaches (Fig. 1, Table 1)."

Table 1 is useful as a reference, but takes up a lot of space (in particular since some cells are large in only a subset of rows). I think it is more useful for the reader to refer back to while reading the results of the paper and not (as currently set) right after the introduction when the results are yet to be detailed.

How Table 2 is organized might be improved? If truly a Glossary, terms could be listed alphabetically. Furthermore, the use of multiple columns is unclear (e.g. not just a term and its definition). If columns are meant to be interpreted as hierarchical, why MOp and its layers would be in the same column. The hierarchy seems present to some extent in the inhibitory and excitatory subclasses.

Table 2: Explanation of the developmental significance of MGE and CGE given in a table defining terminology seems out of place.

Table 2: "The top branches of the CN transcriptomic cell type hierarchy comprising neuronal and non-neuronal cells." Does 'theCN' mean 'the BICCN'?

Table 2: Use of terms ET for extra-telencephalic and CT for corticothalamic. I imagine with such a large consortium, the field is well served to converge on a single nomenclature and perhaps this will be the one, but as the thalamus is a subset of the extratelecephalic structures cortex might target, this naming scheme has a small error. CT could be a special subset of ET, though these read as though intended these to be thought of as two separate groups. Similarly the IT definition might be refined to state "only", and in "Excitatory glutamatergic neuron that projects only to other telencephalic structures" (since some ET neurons might send a collateral). There's a lengthy description of why ET is chosen to replace PT in the supplement, which is useful,

though its placement in the supplement is a problem (since it won't be as well read).

Results, Line 217-223: "These types were grouped into broader subclasses based on shared developmental origin for GABAergic inhibitory neurons [i.e., three caudal ganglionic eminence (CGE)-derived subclasses (Lamp5, Sncg and Vip) and two medial ganglionic eminence (MGE)-derived subclasses (Sst and Pvalb)], layer and projection pattern in mouse for glutamatergic excitatory neurons [i.e., intratelencephalic (IT), extratelencephalic (ET), corticothalamic (CT), near-projecting (NP) and layer 6b (L6b)], and non-neuronal functional subclass (e.g., oligodendrocytes and astrocytes) (Table 2)." This table lists all five of those types of GABAergic neurons in a single line, while having separate lines for IT, ET, CT, etc. Thus, it is difficult to understand the levels assigned to each of these in the hierarchy.

How many types are there? For mouse, there are 90 neuronal t-types, but also listed as 56 neuronal cell types (line 205). Maybe it's not clear to other groups outside your own what this difference is. Also potentially confusing is that this lists numbers for neuronal cell types, but then the consensus number includes 8 non-neuronal cell types (line 217). For mouse, 90 t-types, (but claimed 56 neuronal cell types) (line 205). For marmoset, 94 t-types (line 210). For human, 127 t-types (line 210). Conserved t-types are 45, include 24 GABAergic, 13 glutamatergic, and 8 non-neuronal. For comparison, MERFISH has 42 GABAergic, 39 glutamatergic, and 14 non-neuronal (Line 310). So: could you help me to align these numbers with the taxonomy of Figure 10? (Perhaps counted wrong, I got 88 there.) Separately, I understand how these could differ between methodology, but under one assessment there are almost 2x as many GABAergic types as glutamatergic (t-types), while in MERFISH, there are roughly the same number of GABAergic/glutamatergic types? Why would the number of glutamatergic types seemingly vary this much? Further, when the electrophysiological types are recorded in MOp, these are determined to include 77 t-types (Line 382). Is this greater than the 56 neurons or less than the 90 mouse t-types? There could be many reasons why these numbers don't match but it is somewhat confusing to me since the numbers always vary.

The GABAergic types are grouped by marker genes (Lamp5, Sncg, Vip, Sst, Sst Chodl, Pvalb, and Meis2). I am curious if there are any differences uncovered by this method by interneurons with similar marker genes (such as PV) and the layer in which the soma resides (a L2/3 PV+ interneuron versus a L5 PV+ interneuron). The excitatory cells are grouped by layer in the Table 2 but the interneurons are not. Although the differences might not be as extreme (as IT versus ET or CT), there must be some difference (as they seem to target somata of excitatory cells only in the same layer). (Lines 217-223). For example, in Fig. 4 e-f, it's suggested that there is some continuous variation in the electrical/morphological axis, but it was not as clear to me whether this is accompanied by some gradient in the transcriptional data. They may fall into the same cluster. But within this cluster, then, do the upper, middle, and lower layers cells also separate into branches within this cluster?

Spearman is sometimes in caps (Fig 2), sometimes not (Line 240).

For glutamatergic cell types, "the majority of markers were species-specific (Fig. 2f,g)" (Line 249). What does that mean? "The evolutionary divergence of marker gene expression may reflect species adaptations or relaxed constraints on genes that can be substituted with others for related cellular functions." What are the species-specific genes, then? Is it just a matter of different species using different isoforms of the same genes, or something entirely different. The discussion of this (Lines 1147+) touches on what some explanation might be, but doesn't provide an answer. Is it not a little unexpected that the set of marker genes for these types (illustrated in Fig. 2f Venn diagrams) generally shows little overlap across the three species? I understand that L2/3 IT is clustered based on a lot of data but is it not somewhat remarkable that there aren't more shared genes needed to create that conserved cell type? Does the data mean that, with more species-specific markers, that there are multiple ways to achieve the same general morphology/connectivity? Or that lower levels of other genes are what is needed to achieve this

(such that they don't meet the bar to be marker genes)?

Figure 2: Inconsistent use of the blue color for human?

Figure 2: The recent Berg et al bioRxiv suggested 5 clusters for human L2/3 IT but here there seem to be 6? Or is this a primary motor versus medial temporal difference? (But in that paper, comparing human MTC to rodent VISp is justified by stating there aren't great differences across areas).

Figure 2: What does it mean that L2/3 IT and L6 IT cluster more closely than other glutamatergic cells? These neurons are born at different times in development, so it seems that the clustering is not strictly recapitulating development here (as opposed to the GABAergic cells).

Fig 3: Is there an obvious medial/lateral border at the AGm/AGl interface? This lateral/medial agranular border has been used by others to mark a cytoarchitectural border of MOs and MOp (also called AGm and AGl, or separating whisker M1 and somatic M1 in some publications; Brecht et al., 2004 J Comp Neurol for a rat example).

For Fig 3a, it would be useful to either use some indication of pia/white matter or dorsal/ventral to orient the reader. These panels look at though they are rotated compared to the earlier panels in the same figure but it isn't clear to me what rotating the layer structure is offering nor which direction is medial/lateral within M1. Medial/lateral might help interpret the difference in location of the SSp versus MOs projecting cells, though there would also be some use in understanding how the complete the retrograde labeling is (since it looks like you could argue from this image that SSp and MOs projecting L2/3 cells don't spatially overlap ... though I don't know if that has been proposed in the literature).

Minor point, but the use of color – which is spectacular – will be less accessible to ~8% of male readers (colorblindness). I don't have ideal suggestions about how to handle this, but it strikes me in aligning the colors of the neurons with the clusters (you may very well have picked colors that are easy to separate, but I don't have a viewer for pdf that toggles between RGB view and some simulation of deuteranopia). (Fig 2 and 3 for example; in Fig 4 there are symbols besides the neurons to match to clusters).

For Figures 2b and 3b, is there any kind of certainty that can be ascribed to the nodes in the tree/dendrogram? Although this sort of categorization is maybe newer to classifying neuron types, there are some tests that evolutionary biology-type researchers apply to evaluate the certainty of the branches in a tree (resample the data by bootstrap 10,000 times and ask how often a certain node emerges, for example; maybe other tests exist). This is only of minor importance, but it is not clear from the data alone if such a large sample of expression data provides categorization that is so clean that noise in the expression data is no issue (which we hope is the case ... but maybe you who have examined the data know better), or if we are accepting the categorization as-is because we have no means to test if other groupings work too.

Fig 4a, any idea as to why some of the points sampled by physiology seem so sparse compared to the density of t-type data ("z" and "-" have a lot of black points; "r" and "x" and "y" have less)? Maybe this just means layer 4 and 6 are boring places to record.

Fig 4b,c, the electrophysiological recordings aren't mentioned in the legend in great detail. Do the overlaid traces represent three current steps? Hyperpolarizing, near threshold, and maximum firing rate (I think this is specified in other papers from the same group but not here.)

Fig 4h, the small part with so many labels gets hard to follow. Also the island of L5/6 NP seems to be split into two groups (but the cut is so clean it looks like an editing error instead of the data).

Line 392-394, "excitatory t-types from the IT subclasses with more similar transcriptomes were located also at adjacent cortical depths, suggesting that distances in t-space co-varied with distances in the me-space, even within a layer". I believe this is true. Fig 3b makes a case for the laminar claim (with the interesting exceptions of L6 IT grouping with L2/3 to 5 IT and L5 ET with L6 CT). Fig 2b has some agreement but puts L6 IT in the middle of the other IT-types (and not all in the same location), while L5 ET is grouped with IT away from L6 CT. This topographic arrangement suggests that maybe the organization of these groups is not as clear. What do the differences between these hierarchies mean? Since MERFISH and t-types seem to me that they are both examining gene expression, it is not clear that this is a methodological difference. But is it noise, and are these the only nodes that the hierarchical clustering has less certainty assigning?

Similarly, the consensus hierarchy groups all IT cells (including L5 and L6) together, with L5 ET as the nearest outgroup. This immediately proposes the developmental hypothesis that either (a) In order for L6 IT to be transcriptionally similar to the other L2/3, L4/5, L5 IT type cells, it should be born from progenitor neurons at a similar time (perhaps immediately preceding L5 IT generation?), while the ET neurons (the nearest outgroup in this hierarchy) or CT neurons (even further out) would be born at a different time, or that (b) Hierarchy based on expression is not so strongly reflective of developmental origin and cells can differentiate into terminal neurons with similar expression states despite being born at different times. (a) would be interesting to know (since it would revise the cortical development rule of pyramidal neurons being born 'inside-out'). There is some exploration of this in the companion paper Matho et al, is this known? In trying to ascribe some meaning to the hierarchy, this I think is one potentially interesting question.

The cortical layer inset in Fig 10 gives the average or median laminar position of interneurons and pyramidal cells. It is useful in showing how the pyramidal neurons are considered with respect to layer. For interneurons, it is not as clear the use of having a median/average layer position. This emphasizes to me that the interneurons don't subdivide as well into layers (perhaps that there's a good deal of similarity for each cell type across whatever layer it falls into). I'm still curious if there is some kind of layer differences within each interneuron type. But perhaps this is beyond the scope.

Line 392-394, "excitatory t-types from the IT subclasses with more similar transcriptomes were located also at adjacent cortical depths, suggesting that distances in t-space co-varied with distances in the me-space, even within a layer". Fig 4g is not as convincing to me that this is true since the electrophysiological parameters that might be similar at neighboring depths aren't presented in a manner that shows adjacent populations with similar characteristics. The ordering of the current injection traces according to color code aren't presented in the same order as the pyramids beneath, for example, and the parameter to extract from the traces to look for co-variation is not pointed out.

Line 397 "At the level of single t-types, we found that some t-types showed layer-adapting morphologies across layers (Fig. 4e,f)" Ok. But doesn't this then suggest that if the same t-type results in variable morphology (Fig 4f) that the threshold for clustering those t-types together is not ideal and there is some transcriptomic difference between a Sst+ neuron in L6 and L2/3 that might explain whether the dendrites are broad or restricted in the same layer?

For the Betz cell comparison, I appreciate this data and find it interesting. Because of the enormous soma size of Betz cells, and the absence of neurons fitting this in mice, I am curious how the mouse cells to compare were chosen? There are (at least) two subtypes of ET-type L5 neurons in mice (Economo et al, 2018), which seem to project to either thalamus or medulla. I am curious which have more similar properties to the Betz cells (or if it is known which mouse ET type you use for comparison). Since these are somewhat unique, but also seem to fall into similar

transcriptomic cluster as non-Betz layer 5 cells, are they similar in e-m properties corresponding layer ET cells? This might be a more fair comparison than mouse – for example, do other layer 5 ET types cells in the primate also have high expression of the Kv channels of Fig. 4L?

Line 488 “Although neurons projecting to different target regions were not completely separated on t-SNE” Does this mean that expression pattern alone is insufficient to tell if a neuron is IT/ET or IT projecting to a given area? Is there a way to distinguish between the retrograde approach “reflecting the high quality of retrograde-labeling of neuronal nuclei” instead of just inadvertently including unlabeled nuclei in the analysis?

Fig 5, one weakness of not simultaneously reviewing the Epi-Retro-Seq paper based on the new method is that the figures are more challenging. In Fig 5a, there is a box “P5” in the NeuN/GFP sorting. Is this the retrogradely labeled sample for analysis?

Fig 5c and Fig 5d. I find the results of Fig 5d clear and mostly easy to understand. All the ET projecting cells fall into a single category, two if you count the CT and ET categories in thalamus. Does this reject the Economo et al (2018) result of two separate ET-type populations? In contrast, the data presented in 5c are not as easy to grasp (perhaps because of similar colors), and it leaves the impression there is some signal from every target in every cluster! (Which the more quantitative 5d seems to reject).

I’m not clear on the less well-defined clusters of IT neurons based on projection patterns. Zhang et al (2020 – bioRxiv, your reference 79) reads as though distinguishing neurons into different groups based on their targets from epigenomic data should be feasible (“Based on their epigenomes, intra-telencephalic (IT) cells projecting to different cortical targets could be further distinguished, and some layer 5 neurons projecting to extra-telencephalic targets (L5-ET) formed separate subclusters that aligned with their axonal projections.”). But the overlap of ACA and S1 projecting data in 5b-d makes it look as though this is not a settled question.

One of the great things about the collection of this data is that it has been done rigorously, with defined inclusion/exclusion criteria. Because most of the paper refers to mouse motor cortex as MOp, it gives the impression that most of the mouse data is collected in a stereotaxic area that is uniform across the mice. When reaching the section about “MOp projection neuron types and input-output wiring diagram”, the language shifts to specify MOp-ul. Maybe this is just to be more specific or to define a subset of MOp examined in detail for this companion paper. Is there a reason to suspect these results wouldn’t generalize to MOp in general?

One problem with Fig 6 is that a lot of the experiments have already been done, though perhaps not all Cre lines by the same lab in the same publication. But the overall findings are all expected – generally it felt as though the results summarized in Lines 573-605 were known to those working in this field. (“For example, the L6 Ntsr1 line revealed a typical CT projection pattern with dense projections specific to thalamic nuclei. This result is consistent with labeling from retrograde injections in various thalamic nuclei (PO, VAL, VM) and cortical areas such as MOs and SSp (Fig. 6b, top).” And “Confirming and extending previous reports 86, we characterized detailed axonal trajectories and terminations of two major types of L5b ET cells, namely medulla-projecting and non-medulla projecting neurons; both types may collateralize in the thalamus and terminate in the midbrain (Fig. 6d). Individual IT cells across L2-L6 also generate richly diverse axonal trajectories”). This raised the question of what this adds when done in the context of the other results in the paper. I felt like there wasn’t a new insight in this figure. That said, the figure is a nice visual summary.

It is somewhat awkward to review a paper which includes “Stafford, Daigle, Chance et al., companion manuscript in preparation (Line 664). Some of the excitement of that paper is could be the future access to cell type specific mouse line. Certainly there will be interest in a L5 PT-medulla line. But also challenging as a reviewer when the full story is presented elsewhere. Part of the

desire to be able to give this work a more thorough review falls from wanting to place Npnt and Slco2a1 in context with the Matho et al paper (also describing strategies to get at pyramidal neurons), but doesn't have (if I recall) a PT-medulla only line. This results in a situation where half of Fig 7 is summarizing Stafford et al, while the other in about Matho et al and the list of resources for L5 ET, for example (Fig 7e), isn't even merged to include Slco2, Npnt, etc.

Although it's presented as a new finding here, it's been published that there are L4-like cells in primary motor cortex of primates (Area 4): García-Cabezas and Barbas, *Eur J Neurosci* (2014). (Line 812: "L4-like neurons may also exist in human and marmoset M1.") And, as cited here, shown by Yamawaki et al, *eLife* (2014). Thus, it would be better to focus on how this is extending what's previously known. At least part of it is the molecular characterization (Fig 8b+). I would be interested to see how the MERFISH in Fig 8c, for example, looks if it were done on a slice extending from (more lateral) SSp to MOp to MOs (more medial), as I hypothesize – if this distribution looks like RORb, we would see cells in clearly defined layers (SSp) become less complete layers (as L4 cytoarchitecturally goes away in MOp) and then the L4 cells become less dense (in MOs).

This is beyond the scope of the present paper, but I am curious how the L4-like cells in MOp compare to those in SSp and SSs? There's a recent paper suggesting SSs L4 cells make projections outside their home column (Minamisawa et al, *Cell Reports* 2018). Made me wonder which pair of L4 cells is most similar (if SSp and MOp were similar with SSs being the most transcriptionally unique).

Similarly, the next outcome highlighted (Lines 839-841) "Here as the second case study, through integrated multimodal characterization we demonstrate that L5 ET neurons in MOp can also be divided into MY-projecting and non-MY-projecting types" is also published (Economo et al., 2018) and it appears that the one class described there can be even further subdivided into two groups (Winnubst et al, 2019 – ref 26). The molecular characterization identifies one marker as Slco2a1 for medulla and Hpgd and Npsr1 for thalamus-projecting types (though these are also the markers highlighted by Economo et al Fig 1.). Is the advance that this distinction applies to MOp in addition to frontal areas?

Fig 9a is not explained sufficiently for me to understand it. What's the arrow at the side? What does the color intensity mean in the plots (there's no scale/legend, perhaps correlation coefficient)? The legend claims there is good correspondence between the groups, but then the left plot makes it look as though two of the RNA-data based classes are merged in a single MERFISH group, while the last RNA-based grouping includes two of the MERFISH ones, with ET type 4's affinity not as clear.

Fig 9c MERFISH image legend refers to Mouse ET_types 1-4 and 5, but then only shows 1-3 and 5.

Fig 9b compares mouse ET types 1-4 to three human and marmoset types. I am interested if this means that primates have less diversity in some ways in their layer 5 ET cell types? After reading Winnubst (2019) and their description of different subsets of thalamic projecting L5 neurons, I began to wonder if human L5 corticothalamic projection neurons might show a greater diversity, since the human thalamus is bigger and it might thus be possible to differentiate corticothalamic neurons projecting of many different subdivisions (of, for example, motor thalamus – where there is a much more fine division of subregions than in mice).

Fig 10. Lovely figure. The tree at the top has some of the connections labeled with different thicknesses of lines (and different colors). What is the significance of this? I am inferring it might be something about the certainty of the node or perhaps described in a different location.

Lines 1068+ : "This classification provides strong evidence for the existence of hitherto poorly studied but molecularly distinct subclasses such as the near-projecting (NP) pyramidal neurons,

and many more novel cell types.” To me, this is pretty exciting but it didn’t make the cut of highlighted bullets. (While more obvious things ... t-type and chromatin state give similar maps ... did.)

Line 1162: “The molecular genetic framework of cell type organization established by the current study will provide a robust cellular metric system for cross-species translation of knowledge and insight that bridges levels of organization based on their inherent biological and evolutionary relationships.” Is this a strong argument? I would be concerned that some of the lack of translation to disease would be the feel among human and primate researchers that disorders of frontal areas aren’t well modeled in part because the rodent lacks the expansion of frontal cortex. Similarly, MOp research may be limited because the primary motor cortex of primates has five specialized premotor areas with which it is strongly connected, while rodent MOp has strong reciprocal connections with SSp (qualitatively differing from primates). Thus, while some similarities in intrinsic features might occur, it is difficult to know how this will be useful for studying connections or pathologies that come about with cells embedded in a network (that differs from mouse network). This statement is stronger if disorders were cell-autonomous.

Referee #4 (Remarks to the Author):

This is a remarkable resource of data accompanied with an integrative analysis of neuronal diversity in motor cortex across three different species. The study provides a detailed blueprint for molecular and functional organization of this cortical region and the data will serve as a valuable resource that will be re-analyzed and interpreted by many labs. While it is not expected that the authors perform exhaustive analysis and interpretation of the data, some critical analysis and discussion seem to be lacking from the manuscript.

Below I detail what in my view would make the manuscript more informative and useful to the community. Most importantly, I would like to see more clear discussion of how different datasets correlate, which datasets are orthogonal to each other, and what analysis is most informative. This would serve as an important guidance for follow-up studies by groups that do not have financial resources for comparably exhaustive cell profiling. In addition, comparisons across species is rather limited and the study does not offer a clear picture whether brain evolution is associated with quantitative changes in cell type distribution or emergence of new cell types.

Major points to address:

1. The authors should examine and discuss more in depth orthogonality of individual datasets. Differential connectivity and physiology were historically considered as the key functional determinants of neuronal cell identity. Yet, the molecular data do not clearly resolve such functionally distinct neuronal types (described as continuous variation). Does this mean that the two types of neuronal classification are to large extent orthogonal to each other and therefore molecular classification will always need to be accompanied with anatomical and physiological characterization of neuronal cell types? Are spatial location, DNA methylation, projection, morphology, physiology strongly correlated with gene expression or do they provide additional information? Further, how much of the gene expression variability within clusters can be explained by these different parameters? This can be examined, if not for all cells, for IT cells where integration across different modalities has already been done.
2. The authors should discuss which type of data are most informative when categorizing neuronal cell types. For example how do the RNA expression, accessibility, and methylation profiles compare in terms of their ability to stratify data into meaningful cell types?
3. In this context, the authors should clarify how they define class, subclass, and type of neurons? It would be helpful to propose and implement a unified neuronal cell type nomenclature that could

be applied to additional datasets coming from different labs.

4. The authors propose that variable projections of neurons belonging to the same molecular type could result from activity based pruning during development. Alternative possibility is that axonal pathfinding is controlled by subtype specific genetic programs active during early developmental stages when neurons are establishing their connectivity, but these programs become extinguished in the mature CNS. These fundamentally distinct mechanisms should be discussed as they would have important implications for the interpretation of the adult molecular data.

5. The authors should provide more direct integrative analysis of data across different species. Can the data be shown as degree of overlap between species-integrated clusters and each species-specific cluster? Considering that homologous cell types exhibit great degree of molecular divergence, it would be interesting to see if canonical correlation analysis or manifold alignment can be applied to compare data across different species. Such direct comparison could help to identify truly species specific cell types. It is of interest that number of Parvalbumin cell types seem to be expanded in primates relative to mouse, even with fewer profiled cells. Also there are some human/marmoset clusters that have low overlap with any given mouse cluster; does this suggest a human or primate-specific cell type? Can you get more fine-grained resolution if you re-do this analysis with subsets of data restricted to a specific lineage?

Minor points:

1. Figure 7A: what are the different dot colors?

2. Figure 8A: It would be interesting to provide similar type of representation for all cells, not only IT neurons

3. Figure 8C: Include a color scale

4. Figure 8E: It looks like L4 and L4/5 cells are distinct. Is it fair to conclude that these are the same cell type? Is Rspo expressed to the same level in both subsets?

5. Figure 10: Color scale is missing

6. Was spatial location considered in MERFISH clustering?

Referee #5 (Remarks to the Author):

This manuscript by the BRAIN Initiative Cell Census Network represents a monumental collective undertaking by participating labs to create a multimodal taxonomy of the mammalian motor cortex. The results not only advance our understanding of the organization of the cortex but also provide a framework to integrate information from transcriptomic, epigenomic, physiological and anatomical data modalities to create a unified cell type hierarchy. By extending their analysis across different species they also show the evolutionary conservation and variation of these neuronal cell types. Furthermore, the inclusion of open chromatin and DNA methylation information reveal new future possibilities for studying the establishment and maintenance of neuronal cell types in relation to gene regulatory mechanisms. For these reasons I am confident that the manuscript and its related resources will be of major interest to many readers across diverse fields in neuroscience and biology.

While the authors did a commendable job taking the abundance of data from these different manuscripts and turning it into a coherent whole, the direct connection between the transgenic tools, the input-output connectivity map and the described consensus cell types was somewhat

lacking. In other parts the data could be presented in a different way to better support the main conclusions.

Major Points

The novel genetic driver lines (Figure 6-7) are not directly aligned to the consensus types that are described in the rest of the paper. This makes it hard to interpret how well these tools give us access to the different cell types of the taxonomy put forth in the paper. The authors do acknowledge in the manuscript that this is a major future goal (line 611) but it could be discussed more thoroughly. Furthermore, the authors claim they identified 25 distinct neuron projection types based on unique combinations of axonal targets and laminar distribution. However, this classification is not brought up again or used to classify the coverage of the driver lines. The conclusion that the authors 'provide a comprehensive identification of major projection neuron types with correspondence to molecular markers' (line 607) therefore seems somewhat overstated.

The final section on MY, and non-MY-projecting L5-ET types is supposed to be a case study to 'demonstrate the convergence of properties onto specific cell types' (line 751). However, as the authors note, these observations on distinct molecular, morphological, and laminar distribution have been previously reported in ALM (Economo et al. 2018, Tasic et al. 2018) and have also been partially confirmed in MOp (Winnubst et al. 2019). What new insights have been gained from aligning the data from these different modalities?

What is the degree of accuracy of the Epi-Retro-Seq approach? In other words, what proportion of cells are classified into t-types that do not match their projection pattern. Looking at figure 5c-d we can see for instance 'near projecting' and IT cells with extratelencephalic projections. It also seems that there is a cell type specific bias. L6CT cells are overrepresented in the unbiased sample (figure 5d) but are only marginally enriched in the TH-projecting sample. What does this mean for interpretation of some of the other proportional data shown in figure 5d?

An interesting question that arises from the results presented by the authors is to what extent IT neurons should be viewed as a continuum vs. discrete cell types. The authors note in several instances the relationship between expression, morphology and functional profiles and the cortical depth of IT neurons but don't show any direct quantification or plots for this correlation (similar to figure 5c in Munoz-Castaneda et al. 2020). This seems like an important conceptual advance and it would be informative to see an aggregated analysis across modalities where depth information can be obtained.

To what extent do GABAergic consensus types have more specific marker expression compared to other types (Line 255)? The data presentation between 2G and 2H is so different that it's hard to draw this conclusion.

How do the results in Figure 3e match with the statement in line 340: "each IT cluster projected to multiple regions, with each region receiving input from a different composition of IT clusters". Why are there no double labeled cells? Could the authors provide a breakdown of these numbers to support this important observation? Finally, the data is not shown in a very clear way as it's hard to align the cell types with the projection type and there are no cortical layer indicators.

The authors claim that putative L4 neurons mostly project upward into layer 2/3, while L2 and L3 neurons mostly project downward to layer 5 (line 799). However, looking at the examples shown in figure 8g it seems that this distinction isn't that clear cut. The authors should provide an analysis of normalized axon length per layer to back up this claim.

Minor Points

In general, there are several cases where there are no colorbars for the heatmaps and it's not always clearly indicated what they mean or how they are normalized in the figure legend.

Table 2. The text "The top branches of theCN transcriptomic" should be "The top branches of the transcriptomic". For clarity spell out "OPC cells (OPC)" as "Oligodendrocyte progenitor cell (OPC)".

Figure 1a. Would be informative to see a cross section for all three species with MOp/M1 clearly highlighted.

Figure 3c. Very confusing color legend. Can the same color indicate different cell types? Why not color by cell subclass?

Figure 4h. Very difficult to see distribution of individual cells. Can we see an inset with a zoomed view of the L5 ET cluster?. Also a bit confusing that the colors for the species are so similar to the t-types groups.

Figure 8e. Add a color legend in the figure for MOp:red, VISp: blue.

Ext Fig. 2b. What are the colors on the left? Why are these not M1-14?

Line 194. For readability it would be useful to mention data modality of each technique.

Line 247-248. This seems like something for the methods section.

Line 602. Missing closing parenthesis.

Line 704. cp is not shown in the figure.

Line 861. "with cluster 0 predominantly projecting to MY while other clusters having variable" → "have"

Referee #6 (Remarks to the Author):

The authors present highly rigorous and expansive multimodal molecular characterization of the primary motor cortex in mouse, marmoset and human. The paper is extremely rich in information, extremely dense in content, and brings together nearly every type of genomic information currently accessible for brain in an integrated fashion. Every single figure brings together yet another dimension of integration, and yet another insightful conclusion about what these integrations can teach us about the human brain. As such, I strongly recommend acceptance of the manuscript, as both the dataset and the integrative analyses detailed will undoubtedly serve as an invaluable source of information for the neuroscience and genomics communities.

The study is not without flaws, utilizing non-state-of-the-art analytical tools and improperly executed analyses, although these are minor in the grand scope of the paper and should be easy to correct. I strongly recommend acceptance pending a few corrections.

Analytical Concerns

Transcription Type Classification: In most of the computational analysis across methods the data is heavily over resolved, with "transcription type" classifications being derived from clustering at arbitrary resolutions rather than any known molecular signatures. For example, the authors claim

to identify 24 distinct subtypes of cortical GABAergic neurons, a cell class that is known to contain 5 major distinct populations, only to immediately follow that all 24 can be classified into one of the known 5. The t-types result from the specific clustering paradigm used and most would not be reproducible by a different algorithm. Further, as granularity increases, the signature of any t-type is more likely to be driven by batch-specific and technical variance rather than true biological differences across subpopulations and would be difficult to reproduce in other datasets. Finally, such arbitrary classification is not particularly valuable to the reader who identify or target a particular population in their dataset or experiment, and they acknowledge this as a source of error when attempting to map types across species. It would be better for the authors to compare across known or sufficiently distinct subpopulations. For example, instead of the arbitrary Pvalb_1 and _2, or Astro_1 and Astro_2 perhaps distinguish between protoplasmic/interlaminar/fibrous/etc. and perhaps sub-partition in a non-algorithm-specific fashion if there is enough heterogeneity in the population to justify it.

Differential Expression Analysis: The authors use outdated and inappropriate methods for calling DEGs across cell populations and species. These methods treat each cell as a unique observation rather than each sample and thus are both heavily overpowered and susceptible to batch variance. While this may not be too much of an issue for isogenic mouse lines, the analysis does not consider sex, age or RNA quality of the primate samples, especially human, all of which are major drivers of gene expression differences. The authors should repeat their analysis using an alternative approach that permits complex design. I recommend the authors consider pseudo bulk methods (perhaps within t-types) that can give more accurate estimations of power as opposed to the nonsensical p-values produced by t-test and Wilcoxon when applied to single cell data, as this confounds the comparison of DEG divergence across species, though there may be other ways to correct for this as well.

Small Sample Size: Although it was difficult to parse together which data originated from which sample across the various companion papers, I am under the impression that for each modality the human and marmoset data originated from 2-3 samples each for several of these experiments. If this truly is the case, this is an acceptable sample size, regardless of the number of cells recovered from each tissue sample. The extraordinarily variable nature of primate single-cell data, coupled with the failure to incorporate critical covariates mentioned above would jeopardize replicability, which is already one of the biggest concerns in the field of single-cell genomics.

Presentation concerns

The manuscript felt a bit redundant at times, with many paragraphs devoted to introducing yet another data type, and re-listing all the previous data types it was going to be integrated with, and then summarizing once more all the data types that it was just incorporated with. Even though these integrations are indeed very impressive, I would recommend foregoing some of these redundant sentences and paragraphs to leave more space for the actual insights, and for more biologically meaningful conclusions, rather than re-hashing and re-stating the impact of what was just done.

Figure 10 struck me as more well-suited for extended data.

Several of the figures could be further condensed, pushing some of the information to extended data to streamline the presentation in the main text.

Author Rebuttals to Initial Comments:

I. Overall response

We thank the six referees for their extensive comments on this manuscript, the flagship paper of the BICCN publication package. The comments have been extremely helpful in our effort to improve the manuscript. All referees positively recognized the value of this “massive” and “monumental” collaborative effort in a deep-dive investigation of cell types in the mammalian primary motor cortex using by far the most comprehensive set of state-of-the-art approaches, providing to the community not only an unprecedentedly rich set of resources but also a conceptual framework to guide future efforts in understanding brain cell types and their functions. At the same time, the referees also provided numerous constructive critiques and suggestions, emphasizing many of the critical issues around defining cell types and understanding the relationships between different modalities of cellular properties. In the revised manuscript, we have attempted to address all the points raised by the referees. Here we provide a brief summary of our responses and the major changes we have made to the manuscript:

For this response letter, our detailed point-by-point responses to the referees’ comments are shown in two sections below. In the first section, given the large number of comments the referees gave and many key points repeatedly raised by different referees, we have reorganized all the comments into three groups - general major, general minor and figure specific, with similar comments arranged close to each other so that we can address them collectively. In the second section, our responses are presented in the traditional manner - under each referee’s comments in their original order. Referees can choose to read our responses in either or both sections.

For the manuscript itself, major changes include the following:

1. We simplified the description of individual companion studies associated with each figure, and tried to better synthesize all the major findings together and deliver a clearer message in Discussion.
2. In the revised Discussion, we have now addressed several key conceptual issues raised by the referees, including how to define cell types, the hierarchical organization of cell types in the motor cortex (and deemphasizing specific numbers of types), the co-existence of discrete and continuous variations, the relationship among different modalities and how they contribute to the description of cell types, and the developmental and evolutionary underpinnings of the observed patterns and rules of cell types.
3. We substantially revised the final summarizing figure, Figure 10, to highlight the above key points.
4. To better integrate the newly generated genetic tools into the characterization and understanding of cell types, we revised and reorganized the original Figure 7 which contained two separate approaches for generating cell type targeting tools. We integrated the first part about generating L5 ET transgenic lines into Figure 9 (multimodal

characterization of L5 ET cell types). We substantially revised the remaining part of original Figure 7 which focuses on a set of excitatory neuron targeting lines based on developmental timing, and put it in front of original Figure 6, to allow a better narrative because some of the lines generated were used to map connectivity.

5. We refined our analyses on the robustness of our cell type taxonomies and on differential marker gene expression across species. We refer referees to our companion papers (especially Yao et al 2020 on integrated mouse molecular taxonomy) for more in-depth validation analysis of our cell type taxonomies.
6. We revised Table 2 to make it simpler with a better description of different categories of cell types.

With these changes and clarifications, we hope the referees agree that we have revised our manuscript satisfactorily for publication in Nature. We would be happy to address any further comments they may have.

II. Critiques

A. General - Major

1. *Referee 2*: A main goal of the paper in my view is not to delineate exactly how many cell types there are in the mammalian motor cortex, but to provide insight into how multi-omics information can help to consolidate cell type classifications and their stability/plasticity across species. In other words, which are the most important defining features for cell types and which others are confirmatory or contradicting the dimension of one type of classification. These questions are particularly important if one wants to argue that this work is in the long term useful not only to understand the biology of motor cortex, but also to have access to human cortical cell types with the idea to intervene in disorders where information on promoters and cell types will no doubt be of key importance, but experimental access is very limited for obvious reasons. To that point, the repetitive description of the number of cell types is difficult to follow since the cell type numbers differ significantly across different sections of the paper, which makes it confusing because one begins to wonder what these numbers really mean.

We thank the referee for this excellent point and fully agree with it. We have now deemphasized the specific number of clusters from each data type and added text in Discussion to emphasize the overall structure of cell types which is in a hierarchical organization. In this hierarchy, the higher branches (cell classes and subclasses) are the most robustly distinct from each other, the most conserved across species, and have the most

consistent properties across all modalities examined. The numbers of classes and subclasses are also invariable. At the lower branch or terminal leaf level (clusters), the distinction among clusters can be less clear and more continuous, and thus the exact numbers of clusters are less meaningful. The cross-modality correspondence at the leaf node level can also be weaker and differential variations among different modalities can be observed.

As we are at the early stage of defining cell types, we do not intend to use a single modality of features to define cell types and treat the others as only providing supporting evidence. Rather, we strive to characterize cells in as many different ways as we can, and use cross-modality consistency to define consensus cell types. Through this process, we will also begin to learn which modality is more predictive of the consensus cell types. On the other hand, practically at the current stage, it is likely that data types/modalities with substantially richer feature contents (e.g. transcriptomics) will be better suited for defining and predicting cell types than other data types containing fewer features due to technical limitations (e.g. electrophysiology). Another approach is to define cell types based on molecular genetic programs as they are underlying the origin of cell types. In this respect, an important issue to address is whether cell types defined by molecular genetic programs are also functionally specific (in other words, whether they are consistent with functionally defined cell types).

2. *Referee 2*: The value of this paper beyond pointing people towards these other studies is to present and synthesize a coherent picture with the main messages conveyed by the other studies that people can then read to understand the experimental and computational basis for the claims. This study should be able to synthesize all main findings of these combined studies with Figures well capturing these findings and messages, so that readers who are less interested in the details can extract the main take home messages of this large enterprise. It would be very useful if the authors could still work towards making these main insights and the arguments for how they reached them more accessible to readers only interested in getting an overview picture of the findings. This should be at the expense of some detailed information that might be less interesting for the general reader.

We agree with the referee on this point. Indeed, our focus of this flagship paper was to synthesize the main findings from the other studies described in the core companion papers together. We did this by starting with a succinct list of key insights we obtained at the end of Introduction, then highlighting the major findings in each modality of cell type characterization through a series of figures in the Results section, and finally providing a synthesis of what we have learned about cell types through this unprecedentedly comprehensive effort in Discussion. In the revised manuscript we have now further simplified the detailed descriptions and discussed a number of key points raised by the referees that are indeed central to the understanding of brain cell types in Discussion.

3. *Referee 2*: The last figures show examples of integrative analysis across the different information channels, yet one lacks information on how starting from one source of information and adding more data contributes to the certainty of assigning cell types. The case studies presented read a bit like a self-fulfilling prophecy when one would rather like to know how important which contribution is to the cell types profile, e.g. are we 80% certain with the transcriptome what cell type it is or do we need the single cell morphology and ephys data because we are only 30% sure after the transcriptome? If some sort of synthetic analysis like this would be presented at the overall level (e.g. for these X cell types, we are 80% confident with just the transcriptome, morphology adds another 10% etc, for these Y cell types...present in form of table should be possible), this would be extremely valuable, and would interest the general reader much more than individual case studies which might be more interesting for a specialized reader (in addition, the case studies are rather confirmatory than providing novel insight). The same comment holds for the certainty of cell type identification across species, in particular humans, which will be key for possible medical applications.

By conducting multimodal and cross-species studies, we can now say that we are near 100% certain about the top branches of the transcriptomic taxonomy - the classes (e.g. glutamatergic and GABAergic) and subclasses (within GABAergic: Lamp5, Sncg, Vip, Sst-Chodl, Sst, Pvalb; within glutamatergic: L2/3 IT, L4/5 IT, L5-6 IT, L6 Car3, L5 ET, L5/6 NP, L6 CT and L6b). Below that, several major types can also be accurately predicted (near 100%) by transcriptomics and confirmed by morphology and projection (e.g. Pvalb-Vipr2 chandelier cells, L5 ET_1 medulla-projecting cells), while in other cases the distinction between neighboring transcriptomic clusters is less clear. Additional evidence may break the continuity, e.g. L4/5 IT_1 type turns out to be L4 cells, Sst Hpse type represents those Sst cells with L4-targeting axons. But more often additional evidence also suggests that groups of neighboring clusters exhibit similar morphological, electrical and projectional properties with continuous variations (Fig 10), based on our Patch-seq studies (companion paper Scala et al bioRxiv 2020; Gouwens et al Cell 2020), retrograde tracing studies (companion papers Zhang M et al bioRxiv 2020, Zhang Z et al bioRxiv 2020) and full morphology studies (companion paper Peng et al bioRxiv 2020). It should be noted that our phenotypic characterization is not exhaustive, future studies may further resolve the difference among clusters. Thus, our overall conclusion is that for cell subclasses and some highly distinct cell types, transcriptomics-based definition is highly accurate and predictive of other properties, but at more fine-grained levels one needs multimodal information to define cell types with proper granularity.

4. *Referee 3*: The huge range of dataset going into the companion papers are obviously impressive and will be of great use to the community at large. These are collected by experts in the field and the experiments are done to high standards to include the data. Reviewing this paper is a challenge since (a) the range of techniques in the companion papers is so broad as to make it tough to be an expert in all of them, (b) some of the methods in the companion papers are novel and it is uncertain whether they should be evaluated here or by

reviewers for the specific papers, and (c) the concept of a flagship paper and what it should or should not include is new to me and I am not sure what to evaluate. Is it a review? Is it primary literature? I come away feeling the main point is for the reader to leave the paper understanding the basis for the Mouse Consensus Transcriptomic Taxonomy and its relationship to the primate cell types. It is my guess that, once published, those companion papers will be the standard citation for the applicable individual datasets. My internally generated expectation, then, is that this paper must:

- present a basis for the hierarchy of cell types in these three species (“Justify Figure 10”);
- summarize those results in a manner that makes them accessible to a broad audience;
- emphasize the new findings from such a large body of work

We appreciate the referee’s assessment of the relationship between this flagship paper and the other companion papers. We have now organized our Discussion section according to these suggestions.

5. *Referee 3:* I am less convinced that the results were summarized and accessible. The paper feels at times like each companion paper got a figure to show its highlights and a section in the results, but without making the results section feel like it came from a single author (probably not unexpected with so many contributors). I think this would be improved if the explanations were briefer and provided at a higher level (this assumes all the companion papers get published). Lastly, I feel like the emphasis on new findings was limited because some of the points called out felt like they were already known. So, what is the great insight that we get? I think it’s the hierarchy.

We believe that in the flagship paper we do need to introduce and summarize the different types of studies from the companion papers and the major conclusions from them, in order to build up the cases for our overall perspectives on cell types which cover a wide range of issues. We would like to point out that each figure is a high-level summary of the major findings from a particular study (which by itself is already a large-scale one). Nonetheless, we have now tried to simplify the descriptions of these individual studies as much as we can.

We agree with the referee that the great insight is indeed the hierarchy of the cell type organization, and how different modalities of cell types properties are related to each other at the different levels of this hierarchy. We have discussed this point in detail in our responses to referees’ comments above, so will not repeat here. We have now also elaborated this point in the Discussion section of our revised manuscript.

6. *Referee 3*: Lines 45+: Would you be willing to venture a definition of “cell types”? The presentation of the data seems influenced in part by biological systematics. In that field, each of the terminal taxa are generally species. Maybe there’s not 100% consensus about what defines a species, but Ernst Mayr was effective in popularizing a biological definition (populations that can reproduce with one another AND are reproductively isolated from others, for example). Even if not everyone agrees, can we advance a definition so we have something to argue about? (I mean something like: ‘All the cells of a given cell type must be located in a similar standardized anatomical location/cortical area/layer in the brain of that species, share inputs and deliver outputs to the same targets, have similar intrinsic excitability, and share similar transcriptional and epigenetic state.’ And then, ‘similar’ means ...) Or is it simply that cell types are ‘cells that group together at the lowest possible hierarchical level in the integrated transcriptomic classification’ with the other features (morphology, connectivity, etc) only being found in cells whose transcriptional state matches? Defining this could address whether cells in different cortical areas are the same type and whether similar cells in different layers are the same type. (Are layer 2/3 and layer 5 fast-spiking PV+ interneurons the same cell type? Seems like yes in Figure 10 but no by my definition above. Are medulla-projecting layer 5 pyramids in MOp, MOs, or SSp the same cell type?) Perhaps I am rude for asking, as those paragraphs address the issue but don’t come to a snappy conclusion. (For a contrasting definition, from your reference 105: “Here, we define a cell type as ‘a set of cells in an organism that change in evolution together, partially independent of other cells, and are evolutionarily more closely related to each other than to other cells’. That is, cell types are evolutionary units with the potential for independent evolutionary change.” I find this definition not as helpful for neuroscience.)

We believe cell types should be defined by taking into account all these factors: evolutionary/developmental origin, correspondence across modalities, and anatomical location (region, layer, etc.), but place relative emphasis in such order. These factors interact with each other and their relationships will be reflected in the hierarchical organization of cell types.

We propose that cell types of the cerebral cortex are best explained by a combination of specification through evolutionarily-driven and developmentally-regulated genetic mechanisms, and refinement of cellular identities through intercellular interactions within the networks the cells are embedded in. The genetic mechanisms drive intrinsic or cell-autonomous determination of cell fate, and progressive temporal generation of cell types from common progenitor pools that explain global similarities and continuous features of cellular phenotypes that reflect developmental gradients. Network influences can drive further phenotypic refinement that is not reflected in the adult genetic signature, for example for axonal projections and synaptic connectivity that may reflect transient or stochastic developmental events, region-specific and/or activity-dependent refinement. Such an evolutionary, developmental and experience-dependent framework largely explains the overall hierarchical organization, strong conservation across species, and strong

correlation among cellular phenotypes to a certain level with further network-driven or experience-dependent refinement.

7. *Referee 3*: How many types are there? For mouse, there are 90 neuronal t-types, but also listed as 56 neuronal cell types (line 205). Maybe it's not clear to other groups outside your own what this difference is. Also potentially confusing is that this lists numbers for neuronal cell types, but then the consensus number includes 8 non-neuronal cell types (line 217). For mouse, 90 t-types, (but claimed 56 neuronal cell types) (line 205). For marmoset, 94 t-types (line 210). For human, 127 t-types (line 210). Conserved t-types are 45, include 24 GABAergic, 13 glutamatergic, and 8 non-neuronal. For comparison, MERFISH has 42 GABAergic, 39 glutamatergic, and 14 non-neuronal (Line 310). So: could you help me to align these numbers with the taxonomy of Figure 10? (Perhaps counted wrong, I got 88 there.) Separately, I understand how these could differ between methodology, but under one assessment there are almost 2x as many GABAergic types as glutamatergic (t-types), while in MERFISH, there are roughly the same number of GABAergic/glutamatergic types? Why would the number of glutamatergic types seemingly vary this much? Further, when the electrophysiological types are recorded in MOp, these are determined to include 77 t-types (Line 382). Is this greater than the 56 neurons or less than the 90 mouse t-types? There could be many reasons why these numbers don't match but it is somewhat confusing to me since the numbers always vary.

We recognize that all those different numbers of "types" or clusters are indeed very confusing! This is why in our manuscript we presented various correspondence/confusion matrices to show the alignment of clusters across different data types (see for example the original Fig 10, which is now Extended Data Figure 2). These matrices show various degrees of one-to-one, one-to-several, or several-to-several correspondences between clusters of different modalities. They can serve as references for those readers who are interested in detailed comparison of specific cell types.

The numbers of clusters do vary across data types and the exact numbers do not matter much. This is because in the hierarchical tree of cell types, the distinction between subclasses and major types is highly discrete whereas the distinction between leaf-node clusters can be fuzzy and continuous. Intrinsic and/or methodological differences among different data types can lead to varying numbers of clusters derived from a relatively continuous landscape of cellular phenotypes. The above mentioned correspondence matrices help to align clusters across data types. Co-existence of discreteness and continuity is a key feature in cell type diversity, and it makes it difficult to define an exact number of cell types that can be consistent across modalities. In the revised manuscript we have now emphasized this point.

8. *Referee 3*: More subtly, it will be interesting to understand how “continuously varying properties” (Line 161) in transcriptomics relate to physiology and morphology. Or how to distinguish whether there are correlated or causal. Although I think this alludes to a gradient within a given cortical area, such as from pia to white matter within a given layer (as discussed in Berg et al., 2020 bioRxiv for some L2/3 cell types), I would also be curious if anything is known about the tangential gradient as we move across topographical areas in MOp – from limb to face to trunk. It would seem there might be some gradient needed to help maintain topography in connections to homotopic areas within M1 (well at least in primate) or other motor cortical areas or long-range targets.

Indeed, the continuous variation in transcriptomics does correlate with variation in morpho-electrical properties (Scala et al bioRxiv 2020) and spatial distribution across layers by MERFISH (Zhang M et al bioRxiv 2020). We also observe more extensive variation in long-range axon projection patterns within transcriptomic clusters (Peng et al bioRxiv 2020). Our current studies do not distinguish whether these are correlated or causal.

We have examined tangential variation along anterior-posterior and medial-lateral axes in MOp by MERFISH, and found spatial distribution variations along these axes for some glutamatergic clusters as described in more detail in the companion paper (Zhang M et al bioRxiv 2020).

Another example of continuity versus discreteness is seen in the human cortex, where the expansion of the supragranular layers, consisting of layer 2 and 3 IT excitatory neurons, is also seen as an extension of the IT gradient described in mouse cortex. In the Berg et al bioRxiv 2020 companion paper we describe that there is a general depth-dependence to the IT neurons as a whole along that transcriptomic axis with corresponding changes in neuronal size related to the corresponding length of the apical dendrite. However, we find that there are discriminable cell types that are phenotypically distinct at the ends of this global transcriptomic gradient. Only one of the types shows a very strong within-type gradient, and we show that this transcriptomic gradient is accompanied by similar gradients in anatomical and physiological features. Thus there appears to be a higher-level organization within the transcriptomic space for IT types, with elements of both continuity and separable discreteness that differ for different types.

9. *Referee 3*: Line 392-394, “excitatory t-types from the IT subclasses with more similar transcriptomes were located also at adjacent cortical depths, suggesting that distances in t-space co-varied with distances in the me-space, even within a layer”. I believe this is true. Fig 3b makes a case for the laminar claim (with the interesting exceptions of L6 IT grouping with L2/3 to 5 IT and L5 ET with L6 CT). Fig 2b has some agreement but puts L6 IT in the middle of the other IT-types (and not all in the same location), while L5 ET is grouped with IT away

from L6 CT. This topographic arrangement suggests that maybe the organization of these groups is not as clear. What do the differences between these hierarchies mean? Since MERFISH and t-types seem to me that they are both examining gene expression, it is not clear that this is a methodological difference. But is it noise, and are these the only nodes that the hierarchical clustering has less certainty assigning?

We should emphasize that since the hierarchical tree is one-dimensional, it does not capture the full extent of the complex multidimensional relationships among cell types. Although the branch length (vertical) does reflect the distance between clusters, the order of the branches may not, as they can be flipped around. Furthermore, the hierarchical clustering and calculation of branch lengths are based on differentially expressed (DE) genes between every pair of clusters. The set of DE genes can be different in different data types (e.g. mouse RNA-seq, mouse MERFISH, human RNA-seq), which can lead to somewhat different branching patterns in different hierarchical trees.

We use both hierarchical trees (one-dimensional) and UMAPs (two-dimensional) to illustrate the relationships among cell types. In terms of the distinction among IT, L5 ET and L6 CT, they are highly distinct among each other, but there is a tendency that L5 ET is closer to all IT subclasses, whereas L6 CT forms a more distinct branch along with L6b and L5/6 NP. Within the IT subclasses, individual clusters do vary along the cortical depth across layers, however, we also observe an interesting closeness (with gene signatures) between L6 IT and L2/3 IT types that brings them together (also can be seen in the bent shape of the IT cell collection in the UMAP of Fig 8a, as well as the bent shape of the UMAP of IT cells in Figure 4a of the MERFISH companion paper (Zhang M et al bioRxiv 2020)).

10. *Referee 3*: Similarly, the consensus hierarchy groups all IT cells (including L5 and L6) together, with L5 ET as the nearest outgroup. This immediately proposes the developmental hypothesis that either (a) In order for L6 IT to be transcriptionally similar to the other L2/3, L4/5, L5 IT type cells, it should be born from progenitor neurons at a similar time (perhaps immediately preceding L5 IT generation?), while the ET neurons (the nearest outgroup in this hierarchy) or CT neurons (even further out) would be born at a different time, or that (b) Hierarchy based on expression is not so strongly reflective of developmental origin and cells can differentiate into terminal neurons with similar expression states despite being born at different times. (a) would be interesting to know (since it would revise the cortical development rule of pyramidal neurons being born 'inside-out'). There is some exploration of this in the companion paper Matho et al, is this known? In trying to ascribe some meaning to the hierarchy, this I think is one potentially interesting question.

The referee raised another important issue - the relationship between transcriptomic types/clusters and their developmental origin/basis. We think this issue should be considered more broadly under the framework of the *developmental trajectory* of

transcriptomic types, i.e. how transcriptomic cell types and hierarchy relate to, or are derived from, genetic programs and developmental origins that include progenitor types, lineage progression, and cell birth order/time. We would like to clarify that birth order (i.e. cell division along lineage progression) is more important than birth date, and birth order is only meaningful if cells are generated from the same progenitor. However, this is a fundamental issue that is currently unsolved in the field.

Therefore, the issue raised by Ref 3 here cannot be resolved for at least two reasons. First, the diversity of radial glia progenitor types, whether they are multipotent or fate restricted, remains unresolved. Second, lineage relationships among IT, ET, and CT types are yet to be clarified.

It is quite reasonable to hypothesize that similar transcriptomic types would have similar gene regulatory programs, which may be shaped by common progenitor origin and lineage; sequential birth time (i.e. birth order) could then give rise to related transcriptomic types deployed to different cortical layers in an inside-out sequence. In this scheme, L6, L5, L2/3 IT types are born sequentially and deployed inside-out. We (new Fig 6, and companion paper Matho et al bioRxiv 2020) do provide some support for this scheme. An important issue is how ET and CT types relate to IT and the overall scheme. This is a fundamental issue to be resolved and is beyond the scope of the current paper. We do want to point out that the genetic tools reported in Fig 6 and Matho 2020 are well suited to address this issue by fate mapping the trajectories of cell types from progenitor types (defined by transcription factor expression) to their progeny types defined by multi-modal parameters including transcriptomes.

11. *Referee 3*: The GABAergic types are grouped by marker genes (Lamp5, Sncg, Vip, Sst, Sst Chodl, Pvalb, and Meis2). I am curious if there are any differences uncovered by this method by interneurons with similar marker genes (such as PV) and the layer in which the soma resides (a L2/3 PV+ interneuron versus a L5 PV+ interneuron). The excitatory cells are grouped by layer in the Table 2 but the interneurons are not. Although the differences might not be as extreme (as IT versus ET or CT), there must be some difference (as they seem to target somata of excitatory cells only in the same layer). (Lines 217-223). For example, in Fig. 4 e-f, it's suggested that there is some continuous variation in the electrical/morphological axis, but it was not as clear to me whether this is accompanied by some gradient in the transcriptional data. They may fall into the same cluster. But within this cluster, then, do the upper, middle, and lower layers cells also separate into branches within this cluster?

We have indeed observed that some GABAergic interneuron clusters are enriched in certain layers while others are distributed across multiple layers, as shown in much greater detail in our companion MERFISH (Fig 2 in Zhang M et al 2020) and Patch-seq (Fig 1 in Scala et al 2020) studies, as well as a separate Patch-seq study from the mouse visual cortex (Figures 2, 4 and 6 in Gouwens et al Cell 2020). The layer-enriched distribution is particularly

pronounced in Sst and Pvalb subclasses. There is also a corresponding gradual transition across layers between transcriptomic and morphoelectrical axes, for example, Pvalb+ basket cells in upper, middle and lower layers are indeed segregated into different transcriptomic clusters (Scala et al 2020, Gouwens et al Cell 2020).

12. *Referee 3*: The cortical layer inset in Fig 10 gives the average or median laminar position of interneurons and pyramidal cells. It is useful in showing how the pyramidal neurons are considered with respect to layer. For interneurons, it is not as clear the use of having a median/average layer position. This emphasizes to me that the interneurons don't subdivide as well into layers (perhaps that there's a good deal of similarity for each cell type across whatever layer it falls into). I'm still curious if there is some kind of layer differences within each interneuron type. But perhaps this is beyond the scope.

We have now moved the original Fig 10, including the MERFISH panel and its cortical layer inset, to Extended Data Fig 2, because it does not represent the full picture as the referee pointed out here. Instead, we provide a brief description of layer distribution of interneuron and excitatory neurons in the text and reference our MERFISH and Patch-seq companion papers for more detailed discussions (see our response immediately above this one). The MERFISH data showed that many of the GABAergic clusters showed laminar distributions and preferentially reside within one or two cortical layers (Zhang M et al 2020).

13. *Referee 3*: The first major finding is that "Combined single-cell transcriptomic and epigenomic analysis reveals a unified molecular genetic landscape of adult cortical cell types that integrates gene expression, chromatin state and DNA methylation maps." I am wondering what this bullet means? Transcriptomic data give some idea of which genes are being expressed (and how highly expressed) at a given time – so I understand this will help clarify what needs to be expressed differentially to define cell types. But epigenomics is modifications of the DNA that can be expected to affect gene expression, so it is not surprising that these would describe a similar landscape. Similarly, chromatin state is expected to vary with gene expression, 'opening' up the chromatin to permit expression. DNA methylation might act, for example, to suppress gene expression. Perhaps there is some deeper meaning here, but this bullet seems redundant to me.

The reviewer is correct in thinking that, in principle, transcriptome and epigenome are expected to be concordant. However, this point cannot be taken for granted ("not surprising") for technical and conceptual reasons. From the technical perspective, the transcriptome, chromatin, and DNA methylome datasets are generated by different groups using different technical platforms. There are multiple reasons that the outcome may not be consistent in defining cell types. The fact that they do validate our collective methods and approach, and suggest strong underlying biological mechanisms. Therefore, this outcome does not justify the notion that using multi-modality information is redundant.

From the conceptual perspective, the epigenome provides two additional information sources to the transcriptome. First, the epigenome contains developmental signatures that can persist long after the expression has ended. One specific example at the *Lhx9* locus is given in the companion paper (Yao et al bioRxiv 2020). This developmental gene expression is undetectable in the transcriptome but shows cell-type specificity in the epigenomic profiling. Also, the methylation companion paper (Liu et al bioRxiv 2020) shows that putative developmental signatures of past gene regulatory events account for 65% of all DMRs, with only 35% of DMR being associated with open chromatin and gene expression, in agreement with previously described using bulk samples of purified neural populations (Mo 2015).

Second, the whole genome-wide epigenomic profiling contains regulatory elements in the intronic and intergenic regions, which is considered during clustering. Our companion paper (Yao et al bioRxiv 2020) provides a comprehensive overall result to help understand to what extent the regulatory genome agrees with transcriptome when determining cell types, which is not a foregone conclusion. This kind of integrated analysis is technically challenging and important to help understand the underlying regulatory mechanisms, including differential peaks, potential regulators (TF/motifs), and gene-enhancer links found in specific cell types.

Therefore, although the transcriptome provides the most direct measurement on gene expression at the given time, including chromatin state and DNA methylation maps is still necessary to define a unified molecular genetic landscape of brain cell types. Thus this bullet point represent a major conclusion of the paper.

14. *Referee 3*: “Cell type transcriptional and epigenetic signatures can guide the generation of an extensive genetic toolkit for targeting glutamatergic pyramidal neuron types and fate mapping their progenitor types.” (Line 171) I agree, but the idea that this is possible is not a new finding (probably more examples, but Visel et al., 2013 Cell among others). Similarly, Introduction, Line 87: Is there earlier literature (than the bioRxiv references 33-39) from other groups that have used enhancer-based approaches to target for genetic access to specific cell types in mice? I am not an expert in these approaches but I think I have seen seminars exploiting enhancers used in published mouse models since at least a couple years ago. This bullet in my opinion belongs with the publication describing the models in which this is described.

The idea of using genetic elements to target cell types has been around for several decades. However, how to achieve the level of specificity and comprehensiveness that are necessary for dissection cortical circuits has been an enduring challenge. Currently, there are only a handful of mostly BAC transgenic lines available for targeting several pyramidal neuron populations, and even fewer driver lines for neural progenitors. Here, we generated over a dozen temporally inducible mouse Cre and Flp knock-in driver lines to enable combinatorial

targeting of major progenitor types and projection classes. Intersectional converter lines confer viral access to specific subsets defined by developmental origin, marker expression, anatomical location and projection targets. These strategies establish an experimental framework for understanding the hierarchical organization and developmental trajectory of PyN subpopulations. We have now substantially revised the text and figure (new Fig 6) to highlight these advances, which are reported in-depth in a revised companion paper (Matho et al bioRxiv 2020).

Regarding the use of enhancers for targeting cell types, again the idea has been around for decades but the progress has been very limited, especially for cell types in the mouse brain. Visel et al., 2013 used putative enhancer regions to label different embryonic progenitor regions, not cell types or progenitor types. These transgenic lines are only assayed in embryonic tissues and are not maintained. One of the key challenges is to systematically discover bona fide neuron type enhancers in the adult brain. Indeed, there has been recent progress in developing enhancer-based AAV vectors to target adult brain cell types, including work from researchers such as Gord Fishell, Michael Greenberg and colleagues, as well as some of us within the BICCN (the cited bioRxiv papers are now published). We have now cited this body of work more comprehensively in Introduction.

The key point we are trying to make here is that the current joint transcriptomic and epigenomic analysis provides an unprecedented opportunity to achieve comprehensive access to a wide range of brain cell types, building on top of previous pioneering work. A major effort is now underway in the BICCN to systematically test and validate a larger number of these enhancer elements. The intended specificity and scale of this effort have not been attempted before.

15. *Referee 4*: I would like to see more clear discussion of how different datasets correlate, which datasets are orthogonal to each other, and what analysis is most informative. This would serve as an important guidance for follow-up studies by groups that do not have financial resources for comparably exhaustive cell profiling. In addition, comparisons across species is rather limited and the study does not offer a clear picture whether brain evolution is associated with quantitative changes in cell type distribution or emergence of new cell types.

We have expanded the discussion of these aspects. Please see our detailed responses above to the comments from referees 2 and 3 regarding how different datasets/modalities correlate and how to define cell types in a hierarchical organization.

Regarding the species comparisons, we have begun to see some of the principles of brain evolution but clearly many more species and brain regions are needed to get a clear and coherent picture. Nevertheless, between this manuscript and the two companion manuscripts (Bakken et al. (2020) bioRxiv) and Berg et al. (2020) bioRxiv), we see an overall strong conservation but with quantitative shifts in gene expression, cell type proportions and spatial distributions. While it is not possible to align every cell type at the finest leaf level, the fact that there are ~100 cell types in mouse, marmoset and human suggests similar diversity that may even be possible to align at a finer level with additional species with intermediate evolutionary distance. However, we do observe a number of things. First, the proportions of homologous cell types can vary between species (and cortical regions). Second, gene expression is most similar between closer evolutionary relatives for any given cell type, suggesting divergence as a function of evolutionary distance. Third, homologous cell types can have divergent features, as we show for the Betz cells. Finally, there can be specializations and expansions of specific cell subclasses across species, as shown for the expanded diversity of supragranular excitatory neuron types in Berg et al. (2020) bioRxiv that correlates with the evolutionary expansion of these layers. Together these would seem to lay the groundwork for future comparative work to expand upon these ideas, but likely have begun to reveal a number of principles and ways in which species specialization can manifest. Furthermore, the evolution-associated changes in gene expression include genes associated with neuronal connectivity and function, suggesting that these changes have important functional or circuit wiring implications that will hopefully become interpretable and testable in the future. We have tried to discuss these findings more clearly.

16. *Referee 4:* The authors should examine and discuss more in depth orthogonality of individual datasets. Differential connectivity and physiology were historically considered as the key functional determinants of neuronal cell identity. Yet, the molecular data do not clearly resolve such functionally distinct neuronal types (described as continuous variation). Does this mean that the two types of neuronal classification are to large extent orthogonal to each other and therefore molecular classification will always need to be accompanied with anatomical and physiological characterization of neuronal cell types? Are spatial location, DNA methylation, projection, morphology, physiology strongly correlated with gene expression or do they provide additional information? Further, how much of the gene expression variability within clusters can be explained by these different parameters? This can be examined, if not for all cells, for IT cells where integration across different modalities has already been done.

The reviewer suggests the characterization of the relationship between molecular identities and functional properties (e.g. connectivity) along the hierarchical organization of cell types. Our study shows that spatial location, DNA methylation, projection, morphology, physiology are strongly correlated with gene expression at subclass level, and for some major cell types as well. Thus it is not the case that the two types of classification are to large extent orthogonal to each other.

However, at more granular levels, we do observe a certain degree of differential variations, for example, different morphologically defined PV interneuron “types” (small or large basket cells, translaminal cells, shrub cells, etc.) can be mapped to the same transcriptomic type, appearing as different parts of a continuum (Scala, Kobak et al 2020). In this case, it is possible that previously recognized functional attributes that were considered discrete may not have captured the full spectrum of variation due to the sparseness of the data. The relative completeness of the transcriptomic profiling (due to its tremendous scalability) is able to fill gaps and provide underlying linkage between different features.

On the other hand, we also observe that certain functionally significant features, such as long-range projections, exhibit greater cell-to-cell variability within transcriptomic types (Peng et al bioRxiv 2020), pointing to more studies needed to better understand the relationship of different modalities.

Moreover, our Patch-seq data enables us to relate transcriptomic profiles to morphological, electrophysiological and anatomical (cortical layer location) information. In agreement with the overall results discussed above we find that transcriptomic subclasses (i.e. neurons expressing Sst, Pvalb etc) have also distinct morpho-electric phenotypes. However, within each family the transcriptomic types are not well separated in the morpho-electric space. Within each subclass we find a continuum of morpho-electric variability. Interestingly, we also find that within each subclass neighbouring transcriptomic types show similar morpho-electric features. In summary, these results propose a model where neurons form a hierarchy with non-overlapping branches at the level of subclasses but continuous and correlated transcriptomic and morpho-electrical manifolds within subclasses.

17. *Referee 4:* The authors should discuss which type of data are most informative when categorizing neuronal cell types. For example how do the RNA expression, accessibility, and methylation profiles compare in terms of their ability to stratify data into meaningful cell types?

Both transcriptomic and epigenomic data are reflective of the molecular genetic programs defining cell types, and they provide highly complementary information to enrich our understanding of the molecular genetic programs defining cell types. In general, data types with more comprehensive and in-depth coverage will be more informative. In our study, we used RNA-seq data and transcriptomic taxonomy as the anchor for comparison with other data types.

18. *Referee 4:* In this context, the authors should clarify how they define class, subclass, and type of neurons? It would be helpful to propose and implement a unified neuronal cell type nomenclature that could be applied to additional datasets coming from different labs.

We believe that cell types are best defined in a hierarchical organization. Class, subclass and type define different levels of membership and granularity in this hierarchical organization, which we have now clarified in the revised Table 2.

We strongly agree with the aspiration to propose a unified neuronal cell type nomenclature. While we are not quite at the stage of providing a unified common usage nomenclature for cortical cell types, we have developed a nomenclature system (Miller et al eLife 2021) that can be applied to each dataset in a uniform way that provides a way to cross-reference between projects, and that future studies can be cross-referenced against. Multiple nomenclatures can be applied as aliases to this system. We have added this common nomenclature, and the terms used to describe subclass level cell sets across modalities as a supplemental table.

19. *Referee 4:* The authors propose that variable projections of neurons belonging to the same molecular type could result from activity based pruning during development. Alternative possibility is that axonal pathfinding is controlled by subtype specific genetic programs active during early developmental stages when neurons are establishing their connectivity, but these programs become extinguished in the mature CNS. These fundamentally distinct mechanisms should be discussed as they would have important implications for the interpretation of the adult molecular data.

We fully agree with the reviewer that there could be two distinct developmental mechanisms that define the variable projections of neurons belonging to the same adult-stage molecular type, a genetic program that only exists during development and an activity-dependent shaping process. We have now discussed these two possibilities more explicitly and clearly.

20. *Referee 4:* The authors should provide more direct integrative analysis of data across different species. Can the data be shown as a degree of overlap between species-integrated clusters and each species-specific cluster? Considering that homologous cell types exhibit great degree of molecular divergence, it would be interesting to see if canonical correlation analysis or manifold alignment can be applied to compare data across different species. Such direct comparison could help to identify truly species specific cell types. It is of interest that number of Parvalbumin cell types seem to be expanded in primates relative to mouse, even with fewer profiled cells. Also there are some human/marmoset clusters that have low overlap with any given mouse cluster; does this suggest a human or primate-specific cell

type? Can you get more fine-grained resolution if you re-do this analysis with subsets of data restricted to a specific lineage?

In our companion paper, Bakken et al. 2020, we describe in more detail the definition of cross-species consensus clusters. Briefly, RNA-seq data from all 3 species were integrated with Seurat using canonical correlation analysis, identification of mutual nearest neighbor anchors, and batch correction. Unsupervised clustering of the integrated data identified several hundred metacells that were merged into 45 consensus clusters that contained cells from all species. These consensus types are shown as boxes in Extended Data Figure 1 in this paper. As suggested by the reviewer, we achieved better alignment and finer resolution consensus clusters when we performed this analysis for major cell classes rather than all cells at once. This analysis reveals that the cellular components are largely conserved across these species, and there are no highly distinct types unique to any species.

However, additional cellular diversity is seen in primates for some consensus types, such as Pvalb interneurons, as noted by the reviewer. We also see more specialization of human layer 2 and 3 IT neurons that is described in detail in the companion paper Berg et al. 2020 based on Patch-seq data from human and mouse cortical neurons. We report that two cell types are present in deep L3 of human cortex and do not have matching types found in L2/3 of mouse cortex. Interestingly, these types are most closely related to layer 5/6 mouse glutamatergic types, suggesting that they are either developmentally displaced types or primate-specialized types that are expected to share some cellular properties with deep layer neurons. These changes are likely linked to the expansion of supragranular layers that accompanied the 1000-fold increase in surface area of human cortex relative to mouse and the requirement for greater cortico-cortical connectivity of sub-functionalized areas. These results suggest deep evolutionary conservation of the building blocks of cortex, and species specialization of cell types linked to other structural changes.

21. *Referee 5*: While the authors did a commendable job taking the abundance of data from these different manuscripts and turning it into a coherent whole, the direct connection between the transgenic tools, the input-output connectivity map and the described consensus cell types was somewhat lacking. In other parts the data could be presented in a different way to better support the main conclusions.

We have now revised the text and figure on cell type genetic tools, and have made a stronger and more compelling connection between genetic tools, their use in developmental fate mapping, transcriptomic cell types, and anatomic types (input/output mapping). We also rearranged the order between the original Figs 6 and 7. By strategically targeting the hierarchical organization of excitatory pyramidal neurons (PyNs), focusing on consensus major branches/classes, these transgenic tools provide a much needed experimental

framework to dissect finer types (new Fig 6 and companion paper Matho et al bioRxiv 2020), and thus to characterize and validate (or invalidate) the leaf level transcriptomic types.

Many of these driver lines are now key tools for anatomic input/output mapping with subpopulation resolution (new Fig 7 and companion paper Muñoz-Castañeda et al bioRxiv 2020). In particular, inducible driver-based complete single-cell reconstruction is already driving the anatomic typing incorporating key molecular information (companion paper Peng et al bioRxiv 2020). Furthermore, by targeting major transcription factor defined progenitor types, the driver lines provide key starting points to fate-map the developmental trajectories of major PyN types. These will provide a much-needed developmental genetic basis to validate or invalidate the transcriptomic types.

22. *Referee 5*: An interesting question that arises from the results presented by the authors is to what extent IT neurons should be viewed as a continuum vs. discrete cell types. The authors note in several instances the relationship between expression, morphology and functional profiles and the cortical depth of IT neurons but don't show any direct quantification or plots for this correlation (similar to figure 5c in Muñoz-Castaneda et al. 2020). This seems like an important conceptual advance and it would be informative to see an aggregated analysis across modalities where depth information can be obtained.

The IT neurons do display continuous variation in multiple modalities and across species. Both MERFISH and Patch-seq data showed correlation between gene expression and cortical depth positions of individual cells, as well as with electrophysiological and morphological properties, through more detailed correlation analyses shown in our companion papers (Zhang M et al 2020; Scala et al 2020). In addition to primary motor cortex, a larger transcriptomic study covering the entire mouse isocortex showed that this continuous transition of IT neurons across cortical depth exists in all cortical areas (Yao, Nguyen, et al bioRxiv 2020, ref 53).

Variation across IT neuron types in layers 2 and 3 of human cortex is also described in detail in another companion paper (Berg et al bioRxiv 2020). In the human cortex there is an overall molecular gradient across supragranular IT types, but that can be split into clusters with clearly discriminable properties. One of these clusters displays a particularly strong within-cluster, depth-dependent gradient that is correlated with anatomical and physiological properties. Thus it seems that there are higher order gradients that likely reflect sequential developmental generation from a common progenitor pool. At the same time, there are divisions of this overall gradient with differential laminar positioning, anatomical and physiological properties.

23. *Referee 6*: Transcription Type Classification: In most of the computational analysis across methods the data is heavily over resolved, with “transcription type” classifications being derived from clustering at arbitrary resolutions rather than any known molecular signatures. For example, the authors claim to identify 24 distinct subtypes of cortical GABAergic neurons, a cell class that is known to contain 5 major distinct populations, only to immediately follow that all 24 can be classified into one of the known 5. The t-types result from the specific clustering paradigm used and most would not be reproducible by a different algorithm. Further, as granularity increases, the signature of any t-type is more likely to be driven by batch-specific and technical variance rather than true biological differences across subpopulations and would be difficult to reproduce in other datasets. Finally, such arbitrary classification is not particularly valuable to the reader who identify or target a particular population in their dataset or experiment, and they acknowledge this as a source of error when attempting to map types across species. It would be better for the authors to compare across known or sufficiently distinct subpopulations. For example, instead of the arbitrary Pvalb_1 and _2, or Astro_1 and Astro_2 perhaps distinguish between protoplasmic/interlaminar/fibrous/etc. and perhaps sub-partition in a non-algorithm-specific fashion if there is enough heterogeneity in the population to justify it.

We respectfully disagree with this referee’s assertion that our clustering based transcriptomic classification is arbitrary, algorithm specific, contaminated by batch effect or other technical variance, and thus is not reproducible, generalizable, and does not reflect true biological difference. In this flagship paper as well as our multiple companion papers, we present a large body of evidence demonstrating the biological relevance of the transcriptomic cell type classification. We also provide a well balanced assessment of the relationship between transcriptomic profiles and other modalities of cellular properties, showing high degree of consistency at major cell type level as well as differential variation and fuzziness at more refined levels.

To obtain data-driven, unbiased classification of cell types, clustering of the transcriptomic data must use statistical criteria rather than known molecular signatures. Indeed, the choice of different clustering methods and statistical criteria, as well as the different depths of coverage of the datasets themselves, can lead to different numbers of clusters, but there are many ways to assess the robustness of the classification result. This topic was thoroughly investigated in our companion paper Yao et al bioRxiv 2020. As described in that paper, our transcriptomic classification is based on the integration of 7 datasets collected using different single-cell or single-nucleus RNA-seq methods, and a central theme of that paper is to conduct extensive analysis using a variety of methods to determine the most robust clustering results. Furthermore, our current transcriptomic classification result for the primary motor cortex is highly consistent with previously published work from other cortical areas in mouse and human (e.g. Tasic et al Nature 2018, Hodge et al Nature 2019, Yao, Nguyen, et al bioRxiv 2020).

It is possible that our transcriptomic clusters are over-resolved. The reason for doing this is to maximally represent the transcriptional variations found in the datasets and present them as hypotheses for the field to test and opportunities for discovery. In fact, our transcriptomic classification is associated with many new molecular signatures (marker genes) that can be used to identify and target these cell populations for further studies using other approaches by ourselves and by the community at large. The more granular differences may reflect a variety of cellular phenotypes or potential to respond that were not measured here or reflected in relatively gross anatomical or baseline intrinsic physiological characteristics. Furthermore, even many of these finer cell types show conservation across species, at least supporting the idea that they are meaningful and under evolutionary constraint.

In our current study, we did not intend to prove that every cluster is a real cell type. Instead, we state that cell types are best represented in a hierarchical organization, as discussed above. Cell classes or types at different branching levels can be validated using multimodal evidence. Indeed, for GABAergic interneurons, our Patch-seq study presented here (Fig 4, companion paper Scala Kodak et al 2020) as well as an independent Patch-seq study in mouse visual cortex (Gouwens et al Cell 2020) demonstrate that there are multiple types within each of the 5 major distinct populations the referee mentioned (we call them subclasses) based on morpho-electrical-transcriptomic properties. Critically, these Patch-seq studies establish the strong correlation between the transcriptomic types and historical knowledge about GABAergic interneuron types traditionally defined by morphological and physiological features (as summarized in Fig 10 and more extensively listed in Scala Kodak et al 2020).

24. *Referee 6: Differential Expression Analysis:* The authors use outdated and inappropriate methods for calling DEGs across cell populations and species. These methods treat each cell as a unique observation rather than each sample and thus are both heavily overpowered and susceptible to batch variance. While this may not be too much of an issue for isogenic mouse lines, the analysis does not consider sex, age or RNA quality of the primate samples, especially human, all of which are major drivers of gene expression differences. The authors should repeat their analysis using an alternative approach that permits complex design. I recommend the authors consider pseudo bulk methods (perhaps within t-types) that can give more accurate estimations of power as opposed to the nonsensical p-values produced by t-test and Wilcoxon when applied to single cell data, as this confounds the comparison of DEG divergence across species, though there may be other ways to correct for this as well.

We thank the reviewer for raising this important point. We have revised the differential expression analysis to use a pseudobulk approach as implemented in DESeq2. Briefly, we summed gene UMI counts for each donor and species and compared gene expression between species treating donors as biological replicates using the standard DESeq2 pipeline. We selected genes with adjusted p-values <0.05 and fold-change > 4 to focus on robust differences. 95-99% of markers of each cell subclass that were identified in our initial DE

gene analysis were identified in the pseudobulk analysis, likely because we initially focused on genes with large expression differences. As shown in Figure 2d, we find the same main biological results as initially reported: human and macaque cell types are more transcriptomically similar to each other than to mouse, and non-neuronal cells have more evolutionarily divergent expression than neurons.

25. *Referee 6: Small Sample Size:* Although it was difficult to parse together which data originated from which sample across the various companion papers, I am under the impression that for each modality the human and marmoset data originated from 2-3 samples each for several of these experiments. If this truly is the case, this is an acceptable [sic] (**Note: Reviewer must mean “unacceptable”**) sample size, regardless of the number of cells recovered from each tissue sample. The extraordinarily variable nature of primate single-cell data, coupled with the failure to incorporate critical covariates mentioned above would jeopardize replicability, which is already one of the biggest concerns in the field of single-cell genomics.

We report the number of donors for each species and data modality in Extended Data Figure 1c (table copied below) from the Bakken et al. companion paper. Donor number is listed in parentheses where (p) indicates that nuclei were pooled from multiple mice. We had previously reported expression differences across human donors based on single nucleus RNA-seq of temporal cortex (Hodge et al Nature 2019). Remarkably, donor differences are smaller than cell type differences as seen in the tSNE below (Fig. 1b). However, we recognize that interindividual variation remains an important axis of variation. Therefore, in this study, we focused on cell type and species differences with large magnitude differences in addition to statistical significance. Furthermore, evolutionary conservation of cell type markers defined within species is a strong indication for the replicability of these markers in future samples.

26. *Referee 6*: The manuscript felt a bit redundant at times, with many paragraphs devoted to introducing yet another data type, and re-listing all the previous data types it was going to be integrated with, and then summarizing once more all the data types that it was just incorporated with. Even though these integrations are indeed very impressive, I would recommend foregoing some of these redundant sentences and paragraphs to leave more space for the actual insights, and for more biologically meaningful conclusions, rather than re-hashing and re-stating the impact of what was just done.

We have tried to reduce redundant descriptions and simplify the text as much as we can.

B. General - Minor

1. *Referee 1*: Incorporation of brain disease genetic risk variants (for Parkinson's and Alzheimer's Diseases, for example) using the genomic single cell annotations may set the new data in context of brain abnormalities. Risk variants reside largely in DNA non-coding regions containing multiple SNPs with unclear functionality – the present atlas may inform mechanisms for these disease risks, both inter- and intra-cellularly.

We share the referee's enthusiasm for intersecting genetic risk with genomic cell types and feel this will be a highly fruitful approach to identify the fine locus of brain diseases. In fact we did explore this topic in the companion snATAC-seq paper (Li, Preissl bioRxiv 2020). However, we feel this is a topic for more dedicated analyses rather than effort to classify cell types that is the overarching theme here.

2. *Referee 1*: Mechanisms involved in brain hemispheric asymmetry may be revealed and discussed using the present multidimensional atlas and could expose insight into whole-body disease asymmetry, as well as normal functions such as handedness, for example.

The topic of brain asymmetry has been approached using cruder transcriptomic methods at length with minimal findings to date. In principle such asymmetry must involve at a minimum differential connectivity, which may in turn represent differences in cell types making up those connections. The current analyses were not designed to assess hemispheric variation, but this is a clear area for subsequent exploration to see if the proportions or anatomical/molecular properties of neurons vary between hemispheres. In human however this remains a very difficult question to address due to interindividual

variation and the lack of functional information (e.g. fMRI) guiding sampling within individuals and between hemispheres.

3. *Referee 1*: Although companion papers are available on BioRxiv, more explicit reference to, discussion of, and putting in context the findings of the other papers may help expand the atlas specifics and set the stage of the entire consortium's efforts.

We have added references to all companion papers in the second to last paragraph of Introduction when we first introduced the rationale, design and approaches of our study. This then leads to the final paragraph of a bulleted list of major findings which are largely derived from the companion papers. This indeed helps to integrate this flagship paper with the other companion papers and demonstrate the efforts of the entire consortium.

4. *Referee 3*: Abstract, Line 15: What does “congruently” mean here? The authors's collective vocabulary is better than mine, but I would take it to mean a perfect superposition of one thing onto the other (which would be unlikely to occur in transcriptomic data).

We have removed this word. “Unified molecular genetic landscape” already conveyed what we need to say.

5. *Referee 3*: Introduction, Line 69: The number of citations in the intro are useful for context, but likely could be shorter for publication purposes (examples not needed from all areas where this technique was effectively used ... cortex, hippocampus, hypothalamus, etc.).

We have removed the mentioning of hippocampus and hypothalamus and the associated references to shorten the text.

6. *Referee 3*: What is the molecular specialization of corticomotoneuronal neurons? These are famous in primates and possibly missing in rodents (but see Grinevich, Brecht, and Osten J Neurosci 2005). Do these occur in the data?

We explore this topic on the Betz cells in Figure 4 of the manuscript and in the companion paper (Bakken et al bioRxiv 2020). The primate Betz cells are contained in the layer 5 ET subclass and are homologous to the mouse L5 ET subclass containing the corticospinal projecting neurons. We describe that some of the features of the L5 ET neurons compared to IT neurons are highly conserved, whereas others like size and physiological properties are specialized in primates. As the referee mentions there may be other species specializations

that we did not measure in this neuron subclass, such as whether their axons directly target spinal motoneurons (primates) or not (mice). To look for molecular correlates to species specializations we looked for primate-specific molecular differences or features that varied as a function of evolutionary distance. Remarkably, genes associated with axon guidance showed such species differences, providing hypotheses about molecular mechanisms underlying differential targeting of spinal motoneurons versus interneurons.

7. *Referee 3*: “Comparative analysis of mouse, marmoset and human transcriptomic types achieves a unified cross-species taxonomy of cortical cell types with their hierarchical organization that reflects developmental origins; transcriptional similarity of cell type granularity across species varies as a function of evolutionary distance.” (Line 151+) It would be useful to know if there are exceptions to this rule for evolutionary distance. The example given is “Three non-neuronal types had greater [S]pearman correlations of overall gene expression (Fig. 2d, right columns) between marmoset and mouse likely because non-neuronal cells were undersampled in human M1 resulting in fewer rare types.” (Lines 239-242) but the explanation included suggests these exceptions weren’t evaluated much. This is not unexpected, but when the finding is one of the major bullets and contradictory data isn’t assessed in great detail (because it’s believed to be wrong), then the finding is not so surprising.

We agree with the reviewer that these exceptions to the rule are important to study. In light of another referee comment, we have used a more robust method (pseudo bulk analysis) that treats donors rather than individual nuclei as biological replicates to define differentially expressed genes. Using this refined analysis, we find that endothelial cells and microglia are more similar between human and marmoset, as expected. Only vascular and leptomeningeal cells (VLMCs) are more similar between marmoset and mouse. VLMCs were highly undersampled in human (n = 40 nuclei; 1 type) compared to marmoset (n = 463; 4 types) or mouse (n = 2329; 7 types), and it is likely that average expression is not adequately estimated in human, which reduces the correlation with marmoset. Alternatively, there could be evolutionary changes in the expression or relative proportions of VLMC subtypes in human that we will be able to resolve in future work with deeper sampling of rare non-neuronal cells.

8. *Referee 3*: How should we interpret that many marker genes for glutamatergic subtypes are species specific? In that context, it is a bit inconsistent with the overall pattern of IT, ET, CT, and NC types that do seem to be conserved. Does this mean that no specific marker is needed to make such types, but a suite of them that is robust to substitutions?

Each glutamatergic subtype (and other cell types) expresses thousands of genes, and a large fraction of these genes are expressed in all three species, which enables identification of cell type homologies. For each species, approximately 50 to 350 genes have highly enriched

expression relative to other subclasses and are considered markers, and as we reported only a minority of these markers are conserved across species (see barplots below). While seemingly at odds, it is quite possible for there to be both strong homology and also for there to be a great deal of species specialization as for many other species phenotypes.

It would seem likely that those highly cell type selective genes that are highly conserved are of particular importance to their identity and canonical phenotypes. We hypothesize that conserved markers are under strong evolutionary constraint and are causally related to the core identity of each subclass. For example, in our companion paper (Bakken et al bioRxiv 2020), we identify a transcription factor *RORA* that is a conserved marker of chandelier cells across 3 species and has enriched binding sites in open chromatin. Interestingly, non-conserved markers are often still expressed in other species but have lost their subclass specificity, as shown in our companion paper for GABAergic subclasses (ED Fig. 3a, Bakken et al.).

9. *Referee 3*: Introduction, Line 109-111: The style of present day articles seems to include presaging the results in the end of the introduction, but here you cite the first figure and table! Only an opinion for style purposes, but to me the results begin here. "Here we present the cell census and atlas of cell types in one region of the mammalian brain, the primary motor cortex (MOp or M1) of mouse, marmoset and human, through an analysis with unprecedented scope, depth and range of approaches (Fig. 1, Table 1)."

We appreciate the referee's point here. However, since we want to state the major findings (in bullet points) from this massive paper in Introduction first before diving into the detailed descriptions in the Results section, we need to briefly introduce the design and approaches taken in our study BEFORE stating the major findings so that readers can better understand the overall picture of our study. Thus, we have decided to keep the Introduction section as is.

10. *Referee 3*: Table 1 is useful as a reference, but takes up a lot of space (in particular since some cells are large in only a subset of rows). I think it is more useful for the reader to refer back to while reading the results of the paper and not (as currently set) right after the introduction when the results are yet to be detailed.

We respectfully disagree and feel, as did other referees, that this table provides an important grounding for the datasets that were used in the consortium effort as a whole, particularly given the wide breadth of methods used and species they were applied to. We are confident the space taken will be dramatically reduced upon typesetting so that it will not appear as distracting in the final version.

11. *Referee 3*: How Table 2 is organized might be improved? If truly a Glossary, terms could be listed alphabetically. Furthermore, the use of multiple columns is unclear (e.g. not just a term and its definition). If columns are meant to be interpreted as hierarchical, why MOp and its layers would be in the same column. The hierarchy seems present to some extent in the inhibitory and excitatory subclasses. Explanation of the developmental significance of MGE and CGE given in a table defining terminology seems out of place. "The top branches of theCN transcriptomic cell type hierarchy comprising neuronal and non-neuronal cells." Does 'theCN' mean 'the BICCN'?

We appreciate the helpful suggestions on Table 2 organization and have attempted to improve the hierarchical organization. "theCN" was an unfortunate typo that has been corrected.

We have revised the table significantly to make it smaller and clearer in its use of hierarchical organization.

12. *Referee 3*: Use of terms ET for extra-telencephalic and CT for corticothalamic. I imagine with such a large consortium, the field is well served to converge on a single nomenclature and perhaps this will be the one, but as the thalamus is a subset of the extratelecephalic structures cortex might target, this naming scheme has a small error. CT could be a special subset of ET, though these read as though intended these to be thought of as two separate groups. Similarly the IT definition might be refined to state "only", and in "Excitatory

glutamatergic neuron that projects only to other telencephalic structures” (since some ET neurons might send a collateral). There’s a lengthy description of why ET is chosen to replace PT in the supplement, which is useful, though its placement in the supplement is a problem (since it won’t be as well read).

We understand that CT could be a special subset of ET, that’s why we did not propose to use ET and CT to name the specific subclass of neurons defined transcriptomically. Instead, we proposed L5 ET and L6 CT as names for these transcriptomic subclasses, using the predominant layers these neuron types reside in to define them in a mutually exclusive manner. We agree that adding “only” to the IT definition is better to distinguish it from ET and CT cells because the latter also have intracortical collaterals, and we have now done so. We thank the reviewer for pointing out putting the ET nomenclature in the supplement is a problem as it won’t be as well read. Thus we have moved it to the Online Methods section so that it will appear in the downloadable PDF file of the paper.

13. *Referee 3*: Results, Line 217-223: “These types were grouped into broader subclasses based on shared developmental origin for GABAergic inhibitory neurons [i.e., three caudal ganglionic eminence (CGE)-derived subclasses (Lamp5, Sncg and Vip) and two medial ganglionic eminence (MGE)-derived subclasses (Sst and Pvalb)], layer and projection pattern in mouse for glutamatergic excitatory neurons [i.e., intratelencephalic (IT), extratelencephalic (ET), corticothalamic (CT), near-projecting (NP) and layer 6b (L6b)], and non-neuronal functional subclass (e.g., oligodendrocytes and astrocytes) (Table 2).” This table lists all five of those types of GABAergic neurons in a single line, while having separate lines for IT, ET, CT, etc. Thus, it is difficult to understand the levels assigned to each of these in the hierarchy.

We thank the referee for the comments on table organization and have reorganized it to be more consistent across subtypes.

14. *Referee 3*: Minor point, but the use of color – which is spectacular – will be less accessible to ~8% of male readers (colorblindness). I don’t have ideal suggestions about how to handle this, but it strikes me in aligning the colors of the neurons with the clusters (you may very well have picked colors that are easy to separate, but I don’t have a viewer for pdf that toggles between RGB view and some simulation of deuteranopia). (Fig 2 and 3 for example; in Fig 4 there are symbols besides the neurons to match to clusters).

Due to the large number of cell clusters, it is impossible to introduce symbols in Figure 3 to label the cell clusters. In Figure 3c, we replaced the color scheme so that similar cell clusters (cell clusters that belong to the same subclass) use similar colors now to emphasize the

spatial distribution of the distinct subclasses and the laminar organization of the cells in cortex.

15. *Referee 3*: Lines 1068+ : “This classification provides strong evidence for the existence of hitherto poorly studied but molecularly distinct subclasses such as the near-projecting (NP) pyramidal neurons, and many more novel cell types.” To me, this is pretty exciting but it didn’t make the cut of highlighted bullets. (While more obvious things ... t-type and chromatin state give similar maps ... did.)

We appreciated the referee’s interest in the NP neurons, as their connectivity and function is still poorly understood. The NP neurons were first described by Callaway and colleagues as a third type of L5 pyramidal neurons in the mouse visual cortex that is morphologically distinct from the L5 ET and L5 IT neurons (Kim et al *Neuron* 2015). The NP transcriptomic subclass was first defined in Tasic et al *Nature* 2018 in mouse visual cortex and anterolateral motor cortex. Because of these prior studies, we did not highlight the NP neurons as a major new finding.

16. *Referee 3*: Line 1162: “The molecular genetic framework of cell type organization established by the current study will provide a robust cellular metric system for cross-species translation of knowledge and insight that bridges levels of organization based on their inherent biological and evolutionary relationships.” Is this a strong argument? I would be concerned that some of the lack of translation to disease would be the feel among human and primate researchers that disorders of frontal areas aren’t well modeled in part because the rodent lacks the expansion of frontal cortex. Similarly, MOp research may be limited because the primary motor cortex of primates has five specialized premotor areas with which it is strongly connected, while rodent MOp has strong reciprocal connections with SSp (qualitatively differing from primates). Thus, while some similarities in intrinsic features might occur, it is difficult to know how this will be useful for studying connections or pathologies that come about with cells embedded in a network (that differs from mouse network). This statement is stronger if disorders were cell-autonomous.

Our main point is that cross species comparison at the cell type resolution will be more informative than at the gross anatomy (areal size and specialization etc) level. A striking result of the studies here is the overall conservation of cellular architecture, but with species variation at the finest levels of granularity and in the specific properties (molecular, anatomical, physiological) of different cell types. The conservation allows comparison of orthologous cell types and a way to measure and understand similarities and differences. At the same time, these differences can be profoundly important, for example for the primate Betz cells and for circuitry as noted by the referee. However, these differences are variations on an overall conserved cellular blueprint, and this provides a quantitative framework for understanding both similarities and differences into the future as the community builds on it

to describe circuitry and relationship to disease among other things. Therefore we argue that the argument is strong in that it sets a strong foundation to understand both conservation and divergence of cellular and circuit organization.

We suggest that cell type level analysis will provide compelling evidence for both the core conservation and substantial divergence of cortical circuits. At the same time, we also agree with the referee that circuit connectivity at multiple levels may be another equally or perhaps even more important factor contributing to the conservation and divergence of brain organization across species. It is likely that we will need the combination of both cell type and connectivity information to gain a full understanding. With more cell type datasets from rodents, NHP and human, and in the long run with more comprehensive circuit connectivity data generation becoming increasingly feasible, we will be in a much advanced position to examine and evaluate what brain disorders can be and cannot be modeled in what animal models.

17. *Referee 5:* In general, there are several cases where there are no colorbars for the heatmaps and it's not always clearly indicated what they mean or how they are normalized in the figure legend.

We are sorry about these oversights and have now corrected them.

18. *Referee 5:* Table 2. The text "The top branches of theCN transcriptomic" should be "The top branches of the transcriptomic". For clarity spell out "OPC cells (OPC)" as "Oligodendrocyte progenitor cell (OPC)".

We thank the referee for the suggestions and corrections which have been done in the revised manuscript.

19. *Referee 5:* Line 194. For readability it would be useful to mention data modality of each technique.

Each data modality is shown in Fig. 2a.

20. *Referee 5:* Line 247-248. This seems like something for the methods section.

We have moved the text describing differential expression analysis to the Methods section.

21. *Referee 5:* Line 602. Missing closing parenthesis.

Corrected.

22. *Referee 5*: Line 704. cp is not shown in the figure.
We have substantially revised this figure and related text.
23. *Referee 5*: Line 861. "with cluster 0 predominantly projecting to MY while other clusters having variable" → "have"
Corrected.
24. *Referee 6*: The study is not without flaws, utilizing non-state-of-the-art analytical tools and improperly executed analyses, although these are minor in the grand scope of the paper and should be easy to correct.

As suggested by the reviewer, we have updated the differential expression analysis to use a pseudo bulk approach that enabled use of the well-validated DESeq2 pipeline. We have quantified the robustness of transcriptomic similarity of species consensus clusters by generating a dendrogram based on bootstrapping over variable genes, and this provided confidence in higher level groupings of types by shared developmental origin and long-range projection targets.

25. *Referee 6*: Several of the figures could be further condensed, pushing some of the information to extended data to streamline the presentation in the main text.
We have now condensed the figures and main text as much as we can.

C. Figure specific

Figure 1

1. *Referee 2*: While the 11 companion papers are cited throughout the paper, I suggest to include a separate listing of References to go with this paper on these 11 studies. There is currently Table 1 on the applied methods citing background and companion papers but since they are not cited in sequence, one always has to go back to the reference list and extract them. Another option is to put these 11 references in sequence when this concept is first mentioned in the introduction, so they would appear in one block and have consecutive citation numbers. Similarly, if there was a way of displaying Ref numbers and which panels refer to which Figures in this paper in Figure 1, it might help the reader to follow the flow.

We have now cited the 11 companion papers together when they are first mentioned in Introduction. This should make it easier to find them.

2. *Referee 5*: Figure 1a. Would be informative to see a cross section for all three species with MOp/M1 clearly highlighted.

We have added cross sections for all three species with MOp/M1 highlighted to Fig 1a.

Figure 2

1. *Referee 2*: Data presented in Figure 2 and associated text aims to determine how many cell types there are in mouse motor cortex, and then compare these cell types to human and marmoset cell types. The accompanying text should make this point at the start and perhaps at the expense of some of the details on mouse profiling give insight into the criteria for concluding which and how many cell types are conserved and why.

We agree with this helpful suggestion and have edited accordingly.

2. *Referee 2*: The authors just very shortly say that over- or under-clustering may be issues but not how this is resolved. It is important that readers can follow how cell types are defined across species, much more so than the details of how many nuclei were profiled in mouse. The same goes for marker definition of main cell types and across species – what are the markers and how conserved are they? These are the main points that should come across in this section.

We have tried to make this more clear in the manuscript. We provide a brief description of the definition of cross-species consensus types in the Methods section and a detailed description in our companion paper Bakken et al. 2020. Briefly, RNA-seq data from all 3 species were integrated with Seurat and unsupervised clustering identified several hundred metacells that were merged into 45 consensus clusters that contained cells from all species. These consensus types are shown as boxes in Extended Data Figure 1 in this paper.

The following has been added to the Methods: “Marker genes were identified for each species’ cluster using Seurat’s FindAllMarkers function with test.use set to ‘roc’, >0.7 classification power.” We provide these markers and conservation as Supplementary Tables

7 and 10 in Bakken et al. 2020. Neuronal subclasses had 50-450 marker genes in each species, and up to 20% of these were conserved across all species, as shown in the barplots below. These results have been added to the manuscript text.

3. *Referee 3*: Spearman is sometimes in caps (Fig 2), sometimes not (Line 240).

We have capitalized Spearman in all locations.

4. *Referee 3*: For glutamatergic cell types, “the majority of markers were species-specific (Fig. 2f,g)” (Line 249). What does that mean? “The evolutionary divergence of marker gene expression may reflect species adaptations or relaxed constraints on genes that can be substituted with others for related cellular functions.” What are the species-specific genes, then? Is it just a matter of different species using different isoforms of the same genes, or something entirely different. The discussion of this (Lines 1147+) touches on what some explanation might be, but doesn’t provide an answer. Is it not a little unexpected that the set of marker genes for these types (illustrated in Fig. 2f Venn diagrams) generally shows little overlap across the three species? I understand that L2/3 IT is clustered based on a lot of data but is it not somewhat remarkable that there aren’t more shared genes needed to create that conserved cell type? Does the data mean that, with more species-specific markers, that there are multiple ways to achieve the same general

morphology/connectivity? Or that lower levels of other genes are what is needed to achieve this (such that they don't meet the bar to be marker genes)?

We thank the reviewer for raising this important topic. We also find it remarkable that we can identify conserved cellular components of M1 in rodents and primates, and yet the expression levels of thousands of genes have diverged between species for homologous cell types. For example, we reported in Fig 6a of Hodge et al. 2019 (see scatter plot below) that the majority of orthologous genes have similar expression in the Sst Chodl type between species with correlation = 0.66. Conserved genes provide sufficient information to identify homologous types. At the same time, 18% of genes expressed in the Sst Chodl type have >10-fold expression difference between human and mouse. We also report in a companion paper (Bakken et al., 2020) that even genes with similar expression levels between human and mouse cell subclasses can involve dramatically different isoform usage. Few marker genes are conserved because they must not only be expressed in the cell type of interest but also retain their cell type specificity across species.

In the companion paper Bakken et al. 2020, we showed that the majority of subclass markers are expressed in all species but with reduced specificity. The plot below shows the proportion of GABAergic subclass markers identified in only one species that have different thresholds of expression enrichment (LogFC = natural log fold change) relative to the other two species. For example, ~90% of human-specific Sst subclass markers have <1 LogFC (<2.7-fold) enrichment in human Sst neurons compared to marmoset and mouse Sst neurons (yellow dot in 4th column from left). Since these Sst markers are still expressed in marmoset and mouse, they must not be conserved because they have lost specificity in Sst compared to non-Sst neurons in marmoset and mouse. We have changed "species-specific" to "species-enriched" in the text to emphasize that these markers are not exclusively expressed in one species.

We hypothesize that the relatively small number of genes that have conserved expression and specificity are central to cell type identity. For example, in Bakken et al. 2020, we identify a transcription factor RORA that is a conserved marker of chandelier cells and may drive the gene regulatory networks that distinguish this type from basket cells. Other genes that are expressed in chandelier cells in all species and have divergent background expression in other cell types contribute to chandelier properties but are less likely to be core drivers of chandelier identity. Analysis of larger numbers of species will likely be necessary to understand this evolutionary variation and what the core components of conserved cellular architecture is.

5. *Referee 3*: Inconsistent use of the blue color for human?

In Figure 2a-h, we have used blue to correspond to results for human. In panels i and j, colors are used to differentiate different data types, and light blue corresponds to ATAC-seq reads.

6. *Referee 3*: The recent Berg et al bioRxiv suggested 5 clusters for human L23 IT but here there seem to be 6? Or is this a primary motor versus medial temporal difference? (But in that paper, comparing human MTC to rodent VISp is justified by stating there aren't great differences across areas).

The referee raises an important point in that there is some discrepancy between the results from human temporal cortex and primary motor cortex in terms of the diversity of supragranular excitatory neurons. Glutamatergic neurons in L2/3 of human cortex, including both motor and temporal cortex, are heterogeneous and grouped into discrete cell types, and types with graded transcriptomic differences across layers. Berg et al used the five cell types previously defined in human middle temporal gyrus (Hodge et al. 2019 Nature). Hodge et al had noted that the FREM3 cluster was highly heterogeneous and could be further split

with adjusted clustering parameters. Indeed, Berg et al. (Figure 5) reported that superficial and deep L2/3 FREM3 neurons had distinct morphology and electrophysiology and that these properties changed gradually with laminar depth.

Transcriptomically these same general organizational features appear to be true across human cortical areas, with similar heterogeneity in motor cortex to temporal cortex and additional heterogeneity beyond mouse in deeper layer 3. However, there do appear to be differences in M1 clusters compared to those in temporal cortex that either reflect regional differences in diversity, or slight changes in thresholding that differentially split the largest FREM3 cluster (which as we reported previously can be split further with more lenient clustering). Different regions of mouse cortex show similar diversity to one another, and consistently less than human cortical regions examined so far, most in that they lack the deep layer 3 complexity seen in human cortex. These results highlight the importance of characterizing cellular taxonomies with the relative similarity of types and heterogeneity within types across a variety of cellular phenotypes (particularly taking advantage of non-human primate as this is very challenging to perform in human). Future work may reveal more discrete properties in different modalities, e.g. connectivity, that can be used to refine the discrete glutamatergic neuron populations in human L2/3 and understand regional variation.

7. *Referee 3*: What does it mean that L2/3 IT and L6 IT cluster more closely than other glutamatergic cells? These neurons are born at different times in development, so it seems that the clustering is not strictly recapitulating development here (as opposed to the GABAergic cells).

We have previously found that L6 IT neurons share a transcriptomic signature with both L2/3 IT and L5 IT neurons in mouse (Tasic et al. 2018 *Nature*) and human (Hodge et al. 2019 *Nature*) cortex (also see the bent UMAP in Fig. 8a). This is now more apparent in an updated dendrogram showing branch robustness as suggested by the reviewer below. Other than one highly distinct IT type (L6 IT 3), all IT types are on the same branch. The finer groupings of IT types (e.g. the location of the L2/3 IT type) depend on the set of genes that are compared, as indicated by the lower robustness (lighter colors) of the IT branches. Thus, although there is an overall recapitulation of developmental origin in the hierarchical organization it is based on a distance metric that is somewhat dependent on the analysis method so we try not to overinterpret the relatedness among the different types here.

Also, to our knowledge, little is known about the developmental origin and process of deep layer IT neurons. As mentioned in the response to another referee comment above, cell type development involves complex genetic programs that include different progenitor types, lineage progression, and cell birth order/time.

8. *Referee 3: (Coordinate with Figure 3)* For Figures 2b and 3b, is there any kind of certainty that can be ascribed to the nodes in the tree/dendrogram? Although this sort of categorization is maybe newer to classifying neuron types, there are some tests that evolutionary biology-type researchers apply to evaluate the certainty of the branches in a tree (resample the data by bootstrap 10,000 times and ask how often a certain node emerges, for example; maybe other tests exist). This is only of minor importance, but it is not clear from the data alone if such a large sample of expression data provides categorization that is so clean that noise in the expression data is no issue (which we hope is the case ... but maybe you who have examined the data know better), or if we are accepting the categorization as-is because we have no means to test if other groupings work too.

We thank the reviewer for the suggested analysis. Branch robustness was assessed by rebuilding the dendrogram 10,000 times with a random 80% subset of variable genes across clusters and calculating the proportion of iterations that clusters were present on the same branch. We have updated the dendrogram in Figure 2b (shown below) to visualize the robustness of all branches. Cell types robustly group by major subclasses and show more

variability at the leaves.

Similar analysis has been made for the dendrogram of mouse MOp derived from the consensus analysis of 7 sets of sc/sn RNA-seq (Figure 10a (shown below), which also shows that the nodes at the class and subclass levels are more robust, with more variability at the lower leaf (cluster) level, likely due to less distinctness and more continuity among the clusters within the same subclasses, as described earlier.

For the MERFISH data, we compared the subclasses and clusters with those derived from the consensus sc/sn RNA-seq data analysis. The confusion matrix showed excellent, essentially one-to-one correspondence between MERFISH and the sc/snRNA-seq data at the subclass level, whereas at the cluster level, good correspondence were still observed but with more off-diagonal elements (see companion paper Zhang M et al bioRxiv 2020, Extended Data Figure 2a and Figure 1e, also shown below). These results are again consistent with conclusions described above.

9. *Referee 5:* To what extent do GABAergic consensus types have more specific marker expression compared to other types (Line 255)? The data presentation between 2G and 2H is so different that it's hard to draw this conclusion.

On average, glutamatergic subclasses actually have more marker genes than GABAergic subclasses in each species. Glutamatergic neurons express more genes than GABAergic neurons (see ED Fig. 1 from the companion paper Bakken et al. 2020 and below), and 1-5% of expressed genes are markers for all neuronal subclasses. Every subclass has conserved markers across all three species, and there are few conserved markers of L5 IT and L6 IT subclasses (see barplots below).

Figure 3

1. *Referee 2:* Using MERFISH, the cell type number presented in Figure 3 and associated text is quite different from the one presented before. It is unclear how the genes that were studied here have been chosen. One would have thought that in a logical flow, the genes to study should have arisen from the data presented in Figure 2, but this does not seem to be the case. The authors at the very least should comment on the overlap of genes used in the two approaches. The fact that the cell type number differs needs an explanation. Both methods are based on transcripts, and ideally should result in matching data and cell type classification.

The genes used in the MERFISH experiment and presented in Figure 3 were indeed selected based on the mouse sc/snRNA sequencing dataset which was presented in Figure 2. However, the MERFISH gene panel included 258 genes, which is fewer than the number of variable genes (3792 genes) used for clustering in the sc/sn-RNAseq data. The methods we used to select the MERFISH gene panel are described in detail in the Methods section of the companion paper (Zhang M et al, bioRxiv 2020). Briefly, the 258 genes consist three subsets of marker genes which were selected by three approaches: (I) Canonical marker genes for the major cell types in the cortex selected based on prior knowledge; (II) Differentially expressed (DE) genes selected based on pairwise differential gene expression analysis of the neuronal clusters identified by the sc/snRNA-seq data; (III) A set of genes which contained the highest mutual information (MI) among the clusters identified by sc/snRNA-seq. Of the 258 genes selected, 91% are among the variable genes used for clustering in the sc/sn-RNAseq data.

As for cell cluster number difference, as we describe above in the response to various reviewer points, the cell classification at the class and subclass levels are more robust, while the cell classification at the lower level (cluster) level are more variable, likely due to less distinctness and more continuity among the clusters within the same subclasses. Hence the exact number of clusters are less meaningful than the number of subclasses and we do not expect the independent clustering analyses of MERFISH and sc/snRNA-seq to generate the same number of clusters. As shown in our response to the review point above, we observed excellent agreement between the subclasses identified by the two methods, and good correspondence between the clusters identified by the two methods.

2. *Referee 3*: Is there an obvious medial/lateral border at the AGm/AGl interface? This lateral/medial agranular border has been used by others to mark a cytoarchitectural border of MOs and MOp (also called AGm and AGl, or separating whisker M1 and somatic M1 in some publications; Brecht et al., 2004 J Comp Neurol for a rat example).

We did not observe an obvious border between MOs and MOp marked by the cell molecular signatures. Almost all cell types are shared in these two regions.

3. *Referee 3*: For Fig 3a, it would be useful to either use some indication of pia/white matter or dorsal/ventral to orient the reader. These panels look at though they are rotated compared to the earlier panels in the same figure but it isn't clear to me what rotating the layer structure is offering nor which direction is medial/lateral within M1. Medial/lateral might help interpret the difference in location of the SSp versus MOs projecting cells, though there would also be some use in understanding how the complete the retrograde labeling is (since

it looks like you could argue from this image that SSp and MOs projecting L23 cells don't spatially overlap ... though I don't know if that has been proposed in the literature).

By Figure 3a, we believe the reviewer meant Figure 3e. The spatial plots Figure 3e in the original submitted manuscript were rotated 90° counterclockwise compared to the panels in Figure 3c.

Because the number of retrograde labeled cells in a single slice is low, visual impression does not necessarily lead to statistically significant conclusions. We have now replaced this example map of a single slice with a summary panel showing quantitatively the projection pattern we observed between MOp neuronal types and the three targeting cortical regions.

We acknowledge the limitation (under-labeling) of the CTb-based retrograde labeling method, but in the meantime, our quantifications of the composition of cell clusters projecting to each target region and the projection patterns of each cell cluster are still most likely accurate, because we imaged and quantified a very large number of cells (~190,000 cells from 2 animal replicates for these projection measurements).

4. *Referee 3: (Coordinate with Figure 2)* For Figures 2b and 3b, is there any kind of certainty that can be ascribed to the nodes in the tree/dendrogram? Although this sort of categorization is maybe newer to classifying neuron types, there are some tests that evolutionary biology-type researchers apply to evaluate the certainty of the branches in a tree (resample the data by bootstrap 10,000 times and ask how often a certain node emerges, for example; maybe other tests exist). This is only of minor importance, but it is not clear from the data alone if such a large sample of expression data provides categorization that is so clean that noise in the expression data is no issue (which we hope is the case ... but maybe you who have examined the data know better), or if we are accepting the categorization as-is because we have no means to test if other groupings work too.

Please see the response above in the Figure 2 part.

5. *Referee 4:* Was spatial location considered in MERFISH clustering?

We classified the cells based on their gene expression profiles alone. The spatial information was not considered in MERFISH clustering

6. *Referee 5*: How do the results in Figure 3e match with the statement in line 340: “each IT cluster projected to multiple regions, with each region receiving input from a different composition of IT clusters”. Why are there no double labeled cells? Could the authors provide a breakdown of these numbers to support this important observation? Finally, the data is not shown in a very clear way as it's hard to align the cell types with the projection type and there are no cortical layer indicators.

As described in our response above, in order to better present our findings in this part, we replaced this panel (Figure 3e) now with a summary plot to quantitatively show the projection pattern described in this statement. Indeed, because CTb-based retrograde labeling method is known to under-label projecting cells, double projecting cells could be under-represented as compared to single-projecting cells, and in our quantitative analysis shown in Figure 3e, only single-projecting cells are considered.

7. *Referee 5*: Figure 3c. Very confusing color legend. Can the same color indicate different cell types? Why not color by cell subclass?

We now changed the color scheme so that the cell clusters that belong to the same subclass use similar colors.

Figure 4

1. *Referee 2*: Figure 4 and associated text presents a heroic effort to patch many cortical cells in different species, subsequently determine their morphology and assess the transcriptome. It would be useful if instead of showing single example cell ephys data in panel d, the authors provided quantitative differences between what they call a cell type based on physiology and whether they in addition take into account transcriptome data. In other words, what does morphology and/or ephys traits add to the transcriptome classification we already know? Does morphology and ephys lead to more diversity, or does it go hand-in-hand with the transcriptome data? The authors should synthesize this new data with previously presented data.

The referee asks an excellent question that is key in understanding the logic behind the diversity of cell types. Our consortium has recently published an article in *Nature* online (companion paper Scala, Kobak et al., 2020) which is focused on relating transcriptomic, electrophysiological and morphological characterization of neurons. We found that at the level of broad families of transcriptomic types (e.g. Vip+, Pvalb+, Sst+ etc) there is a very good

match between morpho-ephys traits and transcriptome. However, within each family there is a diversity of morpho-ephys properties and transcriptomic types and within a family transcriptomic cell types are not well separated in the morpho-ephy space. However, we also find that neighbouring transcriptomic cell types within each family show more similar morpho-electric features. Therefore, in Scala, Kobak et al. we suggest that “the tree of cortical cell types” may look more like a banana tree with a few large leaves, rather than an olive tree with many small ones where neurons follow a hierarchy consisting of distinct, non-overlapping branches at the level of families (large leaves), but with a spectrum of cell forming continuous and correlated transcriptomic and morpho-ephy landscapes within each leaf. Due to lack of space we decided to demonstrate the richness of the data and refer to details in Scala et al., and not to repeat the figures demonstrating these findings here. Another important role the Patch-seq data plays is to link transcriptomic types with traditionally recognized cell types that are defined by morphological and electrophysiological properties which provide a Rosetta stone relating these properties. This is illustrated in Figure 10.

2. *Referee 3:* Fig 4a, any idea as to why some of the points sampled by physiology seem so sparse compared to the density of t-type data (“z” and “-“ have a lot of black points; “r” and “x” and “y” have less)? Maybe this just means layer 4 and 6 are boring places to record.

Patch-seq is a much lower throughput method compared to single cell sequencing from dissociated neurons. Therefore, given a limited budget in terms of the number of neurons that we could profile using Patch-seq, as we describe in great detail in Scala, Kobak et al 2020, using various Cre drive lines we tried to cover as diverse a population of neurons as possible. Thus some of the regions in the transcriptomic map were sampled more sparsely.

3. *Referee 3:* Fig 4b,c, the electrophysiological recordings aren’t mentioned in the legend in great detail. Do the overlaid traces represent three current steps? Hyperpolarizing, near threshold, and maximum firing rate (I think this is specified in other papers from the same group but not here.)

Thanks for pointing this out. We have fixed the legend to clearly describe the electrophysiological traces.

4. *Referee 3:* Fig 4h, the small part with so many labels gets hard to follow. Also the island of L5/6 NP seems to be split into two groups (but the cut is so clean it looks like an editing error instead of the data).

We thank the referee for pointing out this unfortunate editing error that has now been corrected, as well as adding a blowup of the ET island.

5. *Referee 3*: Line 392-394, “excitatory t-types from the IT subclasses with more similar transcriptomes were located also at adjacent cortical depths, suggesting that distances in t-space co-varied with distances in the me-space, even within a layer”. Fig 4g is not as convincing to me that this is true since the electrophysiological parameters that might be similar at neighboring depths aren’t presented in a manner that shows adjacent populations with similar characteristics. The ordering of the current injection traces according to color code aren’t presented in the same order as the pyramids beneath, for example, and the parameter to extract from the traces to look for co-variation is not pointed out.

We apologize since the way the sentence is written was confusing and misleading. For the IT cells we find that transcriptomic distances between t-types were strongly correlated with the average soma depth differences. This result is quantified and described in detail in the companion paper (Scala, Kobak et al., Nature 2020 published online, see Fig. 4 for details). We also performed this analysis for many other t-types and in summary we find that within major transcriptomic families, morpho-electric phenotypes and/or soma depth often varied smoothly across neighbouring t-types, indicating that transcriptomic neighbourhood relationships in many cases corresponded to similarities in other modalities. Due to space limitation in the current manuscript we do not show the quantification and figures for these results but they are shown in detail in Scala, Kobak et al., Nature 2020.

6. *Referee 3*: Line 397 “At the level of single t-types, we found that some t-types showed layer-adapting morphologies across layers (Fig. 4e,f)” Ok. But doesn’t this then suggest that if the same t-type results in variable morphology (Fig 4f) that the threshold for clustering those t-types together is not ideal and there is some transcriptomic difference between a Sst+ neuron in L6 and L2/3 that might explain whether the dendrites are broad or restricted in the same layer?

The referee raises a very good point. To determine if there is a more granular classification in agreement with the differences in morphology across cortical layers we would need much more data from what we currently have available. Moreover, these morphological differences may be defined during development or at the level of isoform gene expression. These are excellent questions and we hope future work will provide answers to these important questions.

7. *Referee 3*: For the Betz cell comparison, I appreciate this data and find it interesting. Because of the enormous soma size of Betz cells, and the absence of neurons fitting this in mice, I am curious how the mouse cells to compare were chosen? There are (at least) two subtypes of

ET-type L5 neurons in mice (Economo et al, 2018), which seem to project to either thalamus or medulla. I am curious which have more similar properties to the Betz cells (or if it is known which mouse ET type you use for comparison). Since these are somewhat unique, but also seem to fall into similar transcriptomic cluster as non-Betz layer 5 cells, are they similar in e-m properties corresponding layer ET cells? This might be a more fair comparison than mouse – for example, do other layer 5 ET types cells in the primate also have high expression of the Kv channels of Fig. 4L?

The referee raises a great point about whether there are distinctive properties in the different L5 ET types and how that relates to cross-species comparisons to primate Betz cells. In this original analysis we included mouse neurons mapping to any of the 4 L5 ET clusters (L5 ET_1 n =8 47.2 Mohm, L5 ET_2 n =4 42.88 Mohm, L5 ET_3 n = 3 46.67 Mohm, L5 ET_4 n=7 43.07 Mohm) that include both the medulla projecting (ET1) and non-medulla projecting types (ET2-4). As shown in the new analysis figure and table below now broken down at the type level, all of these mouse types have higher input resistance than human/macaque L5 ET neurons. Note these finer-grained comparisons were all statistically significant except for the human versus mouse non-medulla projecting types comparison, but this is probably just because of the low n for the human neurons (the effect size is quite large at 1.65). We also note the unique biphasic firing pattern was never observed in the mouse neurons regardless of t type. At the level of gene expression, all L5 ET types in the primate have higher expression of the Kv and Ca²⁺ channels of Fig. 4L. Taken together, many of the observations here may not reflect a Betz specialization per se, but rather a broader specialization of primate L5 ET neurons. Comparison of the morpho-electric properties of different primate L5 ET types will be an exciting avenue for future work, where it will be possible to directly link m-e-t properties to cells directly defined by projection targets by retrograde labeling and/or genetic targeting with cell type-selective enhancers.

8. *Referee 5*: Figure 4h. Very difficult to see distribution of individual cells. Can we see an inset with a zoomed view of the L5 ET cluster?. Also a bit confusing that the colors for the species are so similar to the t-types groups.

We have added a zoomed inset as requested, and we hope that the color scheme is easier to interpret with that increased resolution.

Figure 5

1. *Referee 2*: Figure 5 and associated text is focused on analyzing the methylome of motor cortex neurons, while at the same time taking into account neuronal projections. From the design, it is unclear why the authors would not have acquired the methylome of all motor cortex neurons so they could directly compare this to the transcriptome data, but here introduce the additional dimension of neuronal projections. Nevertheless, for this section in particular, making links to the other sections would be very useful and is currently not well done. Again, the key question here is: what is the added value of having the methylome over the transcriptome data, do we need this information to define cell types? Is the projection information useful in addition? These points should be made clearer at the expense of some of the unnecessary details presented here.

We have indeed generated single-cell methylomes from all motor cortex neurons. This dataset was included in the integrative analysis in Figure 2, as well as in Figure 5 b-d where it was labeled as “unbiased” and described in the text “When co-clustering them with MOp neurons collected without enrichment of specific projections, we observed a precise agreement among all of the major cell subclasses (Fig. 5b,c), demonstrating the robustness of Epi-Retro-Seq to classify cell types.”

Methylome studies enable us to investigate non-coding regulatory DNA sequences including putative enhancers (Fig. 5h) and to make inference of key transcription factors that function in each cell cluster (Fig. 5i), which provides mechanistic insights of transcriptional regulation that is unique to each cell type. Such information is not available in transcriptome data. More discussion on this topic is also included in Fig. 2 and text.

By adding the projection information in Epi-Retro-Seq, we linked the molecular properties of neurons to their connectivity. Therefore, we were able to use the projection information to annotate the molecularly defined clusters by either DNA methylome (Fig. 5g) or transcriptome (Fig. 9e). For example, a molecularly defined subcluster of L5 ET neurons is enriched for MY-projecting neurons, providing functional insights into this subcluster. Together with the electrophysiology and morphology (Fig. 4), and spatial information (Fig. 3), the combined dataset provides a comprehensive view of both molecular and anatomical properties and classes of MOp neurons.

2. *Referee 3*: Line 488 “Although neurons projecting to different target regions were not completely separated on t-SNE” Does this mean that expression pattern alone is insufficient to tell if a neuron is IT/ET or IT projecting to a given area? Is there a way to distinguish between the retrograde approach “reflecting the high quality of retrograde-labeling of neuronal nuclei” instead of just inadvertently including unlabeled nuclei in the analysis?

Thank you for pointing out that this sentence is unclear. We have removed it, as the results were more clearly described in the rest of the paragraph. In short, the IT vs ET neurons are robustly separated on t-SNE; however, IT neurons projecting to different IT areas are intermingled, so are ET neurons projecting to some of the ET structures. The mixture of these projections on t-SNE means that the predominant variance of DNA methylation does not correspond to the projection target. However, using supervised methods, we were able to make predictions of the projection target of a neuron, as described in the companion paper Zhang Z et al bioRxiv 2020.

From previous studies we know that neurons projecting to IT targets mostly fall into molecularly defined IT clusters and neurons projecting to the ET targets are usually restricted to L5 ET clusters. Thus the biased distribution of retrogradely labelled neurons across these subclasses reflects the high quality of retrograde-labeling of neuronal nuclei. We acknowledged that there are a small proportion of unlabelled nuclei (false negatives from FANS) and mislabelled nuclei (along the injection track) inevitably included in the study, and they are not distinguishable at the single-cell level. However, we reasoned that as long as the correctly labelled neurons are highly enriched compared to the noise, it would not affect our conclusions. In the revised companion paper Zhang et al., we added a quality control step to identify and remove FANS experiments showing low signal to noise ratio,

which is based on the cell distribution across on-target or off-target subclasses. All data used in this manuscript passed the quality control criterion.

3. *Referee 3*: Fig 5, one weakness of not simultaneously reviewing the Epi-Retro-Seq paper based on the new method is that the figures are more challenging. In Fig 5a, there is a box “P5” in the NeuN/GFP sorting. Is this the retrogradely labeled sample for analysis?

Yes. We added this parenthetically in the main text.

4. *Referee 3*: Fig 5c and Fig 5d. I find the results of Fig 5d clear and mostly easy to understand. All the ET projecting cells fall into a single category, two if you count the CT and ET categories in thalamus. Does this reject the Economo et al (2018) result of two separate ET-type populations? In contrast, the data presented in 5c are not as easy to grasp (perhaps because of similar colors), and it leaves the impression there is some signal from every target in every cluster! (Which the more quantitative 5d seems to reject).

Fig. 5b-d showed the analysis at subclass level, which considered L5 ET as one subclass. However, Fig. 5e-g further separated L5 ET subclass into clusters where cluster 0 is enriched for MY-projecting neurons. This is consistent with the conclusion in Economo et al 2018, that L5 ET can be divided into subtypes corresponding to MY and non-MY projecting neurons. This was further illustrated by the integrative analysis with MOp transcriptome data (Fig. 9e), and with Economo 2018 RNAseq data (EDFig. 9 in companion paper Zhang et al.).

Fig. 5c and d are plotted using the same data, where the distribution of different cell populations in Fig. 5c is quantified in Fig. 5d. Fig. 5d shows the difference of subclass distribution between IT and ET more explicitly, while Fig. 5c provides an overview of differences between projections and shows that among the IT clusters the different projections are intermingled on t-SNE in this unsupervised analysis. The color scheme of Fig. 5c is consistent with the companion paper Zhang et al. which is in press with corrections to proofs completed, so we prefer not to change it.

5. *Referee 3*: I’m not clear on the less well-defined clusters of IT neurons based on projection patterns. Zhang et al (2020 – bioRxiv, your reference 79) reads as though distinguishing neurons into different groups based on their targets from epigenomic data should be feasible (“Based on their epigenomes, intra-telencephalic (IT) cells projecting to different cortical targets could be further distinguished, and some layer 5 neurons projecting to extra-telencephalic targets (L5-ET) formed separate subclusters that aligned with their axonal

projections.”). But the overlap of ACA and S1 projecting data in 5b-d makes it look as though this is not a settled question.

Thank you for raising this important question, which points to what information is captured in t-SNE vs. a supervised method (Zhang et al). t-SNE uses principal components as input and embeds the data into two dimensions, which will inevitably lose the information from many other dimensions. The mixture of different projections on t-SNE reflects that the predominant variance of DNA methylation does not correspond to the projection target. However, using supervised methods with the projection targets as labels, we were able to capture those more subtle differences in the epigenome, which makes it possible to predict the projection target of a neuron. This is one of the conclusions in Zhang et al., although we did not discuss it in detail in this manuscript. We added a sentence in the last paragraph of the section to make this connection more explicit to the audience.

6. *Referee 5*: What is the degree of accuracy of the Epi-Retro-Seq approach? In other words, what proportion of cells are classified into t-types that do not match their projection pattern. Looking at figure 5c-d we can see for instance ‘near projecting’ and IT cells with extratelencephalic projections. It also seems that there is a cell type specific bias. L6CT cells are overrepresented in the unbiased sample (figure 5d) but are only marginally enriched in the TH-projecting sample. What does this mean for interpretation of some of the other proportional data shown in figure 5d?

While Epi-Retro-Seq captures neurons of the targeted projections, we acknowledge that there is a small proportion of contamination of neurons that are not expected to be projecting to the intended targets. The level and source of contamination varies by experiments. Generally, there are two possible mechanisms that can lead to contamination. 1) Labeling errors. When targeting a deep brain structure, for example TH, cells along the injection track that do not project to the intended target become labeled by AAVretro-Cre. 2) Sorting errors in FANS experiments, where some GFP- nuclei on the boundary between GFP+ and GFP- populations were collected. By carefully examining the cell distribution across subclasses, we identified the potential sources of contamination and removed the FANS experiments with low signal to noise ratio as a further quality control step in the revised companion manuscript Zhang et al. Specifically, we quantified the ratio between the number of cells in expected on-target subclasses (e.g. L5 ET cluster for ET-projecting neurons) vs. in expected off-target subclasses (e.g. IT clusters for ET-projecting neurons), and compared the ratio with the one expected from the unbiased data without enrichment for specific projections. This provides an estimation of signal to noise ratio of each FANS experiment. For IT projections, we used IT subclasses as on-target and L6 CT + inh as off-target, and for ET projections, we used L5 ET as on-target and IT + inh as off-target. With this strategy, we estimated that the correctly labelled cells should be at least 30 (IT) or 200 (ET) fold enriched compared to the noise in this MOp dataset, which makes us confident on the further test of enriched projections in cell clusters. To make this more accessible to the

audience, we added this statement in the text as well as a section “Evaluation of contamination in Epi-Retro-Seq” to the Methods.

Following the same rationale, compared to IT neurons (off-target), L6 CT neurons are greatly enriched in TH-projecting samples than in the unbiased sample. The relative less enrichment in L6 CT compared to L5 ET in TH-projecting samples is due to the observation that the efficiency of uptake of AAVretro is much greater for L5-ET cells than for L6-CT cells. This sampling bias does not affect any of our conclusions.

Figure 6

1. *Referee 2:* Figure 6 and associated text take a look at input and output of Mop-upper limb, as well as single cell morphology. The authors also present a number of Cre driver lines. The authors should try to link this information to the one of filling single neurons earlier in the study if possible, and they should present a coherent quantitative synthesis of target structure or input structure overlap between the different neuron types presented. While no link to the transcriptome and methylation data can be made here, at least information on layer identity and cell morphology should be commented upon.

We thank the referee for this suggestion. In our anatomy companion paper, we defined 25 projection neuron (PN) types of the MOp, based on their stereotyped connectivity, including soma locations and primary projection target regions, and we used their soma distribution to refine the laminar organization of the MOp by identifying 11 distinct layer/sublayer. In the revision of our anatomy companion paper, this new laminar organization was further validated by a quantitative analysis of retrograde labeling, cre-expression, and soma locations of reconstructed neurons. The concept of a considerably more complex laminar and cell type organization, beyond the traditional 6 (or 8) cortical layers and IT, ET and CT neuron type classification, is also supported by our MERFISH data (Figure 3), which revealed 39 clusters of PN neurons suggesting further complexity of cortical laminar organization, as well as two types of TEa-projecting neurons (L2 IT and L5 IT) with distinguished MERFISH-based molecular identities (Figure 3). The high level agreement about markedly more complex cortical organization is further supported by our single neuron reconstructions (in new Figure 7, 9), revealing unique features within defined PN types that likely reflect further organizational principles that will be revealed in future studies, for example, with new methods achieving single cell transcriptomic and single neuron morphology readout in the same tissue.

2. *Referee 3*: One of the great things about the collection of this data is that it has been done rigorously, with defined inclusion/exclusion criteria. Because most of the paper refers to mouse motor cortex as MOp, it gives the impression that most of the mouse data is collected in a stereotaxic area that is uniform across the mice. When reaching the section about “MOp projection neuron types and input-output wiring diagram”, the language shifts to specify MOp-ul. Maybe this is just to be more specific or to define a subset of MOp examined in detail for this companion paper. Is there a reason to suspect these results wouldn't generalize to MOp in general?

In the anatomy companion paper, we performed careful 3D delineations of the MOp-ul borders based on extensive multi-modality data, including Nissl-stained cytoarchitecture, retrograde and anterograde connectivity data, gene expression and cre-expression of ~30 driver lines, all of which were registered in the CCF and uploaded into Neuroglancer. Meanwhile, we also refined the laminar organization of the MOp into 11 layers/sublayers by quantitative analysis of distributions of retrograde labeling, gene and cre-expressions, Barseq, and somas of reconstructed single neurons. Consequently, all of our pathway tracing, viral tracing, and Barseq injections, as well as single neuron reconstructions, are largely centered on the MOp-ul. The other functional domains of the MOp, for example, the lower limb or mouth region, have not been analyzed at the same anatomical detail. Therefore, it is important to make a clear statement that anatomy (new Figure 7) focuses on the MOp-ul. However, it can be assumed that different MOp domains are likely to follow the same general organizing principle of a cell type-specific wiring diagram. For example, in both MOp-ul (Figure 7 here and anatomy companion paper) and the ALM (equivalent to the MOp mouth/nose), two major L5 ET types were identified, medulla-projecting neurons in the deep (or lower) sublayer while thalamus-projecting neurons in the superficial (or upper) sublayer, although their axonal terminals in the medulla or thalamus are topographically different.

3. *Referee 3*: One problem with Fig 6 is that a lot of the experiments have already been done, though perhaps not all Cre lines by the same lab in the same publication. But the overall findings are all expected – generally it felt as though the results summarized in Lines 573-605 were known to those working in this field. (“For example, the L6 Ntsr1 line revealed a typical CT projection pattern with dense projections specific to thalamic nuclei. This result is consistent with labeling from retrograde injections in various thalamic nuclei (PO, VAL, VM) and cortical areas such as MOs and SSp (Fig. 6b, top).” And “Confirming and extending previous reports 86, we characterized detailed axonal trajectories and terminations of two major types of L5b ET cells, namely medulla-projecting and non-medulla projecting neurons; both types may collateralize in the thalamus and terminate in the midbrain (Fig. 6d). Individual IT cells across L2-L6 also generate richly diverse axonal trajectories”). This raised the question of what this adds when done in the context of the other results in the paper. I felt like there wasn't a new insight in this figure. That said, the figure is a nice visual summary.

The MOp was selected through mutual agreement by the entire BICCN as an exemplar brain area to explore the potential of collaborative and integrative studies across molecular, electrophysiological and anatomical disciplines to construct a comprehensive cell type-based description of a mammalian brain structure. Our companion study provides a key complementary anatomical analysis to describe MOp cell types based on their spatial locations and projection targets, deriving a highly comprehensive cellular resolution input-output wiring diagram that anchors the molecular data within the full MOp framework. In this context, our study is an essential part of a roadmap towards creating a combined molecular and cellular description of mammalian brain architecture, enabling future studies of other brain structures at a similar data depth.

While MOp has been extensively studied, these studies lacked the possibility to directly compare and integrate different datasets within the same spatial framework. This general “piecemeal” problem of classic neuroanatomy has led to frequent disagreements between experts, including about such fundamental questions as the delineations between of the MOp areal borders (see more below). Here, our collaborative approach integrates state-of-the-art methods for anatomical labeling, imaging and computational data analysis within the mouse brain common coordinate framework (CCF) to generate an expert consensus-based, encyclopedia-like view of MOp anatomy. This includes the most comprehensive classification of cortical projection neuron types and their laminar distribution to date, derived from retrograde and anterograde tracings with nearly 300 single neuron morphologies, and precise three-dimensional delineation of the MOp-ul borders anchoring the resultant input / output connectivity diagram as an integrated model of MOp brain architecture across three scales:

(1) At the macro-scale: We define for the first time the borders of the MOp upper limb domain (MOp-ul) in 3D based on a cyto-architectural delineation using multimodal data from several traditional Nissl stainings, extensive input / output connectivity data and cell type-specific Cre-driver expression that were analyzed by multiple classic anatomy experts as well as by unbiased machine learning approaches. Importantly, our novel cloud-based Neuroglancer visualization of the eco-registered data within CCF at full spatial resolution allowed expert neuroanatomists from different labs to draw and revise borders dynamically in 3D. This is particularly significant since until now anatomists frequently disagreed about the precise regional borders, including about the MOp delineation drawn for example in the classic Paxinos versus Allen ARA versus Allen CCFv3 atlases. The combined use of these advanced technologies by multiple experts represents a new standard for building 3D brain atlases by wide research community.

(2) At the meso-scale: We defined 25 Mop projection neuron (PN) types based on their stereotyped connectivity to target regions and we used this extended PN classification and the PN somas' distribution to outline novel 11 layer/sublayer MOp organization. This refined

laminar organization was further confirmed by Barseq and single neuron morphology reconstruction data. These data thus significantly extend on the classic view of MOp anatomy, such as the classification of IT, PT, CT projection neuron types across 8 cortical sub-layers (e.g. Harris and Shephard, Nat Neurosc. Feb, 2015). Furthermore, we do not claim that our data is the “final” description of MOp projection neuron type complexity, but rather we propose that the concept of a classic laminar description of a cortical area may need to be revised to include considerably more cellular complexity in a more or less continuous gradient across the classic cortical layers.

(3) At the micro-scale: We analyzed the complexity of nearly 300 single neuron morphologies, all integrated and co-registered within the mouse brain CCF. This allowed us to derive the most detailed view of the richness of neuronal morphology and projection diversity for a mammalian cortical structure. These data strongly suggest that the true diversity of PN cortical neuron types is even more complex than the 25 PN types highlighted at the population / meso-scale level. This concept is also supported by MERFISH data recognizing 39 clusters of PN and BARseq data recognizing 18 PN clusters.

In summary, the input-output wiring diagram of the MOp-ul described in our study integrates a broad range of technologies, including pathway tracing, viral tracing, trans-synaptic tracing, molecular barcoding, single neuron reconstruction, 3D whole-brain imaging, and advanced computational analyses and data visualization tools, across all three macro, meso and micro scale resolutions, deriving the most comprehensive view of a brain structure in the mammalian brain to date. Furthermore, we believe that using our study as a roadmap and applying similar approaches across other brain structures will lead to new conceptual models of how information is organized and processed within cortical circuits to drive behavior. Taken altogether, we hope that the Reviewer agrees with us about the scientific and conceptual significance of our study.

4. *Referee 5: (Coordinate with Figure 7)* The novel genetic driver lines (Figure 6-7) are not directly aligned to the consensus types that are described in the rest of the paper. This makes it hard to interpret how well these tools give us access to the different cell types of the taxonomy put forth in the paper. The authors do acknowledge in the manuscript that this is a major future goal (line 611) but it could be discussed more thoroughly. Furthermore, the authors claim they identified 25 distinct neuron projection types based on unique combinations of axonal targets and laminar distribution. However, this classification is not brought up again or used to classify the coverage of the driver lines. The conclusion that the authors ‘provide a comprehensive identification of major projection neuron types with correspondence to molecular markers’ (line 607) therefore seems somewhat overstated.

Our genetic driver lines are designed to target the major PyN subpopulations according to their hierarchical organization. We focused on key transcription factors with demonstrated

roles in the specification and differentiation of IT, ET, CT classes and subpopulations within, which are largely consistent with the major transcriptomic subclasses. We expect that these driver lines reflect the developmental relationships among PyN subpopulations, and they will provide a reliable and orthogonal set of tools to evaluate, validate, and characterize the transcriptomic types at different levels of the hierarchy and granularity. We have now reversed the order of Figures 6 and 7 and put the driver line generation ahead of the connectivity mapping, to make the utility of the novel driver lines clearer as some of them were used for connectivity mapping from MOp.

Figure 7

1. *Referee 2:* Figure 7 and associated text describe Cre and FLP driver lines for different cell types, and introduces the dimension of development into the paper. The tools generated will no doubt be extremely useful for the community, yet one misses how it links to the rest. In particular, what do we learn from the developmental dimension, i.e. how does it link to the transcriptional cell types defined in the adult in earlier Figures of this paper? Comments on how the genes were picked to generate new mouse lines and linking them to the rest of the paper would be useful (this is only shown for two mouse lines in panel A).

We thank the referee for these comments and suggestions. We have now substantially revised this figure accordingly and moved it to Figure 6 (to be in front of the anatomy figure). For the two mouse lines shown in panels a-c, since they are for specific L5 ET types, we have now integrated them into Fig 9 where multimodal analysis of L5 ET types is presented. We have elaborated on how we selected genes for driver lines and how they relate to the transcriptomic cell types. We also included a component on developmental fate mapping, and combinatorial targeting of highly specific IT subtypes by combining lineage, cell birth order, and anatomy.

2. *Referee 3:* It is somewhat awkward to review a paper which includes “Stafford, Daigle, Chance et al., companion manuscript in preparation (Line 664). Some of the excitement of that paper is could be the future access to cell type specific mouse line. Certainly there will be interest in a L5 PT-medulla line. But also challenging as a reviewer when the full story is presented elsewhere. Part of the desire to be able to give this work a more thorough review falls from wanting to place *Npnt* and *Slco2a1* in context with the Matho et al paper (also describing strategies to get at pyramidal neurons), but doesn’t have (if I recall) a PT-medulla only line. This results in a situation where half of Fig 7 is summarizing Stafford et al, while the other in about Matho et al and the list of resources for L5 ET, for example (Fig 7e), isn’t even merged to include *Slco2*, *Npnt*, etc.

Unfortunately we are not able to complete the Stafford, Daigle, Chance et al. paper in time for this publication package. Thus we have included detailed information about the generation of the two mouse lines for specific L5 ET types (Npnt and Slco2a1) in Supplementary Methods. Taking this reviewer's advice, we have now split the original Figure 7 and integrated the two L5 ET mouse lines into Fig 9 where multimodal analysis of L5 ET types is presented. The remaining part of Fig 7 is only about Matho et al and this figure has been moved to be ahead of the anatomy figure because several of the lines generated in Matho et al have been used for connectivity mapping from MOp. We hope that this rearrangement demonstrates more clearly the actual utilization of these genetic tools.

3. *Referee 4:* Figure 7A: what are the different dot colors?

We have now integrated the original Fig 7a-c into Fig 9. Because the expression specificity of the two genes, Npnt and Slco2a1, to L5 ET types has been shown in Fig.9d, we have now removed the original 7a panel to save space.

4. *Referee 5: (Coordinate with Figure 6)* The novel genetic driver lines (Figure 6-7) are not directly aligned to the consensus types that are described in the rest of the paper. This makes it hard to interpret how well these tools give us access to the different cell types of the taxonomy put forth in the paper. The authors do acknowledge in the manuscript that this is a major future goal (line 611) but it could be discussed more thoroughly. Furthermore, the authors claim they identified 25 distinct neuron projection types based on unique combinations of axonal targets and laminar distribution. However, this classification is not brought up again or used to classify the coverage of the driver lines. The conclusion that the authors 'provide a comprehensive identification of major projection neuron types with correspondence to molecular markers' (line 607) therefore seems somewhat overstated.

We have now substantially revised this figure and related text. Here we designed driver lines for dissecting and fate-mapping glutamatergic pyramidal neuron (PyN) subpopulations largely based on their developmental genetic programs, including key transcription factors and effector genes implicated in the specification and differentiation of PyNs. We focused on key transcription factors with demonstrated roles in the specification and differentiation of IT, ET, CT classes and subpopulations within, which are largely consistent with the major transcriptomic subclasses. These driver lines reflect the developmental relationships among PyN subpopulations, thus they will provide a developmental genetic framework and an orthogonal set of tools to evaluate, validate, and characterize the transcriptomic types at different levels of the hierarchy and granularity.

Figure 8

1. *Referee 3*: Although it's presented as a new finding here, it's been published that there are L4-like cells in primary motor cortex of primates (Area 4): García-Cabezas and Barbas, Eur J Neurosci (2014). (Line 812: "L4-like neurons may also exist in human and marmoset M1.") And, as cited here, shown by Yamawaki et al, eLife (2014). Thus, it would be better to focus on how this is extending what's previously known. At least part of it is the molecular characterization (Fig 8b+). I would be interested to see how the MERFISH in Fig 8c, for example, looks if it were done on a slice extending from (more lateral) SSp to MOp to MOs (more medial), as I hypothesize – if this distribution looks like RORb, we would see cells in clearly defined layers (SSp) become less complete layers (as L4 cytoarchitecturally goes away in MOp) and then the L4 cells become less dense (in MOs).

The novel aspect of Fig 8 here is indeed the presentation of multimodal evidence, including molecular evidence, consistently demonstrating the existence of L4 cells in MOp. Please note that as stated in the text, the most specific marker gene for L4 cells is *Rspo1* (shown in the MERFISH panel in Fig 8c), whereas *Rorb* labels more broadly labels the L4/5 IT subclass of cells.

As suggested by the reviewer, we now show a MERFISH slice below, which covers a larger region of SSp, and extend from SSp to MOp and to MOs, with MOp cells shown in darker gray to mark the MOp region. The left panel shows L4/5 IT cells, the middle and right panels show the scaled expression of two L4 marker genes, *Rspo1* and *Rorb*, respectively. The L4/5 IT cells and the two marker genes indeed mark a denser and thicker L4 in SSp as compared to MOp and MOs.

2. *Referee 3*: This is beyond the scope of the present paper, but I am curious how the L4-like cells in MOp compare to those in SSp and SSs? There's a recent paper suggesting SSs L4 cells make projections outside their home column (Minamisawa et al, Cell Reports 2018). Made

me wonder which pair of L4 cells is most similar (if SSp and MOp were similar with SSs being the most transcriptionally unique).

This is a very interesting question. In our full morphology reconstruction companion paper, Peng et al, 2020, we show that both MOp and SSs L4 cells have long-range intracortical projections outside of their home region, whereas the vast majority of SSp L4 cells do not have long-range projections. In a previous publication from Allen Institute (Harris et al, Nature 2019), we also showed that L4 cells from several cortical areas (e.g. visual and auditory cortex) have long-range projections by both population tracing and single cell full morphology. Thus, the lack of long-projection in SSp L4 cells may be an exception rather than the rule.

On the other hand, in our DNA methylation companion paper, Liu et al, 2020, SSp and SSs L4 cells cluster together, whereas MOp L4 cells cluster separately. This divergence suggests that the major variables in DNA methylation profiles may be driven by factors other than long-range projections.

3. *Referee 4:* Figure 8A: It would be interesting to provide similar type of representation for all cells, not only IT neurons

Please see our companion paper, Yao et al, 2020, for similar transcriptomics-epigenomics representation for all cell types.

4. *Referee 4:* Figure 8C: Include a color scale
Color scale is included now.

5. *Referee 4:* Figure 8E: It looks like L4 and L4/5 cells are distinct. Is it fair to conclude that these are the same cell type? Is Rspo expressed to the same level in both subsets?

L4 cells belong to one cluster of the L4/5 IT subclass, thus the L4 cell type is distinct from the other L4/5 IT cell types. All L4/5 IT cells are marked by genes such as Rorb, whereas Rspo1 is mainly expressed in L4 cells.

6. *Referee 5:* Figure 8e. Add a color legend in the figure for MOp:red, VISP: blue.
Done.

7. *Referee 5*: The authors claim that putative L4 neurons mostly project upward into layer 2/3, while L2 and L3 neurons mostly project downward to layer 5 (line 799). However, looking at the examples shown in figure 8g it seems that this distinction isn't that clear cut. The authors should provide an analysis of normalized axon length per layer to back up this claim.

We have now added quantitative vertical profiles showing the average distribution of local axons along cortical depth for L2/3 and L4-like neurons to Fig 8g. Compared to their soma locations, one can see from these vertical profiles that in addition to the axon collaterals at the same level as the soma, L2/3 neurons have axon projections down to L5 whereas the L4-like neurons have axon projections up to L2/3.

Figure 9

1. *Referee 3*: The outcome highlighted (Lines 839-841) "Here as the second case study, through integrated multimodal characterization we demonstrate that L5 ET neurons in MOp can also be divided into MY-projecting and non-MY-projecting types" is also published (Economo et al., 2018) and it appears that the one class described there can be even further subdivided into two groups (Winnubst et al, 2019 – ref 26). The molecular characterization identifies one marker as *Slco2a1* for medulla and *Hpgd* and *Npsr1* for thalamus-projecting types (though these are also the markers highlighted by Economo et al Fig 1.). Is the advance that this distinction applies to MOp in addition to frontal areas?

Indeed one of the advances is that this distinction also applies to MOp. In addition, we have generated transgenic driver lines targeting these two types of L5 ET cells differentially (new figure panel h moved/modified from original Fig. 7). Moreover, we provide the most comprehensive evidence to show that properties in different modalities are aligned and support the division of these two cell types.

2. *Referee 3*: Fig 9a is not explained sufficiently for me to understand it. What's the arrow at the side? What does the color intensity mean in the plots (there's no scale/legend, perhaps correlation coefficient)? The legend claims there is good correspondence between the groups, but then the left plot makes it look as though two of the RNA-data based classes are merged in a single MERFISH group, while the last RNA-based grouping includes two of the MERFISH ones, with ET type 4's affinity not as clear.

We have now clarified that in Fig 9a confusion matrices show the proportion of cells in each integrated (SCF) or MERFISH cluster mapped to the transcriptomic types. The arrowhead

points to the MY-projecting type. What we meant by good correspondence is between MY-projecting (RNA-seq cluster L5 ET_1) and non-MY-projecting (RNA-seq clusters L5 ET_2-4) types. Indeed, within the non-MY-projecting type, there is not necessarily one-to-one correspondence between RNA-seq, integrated, and MERFISH clusters, likely due to the high degree of similarity and continuity among these clusters in each data modality.

3. *Referee 3*: Fig 9c MERFISH image legend refers to Mouse ET_types 1-4 and 5, but then only shows 1-3 and 5.

L5_ET_4 cells are not found in this particular MERFISH section. We have clarified this point in the legend.

4. *Referee 3*: Fig 9b compares mouse ET types 1-4 to three human and marmoset types. I am interested if this means that primates have less diversity in some ways in their layer 5 ET cell types? After reading Winnubst (2019) and their description of different subsets of thalamic projecting L5 neurons, I began to wonder if human L5 corticothalamic projection neurons might show a greater diversity, since the human thalamus is bigger and it might thus be possible to differentiate corticothalamic neurons projecting of many different subdivisions (of, for example, motor thalamus – where there is a much more fine division of subregions than in mice).

L5 ET cells are much sparser in human cortex (see Hodge et al. (2019) Nature), and their relative proportion appears to mirror evolutionary cortical expansion compared to the subcortical targets of those neurons. It is a very interesting question whether this sparsification means that there is less diversity in primates, and one that remains to be addressed. The sparsification means that it is difficult to sample those rare cells at adequate numbers to have comparable analyses across species to ask that question adequately. We are pursuing ways to enrich for these neurons to ask that type of question but do not have answers at this point. This is a great question about L5 corticothalamic neurons as well that will be very insightful to pursue in non-human primates where these questions of target specificity versus transcriptomic definition can be directly addressed. We hope that this comparative analysis will allow many such questions to be asked in direct, targeted ways now that the foundational classifications are becoming available!

5. *Referee 5*: The final section on MY, and non-MY-projecting L5-ET types is supposed to be a case study to ‘demonstrate the convergence of properties onto specific cell types’ (line 751). However, as the authors note, these observations on distinct molecular, morphological, and laminar distribution have been previously reported in ALM (Economo et al. 2018, Tasic et al. 2018) and have also been partially confirmed in MOp (Winnubst et al. 2019). What new insights have been gained from aligning the data from these different modalities?

We agree with the referee that the transcriptional differences between MY- vs. non-MY-projecting L5 ET neurons have been described. However, we had little knowledge about the gene regulatory information (including the epigenetic landscape and usage of genomic regulatory elements) that drives these different gene expression patterns. With the Epi-Retro-Seq data, we are now able to identify differentially methylated genomic regions that are predictive of active regulatory DNA elements such as enhancers and to computationally infer transcription factor bindings in these regions.

Figure 10

1. *Referee 3:* Fig 10. Lovely figure. The tree at the top has some of the connections labeled with different thicknesses of lines (and different colors). What is the significance of this? I am inferring it might be something about the certainty of the node or perhaps described in a different location.

The thickness and color of the lines in the hierarchical tree indicates the confidence level of each branch. Branch robustness was assessed by rebuilding the dendrogram 10,000 times with a random 80% subset of variable genes across clusters and calculating the proportion of iterations that clusters were present on the same branch.

2. *Referee 4:* Figure 10: Color scale is missing.
We have now revised Fig 10 and there is no color scale anymore. We moved the original Fig 10 which contains the correspondence matrices into Extended Data Figure 2 and added color scale there.
3. *Referee 5:* Ext Fig. 2b. What are the colors on the left? Why are these not M1-14?
The color bar along the heatmap in original Ext Fig. 2b (now Extended Data Figure 3) indeed represents the putative enhancers in different modules (from M1 to M14). We now mark them more clearly in the revised version.
4. *Referee 6:* Figure 10 struck me as more well-suited for extended data.
We have now revised Fig 10 to make it a summary of cell types in MOp. It is thus a central figure for the paper and we prefer to keep it in the main text.

III. Original referee critiques

Referees' comments:

Referee #1 (Remarks to the Author):

Reviewer's Report

This is the flagship paper from the BRAIN Initiative Cell Census Network. As stated at their website this initiative has at its goal: "...a comprehensive understanding of brain cell types [being] essential to understand how neural circuits generate perception and complex behaviors." It represents a "coordinated large-scale analyses [available as open-access] of single-cell transcriptomes, chromatin accessibility, DNA methylomes, spatially resolved single-cell transcriptomes, morphological and electrophysiological properties, and cellular resolution input-output mapping, integrated through cross-modal computational analysis." The brain, being at the pinnacle of the evolutionary tree, is complex in structure, function, regulation and diversity of cell types coordinating behavior. In the introduction the authors provide a roadmap of the findings by this initiative that spans this flagship article along with eleven companion papers. Central to this article is the leverage of modern technologies (e.g. single cell analyses) to realize an atlas of the cellular diversity and function in the brain. Table 1 presents a useful summary of experimental and computational techniques used in the present study and associated datasets. Because complex studies of this kind are littered with an alphabet soup of acronyms, the glossary in table 2 provides for the quick look-up for the general reader. Finally, in the introduction, a bullet-point list of the major findings by the consortium provides a useful summary and figure 1 a schematic of the of experimental and computational approaches of the study overall. In the result section, figure 2 summarizes the motor cortex (MOp) consensus cell type taxonomy. What follows is a detailed exposition of brain mapping and genomic and imaging analyses and the generation of CRISPR/Cas9 edited mice. The authors finally present in figure 10 an overview and integrated atlas of cell types in the primary motor cortex of mouse, marmoset and human. In the discussion, the authors pull all the strands together and provide future directions. The methods used are state-of-the-art, verified by numerous other studies.

I find no serious points of critique. The paper is well-written and relatively easy to follow. The discussion is particularly useful to the general reader.

Three minor points of critique (none are dealbreakers):

1. Although companion papers are available on BioRxiv, more explicit reference to, discussion of, and putting in context the findings of the other papers may help expand the atlas specifics and set the stage of the entire consortium's efforts.

We have added references to all companion papers in the second to last paragraph of Introduction when we first introduced the rationale, design and approaches of our study. This then leads to the final paragraph of a bulleted list of major findings which are largely derived from the companion papers. This indeed helps to integrate this flagship paper with the other companion papers and demonstrate the efforts of the entire consortium.

2. Incorporation of brain disease genetic risk variants (for Parkinson's and Alzheimer's Diseases, for example) using the genomic single cell annotations may set the new data in context of brain abnormalities. Risk variants reside largely in DNA non-coding regions containing multiple SNPs with unclear functionality – the present atlas may inform mechanisms for these disease risks, both inter- and intra-cellularly.

We share the referee's enthusiasm for intersecting genetic risk with genomic cell types and feel this will be a highly fruitful approach to identify the fine locus of brain diseases. In fact we did explore this topic in the companion snATAC-seq paper (Li, Preissl bioRxiv 2020). However, we feel this is a topic for more dedicated analyses rather than effort to classify cell types that is the overarching theme here.

3. Mechanisms involved in brain hemispheric asymmetry may be revealed and discussed using the present multidimensional atlas and could expose insight into whole-body disease asymmetry, as well as normal functions such as handedness, for example.

The topic of brain asymmetry has been approached using cruder transcriptomic methods at length with minimal findings to date. In principle such asymmetry must involve at a minimum differential connectivity, which may in turn represent differences in cell types making up those connections. The current analyses were not designed to assess hemispheric variation, but this is a clear area for subsequent exploration to see if the proportions or anatomical/molecular properties of neurons vary between hemispheres. In human however this remains a very difficult question to address due to interindividual variation and the lack of functional information (e.g. fMRI) guiding sampling within individuals and between hemispheres.

Referee #2 (Remarks to the Author):

In this paper by the BRAIN initiative cell census network, the consortium gives an overall description of a massive approach to categorize cell types in the motor cortex of different mammalian species using a variety of different approaches. I will refrain from summarizing all techniques applied and datasets acquired since it would provide just a repetition of information easily accessible in the paper several times (abstract, introduction, results, discussion).

This study presents an impressive amount of data that will no doubt be extremely valuable for data mining and as a resource for many researchers interested in cell types, cortex and neuroscience. Furthermore, it points out possible entry points for future functional studies on the role of cortex and cortical cell types in behavior and computation. A main goal of the paper in my view is not to delineate exactly how many cell types there are in the mammalian motor cortex, but to provide insight into how multi-omics information can help to consolidate cell type classifications and their stability/plasticity across species. In other words, which are the most important defining features for cell types and which others are confirmatory or contradicting the dimension of one type of classification. These questions are particularly important if one wants to argue that this work is in the long term useful not only to understand the biology of motor cortex, but also to have access to human cortical cell types with the idea to intervene in disorders where information on promoters and cell types will no doubt be of key importance, but experimental access is very limited for obvious reasons. To that point, the repetitive description of the number of cell types is difficult to follow since the cell type numbers differ significantly across different sections of the paper, which makes it confusing because one begins to wonder what these numbers really mean.

We thank the referee for this excellent point and fully agree with it. We have now deemphasized the specific number of clusters from each data type and added text in Discussion to emphasize the overall structure of cell types which is in a hierarchical organization. In this hierarchy, the higher branches (cell classes and subclasses) are the most robustly distinct from each other, the most conserved across species, and have the most consistent properties across all modalities examined. The numbers of classes and subclasses are also invariable. At the lower branch or terminal leaf level (clusters), the distinction among clusters can be less clear and more continuous, and thus the exact numbers of clusters are less meaningful. The cross-modality correspondence at the leaf node level can also be weaker and differential variations among different modalities can be observed.

As we are at the early stage of defining cell types, we do not intend to use a single modality of features to define cell types and treat the others as only providing supporting evidence. Rather, we strive to characterize cells in as many different ways as we can, and use cross-modality consistency to define consensus cell types. Through this process, we will also begin to learn which modality is

more predictive of the consensus cell types. On the other hand, practically at the current stage, it is likely that data types/modalities with substantially richer feature contents (e.g. transcriptomics) will be better suited for defining and predicting cell types than other data types containing fewer features due to technical limitations (e.g. electrophysiology). Another approach is to define cell types based on molecular genetic programs as they are underlying the origin of cell types. In this respect, an important issue to address is whether cell types defined by molecular genetic programs are also functionally specific (in other words, whether they are consistent with functionally defined cell types).

Given the massive amount of different approaches, groups conducting the work and strategies to determine cell type diversity, this paper aims to be a top level paper prompting readers to other studies, each describing a facet of the big project in more details. The value of this paper beyond pointing people towards these other studies is to present and synthesize a coherent picture with the main messages conveyed by the other studies that people can then read to understand the experimental and computational basis for the claims. This study should be able to synthesize all main findings of these combined studies with Figures well capturing these findings and messages, so that readers who are less interested in the details can extract the main take home messages of this large enterprise.

Through parts of this paper, the authors do a very good job in providing these top level insights, but there is room for improvement at several places. It would be very useful if the authors could still work towards making these main insights and the arguments for how they reached them more accessible to readers only interested in getting an overview picture of the findings. This should be at the expense of some detailed information that might be less interesting for the general reader. I will comment on suggestions sequentially according to Figures shown.

We agree with the referee on this point. Indeed, our focus of this flagship paper was to synthesize the main findings from the other studies described in the core companion papers together. We did this by starting with a succinct list of key insights we obtained at the end of Introduction, then highlighting the major findings in each modality of cell type characterization through a series of figures in the Results section, and finally providing a synthesis of what we have learned about cell types through this unprecedentedly comprehensive effort in Discussion. In the revised manuscript we have now further simplified the detailed descriptions and discussed a number of key points raised by the referees that are indeed central to the understanding of brain cell types in Discussion.

(1) While the 11 companion papers are cited throughout the paper, I suggest to include a separate listing of References to go with this paper on these 11 studies. There is currently Table 1 on the applied methods citing background and companion papers but since they are not cited in sequence, one always has to go back to the reference list and extract them. Another option is to put these 11 references in sequence when this concept is first mentioned in the introduction, so they would

appear in one block and have consecutive citation numbers. Similarly, if there was a way of displaying Ref numbers and which panels refer to which Figures in this paper in Figure 1, it might help the reader to follow the flow.

We have now cited the 11 companion papers together when they are first mentioned in Introduction. This should make it easier to find them.

(2) Data presented in Figure 2 and associated text aims to determine how many cell types there are in mouse motor cortex, and then compare these cell types to human and marmoset cell types. The accompanying text should make this point at the start and perhaps at the expense of some of the details on mouse profiling give insight into the criteria for concluding which and how many cell types are conserved and why.

We agree with this helpful suggestion and have edited accordingly.

The authors just very shortly say that over- or underclustering may be issues but not how this is resolved. It is important that readers can follow how cell types are defined across species, much more so than the details of how many nuclei were profiled in mouse. The same goes for marker definition of main cell types and across species – what are the markers and how conserved are they? These are the main points that should come across in this section.

We have tried to make this more clear in the manuscript. We provide a brief description of the definition of cross-species consensus types in the Methods section and a detailed description in our companion paper Bakken et al. 2020. Briefly, RNA-seq data from all 3 species were integrated with Seurat and unsupervised clustering identified several hundred metacells that were merged into 45 consensus clusters that contained cells from all species. These consensus types are shown as boxes in Extended Data Figure 1 in this paper.

The following has been added to the Methods: “Marker genes were identified for each species’ cluster using Seurat’s FindAllMarkers function with test.use set to ‘roc’, >0.7 classification power.” We provide these markers and conservation as Supplementary Tables 7 and 10 in Bakken et al. 2020. Neuronal subclasses had 50-450 marker genes in each species, and up to 20% of these were conserved across all species, as shown in the barplots below. These results have been added to the manuscript text.

(3) Using MERFISH, the cell type number presented in Figure 3 and associated text is quite different from the one presented before. It is unclear how the genes that were studied here have been chosen. One would have thought that in a logical flow, the genes to study should have arisen from the data presented in Figure 2, but this does not seem to be the case. The authors at the very least should comment on the overlap of genes used in the two approaches. The fact that the cell type number differs needs an explanation. Both methods are based on transcripts, and ideally should result in matching data and cell type classification.

The genes used in the MERFISH experiment and presented in Figure 3 were indeed selected based on the mouse sc/snRNA sequencing dataset which was presented in Figure 2. However, the MERFISH gene panel included 258 genes, which is fewer than the number of variable genes (3792 genes) used for clustering in the sc/sn-RNAseq data. The methods we used to select the MERFISH gene panel are described in detail in the Methods section of the companion paper (Zhang M et al, biorxiv2020). Briefly, the 258 genes consist three subsets of marker genes which were selected by three approaches: (I) Canonical marker genes for the major cell types in the cortex selected based on prior knowledge; (II) Differentially expressed (DE) genes selected based on pairwise differential gene expression analysis of the neuronal clusters identified by the sc/snRNA-seq data; (III) A set of genes which contained the highest mutual information (MI) among the clusters identified by sc/snRNA-seq. Of the 258 genes selected, 91% are among the variable genes used for clustering in the sc/sn-RNAseq data.

As for cell cluster number difference, as we describe above in the response to various reviewer points, the cell classification at the class and subclass levels are more robust, while the cell classification at the lower level (cluster) level are more variable, likely due to less distinctness and more continuity among the clusters within the same subclasses. Hence the exact number of clusters are less meaningful than the number of subclasses and we do not expect the independent clustering analyses of MERFISH and sc/snRNA-seq to generate the same number of clusters. As shown in our response to the review point above, we observed excellent agreement between the subclasses

identified by the two methods, and good correspondence between the clusters identified by the two methods.

(4) Figure 4 and associated text presents a heroic effort to patch many cortical cells in different species, subsequently determine their morphology and assess the transcriptome. It would be useful if instead of showing single example cell ephys data in panel d, the authors provided quantitative differences between what they call a cell type based on physiology and whether they in addition take into account transcriptome data. In other words, what does morphology and/or ephys traits add to the transcriptome classification we already know? Does morphology and ephys lead to more diversity, or does it go hand-in-hand with the transcriptome data? The authors should synthesize this new data with previously presented data.

The referee asks an excellent question that is key in understanding the logic behind the diversity of cell types. Our consortium has recently published an article in *Nature* online (companion paper Scala, Kobak et al., 2020) which is focused on relating transcriptomic, electrophysiological and morphological characterization of neurons. We found that at the level of broad families of transcriptomic types (e.g. Vip+, Pvalb+, Sst+ etc) there is a very good match between morpho-ephy traits and transcriptome. However, within each family there is a diversity of morho-ephy properties and trascriptomic types and within a family transcriptomic cell types are not well separated in the morpho-ephy space. However, we also find that neighbouring transcriptomic cell types within each family show more similar morpho-electric features. Therefore, in Scala, Kobak et al. we suggest that “the tree of cortical cell types” may look more like a banana tree with a few large leaves, rather than an olive tree with many small ones where neurons follow a hierarchy consisting of distinct, non-overlapping branches at the level of families (large leaves), but with a spectrum of cell forming continuous and correlated transcriptomic and morho-ephy landscapes within each leaf. Due to lack of space we decided to demonstrate the richness of the data and refer to details in Scala et al., and not to repeat the figures demonstrating these findings here. Another important role the Patch-seq data plays is to link transcriptomic types with traditionally recognized cell types that are defined by morphological and electrophysiological properties which provide a Rosetta stone relating these properties. This is illustrated in Figure 10.

(5) Figure 5 and associated text is focused on analyzing the methylome of motor cortex neurons, while at the same time taking into account neuronal projections. From the design, it is unclear why the authors would not have acquired the methylome of all motor cortex neurons so they could directly compare this to the transcriptome data, but here introduce the additional dimension of neuronal projections. Nevertheless, for this section in particular, making links to the other sections would be very useful and is currently not well done. Again, the key question here is: what is the added value of having the methylome over the transcriptome data, do we need this information to define cell types? Is the projection information useful in addition? These points should be made clearer at the expense of some of the unnecessary details presented here.

We have indeed generated single-cell methylomes from all motor cortex neurons. This dataset was included in the integrative analysis in Figure 2, as well as in Figure 5 b-d where it was labeled as “unbiased” and described in the text “When co-clustering them with MOp neurons collected without enrichment of specific projections, we observed a precise agreement among all of the major cell subclasses (Fig. 5b,c), demonstrating the robustness of Epi-Retro-Seq to classify cell types.”

Methylome studies enable us to investigate non-coding regulatory DNA sequences including putative enhancers (Fig. 5h) and to make inference of key transcription factors that function in each cell cluster (Fig. 5i), which provides mechanistic insights of transcriptional regulation that is unique to each cell type. Such information is not available in transcriptome data. More discussion on this topic is also included in Fig. 2 and text.

By adding the projection information in Epi-Retro-Seq, we linked the molecular properties of neurons to their connectivity. Therefore, we were able to use the projection information to annotate the molecularly defined clusters by either DNA methylome (Fig. 5g) or transcriptome (Fig. 9e). For example, a molecularly defined subcluster of L5 ET neurons is enriched for MY-projecting neurons, providing functional insights into this subcluster. Together with the electrophysiology and morphology (Fig. 4), and spatial information (Fig. 3), the combined dataset provides a comprehensive view of both molecular and anatomical properties and classes of MOp neurons.

(6) Figure 6 and associated text take a look at input and output of Mop-upper limb, as well as single cell morphology. The authors also present a number of Cre driver lines. The authors should try to link this information to the one of filling single neurons earlier in the study if possible, and they should present a coherent quantitative synthesis of target structure or input structure overlap between the different neuron types presented. While no link to the transcriptome and methylome data can be made here, at least information on layer identity and cell morphology should be commented upon.

We thank the referee for this suggestion. In our anatomy companion paper, we defined 25 projection neuron (PN) types of the MOp, based on their stereotyped connectivity, including soma locations and primary projection target regions, and we used their soma distribution to refine the laminar organization of the MOp by identifying 11 distinct layer/sublayer. In the revision of our anatomy companion paper, this new laminar organization was further validated by a quantitative analysis of retrograde labeling, cre-expression, and soma locations of reconstructed neurons. The concept of a considerably more complex laminar and cell type organization, beyond the traditional 6 (or 8) cortical layers and IT, ET and CT neuron type classification, is also supported by our MERFISH data (Figure 3), which revealed 39 clusters of PN neurons suggesting further complexity of cortical laminar organization, as well as two types of TEa-projecting neurons (L2 IT and L5 IT) with

distinguished MERFISH-based molecular identities (Figure 3). The high level agreement about markedly more complex cortical organization is further supported by our single neuron reconstructions (in new Figure 7, 9), revealing unique features within defined PN types that likely reflect further organizational principles that will be revealed in future studies, for example, with new methods achieving single cell transcriptomic and single neuron morphology readout in the same tissue.

(7) Figure 7 and associated text describe Cre and FLP driver lines for different cell types, and introduces the dimension of development into the paper. The tools generated will no doubt be extremely useful for the community, yet one misses how it links to the rest. In particular, what do we learn from the developmental dimension, i.e. how does it link to the transcriptional cell types defined in the adult in earlier Figures of this paper? Comments on how the genes were picked to generate new mouse lines and linking them to the rest of the paper would be useful (this is only shown for two mouse lines in panel A).

We thank the referee for these comments and suggestions. We have now substantially revised this figure accordingly and moved it to Figure 6 (to be in front of the anatomy figure). For the two mouse lines shown in panels a-c, since they are for specific L5 ET types, we have now integrated them into Fig 9 where multimodal analysis of L5 ET types is presented. We have elaborated on how we selected genes for driver lines and how they relate to the transcriptomic cell types. We also included a component on developmental fate mapping, and combinatorial targeting of highly specific IT subtypes by combining lineage, cell birth order, and anatomy.

(8) The last figures show examples of integrative analysis across the different information channels, yet one lacks information on how starting from one source of information and adding more data contributes to the certainty of assigning cell types. The case studies presented read a bit like a self-fulfilling prophecy when one would rather like to know how important which contribution is to the cell types profile, e.g. are we 80% certain with the transcriptome what cell type it is or do we need the single cell morphology and ephys data because we are only 30% sure after the transcriptome? If some sort of synthetic analysis like this would be presented at the overall level (e.g. for these X cell types, we are 80% confident with just the transcriptome, morphology adds another 10% etc, for these Y cell types...present in form of table should be possible), this would be extremely valuable, and would interest the general reader much more than individual case studies which might be more interesting for a specialized reader (in addition, the case studies are rather confirmatory than providing novel insight). The same comment holds for the certainty of cell type identification across species, in particular humans, which will be key for possible medical applications.

By conducting multimodal and cross-species studies, we can now say that we are near 100% certain about the top branches of the transcriptomic taxonomy - the classes (e.g. glutamatergic and

GABAergic) and subclasses (within GABAergic: Lamp5, Sncg, Vip, Sst-Chodl, Sst, Pvalb; within glutamatergic: L2/3 IT, L4/5 IT, L5-6 IT, L6 Car3, L5 ET, L5/6 NP, L6 CT and L6b). Below that, several major types can also be accurately predicted (near 100%) by transcriptomics and confirmed by morphology and projection (e.g. Pvalb-Vipr2 chandelier cells, L5 ET_1 medulla-projecting cells), while in other cases the distinction between neighboring transcriptomic clusters is less clear. Additional evidence may break the continuity, e.g. L4/5 IT_1 type turns out to be L4 cells, Sst Hpse type represents those Sst cells with L4-targeting axons. But more often additional evidence also suggests that groups of neighboring clusters exhibit similar morphological, electrical and projectional properties with continuous variations (Fig 10), based on our Patch-seq studies (companion paper Scala et al bioRxiv 2020; Gouwens et al Cell 2020), retrograde tracing studies (companion papers Zhang M et al bioRxiv 2020, Zhang Z et al bioRxiv 2020) and full morphology studies (companion paper Peng et al bioRxiv 2020). It should be noted that our phenotypic characterization is not exhaustive, future studies may further resolve the difference among clusters. Thus, our overall conclusion is that for cell subclasses and some highly distinct cell types, transcriptomics-based definition is highly accurate and predictive of other properties, but at more fine-grained levels one needs multimodal information to define cell types with proper granularity.

Referee #3 (Remarks to the Author):

Comments on “A multimodal cell census and atlas of the mammalian primary motor cortex”

The huge range of dataset going into the companion papers are obviously impressive and will be of great use to the community at large. These are collected by experts in the field and the experiments are done to high standards to include the data.

Reviewing this paper is a challenge since (a) the range of techniques in the companion papers is so broad as to make it tough to be an expert in all of them, (b) some of the methods in the companion papers are novel and it is uncertain whether they should be evaluated here or by reviewers for the specific papers, and (c) the concept of a flagship paper and what it should or should not include is new to me and I am not sure what to evaluate. Is it a review? Is it primary literature? I come away feeling the main point is for the reader to leave the paper understanding the basis for the Mouse Consensus Transcriptomic Taxonomy and its relationship to the primate cell types. It is my guess that, once published, those companion papers will be the standard citation for the applicable individual datasets. My internally generated expectation, then, is that this paper must:

- present a basis for the hierarchy of cell types in these three species (“Justify Figure 10”);
- summarize those results in a manner that makes them accessible to a broad audience; and
- emphasize the new findings from such a large body of work

We appreciate the referee's assessment of the relationship between this flagship paper and the other companion papers. We have now organized our Discussion section according to these suggestions.

So, does it work? I think it mostly does. Though I provide a long list of minor comments, these are intended to demonstrate my respect for and interest in the work instead of as obstacles to publication. I think I understand the basis for the hierarchy. I am less convinced that the results were summarized and accessible. The paper feels at times like each companion paper got a figure to show its highlights and a section in the results, but without making the results section feel like it came from a single author (probably not unexpected with so many contributors). I think this would be improved if the explanations were briefer and provided at a higher level (this assumes all the companion papers get published). Lastly, I feel like the emphasis on new findings was limited because some of the points called out felt like they were already known. So, what is the great insight that we get? I think it's the hierarchy.

We believe that in the flagship paper we do need to introduce and summarize the different types of studies from the companion papers and the major conclusions from them, in order to build up the cases for our overall perspectives on cell types which cover a wide range of issues. We would like to point out that each figure is a high-level summary of the major findings from a particular study (which by itself is already a large-scale one). Nonetheless, we have now tried to simplify the descriptions of these individual studies as much as we can.

We agree with the referee that the great insight is indeed the hierarchy of the cell type organization, and how different modalities of cell type properties are related to each other at the different levels of this hierarchy. We have discussed this point in detail in our responses to referees' comments above, so will not repeat here. We have now also elaborated this point in the Discussion section of our revised manuscript.

Overall, I would recommend publication or publication with minor revision.

Major point:

Lines 45+: Would you be willing to venture a definition of "cell types"? The presentation of the data seems influenced in part by biological systematics. In that field, each of the terminal taxa are generally species. Maybe there's not 100% consensus about what defines a species, but Ernst Mayr was effective in popularizing a biological definition (populations that can reproduce with one another AND are reproductively isolated from others, for example). Even if not everyone agrees, can

we advance a definition so we have something to argue about? (I mean something like: 'All the cells of a given cell type must be located in a similar standardized anatomical location/cortical area/layer in the brain of that species, share inputs and deliver outputs to the same targets, have similar intrinsic excitability, and share similar transcriptional and epigenetic state.' And then, 'similar' means ...) Or is it simply that cell types are 'cells that group together at the lowest possible hierarchical level in the integrated transcriptomic classification' with the other features (morphology, connectivity, etc) only being found in cells whose transcriptional state matches? Defining this could address whether cells in different cortical areas are the same type and whether similar cells in different layers are the same type. (Are layer 2/3 and layer 5 fast spiking PV+ interneurons the same cell type? Seems like yes in Figure 10 but no by my definition above. Are medulla-projecting layer 5 pyramids in MOp, MOs, or SSp the same cell type?) Perhaps I am rude for asking, as those paragraphs address the issue but don't come to a snappy conclusion.

(For a contrasting definition, from your reference 105: "Here, we define a cell type as 'a set of cells in an organism that change in evolution together, partially independent of other cells, and are evolutionarily more closely related to each other than to other cells'. That is, cell types are evolutionary units with the potential for independent evolutionary change." I find this definition not as helpful for neuroscience.)

We believe cell types should be defined by taking into account all these factors: evolutionary/developmental origin, correspondence across modalities, and anatomical location (region, layer, etc.), but place relative emphasis in such order. These factors interact with each other and their relationships will be reflected in the hierarchical organization of cell types.

We propose that cell types of the cerebral cortex are best explained by a combination of specification through evolutionarily-driven and developmentally-regulated genetic mechanisms, and refinement of cellular identities through intercellular interactions within the networks the cells are embedded in. The genetic mechanisms drive intrinsic or cell-autonomous determination of cell fate, and progressive temporal generation of cell types from common progenitor pools that explain global similarities and continuous features of cellular phenotypes that reflect developmental gradients. Network influences can drive further phenotypic refinement that is not reflected in the adult genetic signature, for example for axonal projections and synaptic connectivity that may reflect transient or stochastic developmental events, region-specific and/or activity-dependent refinement. Such an evolutionary, developmental and experience-dependent framework largely explains the overall hierarchical organization, strong conservation across species, and strong correlation among cellular phenotypes to a certain level with further network-driven or experience-dependent refinement.

Minor points (sorry for the length):

Abstract, Line 15: What does “congruently” mean here? The authors’s collective vocabulary is better than mine, but I would take it to mean a perfect superposition of one thing onto the other (which would be unlikely to occur in transcriptomic data).

We have removed this word. “Unified molecular genetic landscape” already conveyed what we need to say.

Introduction, Line 69: The number of citations in the intro are useful for context, but likely could be shorter for publication purposes (examples not needed from all areas where this technique was effectively used ... cortex, hippocampus, hypothalamus, etc.).

We have removed the mentioning of hippocampus and hypothalamus and the associated references to shorten the text.

The first major finding is that “Combined single-cell transcriptomic and epigenomic analysis reveals a unified molecular genetic landscape of adult cortical cell types that integrates gene expression, chromatin state and DNA methylation maps.” I am wondering what this bullet means? Transcriptomic data give some idea of which genes are being expressed (and how highly expressed) at a given time – so I understand this will help clarify what needs to be expressed differentially to define cell types. But epigenomics is modifications of the DNA that can be expected to affect gene expression, so it is not surprising that these would describe a similar landscape. Similarly, chromatin state is expected to vary with gene expression, ‘opening’ up the chromatin to permit expression. DNA methylation might act, for example, to suppress gene expression. Perhaps there is some deeper meaning here, but this bullet seems redundant to me.

The reviewer is correct in thinking that, in principle, transcriptome and epigenome are expected to be concordant. However, this point cannot be taken for granted (“not surprising”) for technical and conceptual reasons. From the technical perspective, the transcriptome, chromatin, and DNA methylome datasets are generated by different groups using different technical platforms. There are multiple reasons that the outcome may not be consistent in defining cell types. The fact that they do validate our collective methods and approach, and suggest strong underlying biological mechanisms. Therefore, this outcome does not justify the notion that using multi-modality information is redundant.

From the conceptual perspective, the epigenome provides two additional information sources to the transcriptome. First, the epigenome contains developmental signatures that can persist long after the expression has ended. One specific example at the *Lhx9* locus is given in the companion paper

(Yao et al bioRxiv 2020). This developmental gene expression is undetectable in the transcriptome but shows cell-type specificity in the epigenomic profiling. Also, the methylation companion paper (Liu et al bioRxiv 2020) shows that putative developmental signatures of past gene regulatory events account for 65% of all DMRs, with only 35% of DMR being associated with open chromatin and gene expression, in agreement with previously described using bulk samples of purified neural populations (Mo 2015).

Second, the whole genome-wide epigenomic profiling contains regulatory elements in the intronic and intergenic regions, which is considered during clustering. Our companion paper (Yao et al bioRxiv 2020) provides a comprehensive overall result to help understand to what extent the regulatory genome agrees with transcriptome when determining cell types, which is not a foregone conclusion. This kind of integrated analysis is technically challenging and important to help understand the underlying regulatory mechanisms, including differential peaks, potential regulators (TF/motifs), and gene-enhancer links found in specific cell types.

Therefore, although the transcriptome provides the most direct measurement on gene expression at the given time, including chromatin state and DNA methylation maps is still necessary to define a unified molecular genetic landscape of brain cell types. Thus this bullet point represent a major conclusion of the paper.

More subtly, it will be interesting to understand how “continuously varying properties” (Line 161) in transcriptomics relate to physiology and morphology. Or how to distinguish whether there are correlated or causal. Although I think this alludes to a gradient within a given cortical area, such as from pia to white matter within a given layer (as discussed in Berg et al., 2020 bioRxiv for some L2/3 cell types), I would also be curious if anything is known about the tangential gradient as we move across topographical areas in MOp – from limb to face to trunk. It would seem there might be some gradient needed to help maintain topography in connections to homotopic areas within M1 (well at least in primate) or other motor cortical areas or long-range targets.

Indeed, the continuous variation in transcriptomics does correlate with variation in morpho-electrical properties (Scala et al bioRxiv 2020) and spatial distribution across layers by MERFISH (Zhang M et al bioRxiv 2020). We also observe more extensive variation in long-range axon projection patterns within transcriptomic clusters (Peng et al bioRxiv 2020). Our current studies do not distinguish whether these are correlated or causal.

We have examined tangential variation along anterior-posterior and medial-lateral axes in MOp by MERFISH, and found spatial distribution variations along these axes for some glutamatergic clusters as described in more detail in the companion paper (Zhang M et al bioRxiv 2020).

Another example of continuity versus discreteness is seen in the human cortex, where the expansion of the supragranular layers, consisting of layer 2 and 3 IT excitatory neurons, is also seen as an extension of the IT gradient described in mouse cortex. In the Berg et al bioRxiv 2020 companion paper we describe that there is a general depth-dependence to the IT neurons as a whole along that transcriptomic axis with corresponding changes in neuronal size related to the corresponding length of the apical dendrite. However, we find that there are discriminable cell types that are phenotypically distinct at the ends of this global transcriptomic gradient. Only one of the types shows a very strong within-type gradient, and we show that this transcriptomic gradient is accompanied by similar gradients in anatomical and physiological features. Thus there appears to be a higher-level organization within the transcriptomic space for IT types, with elements of both continuity and separable discreteness that differ for different types.

What is the molecular specialization of corticomotoneuronal neurons? These are famous in primates and possibly missing in rodents (but see Grinevich, Brecht, and Osten J Neurosci 2005). Do these occur in the data?

We explore this topic on the Betz cells in Figure 4 of the manuscript and in the companion paper (Bakken et al bioRxiv 2020). The primate Betz cells are contained in the layer 5 ET subclass and are homologous to the mouse L5 ET subclass containing the corticospinal projecting neurons. We describe that some of the features of the L5 ET neurons compared to IT neurons are highly conserved, whereas others like size and physiological properties are specialized in primates. As the referee mentions there may be other species specializations that we did not measure in this neuron subclass, such as whether their axons directly target spinal motoneurons (primates) or not (mice). To look for molecular correlates to species specializations we looked for primate-specific molecular differences or features that varied as a function of evolutionary distance. Remarkably, genes associated with axon guidance showed such species differences, providing hypotheses about molecular mechanisms underlying differential targeting of spinal motoneurons versus interneurons.

Similarly, “Cell type transcriptional and epigenetic signatures can guide the generation of an extensive genetic toolkit for targeting glutamatergic pyramidal neuron types and fate mapping their progenitor types.” (Line 171) I agree, but the idea that this is possible is not a new finding (probably more examples, but Visel et al., 2013 Cell among others). Similarly, Introduction, Line 87: Is there earlier literature (than the bioRxiv references 33-39) from other groups that have used enhanced-based approaches to target for genetic access to specific cell types in mice? I am not an expert in these approaches but I think I have seen seminars exploiting enhancers used in published mouse models since at least a couple years ago. This bullet in my opinion belongs with the publication describing the models in which this is described.

The idea of using genetic elements to target cell types has been around for several decades. However, how to achieve the level of specificity and comprehensiveness that are necessary for

dissection cortical circuits has been an enduring challenge. Currently, there are only a handful of mostly BAC transgenic lines available for targeting several pyramidal neuron populations, and even fewer driver lines for neural progenitors. Here, we generated over a dozen temporally inducible mouse Cre and Flp knock-in driver lines to enable combinatorial targeting of major progenitor types and projection classes. Intersectional converter lines confer viral access to specific subsets defined by developmental origin, marker expression, anatomical location and projection targets. These strategies establish an experimental framework for understanding the hierarchical organization and developmental trajectory of PyN subpopulations. We have now substantially revised the text and figure (new Fig 6) to highlight these advances, which are reported in-depth in a revised companion paper (Matho et al bioRxiv 2020).

Regarding the use of enhancers for targeting cell types, again the idea has been around for decades but the progress has been very limited, especially for cell types in the mouse brain. Visel et al., 2013 used putative enhancer regions to label different embryonic progenitor regions, not cell types or progenitor types. These transgenic lines are only assayed in embryonic tissues and are not maintained. One of the key challenges is to systematically discover bona fide neuron type enhancers in the adult brain. Indeed, there has been recent progress in developing enhancer-based AAV vectors to target adult brain cell types, including work from researchers such as Gord Fishell, Michael Greenberg and colleagues, as well as some of us within the BICCN (the cited bioRxiv papers are now published). We have now cited this body of work more comprehensively in Introduction.

The key point we are trying to make here is that the current joint transcriptomic and epigenomic analysis provides an unprecedented opportunity to achieve comprehensive access to a wide range of brain cell types, building on top of previous pioneering work. A major effort is now underway in the BICCN to systematically test and validate a larger number of these enhancer elements. The intended specificity and scale of this effort have not been attempted before.

“Comparative analysis of mouse, marmoset and human transcriptomic types achieves a unified cross-species taxonomy of cortical cell types with their hierarchical organization that reflects developmental origins; transcriptional similarity of cell type granularity across species varies as a function of evolutionary distance.” (Line 151+) It would be useful to know if there are exceptions to this rule for evolutionary distance. The example given is “Three non-neuronal types had greater [S]pearman correlations of overall gene expression (Fig. 2d, right columns) between marmoset and mouse likely because non-neuronal cells were undersampled in human M1 resulting in fewer rare types.” (Lines 239-242) but the explanation included suggests these exceptions weren’t evaluated much. This is not unexpected, but when the finding is one of the major bullets and contradictory data isn’t assessed in great detail (because it’s believed to be wrong), then the finding is not so surprising.

We agree with the reviewer that these exceptions to the rule are important to study. In light of another referee comment, we have used a more robust method (pseudo bulk analysis) that treats donors rather than individual nuclei as biological replicates to define differentially expressed genes. Using this refined analysis, we find that endothelial cells and microglia are more similar between human and marmoset, as expected. Only vascular and leptomenigeal cells (VLMCs) are more similar between marmoset and mouse. VLMCs were highly undersampled in human (n = 40 nuclei; 1 type) compared to marmoset (n = 463; 4 types) or mouse (n = 2329; 7 types), and it is likely that average expression is not adequately estimated in human, which reduces the correlation with marmoset. Alternatively, there could be evolutionary changes in the expression or relative proportions of VLMC subtypes in human that we will be able to resolve in future work with deeper sampling of rare non-neuronal cells.

How should we interpret that many marker genes for glutamatergic subtypes are species specific? In that context, it is a bit inconsistent with the overall pattern of IT, ET, CT, and NC types that do seem to be conserved. Does this mean that no specific marker is needed to make such types, but a suite of them that is robust to substitutions?

Each glutamatergic subtype (and other cell types) expresses thousands of genes, and a large fraction of these genes are expressed in all three species, which enables identification of cell type homologies. For each species, approximately 50 to 350 genes have highly enriched expression relative to other subclasses and are considered markers, and as we reported only a minority of these markers are conserved across species (see barplots below). While seemingly at odds, it is quite possible for there to be both strong homology and also for there to be a great deal of species specialization as for many other species phenotypes.

It would seem likely that those highly cell type selective genes that are highly conserved are of particular importance to their identity and canonical phenotypes. We hypothesize that conserved markers are under strong evolutionary constraint and are causally related to the core identity of each subclass. For example, in our companion paper (Bakken et al bioRxiv 2020), we identify a transcription factor *RORA* that is a conserved marker of chandelier cells across 3 species and has enriched binding sites in open chromatin. Interestingly, non-conserved markers are often still

expressed in other species but have lost their subclass specificity, as shown in our companion paper for GABAergic subclasses (ED Fig. 3a, Bakken et al.).

Introduction, Line 109-111: The style of present day articles seems to include presaging the results in the end of the introduction, but here you cite the first figure and table! Only an opinion for style purposes, but to me the results begin here. “Here we present the cell census and atlas of cell types in one region of the mammalian brain, the primary motor cortex (MOp or M1) of mouse, marmoset and human, through an analysis with unprecedented scope, depth and range of approaches (Fig. 1, Table 1).”

We appreciate the referee’s point here. However, since we want to state the major findings (in bullet points) from this massive paper in Introduction first before diving into the detailed descriptions in the Results section, we need to briefly introduce the design and approaches taken in our study BEFORE stating the major findings so that readers can better understand the overall picture of our study. Thus, we have decided to keep the Introduction section as is.

Table 1 is useful as a reference, but takes up a lot of space (in particular since some cells are large in only a subset of rows). I think it is more useful for the reader to refer back to while reading the results of the paper and not (as currently set) right after the introduction when the results are yet to be detailed.

We respectfully disagree and feel, as did other referees, that this table provides an important grounding for the datasets that were used in the consortium effort as a whole, particularly given the wide breadth of methods used and species they were applied to. We are confident the space taken will be dramatically reduced upon typesetting so that it will not appear as distracting in the final version.

How Table 2 is organized might be improved? If truly a Glossary, terms could be listed alphabetically. Furthermore, the use of multiple columns is unclear (e.g. not just a term and its definition). If columns are meant to be interpreted as hierarchical, why MOp and its layers would be in the same column. The hierarchy seems present to some extent in the inhibitory and excitatory subclasses.

Table 2: Explanation of the developmental significance of MGE and CGE given in a table defining terminology seems out of place.

Table 2: “The top branches of the CN transcriptomic cell type hierarchy comprising neuronal and non-neuronal cells.” Does ‘theCN’ mean ‘the BICCN’?

We appreciate the helpful suggestions on Table 2 organization and have attempted to improve the hierarchical organization. “theCN” was an unfortunate typo that has been corrected.

We have revised the table significantly to make it smaller and clearer in its use of hierarchical organization.

Table 2: Use of terms ET for extra-telencephalic and CT for corticothalamic. I imagine with such a large consortium, the field is well served to converge on a single nomenclature and perhaps this will be the one, but as the thalamus is a subset of the extratelecephalic structures cortex might target, this naming scheme has a small error. CT could be a special subset of ET, though these read as though intended these to be thought of as two separate groups. Similarly the IT definition might be refined to state “only”, and in “Excitatory glutamatergic neuron that projects only to other telencephalic structures” (since some ET neurons might send a collateral). There’s a lengthy description of why ET is chosen to replace PT in the supplement, which is useful, though its placement in the supplement is a problem (since it won’t be as well read).

We understand that CT could be a special subset of ET, that’s why we did not propose to use ET and CT to name the specific subclass of neurons defined transcriptomically. Instead, we proposed L5 ET and L6 CT as names for these transcriptomic subclasses, using the predominant layers these neuron types reside in to define them in a mutually exclusive manner. We agree that adding “only” to the IT definition is better to distinguish it from ET and CT cells because the latter also have intracortical collaterals, and we have now done so. We thank the reviewer for pointing out putting the ET nomenclature in the supplement is a problem as it won’t be as well read. Thus we have moved it to the Online Methods section so that it will appear in the downloadable PDF file of the paper.

Results, Line 217-223: “These types were grouped into broader subclasses based on shared developmental origin for GABAergic inhibitory neurons [i.e., three caudal ganglionic eminence (CGE)-derived subclasses (Lamp5, Sncg and Vip) and two medial ganglionic eminence (MGE)-derived subclasses (Sst and Pvalb)], layer and projection pattern in mouse for glutamatergic excitatory neurons [i.e., intratelencephalic (IT), extratelencephalic (ET), corticothalamic (CT), near-projecting (NP) and layer 6b (L6b)], and non-neuronal functional subclass (e.g., oligodendrocytes and astrocytes) (Table 2).” This table lists all five of those types of GABAergic neurons in a single line, while having separate lines for IT, ET, CT, etc. Thus, it is difficult to understand the levels assigned to each of these in the hierarchy.

We thank the referee for the comments on table organization and have reorganized it to be more consistent across subtypes.

How many types are there? For mouse, there are 90 neuronal t-types, but also listed as 56 neuronal cell types (line 205). Maybe it's not clear to other groups outside your own what this difference is. Also potentially confusing is that this lists numbers for neuronal cell types, but then the consensus number includes 8 non-neuronal cell types (line 217). For mouse, 90 t-types, (but claimed 56 neuronal cell types) (line 205). For marmoset, 94 t-types (line 210). For human, 127 t-types (line 210). Conserved t-types are 45, include 24 GABAergic, 13 glutamatergic, and 8 non-neuronal. For comparison, MERFISH has 42 GABAergic, 39 glutamatergic, and 14 non-neuronal (Line 310). So: could you help me to align these numbers with the taxonomy of Figure 10? (Perhaps counted wrong, I got 88 there.) Separately, I understand how these could differ between methodology, but under one assessment there are almost 2x as many GABAergic types as glutamatergic (t-types), while in MERFISH, there are

roughly the same number of GABAergic/glutamatergic types? Why would the number of glutamatergic types seemingly vary this much? Further, when the electrophysiological types are recorded in MOp, these are determined to include 77 t-types (Line 382). Is this greater than the 56 neurons or less than the 90 mouse t-types? There could be many reasons why these numbers don't match but it is somewhat confusing to me since the numbers always vary.

We recognize that all those different numbers of "types" or clusters are indeed very confusing! This is why in our manuscript we presented various correspondence/confusion matrices to show the alignment of clusters across different data types (see for example the original Fig 10, which is now Extended Data Figure 2). These matrices show various degrees of one-to-one, one-to-several, or several-to-several correspondences between clusters of different modalities. They can serve as references for those readers who are interested in detailed comparison of specific cell types.

The numbers of clusters do vary across data types and the exact numbers do not matter much. This is because in the hierarchical tree of cell types, the distinction between subclasses and major types is highly discrete whereas the distinction between leaf-node clusters can be fuzzy and continuous. Intrinsic and/or methodological differences among different data types can lead to varying numbers of clusters derived from a relatively continuous landscape of cellular phenotypes. The above mentioned correspondence matrices help to align clusters across data types. Co-existence of discreteness and continuity is a key feature in cell type diversity, and it makes it difficult to define an exact number of cell types that can be consistent across modalities. In the revised manuscript we have now emphasized this point.

The GABAergic types are grouped by marker genes (Lamp5, Sncg, Vip, Sst, Sst Chodl, Pvalb, and Meis2). I am curious if there are any differences uncovered by this method by interneurons with

similar marker genes (such as PV) and the layer in which the soma resides (a L2/3 PV+ interneuron versus a L5 PV+ interneuron). The excitatory cells are grouped by layer in the Table 2 but the interneurons are not. Although the differences might not be as extreme (as IT versus ET or CT), there must be some difference (as they seem to target somata of excitatory cells only in the same layer). (Lines 217-223). For example, in Fig. 4 e-f, it's suggested that there is some continuous variation in the electrical/morphological axis, but it was not as clear to me whether this is accompanied by some gradient in the transcriptional data. They may fall into the same cluster. But within this cluster, then, do the upper, middle, and lower layers cells also separate into branches within this cluster?

We have indeed observed that some GABAergic interneuron clusters are enriched in certain layers while others are distributed across multiple layers, as shown in much greater detail in our companion MERFISH (Fig 2 in Zhang M et al 2020) and Patch-seq (Fig 1 in Scala et al 2020) studies, as well as a separate Patch-seq study from the mouse visual cortex (Figures 2, 4 and 6 in Gouwens et al Cell 2020). The layer-enriched distribution is particularly pronounced in Sst and Pvalb subclasses. There is also a corresponding gradual transition across layers between transcriptomic and morphoelectrical axes, for example, Pvalb+ basket cells in upper, middle and lower layers are indeed segregated into different transcriptomic clusters (Scala et al 2020, Gouwens et al Cell 2020).

Spearman is sometimes in caps (Fig 2), sometimes not (Line 240).

We have capitalized Spearman in all locations.

For glutamatergic cell types, "the majority of markers were species-specific (Fig. 2f,g)" (Line 249). What does that mean? "The evolutionary divergence of marker gene expression may reflect species adaptations or relaxed constraints on genes that can be substituted with others for related cellular functions." What are the species-specific genes, then? Is it just a matter of different species using different isoforms of the same genes, or something entirely different. The discussion of this (Lines 1147+) touches on what some explanation might be, but doesn't provide an answer. Is it not a little unexpected that the set of marker genes for these types (illustrated in Fig. 2f Venn diagrams) generally shows little overlap across the three species? I understand that L2/3 IT is clustered based on a lot of data but is it not somewhat remarkable that there aren't more shared genes needed to create that conserved cell type? Does the data mean that, with more species-specific markers, that there are multiple ways to achieve the same general morphology/connectivity? Or that lower levels of other genes are what is needed to achieve this (such that they don't meet the bar to be marker genes)?

We thank the reviewer for raising this important topic. We also find it remarkable that we can identify conserved cellular components of M1 in rodents and primates, and yet the expression levels

of thousands of genes have diverged between species for homologous cell types. For example, we reported in Fig 6a of Hodge et al. 2019 (see scatter plot below) that the majority of orthologous genes have similar expression in the Sst Chodl type between species with correlation = 0.66. Conserved genes provide sufficient information to identify homologous types. At the same time, 18% of genes expressed in the Sst Chodl type have >10-fold expression difference between human and mouse. We also report in a companion paper (Bakken et al., 2020) that even genes with similar expression levels between human and mouse cell subclasses can involve dramatically different isoform usage. Few marker genes are conserved because they must not only be expressed in the cell type of interest but also retain their cell type specificity across species.

In the companion paper Bakken et al. 2020, we showed that the majority of subclass markers are expressed in all species but with reduced specificity. The plot below shows the proportion of GABAergic subclass markers identified in only one species that have different thresholds of expression enrichment (LogFC = natural log fold change) relative to the other two species. For example, ~90% of human-specific Sst subclass markers have <1 LogFC (<2.7-fold) enrichment in human Sst neurons compared to marmoset and mouse Sst neurons (yellow dot in 4th column from left). Since these Sst markers are still expressed in marmoset and mouse, they must not be conserved because they have lost specificity in Sst compared to non-Sst neurons in marmoset and mouse. We have changed “species-specific” to “species-enriched” in the text to emphasize that these markers are not exclusively expressed in one species.

We hypothesize that the relatively small number of genes that have conserved expression and specificity are central to cell type identity. For example, in Bakken et al. 2020, we identify a transcription factor RORA that is a conserved marker of chandelier cells and may drive the gene regulatory networks that distinguish this type from basket cells. Other genes that are expressed in chandelier cells in all species and have divergent background expression in other cell types contribute to chandelier properties but are less likely to be core drivers of chandelier identity. Analysis of larger numbers of species will likely be necessary to understand this evolutionary variation and what the core components of conserved cellular architecture is.

Figure 2: Inconsistent use of the blue color for human?

In Figure 2a-h, we have used blue to correspond to results for human. In panels i and j, colors are used to differentiate different data types, and light blue corresponds to ATAC-seq reads.

Figure 2: The recent Berg et al bioRxiv suggested 5 clusters for human L23 IT but here there seem to be 6? Or is this a primary motor versus medial temporal difference? (But in that paper, comparing human MTC to rodent VISp is justified by stating there aren't great differences across areas).

The referee raises an important point in that there is some discrepancy between the results from human temporal cortex and primary motor cortex in terms of the diversity of supragranular excitatory neurons. Glutamatergic neurons in L2/3 of human cortex, including both motor and temporal cortex, are heterogeneous and grouped into discrete cell types, and types with graded transcriptomic differences across layers. Berg et al used the five cell types previously defined in human middle temporal gyrus (Hodge et al. 2019 Nature). Hodge et al had noted that the FREM3 cluster was highly heterogeneous and could be further split with adjusted clustering parameters. Indeed, Berg et al. (Figure 5) reported that superficial and deep L2/3 FREM3 neurons had distinct morphology and electrophysiology and that these properties changed gradually with laminar depth.

Transcriptomically these same general organizational features appear to be true across human cortical areas, with similar heterogeneity in motor cortex to temporal cortex and additional heterogeneity beyond mouse in deeper layer 3. However, there do appear to be differences in M1 clusters compared to those in temporal cortex that either reflect regional differences in diversity, or slight changes in thresholding that differentially split the largest FREM3 cluster (which as we reported previously can be split further with more lenient clustering). Different regions of mouse cortex show similar diversity to one another, and consistently less than human cortical regions examined so far, most in that they lack the deep layer 3 complexity seen in human cortex. These results highlight the importance of characterizing cellular taxonomies with the relative similarity of types and heterogeneity within types across a variety of cellular phenotypes (particularly taking

advantage of non-human primate as this is very challenging to perform in human). Future work may reveal more discrete properties in different modalities, e.g. connectivity, that can be used to refine the discrete glutamatergic neuron populations in human L2/3 and understand regional variation.

Figure 2: What does it mean that L2/3 IT and L6 IT cluster more closely than other glutamatergic cells? These neurons are born at different times in development, so it seems that the clustering is not strictly recapitulating development here (as opposed to the GABAergic cells).

We have previously found that L6 IT neurons share a transcriptomic signature with both L2/3 IT and L5 IT neurons in mouse (Tasic et al. 2018 *Nature*) and human (Hodge et al. 2019 *Nature*) cortex (also see the bent UMAP in Fig. 8a). This is now more apparent in an updated dendrogram showing branch robustness as suggested by the reviewer below. Other than one highly distinct IT type (L6 IT 3), all IT types are on the same branch. The finer groupings of IT types (e.g. the location of the L2/3 IT type) depend on the set of genes that are compared, as indicated by the lower robustness (lighter colors) of the IT branches. Thus, although there is an overall recapitulation of developmental origin in the hierarchical organization it is based on a distance metric that is somewhat dependent on the analysis method so we try not to overinterpret the relatedness among the different types here.

Also, to our knowledge, little is known about the developmental origin and process of deep layer IT neurons. As mentioned in the response to another referee comment above, cell type development involves complex genetic programs that include different progenitor types, lineage progression, and cell birth order/time.

Fig 3: Is there an obvious medial/lateral border at the AGm/AGl interface? This lateral/medial agranular border has been used by others to mark a cytoarchitectural border of MOs and MOp (also called AGm and AGl, or separating whisker M1 and somatic M1 in some publications; Brecht et al., 2004 *J Comp Neurol* for a rat example).

We did not observe an obvious border between MOs and MOp marked by the cell molecular signatures. Almost all cell types are shared in these two regions.

For Fig 3a, it would be useful to either use some indication of pia/white matter or dorsal/ventral to orient the reader. These panels look at though they are rotated compared to the earlier panels in the same figure but it isn't clear to me what rotating the layer structure is offering nor which direction is medial/lateral within M1. Medial/lateral might help interpret the difference in location of the SSp versus MOs projecting cells, though there would also be some use in understanding how the complete the retrograde labeling is (since it looks like you could argue from this image that SSp

and MOs projecting L23 cells don't spatially overlap ... though I don't know if that has been proposed in the literature).

By Figure 3a, we believe the reviewer meant Figure 3e. The spatial plots Figure 3e in the original submitted manuscript were rotated 90° counterclockwise compared to the panels in Figure 3c.

Because the number of retrograde labeled cells in a single slice is low, visual impression does not necessarily lead to statistically significant conclusions. We have now replaced this example map of a single slice with a summary panel showing quantitatively the projection pattern we observed between MOp neuronal types and the three targeting cortical regions.

We acknowledge the limitation (under-labeling) of the CTb-based retrograde labeling method, but in the meantime, our quantifications of the composition of cell clusters projecting to each target region and the projection patterns of each cell cluster are still most likely accurate, because we imaged and quantified a very large number of cells (~190,000 cells from 2 animal replicates for these projection measurements).

Minor point, but the use of color – which is spectacular – will be less accessible to ~8% of male readers (colorblindness). I don't have ideal suggestions about how to handle this, but it strikes me in aligning the colors of the neurons with the clusters (you may very well have picked colors that are easy to separate, but I don't have a viewer for pdf that toggles between RGB view and some simulation of deuteranopia). (Fig 2 and 3 for example; in Fig 4 there are symbols besides the neurons to match to clusters).

Due to the large number of cell clusters, it is impossible to introduce symbols in Figure 3 to label the cell clusters. In Figure 3c, we replaced the color scheme so that similar cell clusters (cell clusters that belong to the same subclass) use similar colors now to emphasize the spatial distribution of the distinct subclasses and the laminar organization of the cells in cortex.

For Figures 2b and 3b, is there any kind of certainty that can be ascribed to the nodes in the tree/dendrogram? Although this sort of categorization is maybe newer to classifying neuron types, there are some tests that evolutionary biology-type researchers apply to evaluate the certainty of the branches in a tree (resample the data by bootstrap 10,000 times and ask how often a certain node emerges, for example; maybe other tests exist). This is only of minor importance, but it is not clear from the data alone if such a large sample of expression data provides categorization that is so clean that noise in the expression data is no issue (which we hope is the case ... but maybe you who

have examined the data know better), or if we are accepting the categorization as-is because we have no means to test if other groupings work too.

We thank the reviewer for the suggested analysis. Branch robustness was assessed by rebuilding the dendrogram 10,000 times with a random 80% subset of variable genes across clusters and calculating the proportion of iterations that clusters were present on the same branch. We have updated the dendrogram in Figure 2b (shown below) to visualize the robustness of all branches. Cell types robustly group by major subclasses and show more variability at the leaves.

Similar analysis has been made for the dendrogram of mouse MOp derived from the consensus analysis of 7 sets of sc/sn RNA-seq (Figure 10a (shown below), which also shows that the nodes at the class and subclass levels are more robust, with more variability at the lower leaf (cluster) level, likely due to less distinctness and more continuity among the clusters within the same subclasses, as described earlier.

For the MERFISH data, we compared the subclasses and clusters with those derived from the consensus sc/sn RNA-seq data analysis. The confusion matrix showed excellent, essentially one-to-one correspondence between MERFISH and the sc/snRNA-seq data at the subclass level, whereas at the cluster level, good correspondence were still observed but with more off-diagonal elements (see companion paper Zhang M et al bioRxiv 2020, Extended Data Figure 2a and Figure 1e, also shown below). These results are again consistent with conclusions described above.

Fig 4a, any idea as to why some of the points sampled by physiology seem so sparse compared to the density of t-type data (“z” and “-“ have a lot of black points; “r” and “x” and “y” have less)? Maybe this just means layer 4 and 6 are boring places to record.

Patch-seq is a much lower throughput method compared to single cell sequencing from dissociated neurons. Therefore, given a limited budget in terms of the number of neurons that we could profile using Patch-seq, as we describe in great detail in Scala, Kobak et al 2020, using various Cre drive lines we tried to cover as diverse a population of neurons as possible. Thus some of the regions in the transcriptomic map were sampled more sparsely.

Fig 4b,c, the electrophysiological recordings aren't mentioned in the legend in great detail. Do the overlaid traces represent three current steps? Hyperpolarizing, near threshold, and maximum firing rate (I think this is specified in other papers from the same group but not here.)

Thanks for pointing this out. We have fixed the legend to clearly describe the electrophysiological traces.

Fig 4h, the small part with so many labels gets hard to follow. Also the island of L5/6 NP seems to be split into two groups (but the cut is so clean it looks like an editing error instead of the data).

We thank the referee for pointing out this unfortunate editing error that has now been corrected, as well as adding a blowup of the ET island.

Line 392-394, “excitatory t-types from the IT subclasses with more similar transcriptomes were located also at adjacent cortical depths, suggesting that distances in t-space co-varied with distances in the me-space, even within a layer”. I believe this is true. Fig 3b makes a case for the laminar claim (with the interesting exceptions of L6 IT grouping with L2/3 to 5 IT and L5 ET with L6 CT). Fig 2b has some agreement but puts L6 IT in the middle of the other IT-types (and not all in the same location), while L5 ET is grouped with IT away from L6 CT. This topographic arrangement suggests that maybe the organization of these groups is not as clear. What do the differences between these hierarchies mean? Since MERFISH and t-types seem to me that they are both examining gene expression, it is not clear that this is a methodological difference. But is it noise, and are these the only nodes that the hierarchical clustering has less certainty assigning?

We should emphasize that since the hierarchical tree is one-dimensional, it does not capture the full extent of the complex multidimensional relationships among cell types. Although the branch length (vertical) does reflect the distance between clusters, the order of the branches may not, as they can be flipped around. Furthermore, the hierarchical clustering and calculation of branch lengths are based on differentially expressed (DE) genes between every pair of clusters. The set of DE genes can be different in different data types (e.g. mouse RNA-seq, mouse MERFISH, human RNA-seq), which can lead to somewhat different branching patterns in different hierarchical trees.

We use both hierarchical trees (one-dimensional) and UMAPs (two-dimensional) to illustrate the relationships among cell types. In terms of the distinction among IT, L5 ET and L6 CT, they are highly distinct among each other, but there is a tendency that L5 ET is closer to all IT subclasses, whereas L6 CT forms a more distinct branch along with L6b and L5/6 NP. Within the IT subclasses, individual clusters do vary along the cortical depth across layers, however, we also observe an interesting closeness (with gene signatures) between L6 IT and L2/3 IT types that brings them together (also can be seen in the bent shape of the IT cell collection in the UMAP of Fig 8a, as well as the bent shape of the UMAP of IT cells in Figure 4a of the MERFISH companion paper (Zhang M et al bioRxiv 2020)).

Similarly, the consensus hierarchy groups all IT cells (including L5 and L6) together, with L5 ET as the nearest outgroup. This immediately proposes the developmental hypothesis that either (a) In order for L6 IT to be transcriptionally similar to the other L2/3, L4/5, L5 IT type cells, it should be born from progenitor neurons at a similar time (perhaps immediately preceding L5 IT generation?), while the

ET neurons (the nearest outgroup in this hierarchy) or CT neurons (even further out) would be born at a different time, or that (b) Hierarchy based on expression is not so strongly reflective of developmental origin and cells can differentiate into terminal neurons with similar expression states despite being born at different times. (a) would be interesting to know (since it would revise the cortical development rule of pyramidal neurons being born 'inside-out'). There is some exploration of this in the companion paper Matho et al, is this known? In trying to ascribe some meaning to the hierarchy, this I think is one potentially interesting question.

The referee raised another important issue - the relationship between transcriptomic types/clusters and their developmental origin/basis. We think this issue should be considered more broadly under the framework of the developmental trajectory of transcriptomic types, i.e. how transcriptomic cell types and hierarchy relate to, or are derived from, genetic programs and developmental origins that include progenitor types, lineage progression, and cell birth order/time. We would like to clarify that birth order (i.e. cell division along lineage progression) is more important than birth date, and birth order is only meaningful if cells are generated from the same progenitor. However, this is a fundamental issue that is currently unsolved in the field. Therefore, the issue raised by Ref 3 here cannot be resolved for at least two reasons. First, the diversity of radial glia progenitor types, whether they are multipotent or fate restricted, remains unresolved. Second, lineage relationships among IT, ET, and CT types are yet to be clarified.

It is quite reasonable to hypothesize that similar transcriptomic types would have similar gene regulatory programs, which may be shaped by common progenitor origin and lineage; sequential birth time (i.e. birth order) could then give rise to related transcriptomic types deployed to different cortical layers in an inside-out sequence. In this scheme, L6, L5, L2/3 IT types are born sequentially and deployed inside-out. We (new Fig 6, and companion paper Matho et al bioRxiv 2020) do provide some support for this scheme. An important issue is how ET and CT types relate to IT and the overall scheme. This is a fundamental issue to be resolved and is beyond the scope of the current paper. We do want to point out that the genetic tools reported in Fig 6 and Matho 2020 are well suited to address this issue by fate mapping the trajectories of cell types from progenitor types (defined by transcription factor expression) to their progeny types defined by multi-modal parameters including transcriptomes.

The cortical layer inset in Fig 10 gives the average or median laminar position of interneurons and pyramidal cells. It is useful in showing how the pyramidal neurons are considered with respect to layer. For interneurons, it is not as clear the use of having a median/average layer position. This emphasizes to me that the interneurons don't subdivide as well into layers (perhaps that there's a good deal of similarity for each cell type across whatever layer it falls into). I'm still curious if there is some kind of layer differences within each interneuron type. But perhaps this is beyond the scope.

We have now moved the original Fig 10, including the MERFISH panel and its cortical layer inset, to Extended Data Fig 2, because it does not represent the full picture as the referee pointed out here. Instead, we provide a brief description of layer distribution of interneuron and excitatory neurons in the text and reference our MERFISH and Patch-seq companion papers for more detailed discussions (see our response immediately above this one). The MERFISH data showed that many of the GABAergic clusters showed laminar distributions and preferentially reside within one or two cortical layers (Zhang M et al 2020).

Line 392-394, “excitatory t-types from the IT subclasses with more similar transcriptomes were located also at adjacent cortical depths, suggesting that distances in t-space co-varied with distances in the me-space, even within a layer”. Fig 4g is not as convincing to me that this is true since the electrophysiological parameters that might be similar at neighboring depths aren’t presented in a manner that shows adjacent populations with similar characteristics. The ordering of the current injection traces according to color code aren’t presented in the same order as the pyramids beneath, for example, and the parameter to extract from the traces to look for co-variation is not pointed out.

We apologize since the way the sentence is written was confusing and misleading. For the IT cells we find that transcriptomic distances between t-types were strongly correlated with the average soma depth differences. This result is quantified and described in detail in the companion paper (Scala, Kobak et al., Nature 2020 published online, see Fig. 4 for details). We also performed this analysis for many other t-types and in summary we find that within major transcriptomic families, morpho-electric phenotypes and/or soma depth often varied smoothly across neighbouring t-types, indicating that transcriptomic neighbourhood relationships in many cases corresponded to similarities in other modalities. Due to space limitation in the current manuscript we do not show the quantification and figures for these results but they are shown in detail in Scala, Kobak et al., Nature 2020.

Line 397 “At the level of single t-types, we found that some t-types showed layer-adapting morphologies across layers (Fig. 4e,f)” Ok. But doesn’t this then suggest that if the same t-type results in variable morphology (Fig 4f) that the threshold for clustering those t-types together is not ideal and there is some transcriptomic difference between a Sst+ neuron in L6 and L2/3 that might explain whether the dendrites are broad or restricted in the same layer?

The referee raises a very good point. To determine if there is a more granular classification in agreement with the differences in morphology across cortical layers we would need much more data from what we currently have available. Moreover, these morphological differences may be defined during development or at the level of isoform gene expression. These are excellent questions and we hope future work will provide answers to these important questions.

For the Betz cell comparison, I appreciate this data and find it interesting. Because of the enormous soma size of Betz cells, and the absence of neurons fitting this in mice, I am curious how the mouse cells to compare were chosen? There are (at least) two subtypes of ET-type L5 neurons in mice (Economo et al, 2018), which seem to project to either thalamus or medulla. I am curious which have more similar properties to the Betz cells (or if it is known which mouse ET type you use for comparison). Since these are somewhat unique, but also seem to fall into similar transcriptomic cluster as non-Betz layer 5 cells, are they similar in e-m properties corresponding layer ET cells? This might be a more fair comparison than mouse – for example, do other layer 5 ET types cells in the primate also have high expression of the Kv channels of Fig. 4L?

The referee raises a great point about whether there are distinctive properties in the different L5 ET types and how that relates to cross-species comparisons to primate Betz cells. In this original analysis we included mouse neurons mapping to any of the 4 L5 ET clusters (L5 ET_1 n=8 47.2 Mohm, L5 ET_2 n=4 42.88 Mohm, L5 ET_3 n=3 46.67 Mohm, L5 ET_4 n=7 43.07 Mohm) that include both the medulla projecting (ET1) and non-medulla projecting types (ET2-4). As shown in the new analysis figure and table below now broken down at the type level, all of these mouse types have higher input resistance than human/macaque L5 ET neurons. Note these finer-grained comparisons were all statistically significant except for the human versus mouse non-medulla projecting types comparison, but this is probably just because of the low n for the human neurons (the effect size is quite large at 1.65). We also note the unique biphasic firing pattern was never observed in the mouse neurons regardless of t type. At the level of gene expression, all L5 ET types in the primate have higher expression of the Kv and Ca²⁺ channels of Fig. 4L. Taken together, many of the observations here may not reflect a Betz specialization per se, but rather a broader specialization of primate L5 ET neurons. Comparison of the morpho-electric properties of different primate L5 ET types will be an exciting avenue for future work, where it will be possible to directly link m-e-t properties to cells directly defined by projection targets by retrograde labeling and/or genetic targeting with cell type-selective enhancers.

Line 488 “Although neurons projecting to different target regions were not completely separated on t-SNE” Does this mean that expression pattern alone is insufficient to tell if a neuron is IT/ET or IT projecting to a given area? Is there a way to distinguish between the retrograde approach “reflecting the high quality of retrograde-labeling of neuronal nuclei” instead of just inadvertently including unlabeled nuclei in the analysis?

Thank you for pointing out that this sentence is unclear. We have removed it, as the results were more clearly described in the rest of the paragraph. In short, the IT vs ET neurons are robustly separated on t-SNE; however, IT neurons projecting to different IT areas are intermingled, so are ET neurons projecting to some of the ET structures. The mixture of these projections on t-SNE means that the predominant variance of DNA methylation does not correspond to the projection target. However, using supervised methods, we were able to make predictions of the projection target of a neuron, as described in the companion paper Zhang Z et al bioRxiv 2020.

From previous studies we know that neurons projecting to IT targets mostly fall into molecularly defined IT clusters and neurons projecting to the ET targets are usually restricted to L5 ET clusters. Thus the biased distribution of retrogradely labelled neurons across these subclasses reflects the high quality of retrograde-labeling of neuronal nuclei. We acknowledged that there are a small proportion of unlabelled nuclei (false negatives from FANS) and mislabelled nuclei (along the injection track) inevitably included in the study, and they are not distinguishable at the single-cell level. However, we reasoned that as long as the correctly labelled neurons are highly enriched compared to the noise, it would not affect our conclusions. In the revised companion paper Zhang et al., we added a quality control step to identify and remove FANS experiments showing low signal to noise ratio, which is based on the cell distribution across on-target or off-target subclasses. All data used in this manuscript passed the quality control criterion.

Fig 5, one weakness of not simultaneously reviewing the Epi-Retro-Seq paper based on the new method is that the figures are more challenging. In Fig 5a, there is a box “P5” in the NeuN/GFP sorting. Is this the retrogradely labeled sample for analysis?

Yes. We added this parenthetically in the main text.

Fig 5c and Fig 5d. I find the results of Fig 5d clear and mostly easy to understand. All the ET projecting cells fall into a single category, two if you count the CT and ET categories in thalamus. Does this reject the Economo et al (2018) result of two separate ET-type populations? In contrast, the data presented in 5c are not as easy to grasp (perhaps because of similar colors), and it leaves the impression there is some signal from every target in every cluster! (Which the more quantitative 5d seems to reject).

Fig. 5b-d showed the analysis at subclass level, which considered L5 ET as one subclass. However, Fig. 5e-g further separated L5 ET subclass into clusters where cluster 0 is enriched for MY-projecting neurons. This is consistent with the conclusion in Economo et al 2018, that L5 ET can be divided into subtypes corresponding to MY and non-MY projecting neurons. This was further illustrated by the integrative analysis with MOp transcriptome data (Fig. 9e), and with Economo 2018 RNAseq data (EDFig. 9 in companion paper Zhang et al.).

Fig. 5c and d are plotted using the same data, where the distribution of different cell populations in Fig. 5c is quantified in Fig. 5d. Fig. 5d shows the difference of subclass distribution between IT and ET more explicitly, while Fig. 5c provides an overview of differences between projections and shows that among the IT clusters the different projections are intermingled on t-SNE in this unsupervised analysis. The color scheme of Fig. 5c is consistent with the companion paper Zhang et al. which is in press with corrections to proofs completed, so we prefer not to change it.

I’m not clear on the less well-defined clusters of IT neurons based on projection patterns. Zhang et al (2020 – bioRxiv, your reference 79) reads as though distinguishing neurons into different groups based on their targets from epigenomic data should be feasible (“Based on their epigenomes, intra-telencephalic (IT) cells projecting to different cortical targets could be further distinguished, and some layer 5 neurons projecting to extra-telencephalic targets (L5-ET) formed separate subclusters that aligned with their axonal projections.”). But the overlap of ACA and S1 projecting data in 5b-d makes it look as though this is not a settled question.

Thank you for raising this important question, which points to what information is captured in t-SNE vs. a supervised method (Zhang et al). t-SNE uses principal components as input and embeds the

data into two dimensions, which will inevitably lose the information from many other dimensions. The mixture of different projections on t-SNE reflects that the predominant variance of DNA methylation does not correspond to the projection target. However, using supervised methods with the projection targets as labels, we were able to capture those more subtle differences in the epigenome, which makes it possible to predict the projection target of a neuron. This is one of the conclusions in Zhang et al., although we did not discuss it in detail in this manuscript. We added a sentence in the last paragraph of the section to make this connection more explicit to the audience.

One of the great things about the collection of this data is that it has been done rigorously, with defined inclusion/exclusion criteria. Because most of the paper refers to mouse motor cortex as MOp, it gives the impression that most of the mouse data is collected in a stereotaxic area that is uniform across the mice. When reaching the section about “MOp projection neuron types and input-output wiring diagram”, the language shifts to specify MOp-ul. Maybe this is just to be more specific or to define a subset of MOp examined in detail for this companion paper. Is there a reason to suspect these results wouldn't generalize to MOp in general?

In the anatomy companion paper, we performed careful 3D delineations of the MOp-ul borders based on extensive multi-modality data, including Nissl-stained cytoarchitecture, retrograde and anterograde connectivity data, gene expression and cre-expression of ~30 driver lines, all of which were registered in the CCF and uploaded into Neuroglancer. Meanwhile, we also refined the laminar organization of the MOp into 11 layers/sublayers by quantitative analysis of distributions of retrograde labeling, gene and cre-expressions, Barseq, and somas of reconstructed single neurons. Consequently, all of our pathway tracing, viral tracing, and Barseq injections, as well as single neuron reconstructions, are largely centered on the MOp-ul. The other functional domains of the MOp, for example, the lower limb or mouth region, have not been analyzed at the same anatomical detail. Therefore, it is important to make a clear statement that anatomy (new Figure 7) focuses on the MOp-ul. However, it can be assumed that different MOp domains are likely to follow the same general organizing principle of a cell type-specific wiring diagram. For example, in both MOp-ul (Figure 7 here and anatomy companion paper) and the ALM (equivalent to the MOp mouth/nose), two major L5 ET types were identified, medulla-projecting neurons in the deep (or lower) sublayer while thalamus-projecting neurons in the superficial (or upper) sublayer, although their axonal terminals in the medulla or thalamus are topographically different.

One problem with Fig 6 is that a lot of the experiments have already been done, though perhaps not all Cre lines by the same lab in the same publication. But the overall findings are all expected – generally it felt as though the results summarized in Lines 573-605 were known to those working in this field. (“For example, the L6 Ntsr1 line revealed a typical CT projection pattern with dense projections specific to thalamic nuclei. This result is consistent with labeling from retrograde injections in various thalamic nuclei (PO, VAL, VM) and cortical areas such as MOs and SSp (Fig. 6b, top).” And “Confirming and extending previous reports 86, we characterized detailed axonal trajectories and terminations of two major types of L5b ET cells, namely medulla-projecting and non-

medulla projecting neurons; both types may collateralize in the thalamus and terminate in the midbrain (Fig. 6d). Individual IT cells across L2-L6 also generate richly diverse axonal trajectories”).

This raised the question of what this adds when done in the context of the other results in the paper. I felt like there wasn't a new insight in this figure. That said, the figure is a nice visual summary.

The MOp was selected through mutual agreement by the entire BICCN as an exemplar brain area to explore the potential of collaborative and integrative studies across molecular, electrophysiological and anatomical disciplines to construct a comprehensive cell type-based description of a mammalian brain structure. Our companion study provides a key complementary anatomical analysis to describe MOp cell types based on their spatial locations and projection targets, deriving a highly comprehensive cellular resolution input-output wiring diagram that anchors the molecular data within the full MOp framework. In this context, our study is an essential part of a roadmap towards creating a combined molecular and cellular description of mammalian brain architecture, enabling future studies of other brain structures at a similar data depth.

While MOp has been extensively studied, these studies lacked the possibility to directly compare and integrate different datasets within the same spatial framework. This general “piecemeal” problem of classic neuroanatomy has led to frequent disagreements between experts, including about such fundamental questions as the delineations between of the MOp areal borders (see more below). Here, our collaborative approach integrates state-of-the-art methods for anatomical labeling, imaging and computational data analysis within the mouse brain common coordinate framework (CCF) to generate an expert consensus-based, encyclopedia-like view of MOp anatomy. This includes the most comprehensive classification of cortical projection neuron types and their laminar distribution to date, derived from retrograde and anterograde tracings with nearly 300 single neuron morphologies, and precise three-dimensional delineation of the MOp-ul borders anchoring the resultant input / output connectivity diagram as an integrated model of MOp brain architecture across three scales:

(1) At the macro-scale: We define for the first time the borders of the MOp upper limb domain (MOp-ul) in 3D based on a cyto-architectural delineation using multimodal data from several traditional Nissl stainings, extensive input / output connectivity data and cell type-specific Cre-driver expression that were analyzed by multiple classic anatomy experts as well as by unbiased machine learning approaches. Importantly, our novel cloud-based Neuroglancer visualization of the eco-registered data within CCF at full spatial resolution allowed expert neuroanatomists from different labs to draw and revise borders dynamically in 3D. This is particularly significant since until now anatomists frequently disagreed about the precise regional borders, including about the MOp delineation drawn for example in the classic Paxinos versus Allen ARA versus Allen CCFv3 atlases. The combined use of these advanced technologies by multiple experts represents a new standard for building 3D brain atlases by wide research community.

(2) At the meso-scale: We defined 25 Mop projection neuron (PN) types based on their stereotyped connectivity to target regions and we used this extended PN classification and the PN somas' distribution to outline novel 11 layer/sublayer MOp organization. This refined laminar organization was further confirmed by Barseq and single neuron morphology reconstruction data. These data thus significantly extend on the classic view of MOp anatomy, such as the classification of IT, PT, CT projection neuron types across 8 cortical sub-layers (e.g. Harris and Shephard, Nat Neurosc. Feb, 2015). Furthermore, we do not claim that our data is the "final" description of MOp projection neuron type complexity, but rather we propose that the concept of a classic laminar description of a cortical area may need to be revised to include considerably more cellular complexity in a more or less continuous gradient across the classic cortical layers.

(3) At the micro-scale: We analyzed the complexity of nearly 300 single neuron morphologies, all integrated and co-registered within the mouse brain CCF. This allowed us to derive the most detailed view of the richness of neuronal morphology and projection diversity for a mammalian cortical structure. These data strongly suggest that the true diversity of PN cortical neuron types is even more complex than the 25 PN types highlighted at the population / meso-scale level. This concept is also supported by MERFISH data recognizing 39 clusters of PN and BARseq data recognizing 18 PN clusters.

In summary, the input-output wiring diagram of the MOp-ul described in our study integrates a broad range of technologies, including pathway tracing, viral tracing, trans-synaptic tracing, molecular barcoding, single neuron reconstruction, 3D whole-brain imaging, and advanced computational analyses and data visualization tools, across all three macro, meso and micro scale resolutions, deriving the most comprehensive view of a brain structure in the mammalian brain to date. Furthermore, we believe that using our study as a roadmap and applying similar approaches across other brain structures will lead to new conceptual models of how information is organized and processed within cortical circuits to drive behavior. Taken altogether, we hope that the Reviewer agrees with us about the scientific and conceptual significance of our study.

It is somewhat awkward to review a paper which includes "Stafford, Daigle, Chance et al., companion manuscript in preparation (Line 664). Some of the excitement of that paper is could be the future access to cell type specific mouse line. Certainly there will be interest in a L5 PT-medulla line. But also challenging as a reviewer when the full story is presented elsewhere. Part of the desire to be able to give this work a more thorough review falls from wanting to place Npnt and Slco2a1 in context with the Matho et al paper (also describing strategies to get at pyramidal neurons), but doesn't have (if I recall) a PT-medulla only line. This results in a situation where half of Fig 7 is summarizing Stafford et al, while the other in about Matho et al and the list of resources for L5 ET, for example (Fig 7e), isn't even merged to include Slco2, Npnt, etc.

Unfortunately we are not able to complete the Stafford, Daigle, Chance et al. paper in time for this publication package. Thus we have included detailed information about the generation of the two

mouse lines for specific L5 ET types (Npnt and Slco2a1) in Supplementary Methods. Taking this reviewer's advice, we have now split the original Figure 7 and integrated the two L5 ET mouse lines into Fig 9 where multimodal analysis of L5 ET types is presented. The remaining part of Fig 7 is only about Matho et al and this figure has been moved to be ahead of the anatomy figure because several of the lines generated in Matho et al have been used for connectivity mapping from MOp. We hope that this rearrangement demonstrates more clearly the actual utilization of these genetic tools.

Although it's presented as a new finding here, it's been published that there are L4-like cells in primary motor cortex of primates (Area 4): García-Cabezas and Barbas, Eur J Neurosci (2014). (Line 812: "L4-like neurons may also exist in human and marmoset M1.") And, as cited here, shown by Yamawaki et al, eLife (2014). Thus, it would be better to focus on how this is extending what's previously known. At least part of it is the molecular characterization (Fig 8b+). I would be interested to see how the MERFISH in Fig 8c, for example, looks if it were done on a slice extending from (more lateral) SSp to MOp to MOs (more medial), as I hypothesize – if this distribution looks like RORb, we would see cells in clearly defined layers (SSp) become less complete layers (as L4 cytoarchitecturally goes away in MOp) and then the L4 cells become less dense (in MOs).

The novel aspect of Fig 8 here is indeed the presentation of multimodal evidence, including molecular evidence, consistently demonstrating the existence of L4 cells in MOp. Please note that as stated in the text, the most specific marker gene for L4 cells is *Rspo1* (shown in the MERFISH panel in Fig 8c), whereas *Rorb* labels more broadly labels the L4/5 IT subclass of cells.

As suggested by the reviewer, we now show a MERFISH slice below, which covers a larger region of SSp, and extend from SSp to MOp and to MOs, with MOp cells shown in darker gray to mark the MOp region. The left panel shows L4/5 IT cells, the middle and right panels show the scaled expression of two L4 marker genes, *Rspo1* and *Rorb*, respectively. The L4/5 IT cells and the two marker genes indeed mark a denser and thicker L4 in SSp as compared to MOp and MOs.

This is beyond the scope of the present paper, but I am curious how the L4-like cells in MOp compare to those in SSp and SSs? There's a recent paper suggesting SSs L4 cells make projections outside their home column (Minamisawa et al, Cell Reports 2018). Made me wonder which pair of L4 cells is most similar (if SSp and MOp were similar with SSs being the most transcriptionally unique).

This is a very interesting question. In our full morphology reconstruction companion paper, Peng et al, 2020, we show that both MOp and SSs L4 cells have long-range intracortical projections outside of their home region, whereas the vast majority of SSp L4 cells do not have long-range projections. In a previous publication from Allen Institute (Harris et al, Nature 2019), we also showed that L4 cells from several cortical areas (e.g. visual and auditory cortex) have long-range projections by both population tracing and single cell full morphology. Thus, the lack of long-projection in SSp L4 cells may be an exception rather than the rule.

On the other hand, in our DNA methylation companion paper, Liu et al, 2020, SSp and SSs L4 cells cluster together, whereas MOp L4 cells cluster separately. This divergence suggests that the major variables in DNA methylation profiles may be driven by factors other than long-range projections.

Similarly, the next outcome highlighted (Lines 839-841) "Here as the second case study, through integrated multimodal characterization we demonstrate that L5 ET neurons in MOp can also be divided into MY-projecting and non-MY-projecting types" is also published (Economo et al., 2018) and it appears that the one class described there can be even further subdivided into two groups (Winnubst et al, 2019 – ref 26). The molecular characterization identifies one marker as Slco2a1 for medulla and Hpgd and Npsr1 for thalamus-projecting types (though these are also the markers highlighted by Economo et al Fig 1.). Is the advance that this distinction applies to MOp in addition to frontal areas?

Indeed one of the advances is that this distinction also applies to MOp. In addition, we have generated transgenic driver lines targeting these two types of L5 ET cells differentially (new figure panel h moved/modified from original Fig. 7). Moreover, we provide the most comprehensive evidence to show that properties in different modalities are aligned and support the division of these two cell types.

Fig 9a is not explained sufficiently for me to understand it. What's the arrow at the side? What does the color intensity mean in the plots (there's no scale/legend, perhaps correlation coefficient)? The legend claims there is good correspondence between the groups, but then the left plot makes it look as though two of the RNA-data based classes are merged in a single MERFISH group, while the last RNA-based grouping includes two of the MERFISH ones, with ET type 4's affinity not as clear.

We have now clarified that in Fig 9a confusion matrices show the proportion of cells in each integrated (SCF) or MERFISH cluster mapped to the transcriptomic types. The arrowhead points to the MY-projecting type. What we meant by good correspondence is between MY-projecting (RNA-seq cluster L5 ET_1) and non-MY-projecting (RNA-seq clusters L5 ET_2-4) types. Indeed, within the non-MY-projecting type, there is not necessarily one-to-one correspondence between RNA-seq, integrated, and MERFISH clusters, likely due to the high degree of similarity and continuity among these clusters in each data modality.

Fig 9c MERFISH image legend refers to Mouse ET_types 1-4 and 5, but then only shows 1-3 and 5.

L5_ET_4 cells are not found in this particular MERFISH section. We have clarified this point in the legend.

Fig 9b compares mouse ET types 1-4 to three human and marmoset types. I am interested if this means that primates have less diversity in some ways in their layer 5 ET cell types? After reading Winnubst (2019) and their description of different subsets of thalamic projecting L5 neurons, I began to wonder if human L5 corticothalamic projection neurons might show a greater diversity, since the human thalamus is bigger and it might thus be possible to differentiate corticothalamic neurons projecting of many different subdivisions (of, for example, motor thalamus – where there is a much more fine division of subregions than in mice).

L5 ET cells are much sparser in human cortex (see Hodge et al. (2019) Nature), and their relative proportion appears to mirror evolutionary cortical expansion compared to the subcortical targets of those neurons. It is a very interesting question whether this sparsification means that there is less diversity in primates, and one that remains to be addressed. The sparsification means that it is difficult to sample those rare cells at adequate numbers to have comparable analyses across species to ask that question adequately. We are pursuing ways to enrich for these neurons to ask that type of question but do not have answers at this point. This is a great question about L5 corticothalamic neurons as well that will be very insightful to pursue in non-human primates where these questions of target specificity versus transcriptomic definition can be directly addressed. We hope that this comparative analysis will allow many such questions to be asked in direct, targeted ways now that the foundational classifications are becoming available!

Fig 10. Lovely figure. The tree at the top has some of the connections labeled with different thicknesses of lines (and different colors). What is the significance of this? I am inferring it might be something about the certainty of the node or perhaps described in a different location.

The thickness and color of the lines in the hierarchical tree indicates the confidence level of each branch. Branch robustness was assessed by rebuilding the dendrogram 10,000 times with a random 80% subset of variable genes across clusters and calculating the proportion of iterations that clusters were present on the same branch.

Lines 1068+ : “This classification provides strong evidence for the existence of hitherto poorly studied but molecularly distinct subclasses such as the near-projecting (NP) pyramidal neurons, and many more novel cell types.” To me, this is pretty exciting but it didn’t make the cut of highlighted bullets. (While more obvious things ... t-type and chromatin state give similar maps ... did.)

We appreciated the referee’s interest in the NP neurons, as their connectivity and function is still poorly understood. The NP neurons were first described by Callaway and colleagues as a third type of L5 pyramidal neurons in the mouse visual cortex that is morphologically distinct from the L5 ET and L5 IT neurons (Kim et al Neuron 2015). The NP transcriptomic subclass was first defined in Tasic et al Nature 2018 in mouse visual cortex and anterolateral motor cortex. Because of these prior studies, we did not highlight the NP neurons as a major new finding.

Line 1162: “The molecular genetic framework of cell type organization established by the current study will provide a robust cellular metric system for cross-species translation of knowledge and insight that bridges levels of organization based on their inherent biological and evolutionary relationships.” Is this a strong argument? I would be concerned that some of the lack of translation to disease would be the feel among human and primate researchers that disorders of frontal areas aren’t well modeled in part because the rodent lacks the expansion of frontal cortex. Similarly, MOp research may be limited because the primary motor cortex of primates has five specialized premotor areas with which it is strongly connected, while rodent MOp has strong reciprocal connections with SSp (qualitatively differing from primates). Thus, while some similarities in intrinsic features might occur, it is difficult to know how this will be useful for studying connections or pathologies that come about with cells embedded in a network (that differs from mouse network). This statement is stronger if disorders were cell-autonomous.

Our main point is that cross species comparison at the cell type resolution will be more informative than at the gross anatomy (areal size and specialization etc) level. A striking result of the studies here is the overall conservation of cellular architecture, but with species variation at the finest levels of granularity and in the specific properties (molecular, anatomical, physiological) of different cell types. The conservation allows comparison of orthologous cell types and a way to measure and understand similarities and differences. At the same time, these differences can be profoundly important, for example for the primate Betz cells and for circuitry as noted by the referee. However, these differences are variations on an overall conserved cellular blueprint, and this provides a

quantitative framework for understanding both similarities and differences into the future as the community builds on it to describe circuitry and relationship to disease among other things. Therefore we argue that the argument is strong in that it sets a strong foundation to understand both conservation and divergence of cellular and circuit organization.

We suggest that cell type level analysis will provide compelling evidence for both the core conservation and substantial divergence of cortical circuits. At the same time, we also agree with the referee that circuit connectivity at multiple levels may be another equally or perhaps even more important factor contributing to the conservation and divergence of brain organization across species. It is likely that we will need the combination of both cell type and connectivity information to gain a full understanding. With more cell type datasets from rodents, NHP and human, and in the long run with more comprehensive circuit connectivity data generation becoming increasingly feasible, we will be in a much advanced position to examine and evaluate what brain disorders can be and cannot be modeled in what animal models.

Referee #4 (Remarks to the Author):

This is a remarkable resource of data accompanied with an integrative analysis of neuronal diversity in motor cortex across three different species. The study provides a detailed blueprint for molecular and functional organization of this cortical region and the data will serve as a valuable resource that will be re-analyzed and interpreted by many labs. While it is not expected that the authors perform exhaustive analysis and interpretation of the data, some critical analysis and discussion seem to be lacking from the manuscript.

Below I detail what in my view would make the manuscript more informative and useful to the community. Most importantly, I would like to see more clear discussion of how different datasets correlate, which datasets are orthogonal to each other, and what analysis is most informative. This would serve as an important guidance for follow-up studies by groups that do not have financial resources for comparably exhaustive cell profiling. In addition, comparisons across species is rather limited and the study does not offer a clear picture whether brain evolution is associated with quantitative changes in cell type distribution or emergence of new cell types.

We have expanded the discussion of these aspects. Please see our detailed responses above to the comments from referees 2 and 3 regarding how different datasets/modalities correlate and how to define cell types in a hierarchical organization.

Regarding the species comparisons, we have begun to see some of the principles of brain evolution but clearly many more species and brain regions are needed to get a clear and coherent picture. Nevertheless, between this manuscript and the two companion manuscripts (Bakken et al. (2020) bioRxiv) and Berg et al. (2020) bioRxiv), we see an overall strong conservation but with quantitative shifts in gene expression, cell type proportions and spatial distributions. While it is not possible to align every cell type at the finest leaf level, the fact that there are ~100 cell types in mouse, marmoset and human suggests similar diversity that may even be possible to align at a finer level with additional species with intermediate evolutionary distance. However, we do observe a number of things. First, the proportions of homologous cell types can vary between species (and cortical regions). Second, gene expression is most similar between closer evolutionary relatives for any given cell type, suggesting divergence as a function of evolutionary distance. Third, homologous cell types can have divergent features, as we show for the Betz cells. Finally, there can be specializations and expansions of specific cell subclasses across species, as shown for the expanded diversity of supragranular excitatory neuron types in Berg et al. (2020) bioRxiv that correlates with the evolutionary expansion of these layers. Together these would seem to lay the groundwork for future comparative work to expand upon these ideas, but likely have begun to reveal a number of principles and ways in which species specialization can manifest. Furthermore, the evolution-associated changes in gene expression include genes associated with neuronal connectivity and function, suggesting that these changes have important functional or circuit wiring implications that will hopefully become interpretable and testable in the future. We have tried to discuss these findings more clearly.

Major points to address:

1. The authors should examine and discuss more in depth orthogonality of individual datasets. Differential connectivity and physiology were historically considered as the key functional determinants of neuronal cell identity. Yet, the molecular data do not clearly resolve such functionally distinct neuronal types (described as continuous variation). Does this mean that the two types of neuronal classification are to large extent orthogonal to each other and therefore molecular classification will always need to be accompanied with anatomical and physiological characterization of neuronal cell types? Are spatial location, DNA methylation, projection, morphology, physiology strongly correlated with gene expression or do they provide additional information? Further, how much of the gene expression variability within clusters can be explained by these different parameters? This can be examined, if not for all cells, for IT cells where integration across different modalities has

already been done.

The reviewer suggests the characterization of the relationship between molecular identities and functional properties (e.g. connectivity) along the hierarchical organization of cell types. Our study shows that spatial location, DNA methylation, projection, morphology, physiology are strongly correlated with gene expression at subclass level, and for some major cell types as well. Thus it is not the case that the two types of classification are to large extent orthogonal to each other.

However, at more granular levels, we do observe a certain degree of differential variations, for example, different morphologically defined PV interneuron “types” (small or large basket cells, translaminal cells, shrunken cells, etc.) can be mapped to the same transcriptomic type, appearing as different parts of a continuum (Scala, Kobak et al 2020). In this case, it is possible that previously recognized functional attributes that were considered discrete may not have captured the full spectrum of variation due to the sparseness of the data. The relative completeness of the transcriptomic profiling (due to its tremendous scalability) is able to fill gaps and provide underlying linkage between different features.

On the other hand, we also observe that certain functionally significant features, such as long-range projections, exhibit greater cell-to-cell variability within transcriptomic types (Peng et al bioRxiv 2020), pointing to more studies needed to better understand the relationship of different modalities.

Moreover, our Patch-seq data enables us to relate transcriptomic profiles to morphological, electrophysiological and anatomical (cortical layer location) information. In agreement with the overall results discussed above we find that transcriptomic subclasses (i.e. neurons expressing Sst, Pvalb etc) have also distinct morpho-electric phenotypes. However, within each family the transcriptomic types are not well separated in the morpho-electric space. Within each subclass we find a continuum of morpho-electric variability. Interestingly, we also find that within each subclass neighbouring transcriptomic types show similar morpho-electric features. In summary, these results propose a model where neurons form a hierarchy with non-overlapping branches at the level of subclasses but continuous and correlated transcriptomic and morpho-electrical manifolds within subclasses.

2. The authors should discuss which type of data are most informative when categorizing neuronal cell types. For example how do the RNA expression, accessibility, and methylation profiles compare in terms of their ability to stratify data into meaningful cell types?

Both transcriptomic and epigenomic data are reflective of the molecular genetic programs defining cell types, and they provide highly complementary information to enrich our understanding of the molecular genetic programs defining cell types. In general, data types with more comprehensive and in-depth coverage will be more informative. In our study, we used RNA-seq data and transcriptomic taxonomy as the anchor for comparison with other data types.

3. In this context, the authors should clarify how they define class, subclass, and type of neurons? It would be helpful to propose and implement a unified neuronal cell type nomenclature that could be applied to additional datasets coming from different labs.

We believe that cell types are best defined in a hierarchical organization. Class, subclass and type define different levels of membership and granularity in this hierarchical organization, which we have now clarified in the revised Table 2.

We strongly agree with the aspiration to propose a unified neuronal cell type nomenclature. While we are not quite at the stage of providing a unified common usage nomenclature for cortical cell types, we have developed a nomenclature system (Miller et al eLife 2021) that can be applied to each dataset in a uniform way that provides a way to cross-reference between projects, and that future studies can be cross-referenced against. Multiple nomenclatures can be applied as aliases to this system. We have added this common nomenclature, and the terms used to describe subclass level cell sets across modalities as a supplemental table.

4. The authors propose that variable projections of neurons belonging to the same molecular type could result from activity based pruning during development. Alternative possibility is that axonal pathfinding is controlled by subtype specific genetic programs active during early developmental stages when neurons are establishing their connectivity, but these programs become extinguished in the mature CNS. These fundamentally distinct mechanisms should be discussed as they would have important implications for the interpretation of the adult molecular data.

We fully agree with the reviewer that there could be two distinct developmental mechanisms that define the variable projections of neurons belonging to the same adult-stage molecular type, a genetic program that only exists during development and an activity-dependent shaping process. We have now discussed these two possibilities more explicitly and clearly.

5. The authors should provide more direct integrative analysis of data across different species. Can the data be shown as degree of overlap between species-integrated clusters and each species-specific cluster? Considering that homologous cell types exhibit great degree of molecular divergence, it would be interesting to see if canonical correlation analysis or manifold alignment can be applied to compare data across different species. Such direct comparison could help to identify truly species specific cell types. It is of interest that number of Parvalbumin cell types seem to be expanded in primates relative to mouse, even with fewer profiled cells. Also there are some human/marmoset clusters that have low overlap with any given mouse cluster; does this suggest a human or primate-specific cell type? Can you get more fine-grained resolution if you re-do this analysis with subsets of data restricted to a specific lineage?

In our companion paper, Bakken et al. 2020, we describe in more detail the definition of cross-species consensus clusters. Briefly, RNA-seq data from all 3 species were integrated with Seurat using canonical correlation analysis, identification of mutual nearest neighbor anchors, and batch correction. Unsupervised clustering of the integrated data identified several hundred metacells that were merged into 45 consensus clusters that contained cells from all species. These consensus types are shown as boxes in Extended Data Figure 1 in this paper. As suggested by the reviewer, we achieved better alignment and finer resolution consensus clusters when we performed this analysis for major cell classes rather than all cells at once. This analysis reveals that the cellular components are largely conserved across these species, and there are no highly distinct types unique to any species.

However, additional cellular diversity is seen in primates for some consensus types, such as Pvalb interneurons, as noted by the reviewer. We also see more specialization of human layer 2 and 3 IT neurons that is described in detail in the companion paper Berg et al. 2020 based on Patch-seq data from human and mouse cortical neurons. We report that two cell types are present in deep L3 of human cortex and do not have matching types found in L2/3 of mouse cortex. Interestingly, these types are most closely related to layer 5/6 mouse glutamatergic types, suggesting that they are either developmentally displaced types or primate-specialized types that are expected to share some cellular properties with deep layer neurons. These changes are likely linked to the expansion of supragranular layers that accompanied the 1000-fold increase in surface area of human cortex relative to mouse and the requirement for greater cortico-cortical connectivity of sub-functionalized areas. These results suggest deep evolutionary conservation of the building blocks of cortex, and species specialization of cell types linked to other structural changes.

Minor points:

1. Figure 7A: what are the different dot colors?

We have now integrated the original Fig 7a-c into Fig 9. Because the expression specificity of the two genes, *Npnt* and *Slco2a1*, to L5 ET types has been shown in Fig.9d, we have now removed the original 7a panel to save space.

2. Figure 8A: It would be interesting to provide similar type of representation for all cells, not only IT neurons

Please see our companion paper, Yao et al, 2020, for similar transcriptomics-epigenomics representation for all cell types.

3. Figure 8C: Include a color scale

Color scale is included now.

4. Figure 8E: It looks like L4 and L4/5 cells are distinct. Is it fair to conclude that these are the same cell type? Is Rspo expressed to the same level in both subsets?

L4 cells belong to one cluster of the L4/5 IT subclass, thus the L4 cell type is distinct from the other L4/5 IT cell types. All L4/5 IT cells are marked by genes such as Rorb, whereas Rspo1 is mainly expressed in L4 cells.

5. Figure 10: Color scale is missing

We have now revised Fig 10 and there is no color scale anymore. We moved the original Fig 10 which contains the correspondence matrices into Extended Data Figure 2 and added color scale there.

6. Was spatial location considered in MERFISH clustering?

We classified the cells based on their gene expression profiles alone. The spatial information was not considered in MERFISH clustering

Referee #5 (Remarks to the Author):

This manuscript by the BRAIN Initiative Cell Census Network represents a monumental collective undertaking by participating labs to create a multimodal taxonomy of the mammalian motor cortex. The results not only advance our understanding of the organization of the cortex but also provide a framework to integrate information from transcriptomic, epigenomic, physiological and anatomical data modalities to create a unified cell type hierarchy. By extending their analysis across different species they also show the evolutionary conservation and variation of these neuronal cell types.

Furthermore, the inclusion of open chromatin and DNA methylation information reveal new future possibilities for studying the establishment and maintenance of neuronal cell types in relation to gene regulatory mechanisms. For these reasons I am confident that the manuscript and its related resources will be of major interest to many readers across diverse fields in neuroscience and biology.

While the authors did a commendable job taking the abundance of data from these different manuscripts and turning it into a coherent whole, the direct connection between the transgenic tools, the input-output connectivity map and the described consensus cell types was somewhat lacking. In other parts the data could be presented in a different way to better support the main conclusions.

We have now revised the text and figure on cell type genetic tools, and have made a stronger and more compelling connection between genetic tools, their use in developmental fate mapping, transcriptomic cell types, and anatomic types (input/output mapping). We also rearranged the order between the original Figs 6 and 7. By strategically targeting the hierarchical organization of excitatory pyramidal neurons (PyNs), focusing on consensus major branches/classes, these transgenic tools provide a much needed experimental framework to dissect finer types (new Fig 6 and companion paper Matho et al bioRxiv 2020), and thus to characterize and validate (or invalidate) the leaf level transcriptomic types.

Many of these driver lines are now key tools for anatomic input/output mapping with subpopulation resolution (new Fig 7 and companion paper Muñoz-Castañeda et al bioRxiv 2020). In particular, inducible driver-based complete single-cell reconstruction is already driving the anatomic typing incorporating key molecular information (companion paper Peng et al bioRxiv 2020). Furthermore, by targeting major transcription factor defined progenitor types, the driver lines provide key starting points to fate-map the developmental trajectories of major PyN types. These will provide a much-needed developmental genetic basis to validate or invalidate the transcriptomic types.

Major Points

The novel genetic driver lines (Figure 6-7) are not directly aligned to the consensus types that are described in the rest of the paper. This makes it hard to interpret how well these tools give us access to the different cell types of the taxonomy put forth in the paper. The authors do acknowledge in the manuscript that this is a major future goal (line 611) but it could be discussed more thoroughly. Furthermore, the authors claim they identified 25 distinct neuron projection types based on unique combinations of axonal targets and laminar distribution. However, this classification is not brought up again or used to classify the coverage of the driver lines. The conclusion that the authors 'provide

a comprehensive identification of major projection neuron types with correspondence to molecular markers' (line 607) therefore seems somewhat overstated.

Our genetic driver lines are designed to target the major PyN subpopulations according to their hierarchical organization. We focused on key transcription factors with demonstrated roles in the specification and differentiation of IT, ET, CT classes and subpopulations within, which are largely consistent with the major transcriptomic subclasses. We expect that these driver lines reflect the developmental relationships among PyN subpopulations, and they will provide a reliable and orthogonal set of tools to evaluate, validate, and characterize the transcriptomic types at different levels of the hierarchy and granularity. We have now reversed the order of Figures 6 and 7 and put the driver line generation ahead of the connectivity mapping, to make the utility of the novel driver lines clearer as some of them were used for connectivity mapping from MOp.

The final section on MY, and non-MY-projecting L5-ET types is supposed to be a case study to 'demonstrate the convergence of properties onto specific cell types' (line 751). However, as the authors note, these observations on distinct molecular, morphological, and laminar distribution have been previously reported in ALM (Economo et al. 2018, Tasic et al. 2018) and have also been partially confirmed in MOp (Winnubst et al. 2019). What new insights have been gained from aligning the data from these different modalities?

We agree with the referee that the transcriptional differences between MY- vs. non-MY-projecting L5 ET neurons have been described. However, we had little knowledge about the gene regulatory information (including the epigenetic landscape and usage of genomic regulatory elements) that drives these different gene expression patterns. With the Epi-Retro-Seq data, we are now able to identify differentially methylated genomic regions that are predictive of active regulatory DNA elements such as enhancers and to computationally infer transcription factor bindings in these regions.

What is the degree of accuracy of the Epi-Retro-Seq approach? In other words, what proportion of cells are classified into t-types that do not match their projection pattern. Looking at figure 5c-d we can see for instance 'near projecting' and IT cells with extratelencephalic projections. It also seems that there is a cell type specific bias. L6CT cells are overrepresented in the unbiased sample (figure 5d) but are only marginally enriched in the TH-projecting sample. What does this mean for interpretation of some of the other proportional data shown in figure 5d?

While Epi-Retro-Seq captures neurons of the targeted projections, we acknowledge that there is a small proportion of contamination of neurons that are not expected to be projecting to the intended targets. The level and source of contamination varies by experiments. Generally, there are two

possible mechanisms that can lead to contamination. 1) Labeling errors. When targeting a deep brain structure, for example TH, cells along the injection track that do not project to the intended target become labeled by AAVretro-Cre. 2) Sorting errors in FANS experiments, where some GFP-nuclei on the boundary between GFP+ and GFP- populations were collected. By carefully examining the cell distribution across subclasses, we identified the potential sources of contamination and removed the FANS experiments with low signal to noise ratio as a further quality control step in the revised companion manuscript Zhang et al. Specifically, we quantified the ratio between the number of cells in expected on-target subclasses (e.g. L5 ET cluster for ET-projecting neurons) vs. in expected off-target subclasses (e.g. IT clusters for ET-projecting neurons), and compared the ratio with the one expected from the unbiased data without enrichment for specific projections. This provides an estimation of signal to noise ratio of each FANS experiment. For IT projections, we used IT subclasses as on-target and L6 CT + inh as off-target, and for ET projections, we used L5 ET as on-target and IT + inh as off-target. With this strategy, we estimated that the correctly labelled cells should be at least 30 (IT) or 200 (ET) fold enriched compared to the noise in this MOp dataset, which makes us confident on the further test of enriched projections in cell clusters. To make this more accessible to the audience, we added this statement in the text as well as a section “Evaluation of contamination in Epi-Retro-Seq” to the Methods.

Following the same rationale, compared to IT neurons (off-target), L6 CT neurons are greatly enriched in TH-projecting samples than in the unbiased sample. The relative less enrichment in L6 CT compared to L5 ET in TH-projecting samples is due to the observation that the efficiency of uptake of AAVretro is much greater for L5-ET cells than for L6-CT cells. This sampling bias does not affect any of our conclusions.

An interesting question that arises from the results presented by the authors is to what extent IT neurons should be viewed as a continuum vs. discrete cell types. The authors note in several instances the relationship between expression, morphology and functional profiles and the cortical depth of IT neurons but don't show any direct quantification or plots for this correlation (similar to figure 5c in Munoz-Castaneda et al. 2020). This seems like an important conceptual advance and it would be informative to see an aggregated analysis across modalities where depth information can be obtained.

The IT neurons do display continuous variation in multiple modalities and across species. Both MERFISH and Patch-seq data showed correlation between gene expression and cortical depth positions of individual cells, as well as with electrophysiological and morphological properties, through more detailed correlation analyses shown in our companion papers (Zhang M et al 2020; Scala et al 2020). In addition to primary motor cortex, a larger transcriptomic study covering the entire mouse isocortex showed that this continuous transition of IT neurons across cortical depth exists in all cortical areas (Yao, Nguyen, et al bioRxiv 2020, ref 53).

Variation across IT neuron types in layers 2 and 3 of human cortex is also described in detail in another companion paper (Berg et al bioRxiv 2020). In the human cortex there is an overall molecular gradient across supragranular IT types, but that can be split into clusters with clearly discriminable properties. One of these clusters displays a particularly strong within-cluster, depth-dependent gradient that is correlated with anatomical and physiological properties. Thus it seems that there are higher order gradients that likely reflect sequential developmental generation from a common progenitor pool. At the same time, there are divisions of this overall gradient with differential laminar positioning, anatomical and physiological properties.

To what extent do GABAergic consensus types have more specific marker expression compared to other types (Line 255)? The data presentation between 2G and 2H is so different that it's hard to draw this conclusion.

On average, glutamatergic subclasses actually have more marker genes than GABAergic subclasses in each species. Glutamatergic neurons express more genes than GABAergic neurons (see ED Fig. 1 from the companion paper Bakken et al. 2020 and below), and 1-5% of expressed genes are markers for all neuronal subclasses. Every subclass has conserved markers across all three species, and there are few conserved markers of L5 IT and L6 IT subclasses (see barplots below).

How do the results in Figure 3e match with the statement in line 340: “each IT cluster projected to multiple regions, with each region receiving input from a different composition of IT clusters”. Why

are there no double labeled cells? Could the authors provide a breakdown of these numbers to support this important observation? Finally, the data is not shown in a very clear way as it's hard to align the cell types with the projection type and there are no cortical layer indicators.

As described in our response above, in order to better present our findings in this part, we replaced this panel (Figure 3e) now with a summary plot to quantitatively show the projection pattern described in this statement. Indeed, because CTb-based retrograde labeling method is known to under-label projecting cells, double projecting cells could be under-represented as compared to single-projecting cells, and in our quantitative analysis shown in Figure 3e, only single-projecting cells are considered.

The authors claim that putative L4 neurons mostly project upward into layer 2/3, while L2 and L3 neurons mostly project downward to layer 5 (line 799). However, looking at the examples shown in figure 8g it seems that this distinction isn't that clear cut. The authors should provide an analysis of normalized axon length per layer to back up this claim.

We have now added quantitative vertical profiles showing the average distribution of local axons along cortical depth for L2/3 and L4-like neurons to Fig 8g. Compared to their soma locations, one can see from these vertical profiles that in addition to the axon collaterals at the same level as the soma, L2/3 neurons have axon projections down to L5 whereas the L4-like neurons have axon projections up to L2/3.

Minor Points

In general, there are several cases where there are no colorbars for the heatmaps and it's not always clearly indicated what they mean or how they are normalized in the figure legend.

We are sorry about these oversights and have now corrected them.

Table 2. The text "The top branches of the CN transcriptomic" should be "The top branches of the transcriptomic". For clarity spell out "OPC cells (OPC)" as "Oligodendrocyte progenitor cell (OPC)".

We thank the referee for the suggestions and corrections which have been done in the revised manuscript.

Figure 1a. Would be informative to see a cross section for all three species with MOp/M1 clearly highlighted.

We have added cross sections for all three species with MOp/M1 highlighted to Fig 1a.

Figure 3c. Very confusing color legend. Can the same color indicate different cell types? Why not color by cell subclass?

We now changed the color scheme so that the cell clusters that belong to the same subclass use similar colors.

Figure 4h. Very difficult to see distribution of individual cells. Can we see an inset with a zoomed view of the L5 ET cluster?. Also a bit confusing that the colors for the species are so similar to the t-types groups.

We have added a zoomed inset as requested, and we hope that the color scheme is easier to interpret with that increased resolution.

Figure 8e. Add a color legend in the figure for MOp:red, VISp: blue.

Done.

Ext Fig. 2b. What are the colors on the left? Why are these not M1-14?

The color bar along the heatmap in original Ext Fig. 2b (now Extended Data Figure 3) indeed represents the putative enhancers in different modules (from M1 to M14). We now mark them more clearly in the revised version.

Line 194. For readability it would be useful to mention data modality of each technique.

Each data modality is shown in Fig. 2a.

Line 247-248. This seems like something for the methods section.

We have moved the text describing differential expression analysis to the Methods section.

Line 602. Missing closing parenthesis.

Corrected.

Line 704. cp is not shown in the figure.

We have substantially revised this figure and related text.

Line 861. "with cluster 0 predominantly projecting to MY while other clusters having variable" → "have"

Corrected.

Referee #6 (Remarks to the Author):

The authors present highly rigorous and expansive multimodal molecular characterization of the primary motor cortex in mouse, marmoset and human. The paper is extremely rich in information, extremely dense in content, and brings together nearly every type of genomic information currently accessible for brain in an integrated fashion. Every single figure brings together yet another dimension of integration, and yet another insightful conclusion about what these integrations can teach us about the human brain. As such, I strongly recommend acceptance of the manuscript, as both the dataset and the integrative analyses detailed will undoubtedly serve as an invaluable source of information for the neuroscience and genomics communities.

The study is not without flaws, utilizing non-state-of-the-art analytical tools and improperly executed analyses, although these are minor in the grand scope of the paper and should be easy to correct. I strongly recommend acceptance pending a few corrections.

As suggested by the reviewer, we have updated the differential expression analysis to use a pseudo bulk approach that enabled use of the well-validated DESeq2 pipeline. We have quantified the robustness of transcriptomic similarity of species consensus clusters by generating a dendrogram based on bootstrapping over variable genes, and this provided confidence in higher level groupings of types by shared developmental origin and long-range projection targets.

Analytical Concerns

Transcription Type Classification: In most of the computational analysis across methods the data is heavily over resolved, with “transcription type” classifications being derived from clustering at arbitrary resolutions rather than any known molecular signatures. For example, the authors claim to identify 24 distinct subtypes of cortical GABAergic neurons, a cell class that is known to contain 5 major distinct populations, only to immediately follow that all 24 can be classified into one of the known 5. The t-types result from the specific clustering paradigm used and most would not be reproducible by a different algorithm. Further, as granularity increases, the signature of any t-type is more likely to be driven by batch-specific and technical variance rather than true biological differences across subpopulations and would be difficult to reproduce in other datasets. Finally, such arbitrary classification is not particularly valuable to the reader who identify or target a particular population in their dataset or experiment, and they acknowledge this as a source of error when attempting to map types across species. It would be better for the authors to compare across known or sufficiently distinct subpopulations. For example, instead of the arbitrary Pvalb_1 and _2, or Astro_1 and Astro_2 perhaps distinguish between protoplasmic/interlaminar/fibrous/etc. and perhaps sub-partition in a non-algorithm-specific fashion if there is enough heterogeneity in the population to justify it.

We respectfully disagree with this referee’s assertion that our clustering based transcriptomic classification is arbitrary, algorithm specific, contaminated by batch effect or other technical variance, and thus is not reproducible, generalizable, and does not reflect true biological difference. In this flagship paper as well as our multiple companion papers, we present a large body of evidence demonstrating the biological relevance of the transcriptomic cell type classification. We also provide a well balanced assessment of the relationship between transcriptomic profiles and other modalities of cellular properties, showing high degree of consistency at major cell type level as well as differential variation and fuzziness at more refined levels.

To obtain data-driven, unbiased classification of cell types, clustering of the transcriptomic data must use statistical criteria rather than known molecular signatures. Indeed, the choice of different clustering methods and statistical criteria, as well as the different depths of coverage of the datasets themselves, can lead to different numbers of clusters, but there are many ways to assess the robustness of the classification result. This topic was thoroughly investigated in our companion paper Yao et al bioRxiv 2020. As described in that paper, our transcriptomic classification is based on the integration of 7 datasets collected using different single-cell or single-nucleus RNA-seq methods, and a central theme of that paper is to conduct extensive analysis using a variety of methods to determine the most robust clustering results. Furthermore, our current transcriptomic classification result for the primary motor cortex is highly consistent with previously published work from other cortical areas in mouse and human (e.g. Tasic et al Nature 2018, Hodge et al Nature 2019, Yao, Nguyen, et al bioRxiv 2020).

It is possible that our transcriptomic clusters are over-resolved. The reason for doing this is to maximally represent the transcriptional variations found in the datasets and present them as

hypotheses for the field to test and opportunities for discovery. In fact, our transcriptomic classification is associated with many new molecular signatures (marker genes) that can be used to identify and target these cell populations for further studies using other approaches by ourselves and by the community at large. The more granular differences may reflect a variety of cellular phenotypes or potential to respond that were not measured here or reflected in relatively gross anatomical or baseline intrinsic physiological characteristics. Furthermore, even many of these finer cell types show conservation across species, at least supporting the idea that they are meaningful and under evolutionary constraint.

In our current study, we did not intend to prove that every cluster is a real cell type. Instead, we state that cell types are best represented in a hierarchical organization, as discussed above. Cell classes or types at different branching levels can be validated using multimodal evidence. Indeed, for GABAergic interneurons, our Patch-seq study presented here (Fig 4, companion paper Scala Kodak et al 2020) as well as an independent Patch-seq study in mouse visual cortex (Gouwens et al Cell 2020) demonstrate that there are multiple types within each of the 5 major distinct populations the referee mentioned (we call them subclasses) based on morpho-electrical-transcriptomic properties. Critically, these Patch-seq studies establish the strong correlation between the transcriptomic types and historical knowledge about GABAergic interneuron types traditionally defined by morphological and physiological features (as summarized in Fig 10 and more extensively listed in Scala Kodak et al 2020).

Differential Expression Analysis: The authors use outdated and inappropriate methods for calling DEGs across cell populations and species. These methods treat each cell as a unique observation rather than each sample and thus are both heavily overpowered and susceptible to batch variance. While this may not be too much of an issue for isogenic mouse lines, the analysis does not consider sex, age or RNA quality of the primate samples, especially human, all of which are major drivers of gene expression differences. The authors should repeat their analysis using an alternative approach that permits complex design. I recommend the authors consider pseudo bulk methods (perhaps within t-types) that can give more accurate estimations of power as opposed to the nonsensical p-values produced by t-test and Wilcoxon when applied to single cell data, as this confounds the comparison of DEG divergence across species, though there may be other ways to correct for this as well.

We thank the reviewer for raising this important point. We have revised the differential expression analysis to use a pseudobulk approach as implemented in DESeq2. Briefly, we summed gene UMI counts for each donor and species and compared gene expression between species treating donors as biological replicates using the standard DESeq2 pipeline. We selected genes with adjusted p-values < 0.05 and fold-change > 4 to focus on robust differences. 95-99% of markers of each cell subclass that were identified in our initial DE gene analysis were identified in the pseudobulk analysis, likely because we initially focused on genes with large expression differences. As shown in Figure 2d, we find the same main biological results as initially reported: human and macaque cell

types are more transcriptomically similar to each other than to mouse, and non-neuronal cells have more evolutionarily divergent expression than neurons.

Small Sample Size: Although it was difficult to parse together which data originated from which sample across the various companion papers, I am under the impression that for each modality the human and marmoset data originated from 2-3 samples each for several of these experiments. If this truly is the case, this is an acceptable sample size, regardless of the number of cells recovered from each tissue sample. The extraordinarily variable nature of primate single-cell data, coupled with the failure to incorporate critical covariates mentioned above would jeopardize replicability, which is already one of the biggest concerns in the field of single-cell genomics.

We report the number of donors for each species and data modality in Extended Data Figure 1c (table copied below) from the Bakken et al. companion paper. Donor number is listed in parentheses where (p) indicates that nuclei were pooled from multiple mice. We had previously reported expression differences across human donors based on single nucleus RNA-seq of temporal cortex (Hodge et al Nature 2019). Remarkably, donor differences are smaller than cell type differences as seen in the tSNE below (Fig. 1b). However, we recognize that interindividual variation remains an important axis of variation. Therefore, in this study, we focused on cell type and species differences with large magnitude differences in addition to statistical significance. Furthermore, evolutionary conservation of cell type markers defined within species is a strong indication for the replicability of these markers in future samples.

The manuscript felt a bit redundant at times, with many paragraphs devoted to introducing yet another data type, and re-listing all the previous data types it was going to be integrated with, and then summarizing once more all the data types that it was just incorporated with. Even though these integrations are indeed very impressive, I would recommend foregoing some of these redundant sentences and paragraphs to leave more space for the actual insights, and for more biologically meaningful conclusions, rather than re-hashing and re-stating the impact of what was just done.

We have tried to reduce redundant descriptions and simplify the text as much as we can.

Figure 10 struck me as more well-suited for extended data.

We have now revised Fig 10 to make it a summary of cell types in MOp. It is thus a central figure for the paper and we prefer to keep it in the main text.

Several of the figures could be further condensed, pushing some of the information to extended data to streamline the presentation in the main text.

We have now condensed the figures and main text as much as we can.

Reviewer Reports on the First Revision:

Referee #2 (Remarks to the Author):

The authors have done a good job in addressing my initial concerns. The flow and synthesis of the paper is much improved. It is of course still daunting to read through this massive amount of work, but this is inherent in the task given by such a synthesis paper. I therefore support publication of this work provided the guidelines given to the authors in terms of length restrictions etc are followed.

Referee #3 (Remarks to the Author):

Quoted from the first review:

>Overall, I would recommend publication or publication with minor revision.

I maintain this opinion. Congratulations to your group and the work presented here and in the companion papers which I imagine were completed during a challenging time for all. It can't have been easy to lead such a large group under these conditions.

All comments below minor:

The paper discussed in Fig 6 (Matho et al; substantial overlap in authorship with the current paper) is also on bioRxiv, but seems to use PT (at least the text header is "PT drivers") instead of ET as the terminology for some subsets of cells (compare your Fig 6 to the Fig 1 in the bioRxiv version uploaded ~March 2021). Because of the text devoted to the rationale for the naming convention, I understand your preference is for ET. Is there some distinction being made that I missed (ET is not replacing PT, but PT is a subset of ET) or is it just some mix of terminology between people who adopt or don't adopt the change? But also PT does show up occasionally in the current paper (Fig 8e).

The treatment of the L4-like neurons in MOp is nice. The image in the rebuttal letter for Rspo and Rorb seems better. The contrast is set better there than in Fig 8c for Rspo1, though it seems to be the same section, and makes it easier to see low non-zero expression in L3 and 5 as well as the loss of granularity of this layer in MOp and MOs. Still, re: "L4-like neurons may also exist in human and marmoset M1" (Line 819), other groups have asserted this before (García-Cabezas and Barbas, Eur J Neurosci 2014 for example) so this is not without some support.

Figure 2b, the confidence color scheme is useful, but it looks like some lines are thicker or thinner (in addition to being different colors, grey is thin and black is thick), if this has meaning, I don't follow. Looks similar in Fig 10, though the confidence scale bar not included this time.

Abstract, Line 10

"single-cell transcriptomes, chromatin accessibility, DNA methylomes, spatially resolved single-cell transcriptomes" These are different data sets but maybe it is not needed to mention both single-cell transcriptomes separately here, as this list is repeated in Line 15-16 ("their transcriptome, open chromatin and DNA methylation maps").

Line 341 "Overall, projection of MOp neurons" = Projection patterns? Projections?

Line 450 Describes example recordings as "exemplary". Do you mean 'example'? (I'm sure your recordings are great, but this implies only the best.)

Line 167-170

The "long-range axon projection patterns of individual glutamatergic excitatory neurons exhibit a complex and diverse range of relationships (between one-to-one and many-to-many) with transcriptomic and epigenetic types, suggesting another level of regulation in defining single-cell connective specificity." Does this bullet mean that transcription/epigenetics doesn't answer why so many long-range projection patterns? Coupled with this, Fig 5 is remarkable to me for how intermingled all the L5 ET types are in Fig 5f (by projection target), as compared to the clusters identified in 5e. I expected that every L5ET would send a collateral to striatum (color scheme makes it hard to tell if some ->STR color is in all of the clusters), but that there might be only 2-3 clusters. Further, still weird that the clusters do not correspond as strongly as I was hoping to projection targets (5g) such that I guess molecular identity is not a foolproof way to ID projection targets, and that this failure is not due to insufficient data (as these datasets are large) but because the biology is hard! Maybe I misunderstand the point here but seems like within this example group that the terminal subgroups of L5ET are difficult to subdivide with this method.

Line 221: snRNAseq finds: "24 GABAergic, 13 glutamatergic, and 8 non-neuronal types"

But then Line 310 "Clustering analysis of the MERFISH-derived single-cell expression profiles resulted in a total of 95 cell clusters in MOp (42 GABAergic, 39 glutamatergic, and 14 non-neuronal)" ... maybe this has to do with how classes are subdivided or clustered, but the vast divergence in numbers stands out while reading. Similarly, in Fig 2 hierarchy, there are ~2 PVALB types (many fewer than somatostatin), while in Fig 3 there are ~12 (more than somatostatin!). I guess I still don't get why.

Referee #4 (Remarks to the Author):

The authors provided explanation to most points raised. While the manuscript remains highly complex and the massive datasets generated did not yield major new insights into the complexity of the nervous system, it will be a useful resource for future, biologically more insightful studies.

Referee #5 (Remarks to the Author):

The authors made significant improvements to the overall structure and presentation of the paper. I especially appreciate the de-emphasize of the exact number of clusters detected (see also comments made by other reviewers). Instead, the focus is now on the observed hierarchical organization with robust cell- and subclasses, that are clearly distinct and preserved across species, and lower level subtypes that are more continuous and variable. The added 'robustness analysis' and the improved figure 10 were especially helpful in this regard.

While the authors addressed most of my comments, I would still argue that the two L5 ET subtypes section is an odd case study since most of the discoveries have been reported before. That being said, I agree with the authors that there is sufficient new data to make this section novel enough. Unfortunately, the authors didn't comment on my question regarding the '25 distinct neuron projection types' that they claim to have identified. Having read it again, I still don't understand how they came upon this number and how meaningful it is. I could imagine other readers being confused as well. It seems to be a rather minor point of the paper though.

I appreciated the new analysis the authors added at my request regarding the enrichment of projection populations using Epi-Retro-Seq. It should be pointed out though that this on-target vs. off-target comparison could obviously not be performed if the projections of the different populations were not known beforehand. While it is not crucial for the specific analysis performed in this paper, this circularity clearly limits the application of this technique in other use cases.

By switching figure 6 and 7 adjusting their content I think the utility of the generated driver lines fit better into the overall structure of the story. It still seems to me though that the true utility of the developmentally induced driver lines is yet to be determined. Studying the developmental relationships amongst PyN subpopulations can therefore be classified as more of a future direction.

Overall, I can strongly recommend the paper for publication.

Author Rebuttals to First Revision:

Referee #2 (Remarks to the Author):

The authors have done a good job in addressing my initial concerns. The flow and synthesis of the paper is much improved. It is of course still daunting to read through this massive amount of work, but this is inherent in the task given by such a synthesis paper. I therefore support publication of this work provided the guidelines given to the authors in terms of length restrictions etc are followed.

We thank the referee for supporting publication of the paper. We have now substantially condensed the paper following the guidelines provided by the editor.

Referee #3 (Remarks to the Author):

Quoted from the first review:

>Overall, I would recommend publication or publication with minor revision.

I maintain this opinion. Congratulations to your group and the work presented here and in the companion papers which I imagine were completed during a challenging time for all. It can't have been easy to lead such a large group under these conditions.

We appreciate the referee's comments. It is indeed not easy to complete the work presented in this flagship paper and all the companion papers. We are glad that we pulled it through.

All comments below minor:

The paper discussed in Fig 6 (Matho et al; substantial overlap in authorship with the current paper) is also on bioRxiv, but seems to use PT (at least the text header is "PT drivers") instead of ET as the terminology for some subsets of cells (compare your Fig 6 to the Fig 1 in the bioRxiv version uploaded ~March 2021). Because of the text devoted to the rationale for the naming convention, I understand your preference is for ET. Is there some distinction being made that I missed (ET is not replacing PT, but PT is a subset of ET) or is it just some mix of terminology between people who adopt or don't adopt the change? But also PT does show up occasionally in the current paper (Fig 8e).

We have now made sure PT is replaced by ET in all places in this flagship paper as well as the Matho et al companion paper.

The treatment of the L4-like neurons in MOp is nice. The image in the rebuttal letter for Rspo and Rorb seems better. The contrast is set better there than in Fig 8c for Rspo1, though it seems to be the same section, and makes it easier to see low non-zero expression in L3 and 5 as well as the loss of granularity of this layer in MOp and MOs. Still, re: "L4-like neurons may also exist in human and marmoset M1" (Line 819), other groups have asserted this before (García-Cabezas and Barbas, Eur J Neurosci 2014 for example) so this is not without some support.

We have now gladly incorporated the MERFISH image for Rspo1 and Rorb shown in the rebuttal letter into the final Fig 7 (old Fig 8). We have also added the Garcia-Cabezas and Barbas ref in the revised text regarding L4-like neurons in human and NHP M1.

Figure 2b, the confidence color scheme is useful, but it looks like some lines are thicker or thinner (in addition to being different colors, grey is thin and black is thick), if this has meaning, I don't follow. Looks similar in Fig 10, though the confidence scale bar not included this time.

We thank the referee for identifying these issues. The line thickness in the taxonomy trees in Fig 2b and Fig 10a (new Fig 1b and Fig 9a) did correlate with the confidence level. However, since it is redundant with the colors, we have now made all the lines at the same thickness. The confidence color scale in Fig 10a (new Fig 9a) is the same as shown in Fig 2b (new Fig 1). We have now added it to the new Fig 9a.

Abstract, Line 10

“single-cell transcriptomes, chromatin accessibility, DNA methylomes, spatially resolved single-cell transcriptomes” These are different data sets but maybe it is not needed to mention both single-cell transcriptomes separately here, as this list is repeated in Line 15-16 (“their transcriptome, open chromatin and DNA methylation maps”).

We prefer to keep the sentences as is, because they do mean different things in these different places.

Line 341 “Overall, projection of MOp neurons” = Projection patterns? Projections?

This is corrected to “projections of MOp neurons”.

Line 450 Describes example recordings as “exemplary”. Do you mean ‘example’? (I’m sure your recordings are great, but this implies only the best.)

This is corrected to “in each cell”, with “exemplary” removed.

Line 167-170

The “long-range axon projection patterns of individual glutamatergic excitatory neurons exhibit a complex and diverse range of relationships (between one-to-one and many-to-many) with transcriptomic and epigenetic types, suggesting another level of regulation in defining single-cell

connectional specificity.” Does this bullet mean that transcription/epigenetics doesn’t answer why so many long-range projection patterns? Coupled with this, Fig 5 is remarkable to me for how intermingled all the L5 ET types are in Fig 5f (by projection target), as compared to the clusters identified in 5e. I expected that every L5ET would send a collateral to striatum (color scheme makes it hard to tell if some ->STR color is in all of the clusters), but that there might be only 2-3 clusters. Further, still weird that the clusters do not correspond as strongly as I was hoping to projection targets (5g) such that I guess molecular identity is not a foolproof way to ID projection targets, and that this failure is not due to insufficient data (as these datasets are large) but because the biology is hard! Maybe I misunderstand the point here but seems like within this example group that the terminal subgroups of L5ET are difficult to subdivide with this method.

Indeed, the current transcriptomic/epigenomic types do not fully explain the many different long-range projection patterns. This is a particularly challenging problem for L5 ET neurons as each of such neurons can have multiple projection targets and thus their full projection patterns are not captured by retrograde labeling from a single target (see our companion paper, Peng et al bioRxiv 2020).

Line 221: snRNAseq finds: “24 GABAergic, 13 glutamatergic, and 8 non-neuronal types”

But then Line 310 “Clustering analysis of the MERFISH-derived single-cell expression profiles resulted in a total of 95 cell clusters in MOp (42 GABAergic, 39 glutamatergic, and 14 non-neuronal)” ... maybe this has to do with how classes are subdivided or clustered, but the vast divergence in numbers stands out while reading. Similarly, in Fig 2 hierarchy, there are ~2 PVALB types (many fewer than somatostatin), while in Fig 3 there are ~12 (more than somatostatin!). I guess I still don’t get why.

As the referees originally pointed out and as we agreed, the exact number of clusters or “types” from each dataset is specific to that dataset and the numbers from different datasets often do not agree with each other, because at the leaf node level of the hierarchical tree the distinction among clusters can be less clear and more continuous, the cross-modality correspondence can also be weaker and differential variations in different modalities can be observed. Thus, in the Discussion section we have now emphasized the point that the exact numbers of clusters are less meaningful. However, in Results, to faithfully represent our analysis result of each dataset, we feel it is necessary to describe the exact numbers of clusters obtained due to the specific nature of each data type and the specific cutoff criteria used for cluster calls. We hope that the higher level and more comprehensive discussion provided in the Discussion section will help readers better understand the nature of cell type organization.

Specifically in the cases the referees mentioned here, the numbers of consensus cell types from cross-species integration (i.e., those from line 221 and Fig 2 hierarchy) are indeed lower than the

numbers of cell types from a single species (i.e., those from line 310 and Fig 3, which are from the mouse MERFISH study), due to the more pronounced differential variations across species (for both biological and technical reasons).

Referee #4 (Remarks to the Author):

The authors provided explanation to most points raised. While the manuscript remains highly complex and the massive datasets generated did not yield major new insights into the complexity of the nervous system, it will be a useful resource for future, biologically more insightful studies.

We thank the referee for the comment. We believe that our study on multimodal cell type characterization did yield major new insights into the complexity of brain cell types, as stated in Abstract, Introduction and Discussion.

Referee #5 (Remarks to the Author):

The authors made significant improvements to the overall structure and presentation of the paper. I especially appreciate the de-emphasize of the exact number of clusters detected (see also comments made by other reviewers). Instead, the focus is now on the observed hierarchical organization with robust cell- and subclasses, that are clearly distinct and preserved across species, and lower level subtypes that are more continuous and variable. The added 'robustness analysis' and the improved figure 10 were especially helpful in this regard.

We thank the referee for recognizing the significant improvements we have made to the paper.

While the authors addressed most of my comments, I would still argue that the two L5 ET subtypes section is an odd case study since most of the discoveries have been reported before. That being said, I agree with the authors that there is sufficient new data to make this section novel enough. Unfortunately, the authors didn't comment on my question regarding the '25 distinct neuron projection types' that they claim to have identified. Having read it again, I still don't understand how they came upon this number and how meaningful it is. I could imagine other readers being confused as well. It seems to be a rather minor point of the paper though.

We agree with the referee that the exact number of projection types is not that meaningful, as it has not been sufficiently validated by different types of studies. Since this is indeed a minor point of the paper, we have now removed the '25 distinct neuron projection types' statement from the paper.

I appreciated the new analysis the authors added at my request regarding the enrichment of projection populations using Epi-Retro-Seq. It should be pointed out though that this on-target vs. off-target comparison could obviously not be performed if the projections of the different populations were not known beforehand. While it is not crucial for the specific analysis performed in this paper, this circularity clearly limits the application of this technique in other use cases.

In cases where the expected results from on- vs. off-target comparisons are unknown, other methods can be applied to evaluate and eliminate off-target labeled cells, such as careful microscopic examination of brain slices during dissection and complete elimination of tissue if there are signs of contamination. Therefore, this does not limit the application of Epi-Retro-Seq in other use cases. We thank the referee for pointing this out, and we have added additional statements in the Methods section "Evaluation of contamination in Epi-Retro-Seq".

By switching figure 6 and 7 adjusting their content I think the utility of the generated driver lines fit better into the overall structure of the story. It still seems to me though that the true utility of the developmentally induced driver lines is yet to be determined. Studying the developmental relationships amongst PyN subpopulations can therefore be classified as more of a future direction.

We agree with the referee that studying the developmental relationships among related cell types is an important future direction. We have stated this in the Discussion section.

Overall, I can strongly recommend the paper for publication.